# Taming Polysemanticity in LLMs: Theory-Grounded Feature Recovery via Sparse Autoencoders

**Siyu Chen**[*]    **Heejune Sheen**[*]    **Xuyuan Xiong**[†]    **Tianhao Wang**[§]    **Zhuoran Yang**[*]

[*]Department of Statistics and Data Science, Yale University
[†]Antai College of Economics and Management, Shanghai Jiao Tong University
[§]Toyota Technological Institute at Chicago

{siyu.chen.sc3226, heejune.sheen, zhuoran.yang}@yale.edu
xxy2021@sjtu.edu.cn    tianhao.wang@ttic.edu

## Abstract

We study the challenge of achieving theoretically grounded feature recovery using Sparse Autoencoders (SAEs) for the interpretation of Large Language Models. Existing SAE training algorithms often lack rigorous mathematical guarantees and suffer from practical limitations such as hyperparameter sensitivity and instability. We rethink this problem from the perspective of neuron activation frequencies, and through controlled experiments, we identify a striking phenomenon we term ***neuron resonance***: neurons reliably learn monosemantic features when their activation frequency matches the feature's occurrence frequency in the data. Building on this finding, we introduce a new SAE training algorithm based on ***bias adaptation***, a technique that adaptively adjusts neural network bias parameters to ensure appropriate activation sparsity. We theoretically prove that this algorithm correctly recovers all monosemantic features when input data is sampled from our proposed statistical model. Furthermore, we develop an improved empirical variant, Group Bias Adaptation (GBA), and demonstrate its superior performance against benchmark methods when applied to LLMs with up to 2 billion parameters. This work represents a foundational step in demystifying SAE training by providing the first SAE algorithm with theoretical recovery guarantees and practical effectiveness for LLM interpretation. Code is made available at https://github.com/FFishy-git/TamingSAE_GBA.

## 1 Introduction

Large Language Models (LLMs) have demonstrated remarkable capabilities across diverse tasks. It is found that LLMs encode vast amounts of information by *superposition* (Lu et al., 2024; Xiong et al., 2024; Elhage et al., 2022; Bengio et al., 2013)—packing multiple concepts into the same weight or activation directions to maximize capacity. This efficiency comes at a cost: individual neurons (or activation vectors) become polysemantic (Scherlis et al., 2022), meaning they respond to several monosemantic features at once, making interpretation challenging.

Dictionary learning has recently been applied to disentangle polysemantic LLM representations, with Sparse Autoencoders (SAEs) emerging as a leading approach (Cunningham et al., 2023; Bricken et al., 2023; Templeton et al., 2024; Gao et al., 2024; Rajamanoharan et al., 2024b). An SAE encodes an LLM's internal activation $x \in \mathbb{R}^d$ into a high-dimensional, sparse code $z = f_{\mathrm{enc}}(x) \in \mathbb{R}^M$ with $M \gg d$, then decodes $\widehat{x} = f_{\mathrm{dec}}(z) \approx x$. By enforcing sparsity—so only a few components of $z$ are nonzero—each active neuron ideally reflects a single interpretable feature. Empirically, SAEs have revealed such monosemantic features in models like Pythia-70M (Cunningham et al., 2023) and Claude 3.5 Sonnet (Templeton et al., 2024).

Despite these promising empirical advances, existing studies on SAEs still *lack rigorous guarantees* regarding feature recovery. Popular SAE training algorithms, which typically minimize a loss function of the form $\mathcal{L}(x, \widehat{x}) = \|x - \widehat{x}\|_2^2 + \lambda \cdot R(z)$ where $R(z)$ is a sparsity regularizer, involve

Figure 1: **Illustration of SAE architecture and neuron resonance (left) and a demo neuron (right) learned using GBA.** Left: SAE architecture and the resonance phenomenon—neurons successfully learn features when their activation frequency $p$ matches the feature occurrence frequency $f$. Right: a neuron that activates for the concept "class".

hyperparameters like $\lambda$. For instance, methods employing $L_p$ regularization for $R(z) = \|z\|_p$ and $p \in \{0, 1\}$. Other strong candidates include the TopK activation method (Makhzani & Frey, 2013; Gao et al., 2024) and gated SAE (Rajamanoharan et al., 2024a). However, these methods exhibit specific drawbacks. For example, $L_1$ regularization is sensitive in the hyperparameter $\lambda$ and often leads to activation shrinkage, where the magnitudes of the learned features are systematically underestimated (Tibshirani, 1996). TopK approaches, while enforcing a hard sparsity constraint, often overlook the fact that different inputs may require varying numbers of active features, and also suffer from *inconsistency* across random seeds (Paulo & Belrose, 2025), which means that they yield sets of learned features that are sensitive to the random initialization (Paulo & Belrose, 2025).

This landscape motivates us to address fundamental questions concerning the reliability and theoretical underpinnings of feature recovery with SAEs:

*What enables neurons to successfully recover features? Can we design a training algorithm that provably recovers features while being practical for modern LLMs?*

Let us consider what makes SAE training successful. In an ideally trained SAE, each neuron learns a distinct monosemantic feature and activates precisely when that feature appears in the input. Thus, the neuron will have an *activation frequency $p$*—the fraction of inputs for which it activates—that matches the *occurrence frequency $f$* of its corresponding feature in the data. This observation raises a natural question: if we control neurons to activate with frequency $p$ matching a feature's frequency $f$, will they reliably learn that feature? Moreover, since we typically cannot know a feature's frequency $f$ in advance, what conditions on a neuron's activation frequency $p$ enable it to learn a feature with unknown frequency $f$?

To investigate these questions, we conducted controlled experiments on synthetic data with known feature frequencies. Our experiments reveal a striking phenomenon we term **neuron resonance**: *Neurons reliably recover features when their activation rate matches the feature's frequency in the data.* Like a radio tuning to a specific frequency for a clear signal, SAE neurons must "resonate" at the right activation rate to capture their target features. Importantly, our theory shows that successful learning requires only that $p$ fall within a *resonance band* around $f$, not an exact match. This flexibility enables practical feature discovery: even without knowing $f$ in advance, we can recover features by ensuring neurons' activation frequencies cover a diverse range.

The resonance principle reveals a fundamental yet intuitive correspondence: common features require frequently active neurons, while rare features need selective, infrequently-firing neurons. Based on this, we develop **Group Bias Adaptation (GBA)**, an algorithm that creates multiple groups of neurons with geometrically-spaced target activation frequencies (e.g., 10%, 5%, ...). Each neuron computes $z_m = \phi(w_m^\top(x - b_{\mathrm{pre}}) + b_m)$, where $w_m \in \mathbb{R}^d$ is the weight vector, $b_m \in \mathbb{R}$ is the bias, $b_{\mathrm{pre}} \in \mathbb{R}^d$ is the shared pre-bias, and $\phi$ is the activation function (e.g., ReLU). GBA dynamically adjusts these biases to match the target frequencies: decreasing bias if fires too frequently to increase selectivity, and increasing bias when rarely fires to encourage activation. The direct frequency control across diverse activation ranges ensures comprehensive feature recovery while circumventing the hyperparameter sensitivity and dead neuron problems in existing methods.

We thus provide affirmative answers to both fundamental questions posed earlier through the following contributions. First, we discover and investigate the **neuron resonance phenomenon**, revealing the principle that governs successful feature learning in SAEs from the view of neuron activation frequency. **Theoretically**, we justify the resonance principle by rigorously showing that neurons with appropriate activation frequencies can provably recover all monosemantic features when data follows a well-defined statistical model. To our best knowledge, this provides the first dynamical

analysis and learning guarantee for SAE training. **Empirically**, we scale GBA to Qwen2.5-1.5B and Gemma2-2B on Pile datasets and demonstrate its superiority: (i) achieving the Pareto frontier in reconstruction-sparsity tradeoff comparable to TopK, (ii) significantly higher cross-seed consistency than TopK, (iii) competitive performance on SAEBench (Karvonen et al., 2025) interpretability metrics while maintaining 99% neuron aliveness, and (iv) remarkable consistency and robustness through ablation study, requiring only simple hyperparameter rules without dataset-specific tuning.

**Related works.** Our work builds on a long history of sparse dictionary learning (Olshausen & Field, 1996; Spielman et al., 2012; Bruckstein et al., 2009). Recently, SAEs have been increasingly used for LLM interpretation, revealing monosemantic features and circuit patterns (Bricken et al., 2023; Dunefsky et al., 2024; Ameisen et al., 2025), cross-run feature stability (Papadimitriou et al., 2025), and behavior steering (Shu et al., 2025). A detailed discussion is deferred to §A.

**Notations.** Let $\mathbb{R}_+$ denote the set of non-negative real numbers. We use standard Big-$O$ and small-$o$ notation and use $a \gtrsim b$ to hide $\mathrm{polylog}(n)$ factor for sufficiently large $n$. We denote by $[n]$ the set $\{1, 2, \ldots, n\}$ for positive integer $n$.

## 2 PRELIMINARIES

**A model for feature recovery.** As a motivating example, consider how a model processes "The detective found a muddy footprint near the broken window." The internal representation mixes monosemantic features:

$$x = h_1 \cdot v_1 + h_2 \cdot v_2 + \ldots, \quad \text{where } v_1 = \text{``muddy footprint''}, \ v_2 = \text{``broken window''}.$$

Here, $h_1, h_2 \geq 0$ are nonnegative coefficients, where negative values would imply contradictory concepts. We formalize this as follows: Let $V \in \mathbb{R}^{n \times d}$ be a feature matrix where each row $v_i$ is a monosemantic feature. For $N$ data points, each row $x_\ell$ of data matrix $X \in \mathbb{R}^{N \times d}$ is an $s$-sparse mixture of features with nonnegative coefficients collected in $H \in \mathbb{R}_+^{N \times n}$:

$$X = HV \in \mathbb{R}^{N \times d}. \tag{2.1}$$

We focus on the superposition regime where $n > d$, meaning features are necessarily linearly dependent (Arora et al., 2018; Olah et al., 2020; Elhage et al., 2022). Our goal is to recover $V$ from $X$ without knowing $H$—a common challenge in model interpretation.

**SAE architecture.** We follow Gao et al. (2024); Cunningham et al. (2023) and use a three-layer neural network for SAE with tied encoding and decoding weights. Let $M$ be the width of the SAE, and for input $x \in \mathbb{R}^d$, its output is

$$f(x; \Theta) = \sum_{m=1}^{M} a_m w_m \phi(w_m^\top (x - b_{\mathrm{pre}}) + b_m) + b_{\mathrm{pre}}. \tag{2.2}$$

where $\Theta = \{w_m, a_m, b_m, b_{\mathrm{pre}}\}_{m=1}^{M}$ denotes the trainable parameters. For each neuron $m \in [M]$: $w_m \in \mathbb{R}^d$ is the tied encoder/decoder weight, $a_m \in \mathbb{R}$ is the output scale, $b_m \in \mathbb{R}$ is the bias, and $b_{\mathrm{pre}} \in \mathbb{R}^d$ centers the input. The pre-activation is $y_m = w_m^\top (x - b_{\mathrm{pre}}) + b_m$, and neuron $m$ is *activated* when $y_m > 0$. When activated, neuron $m$ contributes $a_m \cdot w_m \cdot \phi(y_m)$ to the reconstruction, where the tied weight $w_m$ serves as both detector (encoder) and reconstructor (decoder).

**Existing SAE training methods.** Prior methods minimize reconstruction loss $\mathcal{L}_{\mathrm{rec}}(x; \Theta) = \frac{1}{2}\|f(x; \Theta) - x\|_2^2$ with sparsity constraints. $L_1$ SAE adds penalty $\lambda \sum_{m=1}^{M} \|w_m\|_2 \cdot \phi(y_m)$ but suffers from shrinkage bias (Tibshirani, 1996). TopK SAE (Makhzani & Frey, 2013; Gao et al., 2024) retains only $K$ largest activations but exhibits extreme seed sensitivity (Paulo & Belrose, 2025). Both methods have significant limitations detailed in the introduction.

## 3 RETHINKING HOW SAEs LEARN: NEURON RESONANCE

We rethink SAE training from the perspective of *neuron activation frequency*: how should a neuron's activation frequency $p$ relate to a feature's occurrence frequency $f$ for reliable feature learning?

To investigate this, we conducted controlled experiments using synthetic data generated from (2.1). We construct $s$-sparse coefficient matrices $H$, which have uniform feature occurrence frequency $f = s/n$. The selected features can be viewed as the "concepts" in each data point, and $f$ reflects how often each concept appears in the data. We generate the feature matrix $V$ by randomly sampling

$n$ vectors from the unit sphere in $\mathbb{R}^d$, mimicking independent features in high-dimensional space. To study the relationship between neuron activation frequencies $p$ and feature occurrence frequencies $f$, we train a set of SAEs while systematically controlling $p$ through dynamic bias adaptation. More details can be found in §D.1 and the bias adaptation can be found in §4. We measure feature learning success using the Feature Recovery Rate (FRR), which quantifies the percentage of features learned by at least one neuron (see §C.4). The relationship between $p$, $d$, and FRR is shown in Figure 2.

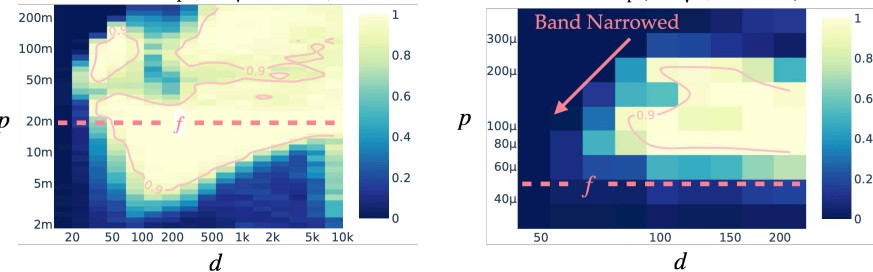

Figure 2: **Feature Recovery Rate (FRR) for varying activation frequencies $p$ and dimensions $d$.** **Left:** Light superposition ($d > \sqrt{n}$, $n = 128$). **Right:** Heavy superposition ($d < \sqrt{n}$, $n = 65536$). In the left panel, the high-FRR region forms a **wide band in $p$ above $f$**. In the right panel, it collapses into a **narrowing diagonal band**, where $p$ must track $f$ tightly. The contrast between wide and narrow resonance bands provides empirical evidence for the theoretical phase transition at $d \approx \sqrt{n}$, showing that the same feature frequency $f$ yields different learning tolerances depending on superposition level. Here, $\mu$ stands for $10^{-6}$ and $m$ stands for $10^{-3}$.

**Neuron resonance phenomenon.** The results reveal a striking pattern we term *neuron resonance*: neurons successfully learn features when their activation frequency $p$ falls within a specific band around the feature's occurrence frequency $f$. The width of this resonance band depends critically on the degree of superposition. In heavy superposition where $d < \sqrt{n}$ (right panel), the band is narrow, requiring $p$ to closely match $f$. In light superposition where $d > \sqrt{n}$ (left panel), particularly when $d > n$, the band widens significantly. This widening is intuitive: when $d > n$, features become nearly orthogonal and easier to separate, allowing neurons with imperfect frequency matching to still converge to individual features due to reduced interference. Since real-world data typically exhibits heavy superposition ($n \gg d$), we expect the resonance phenomenon to persist with a narrow band similar to the right panel of Figure 2. Here, we set $s = 3$, $M = 512$ (left) and $M = 262k$ (right).

In §6, we theoretically characterize a feasible activation frequency range for faithful feature recovery. A feature with occurrence frequency $f$ is learned when neurons' activation frequency $p$ lies in the resonance band $f \lesssim p \lesssim \min\{\sqrt{f}, df\}$ (up to logarithmic factors). With $f = s/n$, a phase transition occurs at $d = \sqrt{n}$: light superposition ($d > \sqrt{n}$) yields a wider band $p \lesssim \sqrt{f}$, while heavy superposition ($d < \sqrt{n}$) constrains it to $p \lesssim df$, narrowing as $d$ decreases. This phase transition and narrowing band in heavy superposition perfectly matches our empirical findings in Figure 2.

**Motivation for frequency-aware training.** Existing methods cannot directly control neuron activation frequencies. They achieve this by imposing sparsity constraints: $L_1$ SAE uses penalty terms while TopK SAE limits the number of active neurons per input. The resonance principle indicates that optimal feature learning requires aligning neuron activation frequencies with the natural feature frequency distribution—ranging from high-frequency features (e.g., common function words) to low-frequency features (e.g., domain-specific terminology). This insight motivates our Group Bias Adaptation algorithm in the next section.

## 4 ALGORITHM: GROUP BIAS ADAPTATION

**From resonance principle to algorithm design.** The neuron resonance phenomenon (§3) reveals that successful feature learning requires matching neuron activation frequencies to feature occurrence rates. This insight motivates our Group Bias Adaptation (GBA) algorithm, which operationalizes the resonance principle through two key design choices:

1. **Direct frequency control**: Instead of relying on indirect penalties ($L_1$) or fixed constraints (TopK), we directly control each neuron's activation frequency through adaptive bias adjustment. When a neuron fires too frequently, we decrease its bias to make it more selective; when it rarely fires, we increase its bias to make it more active.

2. **Multiple frequency bands**: We partition neurons into groups with geometrically-spaced target activation frequencies (e.g., 10%, 5%, 2.5%, ...), creating a spectrum of "resonance bands" that automatically covers the diverse feature frequency range—from common features to rare, specialized ones. Then we use the previously described adaptive bias adjustment within each group to maintain the desired activation frequency.

These design principles ensure: (i) sufficient sparsity for interpretability while avoiding dead neurons by controlling the lowest activation frequencies, and (ii) smooth training dynamics via *adaptive bias adjustment* while maintaining efficient control. The complete algorithm is presented below.

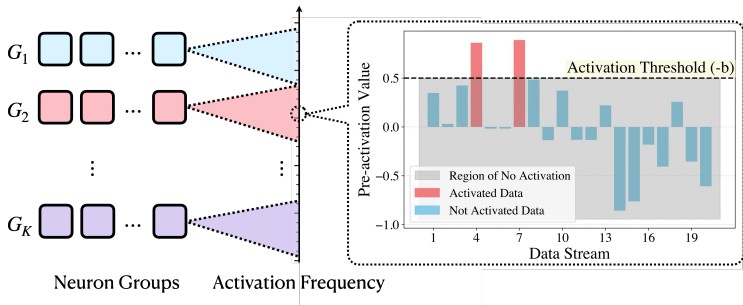

Figure 3: **Illustration of Group Bias Adaptation (GBA). Left**: Neurons are partitioned into $K$ groups with geometrically-spaced target activation frequencies (TAFs) from 10% to 0.01%, creating resonance bands that match the natural feature frequency distribution. **Right**: Bias adaptation mechanism—if a neuron over-activates ($\widehat{p}_m > p_k$), we decrease its bias to make it more selective; if it under-activates ($\widehat{p}_m < \epsilon$), we increase its bias using the group baseline $\bar{r}_k$ to make it more sensitive.

**Neuron grouping strategy.** To cover the diverse feature spectrum, we partition the $M$ neurons into $K$ groups (default $K = 10$) with geometrically-spaced target activation frequencies (TAFs). Specifically, we fix the decaying ratio $p_k/p_{k+1}$, yielding TAFs from 10% down to 0.01%. This geometric spacing naturally matches the long-tail distribution of feature frequencies in language—from common words to rare technical terms (see §C.3 for a detailed justification). Each group contains $M/K$ neurons sharing the same TAF $p_k$ within the group.

---

**Algorithm 1** Group Bias Adaptation (GBA)

---

1: **Input:** data $X$, initialization $\Theta^{(0)}$, neuron groups and desired target activation frequencies $\{G_k, p_k\}_{k=1}^K$, a first-order optimization algorithm $\mathtt{Opt}$
2: **Hyperparameters:** $T$, $L$, $B$, $\gamma_+$, $\gamma_-$, and $\epsilon$
3: For all $m \in [M]$, initialize buffer $\mathcal{B}_m \leftarrow \varnothing$
4: **For** $t = 1, \ldots, T$**:**
5:      ▷ *Forward pass, backward with reconstruction loss, optimizer step with fixed biases.*
6:      Sample mini-batch $X_t \in \mathbb{R}^{L \times d}$, row-normalize, and compute: $\mathcal{L}^{(t)} \leftarrow \mathcal{L}_{\text{rec}}(X_t; \Theta^{(t-1)})$
7:      Backward and optimizer step (exclude biases): $\Theta^{(t)} \leftarrow \mathtt{Opt}(\Theta^{(t-1)} \setminus \{b^{(t-1)}\}, \nabla \mathcal{L}^{(t)})$
8:      Append pre-activations to buffers: $\mathcal{B}_m \leftarrow \mathcal{B}_m \cup \{y_{m,1}^{(t)}, \ldots, y_{m,L}^{(t)}\}$ for all $m$
9:      ▷ *Bias adaptation: when buffers reach size $B$, update biases and clear buffers.*
10:      **If** $|\mathcal{B}_1| \geq B$ **then**
11:          ▷ *Compute per-neuron activation frequency $\widehat{p}_m$ and max pre-activation $r_m$ in buffer.*
12:          Set $\widehat{p}_m \leftarrow |\mathcal{B}_m|^{-1} \sum_{y \in \mathcal{B}_m} \mathbb{1}(y > 0)$ and $r_m \leftarrow \max\{\max_{y \in \mathcal{B}_m} y, 0\}$ for $m \in [M]$
13:          ▷ *Average positive max pre-activation $r_m$ for each group as group baseline $\bar{r}_k$.*
14:          Set $\bar{r}_k \leftarrow \left(\sum_{m \in G_k} \mathbb{1}(r_m > 0)\right)^{-1} \sum_{m \in G_k} r_m$ for $k \in [K]$
15:          ▷ *Adjust biases based on target activation frequency $p_k$.*
16:          **For each group** $k = 1, \ldots, K$ **and each neuron** $m \in G_k$**:**
17:              If $\widehat{p}_m > p_k$, set $b_m \leftarrow \max\{b_m - \gamma_- r_m, -1\}$
18:              If $\widehat{p}_m < \epsilon$, set $b_m \leftarrow \min\{b_m + \gamma_+ \bar{r}_k, 0\}$
19:          Clear buffers: set $\mathcal{B}_m \leftarrow \varnothing$ for all $m$
20: **Return** the final SAE parameters $\Theta^{(T)}$

---

**Tracking activation frequencies.** We measure each neuron's empirical activation frequency using buffered pre-activations. For neuron $m$ with weight $w_m$ and bias $b_m$, the pre-activation is $y_m(x) = w_m^\top(x - b_{\text{pre}}) + b_m$. During training, we accumulate $B$ samples in a buffer and compute the empirical frequency $\widehat{p}_m$ as shown in Algorithm 1 line 12. While the algorithm shows storing full pre-activations for clarity, the implementation is memory-efficient: we only track each neuron's maximum pre-activation $r_m$ and activation count, updating these statistics incrementally.

**Adaptive bias updates.** Biases adapt to maintain target activation frequencies through feedback control. For neuron $m$ in group $k$, we compare its empirical frequency $\widehat{p}_m$ to its target $p_k$:

1. If $\widehat{p}_m > p_k$ (over-active): decrease bias $b_m \leftarrow b_m - \gamma_- r_m$ to make the neuron more selective
2. If $\widehat{p}_m < \epsilon$ (under-active): increase bias $b_m \leftarrow b_m + \gamma_+ \bar{r}_k$ to make it more sensitive

Here $r_m$ is the neuron's max pre-activation for proportional decrease of over-active neurons, while $\bar{r}_k$ is the group average for boosting under-active neurons. We clamp biases to $[-1, 0]$ to prevent extreme values, which should not loss of generality since inputs are normalized. To give a sense of how the bias adapts, we find the rates $\gamma_+ = \gamma_- = 0.01$ provide smooth adjustment to the bias without oscillations in the loss when we perform one adaptation step against every 50 optimizer steps with batch size $L = 512$.

**Summary.** GBA integrates with standard SAE training through periodic bias adaptation. The training alternates between: (1) a gradient phase where a standard optimizer (Adam/AdamW) updates weights $W$ and output scales $a$, and (2) an adaptation phase that adjusts biases when the buffer reaches $B$ samples. Crucially, biases are excluded from gradient updates and controlled only through the adaptation mechanism. Algorithm 1 presents the complete procedure. This design ensures each neuron finds its resonant features through frequency matching. The groups create "resonance bands" covering the feature spectrum, while adaptive bias control maintains target frequencies despite training dynamics. Features naturally migrate to neurons with matching activation rates.

## 5 EXPERIMENTAL EVALUATIONS

To demonstrate the effectiveness of our proposed method, we conduct experiments on the Qwen2.5-1.5B base model (Yang et al., 2024) using Pile Github and Pile Wikipedia datasets (Gao et al., 2020). We train SAEs with 66k hidden neurons attached to the MLP outputs at layers 2, 13, and 26. We evaluate each method using two metrics: (1) reconstruction loss and (2) average fraction of activated neurons per input. All methods employ JumpReLU activation (Erichson et al., 2019; Rajamanoharan et al., 2024b) for optimal performance. We compare GBA against three baselines: $L_1$, TopK, and Bias Adaptation (BA)—a single-group variant of GBA with fixed target activation frequency $p$. Additional details and comparisons between ReLU and JumpReLU are in §D.

**Evaluation metrics.** Beyond reconstruction loss and sparsity, we use the following metrics throughout our experiments (see §C.4 for formal definitions). *Feature Recovery Rate* (FRR) is the fraction of ground-truth features $v_i$ such that at least one learned neuron has cosine similarity above a threshold $\tau_{\text{align}}$: $\text{FRR} = \frac{1}{n}\sum_{i=1}^n \mathbb{1}\big[\exists\, m \in [M] : |\langle w_m, v_i\rangle|/(\|w_m\|_2\|v_i\|_2) \geq \tau_{\text{align}}\big]$. To assess cross-seed consistency, *Maximum Cosine Similarity* (MCS) computes, for each neuron in one run, the highest cosine similarity with any neuron in another run. The *Neuron Z-score* $Z_m^{\max} = (\phi_{m,\max} - \mu_m)/s_m$ quantifies selective activation: high values indicate neurons that fire sharply for specific inputs relative to their baseline. Here, $\phi_{m,\max}$ is the maximum activation of neuron $m$ across inputs, and $\mu_m$ and $s_m$ are the mean and standard deviation of its activations, respectively. In GBA, neurons are partitioned into $K$ groups whose target activation frequencies form a geometric sequence between the *Highest Target Frequency* (HTF) $p_1$ and *Lowest Target Frequency* (LTF) $p_K$. Finally, the *top-$\alpha$ selection rule* restricts analysis to the top $\alpha$ fraction of neurons ranked by a scalar metric (e.g., maximum activation or Z-score), focusing evaluation on the most active or significant neurons.

**Toy example: why frequency-aware training matters.** To isolate the benefit of frequency-aware training, we construct a synthetic dataset with $n = 128$ features in $d = 42$ dimensions where data points are *imbalanced*: half contain $s = 3$ active features (sparse) and half contain $s = 20$ (dense), so all features share the same occurrence frequency $f \approx 0.09$. TopK SAEs assume a fixed sparsity $K$ per input. Since in real data, one never knows the true number of active features, we test TopK with $K \in \{20, 30, 50\}$, which are all reasonable guesses for the average sparsity level. As shown in Table 1, TopK with $K = 20$ achieves near-perfect recovery, but performance degrades sharply as $K$ increases (e.g., FRR at MCS $\geq 0.9$ drops from 98.4% to 23.4% at $K = 50$). In contrast, GBA with full frequency coverage (HTF= 0.5, LTF= 0.001) achieves 100% FRR without any tuning

needed or strong prior knowledge of $f$. This fact demonstrates that frequency-aware training adapts to varying sparsity levels without much manual tuning. See §K.4 for details.

| Method | FRR ($\tau_{\text{align}} \geq 0.8$) | FRR ($\tau_{\text{align}} \geq 0.9$) |
|---|---|---|
| TopK ($K = 20$) | **100.0%** | 98.4% |
| TopK ($K = 30$) | 98.4% | 24.2% |
| TopK ($K = 50$) | 94.5% | 23.4% |
| GBA (full coverage) | **100.0%** | **100.0%** |
| GBA (misspecified) | 38.3% | 3.9% |

Table 1: **Feature Recovery Rate on imbalanced synthetic data** ($n = 128$, $d = 42$, $M = 8192$). TopK's FRR degrades when $K$ does not match the data sparsity, while GBA with proper frequency coverage achieves perfect recovery.

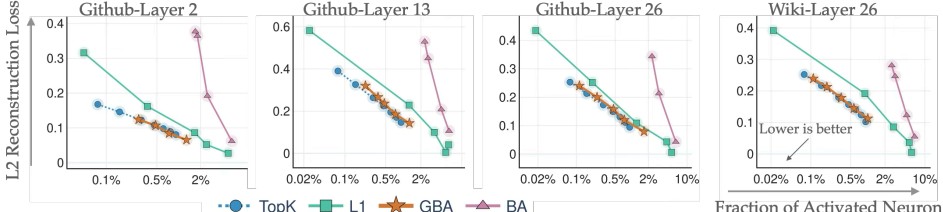

Figure 4: **Curve for reconstruction loss and sparsity** (average fraction of neurons activated per data point). All experiments are conducted using an SAE with 66k neurons. For TopK, we vary $K \in [50, 600]$. For $L_1$, we vary the penalty coefficient $\lambda \in [0.001, 0.1]$. For BA (non-grouped), we vary the target frequency $p \in [0.003, 0.1]$. For GBA, we sample within the range $K \in [3, 20]$, $p_1 \in [0.05, 0.5]$, and $p_K \in [10^{-4}, 5 \times 10^{-3}]$.

**Reconstruction loss and activation sparsity frontier.** We first compare the normalized $\ell_2$ reconstruction loss against the average fraction of activated neurons across different methods. The results are presented in Figure 4, where each benchmark method (TopK, $L_1$, BA) involves varying sparsity-related tuning parameters. Our method performs comparably to the best-performing benchmark, TopK—achieving the lowest reconstruction loss among all methods for a given sparsity level. Specifically, when these methods have the same average fraction of activated neurons, GBA's reconstruction (yellow star) is comparable to TopK's best curve while significantly outperforming both the $L_1$ penalty method and the non-grouped variant BA. The consistent superiority over BA across all experiments provides strong evidence that the grouping mechanism is crucial for achieving both optimal performance.

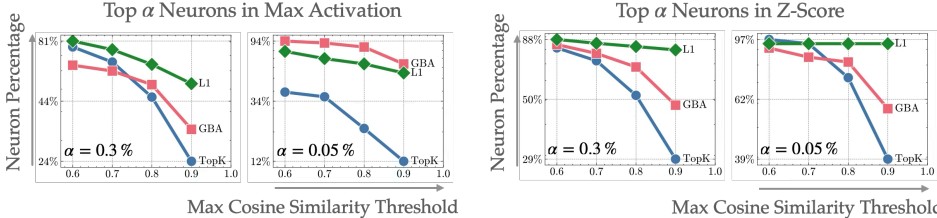

Figure 5: **Fraction of neurons that have max cosine similarity exceeding threshold** for Github-Layer 26, where the max cosine similarity is evaluated for neurons from 6 different runs initialized with different seeds. We take Max Activation and Z-Score as the selecting criteria and plot within a subset of neurons that rank top-$\alpha$ under the criteria in all the neurons with $\alpha$ in $\{0.3\%, 0.05\%\}$ (i.e., top-200 and top-30 neurons out of 66k).

**Consistency of recovered features.** Furthermore, we assess the consistency of the learned features across independent runs with different random seeds. Since ground truth features are unavailable, consistency serves as a proxy for the reliability of the training method. For each neuron in one run, we compute its Maximum Cosine Similarity (MCS) with neurons from another run; a high MCS indicates that a feature is consistently recovered. To avoid the influence of rarely activated neurons, we restrict our analysis to the top-$\alpha$ neurons—selected based on maximum activation or Z-score. The results are presented in Figure 5 and Table 2, and the key findings are shown as follows:

1. As noted in prior work, TopK is seed-sensitive (Paulo & Belrose, 2025). In our experiments, GBA yields a higher percentage of neurons with high MCS. To quantify this effect, Table 2 reports the fraction of consistent neurons (MCS $> 0.9$) under different selection criteria; GBA consistently exceeds TopK. Variability across seeds is small: all runs show tight fluctuations over the $\binom{6}{2} = 15$ pairwise comparisons.
2. The $L_1$ penalty-based SAE is generally more consistent than TopK, and our results confirm this trend: across most selection criteria, $L_1$ achieves higher consistency than both TopK and GBA. However, when focusing on the most active neurons (top-$0.05\%$ by activation), GBA surpasses $L_1$, suggesting stronger recovery of the most salient features.

Additional scatter-plot analyses of neuron-level metrics (Z-score vs. maximum activation, activation fraction, and MCS) for both GBA and TopK are provided in §K.3. Key findings include: (i) neurons with higher Z-scores also exhibit higher MCS, confirming that selectively activated neurons are consistently recovered across seeds; (ii) GBA captures infrequent yet salient features more effectively than TopK, whose high-Z-score neurons concentrate at higher activation fractions.

Combined with the reconstruction-sparsity frontier in Figure 4, in our experimental setting, the proposed GBA method achieves the Pareto frontier in terms of reconstruction fidelity, activation sparsity, and feature consistency.

**Interpretability.** To further evaluate the interpretability of the learned features, we employ a suite of metrics by Karvonen et al. (2025). To ensure a fair comparison, we retrain GBA SAE with 66k neurons on Gemma2-2B (Team et al., 2024) residual stream after layer 12, and compare it against six baselines provided in SAE-Bench: Standard, GatedSAE (Rajamanoharan et al., 2024a), BatchTopK, Matryoshka (Bussmann et al., 2025), TopK, and JumpReLU SAE (Rajamanoharan et al., 2024b), all with $L_0$ between

| Top-$\alpha$ Neurons | GBA (ours) | TopK |
|---|---|---|
| 10% | **0.0366** $\pm$ 0.0010 | 0.0317 $\pm$ 0.0006 |
| 25% | **0.0146** $\pm$ 0.0004 | 0.0127 $\pm$ 0.0002 |
| 50% | **0.0073** $\pm$ 0.0002 | 0.0063 $\pm$ 0.0001 |
| 100% | **0.0037** $\pm$ 0.0001 | 0.0032 $\pm$ 0.0001 |

Table 2: Fraction of neurons with Maximum Cosine Similarity (MCS) $> 0.9$ across different selection percentiles based on the top $\alpha$ selection rule in maximum activation. Results are averaged over $\binom{6}{2} = 15$ pairwise comparisons from 6 random seeds with standard deviations shown. GBA achieves higher consistency than TopK for all $\alpha$.

$300 \sim 400$. The results are summarized in Table 3, demonstrating the competitive interpretability of GBA across all metrics. GBA achieves the best performance in 4 out of 9 metrics, particularly excelling in Explained Variance, Absorption Score, and Alive Fraction. This evidence further supports our theorectical claim to be made in §6 that GBA can reliably recover monosemantic features. More visualizations and detailed explanations of the interpretability metrics are available in §K.5.

| Metric | Standard | GatedSAE | BatchTopK | Matryoshka | TopK | GBA (ours) | JumpReLU |
|---|---|---|---|---|---|---|---|
| $L_0$ | 468.9 | 408.7 | 339.4 | 338.1 | 334.4 | 309.0 | 305.3 |
| Explained Variance ↑ | 0.816 | 0.871 | 0.859 | 0.840 | 0.859 | **0.902** | 0.855 |
| Absorption Score ↓ | 0.1355 | 0.0696 | 0.0347 | 0.0158 | 0.0269 | **0.0041** | 0.0424 |
| SCR Metric ↑ | 0.228 | 0.294 | 0.268 | 0.331 | **0.332** | 0.296 | 0.309 |
| Sparse Probing ↑ | 0.958 | **0.959** | 0.957 | 0.957 | 0.958 | 0.956 | 0.958 |
| TPP Metric ↑ | 0.026 | 0.086 | 0.267 | **0.312** | 0.213 | 0.184 | 0.099 |
| Alive Fraction ↑ | 0.743 | 0.918 | 0.770 | 0.828 | 0.872 | **0.970** | 0.711 |
| RAVEL Disent. ↑ | 0.709 | **0.764** | 0.729 | 0.742 | 0.725 | 0.737 | 0.737 |
| RAVEL Cause ↑ | 0.670 | **0.749** | 0.697 | 0.710 | 0.700 | 0.694 | 0.731 |
| RAVEL Isolation ↑ | 0.748 | 0.778 | 0.761 | 0.773 | 0.749 | **0.779** | 0.742 |

Table 3: **Performance comparison of SAE models with $L_0$ between $300 \sim 400$ on SAEBench.** Arrows indicate whether higher (↑) or lower (↓) values are better. Bold indicates best performance, and underline indicates second best. **GBA (ours)** achieves the best performance in 4 out of 9 metrics, particularly excelling in Explained Variance and Absorption Score.

**Ablation study on GBA hyperparameters.** To assess GBA's sensitivity to hyperparameters, we perform an ablation study varying the number of groups $K$, Highest Target Frequency (HTF) $p_1$, and Lowest Target Frequency (LTF) $p_K$, as shown in Figure 6. The left panel reveals a key pattern: as HTF increases, performance stabilizes—scatter points converge and align with TopK's curve,

especially for $K \geq 10$. Additionally, since low HTF values (e.g., 0.05) hinder recovery of frequent features, it results in higher reconstruction loss. The middle and right panels again confirm that both reconstruction loss and sparsity stabilize when $K \geq 10$, demonstrating insensitivity to the exact number of groups. We observe a slight increase in loss when increasing the number of groups $K$ in the middle panel. This is not detrimental but rather reflects a tradeoff: with few groups (e.g., $K = 3$), many neurons are assigned high target frequencies, resulting in denser activations (right panel) and thus lower reconstruction loss at the expense of interpretability.

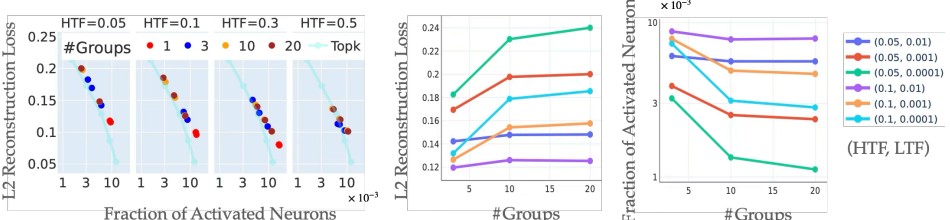

Figure 6: **Ablation study for GBA in terms of** $K$, **HTF, and LTF** for Github-Layer 26. For each run, we partition neurons into $K$ groups with target frequencies as a geometric sequence between HTF and LTF. HTFs: $\{0.05, 0.1, 0.3, 0.5\}$; LTFs: $\{10^{-4}, 10^{-3}, 5 \times 10^{-3}\}$. **Left**: Loss vs sparsity grouped by HTF. Different colors represent different $K$ values; dots of the same color correspond to different LTFs. **Middle & Right**: Loss and sparsity for varying $K$, where each curve represents a pair of HTF and LTF. Results show GBA stabilizes when HTF = 0.5 and $K \geq 10$.

**Simple rule for hyperparameter selection.** These results establish that GBA is nearly tuning-free with a simple selection rule: (1) set HTF = 0.5 as the default upper bound, since randomly initialized neurons with zero bias fire 50% of the time; (2) set LTF = $10^{-3}$ to $10^{-4}$ to cover rare features while preventing dead neurons; (3) use a large number of groups for better frequency coverage and stable performance. This principled setup eliminates the need for dataset-specific tuning—in stark contrast to searching TopK's $K$ across 66k neurons or tuning $L_1$'s penalty coefficient $\lambda$.

## 6 NEURON RESONANCE: A THEORETICAL PERSPECTIVE

The neuron resonance phenomenon observed in §3 raises a fundamental question: *How does frequency matching enable reliable feature recovery, and what determines the resonance band?* We provide a theoretical analysis that justifies this phenomenon with precise recovery conditions.

To rigorously analyze the neuron resonance phenomenon, we study a simplified variant of Algorithm 1 that captures its core mechanism. We consider the *Bias Adaptation* (BA) algorithm, which is essentially GBA with a single neuron group and all neurons share a fixed target activation frequency $p$. The SAE is trained via spherical gradient descent (weight updates normalized to unit sphere). This single-group setting isolates the activation frequency factor from other dynamics, helping us understand how neurons with frequency $p$ selectively learn features with similar occurrence frequency.

For the data model (2.1), we assume $V$ has i.i.d. $\mathcal{N}(0,1)$ entries and simplify $H$ to have exactly $s$-sparse rows: each row $\ell$ has uniform random support $S_\ell$ with $|S_\ell| = s$ for a constant $s$ and entries $H_{\ell,i} = 1/\sqrt{s}$ for $i \in S_\ell$, zero otherwise. This simplified $H$ structure is only for presentation convenience; our analysis captures more general coefficient matrices (see §B.1). The following theorem characterizes the conditions under which BA can recover all features with high probability.

**Theorem 6.1.** *Consider the simplified data model $X = HV$ with data size $N$, feature size $n$ and feature dimension $d$. We train an SAE with $M$ neurons using the BA algorithm with spherical gradient descent, target frequency $p$, and learning rate $\eta \gtrsim (pN)^{-1}$. Under certain regularity conditions on the SAE model (§B.2.1), for any small constants $\varsigma, \varepsilon \in (0,1)$ such that*

$$\text{Network Width:} \quad M \gtrsim n \cdot p^{s/(1-\varepsilon)^2} \tag{6.1}$$

$$\text{Frequency Range:} \quad n^{-1} \lesssim p \lesssim \min\left\{ n^{-(1+s^{-1})/2}, n^{-2(1+\varepsilon)^2/s}, \frac{d^{1-\varsigma}}{n} \right\} \tag{6.2}$$

*with probability at least $1 - n^{-4\varepsilon}$, every feature $i \in [n]$ is recovered by at least one neuron $m_i$ within $T = \varsigma^{-1}$ iterations in the sense that $\langle w_{m_i}^{(T)}, v_i \rangle / \|v_i\|_2 \geq 1 - o(1)$.*

See §B for the full version of the theorem with detailed assumptions and the proof is in §G. To our best knowledge, Theorem 6.1 provides the *first provable guarantee* that an SAE training algorithm can recover monosemantic features within a constant number of iterations. The theorem relies on $V$ being Gaussian for technical convenience; we discuss the role of this assumption and its empirical robustness in §C.1. Our results on both synthetic (§3) and real LLM data (§5) confirm that the BA and GBA algorithms work well when $V$ is non-Gaussian. Moreover, although the theorem only analyzes for a single group with frequency $p$ for clarity of presentation, we can easily extend it to GBA with multiple groups under the same regularity conditions on the SAE model. See §B for detailed discussions.

**Interpreting the theorem.** The theorem reveals two critical factors for successful feature recovery:

1. **Network width:** The required width $M \gtrsim n \cdot p^{-s/(1-\varepsilon)^2}$ shows that $M$ scales linearly with the number of features $n$ but exponentially with sparsity $s$ when $p$ is fixed. The linear scaling with $n$ is intuitive, as each neuron can learn at most one feature. This exponential dependency arises from the challenge of distinguishing features when they co-occur in the same data points. Figure 7(a) experimentally validates this exponential scaling. This result highlights the benefit of overparameterization in SAE training.
2. **Activation frequency range:** The condition on $p$ translates to a "resonance band", where features are most effectively learned when the neuron's activation frequency $p$ falls into the band. The upper bound depends on both the superposition level (controlled by $d/n$) and the feature sparsity $s$. Figure 7(b) visualizes these resonance bands for different sparsity levels. Notably, in our simplified data model, the feature occurrence frequency is $f = s/n = \Theta(n^{-1})$, so for a large constant $s$, we can rewrite the condition as $f \lesssim p \lesssim \min\{\sqrt{f}, fd\}$, as mentioned in §3.

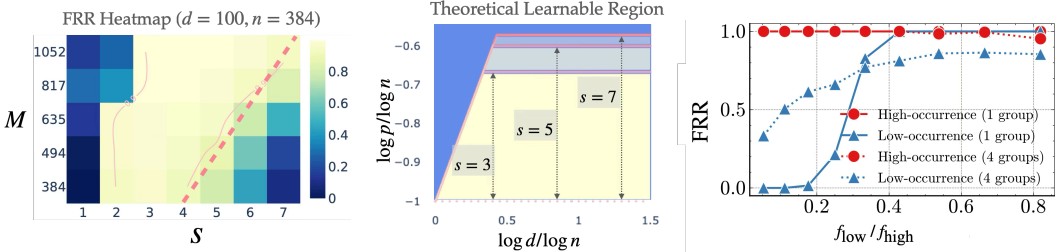

Figure 7: **(a) Network width scaling:** Heatmap of Feature Recovery Rate (FRR) with respect to $(M, s)$ for the GBA algorithm with $M$ axis in log scale, showing exponential dependency on $s$. **(b) Resonance bands:** Theoretical learnable region (yellow) for different sparsity values, demonstrating the transition at $d \approx \sqrt{n}$ between heavy and light superposition regimes. For large $s$, the upper bound approaches $\min\{\sqrt{f}, df\}$ with $f = \Theta(n^{-1})$. **(c) Feature imbalance:** FRR vs. relative occurrence $f_{\text{low}}/f_{\text{high}}$, showing GBA's advantage over BA in handling imbalanced feature frequencies. All experiments use $(n, d) = (384, 100)$. For (c), we use $s = 3$ and $M = 1024$.

This theorem rigorously justifies the neuron resonance phenomenon by proving that neurons with frequency $p$ optimally recover features within a specific frequency band. This insight motivates the GBA algorithm's multi-group design: by creating groups with geometrically decaying target frequencies, we ensure coverage of the entire feasible frequency range, enabling recovery of features with diverse occurrence patterns.

**GBA handles imbalanced features.** As an extension to the discussion above and to build connection to the GBA algorithm, we compare the analyzed BA algorithm with GBA (with 4 groups) on data with imbalanced feature frequencies. To demonstrate the effectiveness of GBA, we construct a dataset with features divided into two groups of equal size: one group with high occurrence frequency $f_{\text{high}}$ and the other with low frequency $f_{\text{low}}$. We vary the imbalance ratio $f_{\text{low}}/f_{\text{high}}$ while keeping the average frequency fixed. The results in Figure 7(c) show that GBA significantly outperforms BA as the imbalance increases, i.e., $f_{\text{low}}/f_{\text{high}} < 0.3$, highlighting GBA's ability to recover features across a wide frequency spectrum, and flexibility to handle real-world data with diverse feature occurrence patterns.

**Reproducibility.** The anonymous source code to this project is available in the supplementary for both data processing and model training. The assumptions and proofs to the main theory can be found in §B and §G, respectively.

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

CONTENTS

## A  RELATED WORKS

**SAE Training Methods.** Many methods have been proposed to train SAEs, addressing the trade-off between reconstruction fidelity and sparsity-induced interpretability from various perspectives. One canonical approach is imposing an $L_1$ penalty on the activations Bricken et al. (2023). Although $L_1$ is a natural surrogate for enforcing $L_0$ sparsity, it typically suffers from activation shrinkage Tibshirani (1996). Several works have attempted to overcome this drawback through alternative techniques Wright & Sharkey (2024); Taggart (2024); Rajamanoharan et al. (2024a); Konda et al. (2014). In particular, Rajamanoharan et al. (2024b) proposed the JumpReLU activation, which achieves state-of-the-art performance despite requiring backpropagation with pseudo-derivatives because of the non-smooth nature of JumpReLU and the need for tuning the kernel density estimation bandwidth. Another representative example is the use of TopK activation Makhzani & Frey (2013), which has proven effective when scaled to large models Gao et al. (2024). However, it has been observed that features learned via TopK activation are quite sensitive to the random seed Paulo & Belrose (2025), raising concerns about their reliability.

**Sparse Dictionary Learning.** Beyond SAE training methods, there is a long history of research on sparse dictionary learning (SDL) dating back to Olshausen & Field (1996); Kreutz-Delgado et al. (2003). Numerous techniques have been developed for applications in signal processing and com-

puter vision (Bruckstein et al., 2009; Rubinstein et al., 2010). For example, Spielman et al. (2012) proposed a polynomial-time algorithm that can accurately recover both the dictionary and its coefficient matrix, under the assumption of sparsity in the coefficients.

**Using SAEs for Model Interpretation.** In recent years, SAEs have gained attention for model interpretation, particularly in the context of large language models (LLMs) (Bricken et al., 2023; Paulo & Belrose, 2025). Notably, Bricken et al. (2023); Dunefsky et al. (2024); Ameisen et al. (2025) have identified several interesting features and circuit patterns learned by SAEs or their variants. Beyond detecting monosemantic features, Papadimitriou et al. (2025) found that groups of SAE-learned features remain remarkably stable across different training runs and encode cross-modal semantics. Additionally, the potential of SAE activations for steering model behavior has been explored (Ameisen et al., 2025; Shu et al., 2025).

# B  FORMAL THEORY OF SAE TRAINING DYNAMICS

In this section, we present a formal theory of SAE training dynamics, providing rigorous guarantees for feature recovery when data follows a well-defined statistical model. However, before delving into the details of the theory, we first need to answer the following fundamental questions:

- What is the precise statistical model for data generation we should consider?
- What does it mean to recover features, and under what conditions is feature recovery even possible?

In this section, we will

- Formalize the statistical model for data generation. State the feature recovery problem and define identifiability of features, which is a pre-requisite for any recovery guarantee
- Present the full set of assumptions and result on SAE training dynamics.

**Notations.** Let $\mathbb{R}_+$ denote the set of non-negative real numbers. For two sets $A$ and $B$, we denote by $A \sqcup B$ the disjoint union of $A$ and $B$. We denote by $\mathbf{1}$ the all-ones vector, whose dimension will be clear from context. In the remaining of the section, we abuse the notation and use $a \gtrsim b$ to denote that $a \geq b + O(\log \log n / \log n)$ for sufficiently large $n$, which differs from what we use in the main text.

## B.1  DATA MODEL

We consider the data model $X = HV$ from (2.1), where data matrix $X \in \mathbb{R}^{N \times d}$ is a sparse, nonnegative combination of monosemantic features $V \in \mathbb{R}^{n \times d}$ with coefficients $H \in \mathbb{R}_+^{N \times n}$. Our statistical framework requires the following *decomposable data* conditions:

**Definition B.1** (Decomposable Data)**.** *We say that the data matrix $X \in \mathbb{R}^{N \times d}$ is decomposable if there exists a positive integer $n \in \mathbb{N}$, a **nonnegative** matrix $H \in \mathbb{R}_+^{N \times n}$ and a feature matrix $V \in \mathbb{R}^{n \times d}$ such that $X = HV$. Moreover, each row of $H$ has unit $\ell_2$ norm and the $\ell_2$ norm of each row of $V$ is $\Theta(\sqrt{d})$. Furthermore, the coefficient matrix $H \in \mathbb{R}^{N \times n}$ satisfies the following three conditions:*

**(H1)** *Row-wise sparsity:* $\max_{\ell \in [N]} \|H_{\ell,:}\|_0 = s$ *with* $s = \Theta(1)$.

**(H2)** *Non-degeneracy: For every $i \in [n]$,* $\|H_{:,i}\|_1 / \|H_{:,i}\|_0 = \Theta(1)$.

**(H3)** *Low co-occurrence:* $\rho_2 := \max_{i \neq j} \langle \mathbb{1}\{H_{:,i} \neq 0\}, \mathbb{1}\{H_{:,j} \neq 0\} \rangle / \|H_{:,i}\|_0 \ll n^{-1/2}$.

*In addition, we further assume that the feature matrix $V \in \mathbb{R}^{n \times d}$ satisfies:*

**(V1)** *Incoherence: For all $i \neq j$,* $|\langle v_i, v_j \rangle| / (\|v_i\|_2 \|v_j\|_2) = o(1)$.

These conditions ensure feature recoverability: *nonnegativity* removes sign ambiguity since opposite directions yield contradictory concepts; *row-wise sparsity* (H1) limits each data point to $s$ features, essential for sparse recovery; *non-degeneracy* (H2) ensures sufficient feature magnitude when present; *low co-occurrence* (H3) and *incoherence* (V1) guarantee features are distinguishable

by occurrence pattern or direction—generalizing the orthogonality assumption common in sparse recovery (Marques et al., 2018; Candès & Plan, 2009). All these conditions will be used in our theoretical analysis of SAE training dynamics.

**Feature recovery problem.** Note that the bilinear representation $X = HV$ has two intrinsic ambiguities: (i) *feature permutation*—reordering features leaves $HV$ unchanged; (ii) *feature scaling*—scaling features while inversely scaling coefficients preserves the product. With the data model, we can now define the feature recovery problem: given data $X$ generated from an unknown decomposable pair $(H, V)$, the goal is to learn an SAE such that for each feature $v_i$ in $V$, there exists a neuron $m_i$ in the SAE with weight vector $w_{m_i}$ satisfying

$$\langle w_{m_i}, v_i \rangle / \|v_i\|_2 \geq 1 - o(1).$$

This means each feature is closely approximated by at least one neuron, up to a small error.

### B.2 SAE Dynamics with Bias Adaptation

In the following, we first introduce a *Bias Adaptation (BA)* algorithm, which is a simplified version of the GBA algorithm with only one group of neurons and a fixed target activation frequency (TAF) $p$. Then, we provide theoretical results on the training dynamics of BA, which is accompanied by synthetic experiments to validate the theoretical findings.

#### B.2.1 Simplification for Theoretical Analysis

We make several simplifications to the setup of SAE to facilitate theoretical analysis.

**Decomposible data with Gaussian features.** We assume that the data matrix $X \in \mathbb{R}^{N \times n}$ is decomposable in the sense of Definition B.1. Moreover, we assume that the feature matrix $V \in \mathbb{R}^{n \times d}$ has i.i.d. entries following $\mathcal{N}(0, 1)$. Such a choice of $V$ satisfies the incoherence condition (V1).

**SAE model.** We consider a simplified version of the SAE model $f(x; \Theta)$ in (2.2), where the only trainable parameters are the weights $\{w_m\}_{m=1}^M$.

- (*Small output scale*) We assume that the output scale $a_m = a$ and $a$ is sufficiently small. When computing the gradient, we rescale the $\nabla \mathcal{L}(\Theta)$ back to its original scale by multiplying $a^{-1}$.

- (*Fixed pre-bias*) We fix the pre-bias $b_{\mathrm{pre}} = 0$, as the data matrix $X$ is centered.

- (*ReLU-like smooth activation*) We use a smooth, ReLU-like activation function $\phi$ (see Definition B.3 for details). One example is the softplus activation $\phi(x) = \log(1 + \exp(x))$.

- (*Fixed bias*) For each neuron $m \in [M]$, we fix the bias $b_m = b < 0$ throughout training, where $b$ is a negative scalar whose value will be specified later. This fixed bias will determine the target activation frequency (TAF) $p$ of all neurons via $p = \Phi(-b)$, where $\Phi(\cdot)$ is the tail probability function for Gaussian distribution. We will detail the intuition behind this choice later.

These simplifications help isolate the core aspects of feature recovery and make the analysis more tractable.

**Bias Adaptation (BA) algorithm.** Recall that Bias Adaptation (BA) algorithm is a special case of GBA algorithm with only one group of neurons and a fixed TAF $p$. As our goal is to systematically understand how neurons with a specific TAF $p$ can recover features with similar occurrence frequency, it is reasonable to try using a version of GBA algorithm with a single group and a fixed TAF $p$. We introduce the algorithm as follows. Here we determine the value of $p$ implicitly by choosing a fixed bias $b < 0$, and they are related by $p = \Phi(-b)$, where $\Phi(\cdot)$ is the tail probability function for Gaussian distribution. That is, $\Phi(t) = \mathbb{P}(Z \geq t)$ for $Z \sim \mathcal{N}(0, 1)$.

Given the data matrix $X$ and the SAE model $f(x; \Theta)$, we can compute the loss function as $\mathcal{L}_{\mathrm{rec}}(\Theta) = \mathrm{Avg}_{x \in X}(\frac{1}{2}\|f(x; \Theta) - x\|_2^2)$, and its gradient with respect to the weights $\{w_m\}_{m=1}^M$. Since only the directions of the features $\{v_i\}_{i=1}^n$ matter, we adopt spherical gradient descent to update the weights.

That is, starting from the initial weights $\{w_m^{(0)}\}_{m\in[M]}$ uniformly sampled from the unit sphere $\mathbb{S}^{d-1}$, for any $t \geq 1$, in the $t$-th iteration, we update each $w_m^{(t-1)}$ by

$$\textbf{BA:} \quad w_m^{(t)} = \frac{w_m^{(t-1)} + \eta\, g_m^{(t)}}{\|w_m^{(t-1)} + \eta\, g_m^{(t)}\|_2}, \quad \text{where} \quad g_m^{(t)} = \lim_{a\to 0} -a^{-1}\nabla_{w_m}\mathcal{L}_{\text{rec}}(\Theta^{(t-1)}). \quad \text{(B.1)}$$

Here, $g_m^{(t)}$ is the rescaled negative gradient of the loss function $\mathcal{L}_{\text{rec}}(\cdot)$ with respect to the weight $w_m$ of neuron $m$ at iteration $t$. We will show that, under proper conditions, for any feature $v_i$, there exists at least one neuron $m_i \in [M]$ such that the alignment between $w_{m_i}^{(T)}$ and $v_i$ is arbitrarily close to one when $T$ is sufficiently large.

Before we proceed to the main theoretical results, we make several remarks on the above simplifications for theoretical analysis and their implications.

**Fixed bias is without loss of generality.** As we consider Gaussian features and always normalize $w_m^{(t)}$ to the unit sphere, it can be shown using the Gaussian conditioning technique that the pre-activations remain approximately Gaussian, i.e., $y_m(x_l) = \langle w_m^{(t)}, x_\ell \rangle + b \sim \mathcal{N}(b, 1)$ for a constant number of iterations $t$. See §B.4 for details. Therefore, to achieve the desired TAF $p$, it is without loss of generality to fix the bias $b < 0$ such that $\Phi(-b) = p$, which means that the pre-activations of each neuron will be non-negative for approximately $p$ fraction of the $N$ data points throughout the training.

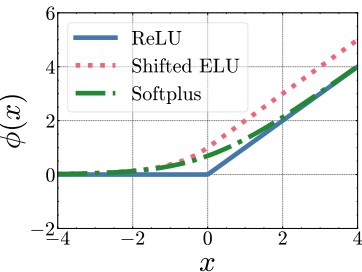

Figure 8: Smooth ReLU-like activations

**Smooth ReLU-like activation approximates ReLU.** We choose a smooth activation function for technical convenience. For definition, we defer to Definition B.3. These activations can be viewed as a smooth approximation to the ReLU function, as illustrated in Figure 8. This class of activations encompasses functions like Softplus and shifted ELU, and closely resembles the standard ReLU activation function. We believe that a more refined analysis can also be applied to the standard ReLU activation, but we leave this as future work.

**Small output scale decouples neuron dynamics.** Following a common paradigm in the literature (see e.g. Lee et al. (2024); Chen et al. (2025)), we assume that the output scale of the SAE is sufficiently small. The benefit of this condition is that it decouples the dynamics among the $M$ neurons, making the analysis more tractable. Specifically, the rescaled negative gradient of the loss $\mathcal{L}(\Theta)$ is given by

$$g_m = -a^{-1}\nabla_{w_m}\mathcal{L}(\Theta) = \sum_{\ell=1}^{N} \big(\ \varphi(w_m^\top x_\ell; b)x_\ell\ -\ \psi_m(x_\ell; \Theta)\ \big) \overset{a\to 0}{=} \sum_{\ell=1}^{N}\ \varphi(w_m^\top x_\ell; b)x_\ell\ , \quad \text{(B.2)}$$

where we define $\varphi(\cdot, \cdot)$ and $\psi_m(\cdot; \Theta)$ as

$$\varphi(u, v) = \phi(u + v) + \phi'(u + v)\cdot u\ ,$$

$$\psi_m(x; \Theta) = \phi'(w_m^\top x + b)\cdot w_m^\top f(x; \Theta)\cdot x + \phi(w_m^\top x + b)\cdot f(x; \Theta)\ .$$

Here, $\varphi : \mathbb{R} \mapsto \mathbb{R}$ is a *decoupled* term that depends only on each individual neuron's weight and bias, while $\psi_m : \mathbb{R}^d \mapsto \mathbb{R}^d$ is a *coupling* term that captures the interaction between the neuron and the rest of the network. Since the scale of $f(x; \Theta)$ is proportional to $a$, this coupling term is negligible when $a$ is small. As a result, when $a$ is infinitesimally small, each neuron $m$ evolves independently of the other neurons. Furthermore, thanks to the decoupled dynamics, the restriction to a single group with a fixed TAF $p$ does not result in any loss of generality, as the analysis of multiple groups is a straightforward extension.

### B.2.2 MAIN THEOREM ON TRAINING DYNAMICS

Intuitively, to recover a feature $v$, it has to appear in sufficiently many data points with sufficiently large coefficients. To characterize this intuition, we introduce two key quantities based on the coefficient matrix $H$. First, for each feature index $i \in [n]$, let $\mathcal{D}_i = \{l \in [N] : H_{l,i} \neq 0\}$ be the set of data indices that contain feature $v_i$. The occurrence of the feature $v_i$ is thus given by $|\mathcal{D}_i|/N$. We define the *maximum feature occurrence* as the largest occurrence among all features, i.e.,

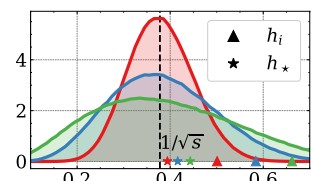

Figure 9: Relationship between $s$, $h_\star$ and $h_i$ with different concentration level in $H$'s non-zero entries' empirical distribution (shadow). A less concentrated $H$ leads to larger $h_\star$ and $h_i$.

$$\rho_1 = \max_{i \in [n]}\{|\mathcal{D}_i|/N\} = \max_{i \in [n]}\left\{1/N \cdot \sum_{l \in [N]} \mathbb{1}\{H_{l,i} \neq 0\}\right\}.$$

To ensure each feature $v_i$ appears in sufficiently many data points, we require that the occurrence of each feature is comparable to $\rho_1$, i.e., $|\mathcal{D}_i|/(\rho_1 N)$ is not too small for each $i \in [n]$.

Second, to measure the magnitude of coefficients associated with each feature, we define the *cut-off* level for the feature $i$ as

$$h_i := \max\left\{h \leq 1 : \frac{1}{|\mathcal{D}_i|}\sum_{l \in \mathcal{D}_i} \mathbb{1}\{H_{l,i} \geq h\} \geq \text{polylog}(n)^{-1}\right\}. \tag{B.3}$$

Intuitively, $h_i$ is a critical threshold such that, among all data points containing $v_i$, at least a $\text{polylog}(n)^{-1}$ fraction of them have coefficients no smaller than $h_i$. In other words, $h_i$ reflects the magnitude of coefficients associated with feature $v_i$, within the subset of data points where $v_i$ is present. Thus, $h_i$ can effectively be viewed as a notion of "signal strength" for feature $v_i$, and we should require that $h_i$ is not too small for each $i \in [N]$.

Furthermore, we additionally introduce a global quantity called the *concentration coefficient* $h_\star = h_\star(H)$, whose definition is technical and deferred to (G.1) in the appendix. Intuitively, $h_\star$ characterizes the global concentration level of nonzero entries in $H$. For now we can intuitively understand it as the variance of the nonzero entries in $H$, and thus $h_\star$ will increase when the nonzero entries in $H$ are less concentrated.

With these definitions, we are now ready to state the main theorem on the training dynamics.

**Theorem B.2.** *Let $X = HV$ be decomposable in the sense of Definition B.1 with $H \in \mathbb{R}^{N \times n}$ satisfying all the conditions therein, and further assume that $V \in \mathbb{R}^{n \times d}$ has i.i.d. entries following $\mathcal{N}(0,1)$. For this $X$, we train the SAE with BA algorithm given in (B.1). Let $\varsigma, \varepsilon \in (0,1)$ be any small constants. We assume that the number of neurons $M$ is sufficiently large:*

$$\boxed{\textit{Network Width:} \quad \frac{\log M}{\log n} \gtrsim \max_{i \in [n]}\left\{\frac{b^2}{2(1-\varepsilon)^2 h_i^2 \log n} + 1\right\}. \tag{B.4}}$$

*Moreover, we assume that the learning rate $\eta$ satisfies $\log \eta \gtrsim (b^2/2 - \log N)$ and that the bias $b < 0$ is set to satisfy the following condition:*

$$\boxed{\textit{Bias Range:} \quad 1 \gtrsim \frac{b^2}{2\log n} \gtrsim \max\left\{\frac{1}{2} + \frac{h_\star^2}{2}, \ 2(1+\varepsilon)^2 h_\star^2, \ 1 - (1-\varsigma) \cdot \frac{\log d}{\log n}\right\}. \tag{B.5}}$$

*Furthermore, we assume the coefficient matrix $H$ satisfies the following feature balance condition:*

$$\boxed{\textit{Feature Balance:} \quad \frac{|\mathcal{D}_i|}{\rho_1 N} \geq \text{polylog}(n)^{-1}, \quad h_i^2 \gg \frac{\log\log(n)}{\log(n)}, \quad \forall i \in [n]. \tag{B.6}}$$

*Then, it holds with probability at least $1 - n^{-4\varepsilon}$ over the randomness of $V$ that for any feature $i \in [n]$, there exists at least one unique neuron $m_i$ such that after at most $T = \varsigma^{-1}$ iterations, the alignment between the weights of neuron $m_i$ and the feature vector $v_i$ satisfies $\langle w_{m_i}^{(T)}, v_i\rangle / \|v_i\|_2 \geq 1 - o(1)$.*

See §G for a detailed proof of this theorem. Theorem B.2 shows that under appropriate conditions, BA provably recovers all monosemantic features within a constant number of iterations. These conditions include that (i) the network is sufficiently wide compared to the number of features as specified in (B.4), (ii) the bias $b$ is chosen within a certain range as specified in (B.5), and (iii) the co-efficient matrix $H$ satisfies the feature balance condition in (B.6), ensuring that each feature appears frequently enough with sufficiently large coefficients. To our best knowledge, this theorem is the first theoretical result that proves a SAE training algorithm can provably recover all monosemantic features.

**Going from one group to multiple groups.** The analysis of BA with a single group can be naturally extended to the case of multiple groups in GBA thanks to the decoupled dynamics among neurons. Specifically, as we have shown in (B.2), each neuron $m$ evolves *independently* when all neurons' output scale $a_m$ is sufficiently small. Therefore, if we have $K$ groups of neurons, each with $M/K$ neurons and bias $b_k$ for group $k \in [K]$ such that the TAF $p_k = \Phi(-b_k)$, then the same analysis can be applied to each group separately, and we will derive the same conditions as in Theorem B.2 for each group. As a result, to learn features of a certain occurrence frequency $f$, we just need to ensure that at least one group $k$ has TAF $p_k$ and the corresponding bias $b_k$ satisfying the Bias Range condition in (B.5).

**Specializing Theorem B.2 to Theorem 6.1.** To relate the above theorem back to the one presented in the main text:

- We first note that the decomposable data condition in Definition B.1 is always satisfied when $H$ is designed to have exactly $s$-sparse rows: each row $\ell$ has uniform random support $S_\ell$ with $|S_\ell| = s$ for a constant $s$ and entries $H_{\ell,i} = 1/\sqrt{s}$ for $i \in S_\ell$, zero otherwise.

- Moreover, the feature balance condition in (B.6) is also satisfied with high probability in this case because each feature appears in $s/n$ fraction of data points, and $|\mathcal{D}_i| = sN/n$ for each $i \in [n]$ by the law of large numbers, and so is $\rho_1 = s/n$.

- Finally, in this case, the concentration coefficient $h_i$ defined in (B.3) is equal to $1/\sqrt{s}$, as the nonzero entries are all equal to $1/\sqrt{s}$.

Hence, the remaining conditions to be checked are the network width condition in (B.4) and the bias range condition in (B.5). We now invoke Theorem G.1, which states that if $H$ has every entries belonging to $\{0, 1/\sqrt{s}\}$, then $h_\star = 1/\sqrt{s}$ as well. Therefore, by substituting $h_i = h_\star = 1/\sqrt{s}$ into (B.4) and (B.5), multiplying both sides of the inequalities by $\log n$, taking exponential, and using the fact that

$$\exp\left(\frac{b^2}{2}\right) = \frac{\Theta(\text{polylog}(n))}{p}$$

by the Gaussian tail estimate for $b > \sqrt{\log n} \gg 1$ (guaranteed by the bias range condition), we recover the conditions in Theorem 6.1.

### B.3 DETAILS ON RELU-LIKE ACTIVATION

In this section, we provide the omitted details for §6. We give a formal definition of ReLU-like activations.

**Definition B.3** (ReLU-like Activation). *For the activation function $\phi : \mathbb{R} \to \mathbb{R}$, we define $\varphi$ as*

$$\varphi(x) = \varphi(x; 0) = \phi(x) + x\,\phi'(x).$$

*We say that $\phi$ is ReLU-like if it satisfies the following:*

1. *(Lipschitzness) The activation function $\phi$ is continuously differentiable, 1-Lipschitz, and $\gamma_1$-smooth with $\gamma_1 = O(\text{polylog}(n))$. Furthermore, $\varphi(x)$ is $\gamma_2$-Lipschitz with $\gamma_2 = O(\text{polylog}(n))$.*
2. *(Monotonicity) The activation function $\phi$ is non-decreasing, and moreover, $\phi'(x) > C_0$ for some constant $C_0 > 0$ and all $x \geq 0$.*
3. *(Diminishing Tail) There exists a threshold $\kappa_0 = O((\log n)^{-1/2})$ and a sufficiently large constant $c_0 > 0$ such that for all $x < -\kappa_0$, $\max\{|\phi(x)|, |\phi'(x)|, |x\,\phi'(x)|\} \leq n^{-c_0}$.*

***Lipschitzness.*** Under the above assumptions, we note that $\varphi(x; b)$ is $L$-Lipschitz in $x$ with $L = (\gamma_2 + |b|\gamma_1) = O(\text{polylog}(n)) > 1$. The Lipschitz property of the function $\varphi$ is pivotal in our analysis since it enables control over error propagation across iterations. However, this property depends on the smoothness of the activation function $\phi$, a condition that the standard ReLU does not satisfy. Fortunately, many common activation functions—such as softplus, noisy ReLU, and shifted ELU (with the limit at $-\infty$ set to 0)—do satisfy this smoothness requirement. In particular, with a large smoothness parameter $\gamma_1 = \text{polylog}(n)$, we can use a smooth activation function to well approximate the ReLU function. For instance, we can take $\phi(x) = \gamma_1^{-1} \log(1 + e^{\gamma_1 x})$ for some $\gamma_1 = \text{polylog}(n)$ as a smooth approximation of the ReLU activation function.

***Monotonicity.*** The monotonicity property ensures that neurons with large pre-activations, which indicate a good alignment with the underlying features, will also have large post-activations. This then guarantees a continuous growth of the corresponding neuron weights.

***Diminishing Tail.*** The diminishing tail condition ensures that both the activation function $\phi$ and its derivative $\phi'$ are negligibly small when the input is below the threshold $-\kappa_0$. This property suppresses unwanted neuron activations, thereby promoting sparsity in the activations—a key factor in the successful training of the SAE.

### B.4  PROOF OVERVIEW

In the following, we provide an overview of the key steps in the proof of Theorem B.2.

#### B.4.1  GOOD INITIALIZATION WITH WIDE NETWORK

By planting a large pool of i.i.d. random neurons at initialization, we can—*with overwhelming probability*—(1) assign to each feature $v_i$ one neuron $m_i$ whose inner product with $v_i$ is already very large, and (2) simultaneously ensure that this same neuron has only weak correlations with *all* the other features. Concretely, we prove that if $M$ grows fast enough relative to $n$, then there exists a choice of distinct neurons $\{m_i\}_{i=1}^n$ such that

$$\textbf{InitCond-1:} \quad \langle v_i, w_{m_i}^{(0)} \rangle \geq (1 - \varepsilon) \sqrt{2 \log \frac{M}{n}},$$

$$\textbf{InitCond-2:} \quad \max_{j \neq i} \left| \langle v_j, w_{m_i}^{(0)} \rangle \right| \leq \sqrt{2}(1 + \varepsilon) \sqrt{2 \log n}. \tag{B.7}$$

These two properties together ensure a *good initialization* for the neuron $m_i$ dedicated to feature $v_i$. With $M \gg n^3$, we deduce that neuron $w_{m_i}$ aligns *exclusively* with feature $v_i$. In fact, as $M$ increases the separation between the two thresholds also increases, so $w_{m_i}^{(0)}$ is ever more strongly aligned with its own feature $v_i$ than with any other $v_j$ at the start. This widening margin precisely captures the *benign over-parameterization* effect: having many neurons actually promotes clean, feature-specific initialization. See Theorem E.1 for more details.

#### B.4.2  PREACTIVATIONS ARE APPROXIMATELY GAUSSIAN

We give a brief overview of how we deal with the challenge of tracking the highly nonlinear dynamics in (B.1). With an abuse of notation, let us denote by $w_t$ and $b_t$ one neuron's weight and bias after iteration $t$. For the first step, the preactivations are Gaussian, i.e., $w_0^\top x_\ell + b_0 \sim \mathcal{N}(b_0, 1)$. For later steps, we expand the gradient descent update for the neuron weights $w_t$ at iteration $t$. Let us denote by $\varphi_t = (\varphi(w_{t-1}^\top x_\ell; b_{t-1}))_{\ell \in [N]}$ and $g_t = X^\top \varphi_t$ the gradient computed in (B.2) at iteration $t$. By the gradient formula in (B.2), we have

$$w_t = \sum_{\tau=1}^t \lambda_\tau \cdot X^\top \varphi_\tau + \lambda_0 \cdot w_0, \quad \text{and} \quad Xw_t = \sum_{\tau=1}^t \lambda_\tau \cdot Xg_t + \lambda_0 \cdot Xw_0, \tag{B.8}$$

for some coefficient $\lambda_\tau$. Let us recall the decomposition $X = HV$. The first equality in (B.8) indicates that $w_{t-1}$ only contains information of $V$ through the $(t-1)$-dimensional projection $\Phi = \text{span}\{\varphi_\tau^\top H\}_{\tau=1}^{t-1}$. For the second equality, the most recent component $Xg_t = HVg_t$ in the preactivations contains information from a new gradient direction—the direction of projecting $g_t$ onto

the orthogonal space of $G = \{w_0, g_1, \ldots, g_{t-1}\}$, which we denote as $g_t^\perp$. Data $X$'s projection onto this new direction can be decomposed as

$$Xg_t = H \cdot (\Phi^\perp V g_t^\perp + \Phi V g_t^\perp),$$

where $\Phi^\perp V g_t^\perp$ is independent of all the previous updates, as the projection is orthogonal to both $\Phi$ and $G$. Therefore, $V g_t^\perp$ is a high-dimensional independent Gaussian vector plus a low-dimensional coupling term $\Phi V g_t^\perp$. The argument holds true for all iteration steps, and if $t \ll d \wedge n$, we approximately have $x_\ell w_t \sim \mathcal{N}(b_t, 1)$ thanks to the normalization of the weight $w_t$. This argument can be made rigorous by use of the *Gaussian conditioning technique* (Wu & Zhou, 2023; Bayati & Montanari, 2011; Montanari & Wu, 2023) in the formal proof. See §E.3 for details.

### B.4.3 WEIGHT DECOMPOSITION AND CONCENTRATION UNDER SPARSITY

For one neuron dedicated to the target feature $v_i$ and satisfying the initialization conditions in (B.7), we decompose the weight $w_t$ into two directions: 1) the projection of $w_t$ onto the 2-dimensional subspace spanned by $w_0$ and $v_i$; 2) the projection of $w_t$ onto the orthogonal space $w_t^\perp$. We define

$$\alpha_t = \frac{\langle w_t, v_i \rangle}{\|v_i\|_2}, \quad \beta_t = \|w_t^\perp\|_2.$$

Using $\alpha_t$ and $\beta_t$, one can compute the first and second moments of the post-activation $\varphi_t$ under the decomposition of the pre-activations (into a high-dimensional Gaussian component and a low-dimensional coupling term) obtained by the Gaussian conditioning technique. The post-activation $\varphi_t$ then gives rise to the next-step $w_{t+1}$, and we thus obtain an induced recursion over $\alpha_t$ and $\beta_t$. As a more concrete example, let us take learning rate $\eta = \infty$, and we can express $\alpha_t$ as

$$\alpha_{-1,t} = \frac{\langle w_t, v_i \rangle}{\|v_i\|_2} = \frac{v_i^\top X^\top \varphi_t}{\|v_i\|_2 \cdot \|X^\top \varphi_t\|_2}$$

Recall that $X = HV$. By a splitting of $V = [V_{-1}; v_i^\top]$ in the row and a splitting of $H = [H_{-i}, H_i]$ in the column, we have

$$\alpha_{-1,t} = \frac{\|v_i\|_2^2 \cdot H_i^\top \varphi_t + v_i^\top V_{-i}^\top \cdot H_{-i}^\top \varphi_t}{\|v_i\|_2 \cdot \|X^\top \varphi_t\|_2} = \underbrace{\frac{\|v_i\|_2^2 \cdot H_i^\top \varphi_t}{\|X^\top \varphi_t\|_2}}_{\text{Signal}} + \underbrace{\frac{v_i^\top V_{-i}^\top \cdot H_{-i}^\top \varphi_t}{\|v_i\|_2 \cdot \|X^\top \varphi_t\|_2}}_{\text{Noise}}.$$

Here, we explicitly separate the signal from the noise. Our goal is to steer the neuron toward the direction of $v_i$, while treating gradient contributions from other features as noise.

**Controlling Moment of the Activations.** To proceed, we must tightly control both the signal and noise terms in the numerator and the denominator. Concretely, this means bounding the first moment of the activation $\varphi_t$ (which enters the numerator) and its second moment (which controls the denominator), all while respecting the sparsity structure of $\varphi_t$. A core difficulty stems from the pre-activation

$$Xw_t = HVw_t,$$

whose entries are not independent—even under a Gaussian approximation—because the feature rows $H_{\ell,:}$ are correlated. This correlation invalidates the assumptions of classical concentration inequalities, such as Bernstein's, and the problem only worsens once we apply the nonlinear activation. Moreover, classical concentration techniques based on the bounded-differences property, such as McDiarmid's inequality (McDiarmid et al., 1989), are not applicable here. This is because the bounded-differences property only

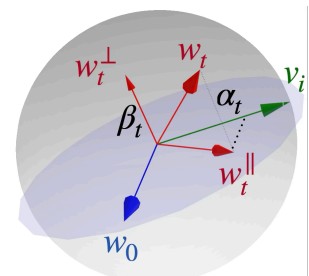

Figure 10: Visualization of the projection of $w_t$ onto 1) the subspace spanned by $w_0$ and $v_i$ and 2) the orthogonal space $w_t^\perp$.

offers a uniform bound on the impact of each individual input change on the output, and it fails to capture that the activations are sparse—remaining zero most of the time.

To overcome these dependencies, we invoke the Efron-Stein inequality (Boucheron et al., 2003). Unlike McDiarmid's bounded-differences inequality, which requires each individual input change

to have a uniformly small impact on $f$, Efron-Stein only demands a bound on the total conditional variance, namely

$$\mathbb{E}\Big[\sum_{i=1}^{n}\big(f(x) - f(x^{(i)})\big)^2 \,\Big|\, x\Big] \;\leq\; V,$$

where $x^{(i)}$ denotes the vector obtained by replacing the $i$th coordinate of $x$ with an independent copy. This variance-based condition is far more flexible in the presence of both correlation and nonlinearity, allowing us to derive the sharp moment bounds we need.

### B.4.4 State Recursion and Convergence

We track at iteration $t$ the *alignment* $\alpha_{-1,t}$ and the *orthogonal component* $\beta_t$ of the neuron weight $w_t$. By exploiting the "Gaussian-like" concentration of the pre-activation $Xw_t = HVw_t$ and applying the refined Efron-Stein inequality to handle both feature correlations and the nonlinearity, one obtains the coupled recurrences

$$\frac{1}{\alpha_{-1,t}} \leq (1 + o(1)) \;+\; \lambda_t\Big(\frac{\Phi(-b)}{\rho_1 d}\,\frac{1}{\alpha_{-1,t-1}} + \widetilde{\xi}_t\Big), \qquad \frac{\beta_t}{\alpha_{-1,t}} \leq \lambda_t\Big(\frac{\beta_{t-1}}{\alpha_{-1,t-1}} + \widetilde{\xi}_t\Big).$$

Here, $\lambda_t \propto \rho_1 N/|\mathcal{D}_i|$, and $\Phi(-b)$ denotes the Gaussian tail probability beyond the threshold $-b$, which captures the activation sparsity. For clarity, we focus on the noiseless regime (i.e., assume $\widetilde{\xi}_t = 0$) so that all noise contributions are neglected. We now elaborate on these recursions in detail:

1. Recall that we require $\beta_t \ll \alpha_{-1,t}$ since the neuron should eventually converge exclusively in the direction of the target feature. In our framework, the minimal growth rate of the ratio $\beta_t/\alpha_{-1,t}$ is intrinsically controlled by $\lambda_t = \widetilde{O}(\rho_1 N/|\mathcal{D}_i|)$. By the definition of $\rho_1$, this ratio is inherently larger than 1. Thus, to prevent an unbounded escalation of $\beta_t/\alpha_{-1,t}$, we must restrict $\lambda_t$ to, at most, a polylogarithmic scale, i.e., $\lambda_t = \widetilde{O}(1)$.

2. If we additionally set $\Phi(-b)/(\rho_1 d) < d^{-\varsigma}$ for some $\varsigma \in (0,1)$, then the map $\alpha_{-1,t}^{-1} \mapsto \alpha_{-1,t+1}^{-1}$ is contractive. Hence $\alpha_{-1,t}$ grows from its initialization $\widetilde{\Theta}(d^{-1/2})$ to $1-o(1)$ in $O(1)$ steps, and the growth rate is much faster than that of $\beta_t/\alpha_{-1,t}$ thanks to the sparsity condition $\Phi(-b)/(\rho_1 d) \ll 1$.

From the above discussions, we already justify the inclusion of the **Individual Feature Occurrence** condition $\frac{|\mathcal{D}_i|}{\rho_1 N} \geq \text{polylog}(n)^{-1}$ in (B.6) and part of the **Bias Range** condition $\frac{b^2}{2\log n} \gtrsim 1 - (1 - \varsigma) \cdot \frac{\log d}{\log n} \Leftrightarrow \Phi(-b) \ll d^{1-\varsigma}/n = \widetilde{O}(d^{-\varsigma} \cdot (\rho_1 d))$ in (B.5). The remaining conditions can be derived based on a more careful analysis, including the noise term $\widetilde{\xi}_t$ and the initialization conditions (B.7).

## C Supplementary Discussions

### C.1 Discussions for Gaussian Feature Assumption

In our current theory, we assume that the features are Gaussian distributed. This is primarily used to obtain clean, closed-form concentration bounds on:

1. (§B.4.1) Inner products between different feature directions, so we can control interference between features when multiple are active, and

2. (§B.4.2) Pre-activations $y = w^\top x$ and their sparsity under the data model $X = HV$.

In specific, we employ the Gaussian conditioning technique, which allows us to decompose a high dimensional Gaussian random vector into components that are explicitly dependent on the conditioning event and components that are independent Gaussian noise. Theoretically extending these probabilistic techniques to non-Gaussian settings is a non-trivial task and out of the scope of this paper.

One generalization of the Gaussian conditioning technique involves using features uniformly sampled from the unit sphere in our synthetic experiment (§3). This setup better reflects real-world

scenarios where features in LLMs are often normalized through layer-normalization. Despite the different feature distribution, we still observe the resonance phenomenon predicted by our theory. This is evident when inspecting the Feature Recovery Rate (FRR) in Figure 2, plotted for varying activation frequencies $p$ and dimensions $d$. This observation suggests that our theory is robust to the specific distribution of features.

In addition, the empirical results on real LLM activations further confirm that the proposed GBA method works well when $V$'s rows are just the features learned by the LLM, which are non-Gaussian.

### C.2 DETAILS ON TOPK AND $L_1$ TRAINING METHODS

We provide here more details on the training methods used in our experiments, including the Sparse Autoencoder (SAE) with TopK activation and SAE with $L_1$ regularization.

**Sparse Autoencoder (SAE) with TopK activation.** In an SAE with TopK activation, sparsity is enforced by selecting only the $K$ neurons with the highest activation values in the hidden layer. Let $y = W(x - b_{\text{pre}}) + b$ be the pre-activation values of the hidden layer. Let $\phi(y)$ be the activations after applying a standard activation function. The TopK selection mechanism, denoted as $S_K(\cdot)$, operates on $\phi(y)$. For a vector $v \in \mathbb{R}^M$, $S_K(v)$ produces a vector $v' \in \mathbb{R}^M$ such that:

$$v'_j = \begin{cases} v_j & \text{if } v_j \text{ is among the } K \text{ largest values in } v, \\ 0 & \text{otherwise} \end{cases}$$

for $j \in [M]$. The post-activation in a TopK SAE is:

$$z = S_K\big(\phi(W(x - b_{\text{pre}}) + b)\big),$$

which by definition is $K$-sparse. The reconstructed output is:

$$\widehat{x} = \text{diag}(a) \cdot W^\top z + b_{\text{pre}}.$$

Let $\Theta = (W, b_{\text{pre}}, b, a)$ be the parameters of the SAE. The loss function for the TopK SAE is the reconstruction loss:

$$\mathcal{L}_{\text{rec}}(x; \Theta) = ||x - \widehat{x}||_2^2.$$

**Sparse Autoencoder (SAE) with $L_1$ regularization.** In an SAE with $L_1$ regularization, sparsity is encouraged by adding a penalty term to the reconstruction loss, proportional to the sum of the absolute values of the hidden layer activations. Let $y = W(x - b_{\text{pre}}) + b$ be the pre-activation values of the hidden layer. Let $z = \phi(y) = \phi(W(x - b_{\text{pre}}) + b)$ be the activations after applying a standard activation function; these are the hidden layer representations that will be encouraged towards sparsity. The reconstructed output is:

$$\widehat{x} = \text{diag}(a) \cdot W^\top z + b_{\text{pre}}.$$

The loss function for the L1 SAE, $\mathcal{L}(x; \Theta)$, incorporates both the reconstruction error and the L1 penalty on the hidden activations $z$:

$$\mathcal{L}(x; \Theta) = \|x - \widehat{x}\|_2^2 + \lambda \cdot \sum_{j=1}^{m} |z_j| \cdot \|w_j\|_2,$$

where $\lambda > 0$ is the sparsity penalty parameter that controls the strength of the regularization, $m$ is the number of neurons in the hidden layer, and $w_j$ is the $j$-th row of the weight matrix $W$.

**JumpReLU.** In our real-data experiments, we also consider the *JumpReLU* activation, a non-smooth, non-monotonic function. Conceptually, it behaves like ReLU for positive inputs but introduces a sharp jump for sufficiently large inputs. In our implementation, we adopt a simplified scalar form adapted to our neuron pre-activation $w_m^\top x + b_m$:

$$\text{JumpReLU}(w_m^\top x; b_m) = \begin{cases} 0, & \text{if } w_m^\top x + b_m < 0, \\ w_m^\top x, & \text{if } w_m^\top x + b_m \geq 0. \end{cases}$$

This activation acts as a hard thresholded identity: it passes the neuron's response only when the pre-activation crosses a bias-controlled threshold. Although JumpReLU does not satisfy the smoothness or Lipschitz conditions required in our theory (see Definition B.3), it is empirically effective and included in our experimental comparisons §5. To train SAEs with JumpReLU activation, we follow Rajamanoharan et al. (2024b) and use straight-through estimators for the gradient of JumpReLU with respect to the bias $b_m$. Specifically, for a small constant $\epsilon > 0$, we approximate the gradients as

$$\frac{\partial \text{JumpReLU}(y; b)}{\partial y} = \begin{cases} 0, & \text{if } y + b < 0 \\ 1, & \text{if } y + b \geq 0 \end{cases}, \quad \frac{\partial \text{JumpReLU}(y; b)}{\partial b} \approx \begin{cases} 0, & \text{if } |y + b| > \frac{\epsilon}{2} \\ \frac{b}{\epsilon}, & \text{if } |y + b| \leq \frac{\epsilon}{2}. \end{cases}$$

The approximation follows the logic: the gradient with respect to $b$ is in essence the gradient of Heaviside step function, which can be approximated by a smoothed version over a small interval around the threshold. Note that for GBA method, we do not apply any gradient for the bias; instead, we update the bias through the frequency control mechanism described in §4.

***Activation sparsity.*** For both the L1 and TopK SAE, we define the sparsity as the number of non-zero entries in the latent $z$, i.e., $\|z\|_0$.

***Minor notational discrepancy.*** In the main text and above definition we express the activation as $\phi(w_m^\top x + b_m)$, whereas in the definition above the JumpReLU activation is indeed as a bivariate function of $w_m^\top x$ and $b_m$. This slight difference is purely notational and does not affect the underlying functionality or the definition of activation sparsity. For simplicity, we always stick to $\phi(w_m^\top x + b_m)$ in the main text, even for the JumpReLU activation.

### C.3 Discussions for Geometric Spacing of Target Activation Frequencies

In our Group Bias Adaptation (GBA) method, we assign neurons into $K$ groups with Target Activation Frequencies (TAFs) $\{p_k\}_{k=1}^K$ that are geometrically spaced between a Highest Target Frequency (HTF) and a Lowest Target Frequency (LTF). This design choice is motivated by several theoretical and practical considerations:

1. **Theoretic guided group allocation.** The theoretical resonance condition depends on $p$ being within **at least a multiplicative** band around $f$ (can be even wider though if we have less superposition). A geometric grid guarantees that for any feature frequency $f$ within $[p_K, p_1]$, there exists some group with TAF $p_k$ within a constant factor of $f$, regardless of the exact exponent of the empirical feature distribution.

2. **Better coverage in log-frequency space.** Geometric spacing minimizes the number of groups $K$ needed to cover a wide frequency range $[p_K, p_1]$ to just logarithmic in the ratio $p_1/p_K$. Empirically, we show in Figure 6 that having $K > 10$ groups is sufficient for covering much of the frequency spectrum. However, other spacing (such as Zipfian) would potentially require many more groups to achieve similar coverage, as the frequency decay is slower than geometric. This would dilute the number of neurons per group, and we might risk missing features in that frequency range.

### C.4 Evaluation Metrics

We explain here the details of the evaluation metrics used in our experiments to assess how well the GBA algorithm recovers the underlying features.

We first introduce the *maximum activation* and *neuron Z-score*, which are used to measure the quality of the learned neurons. Then, we introduce the notion of *Max Cosine Similarity* (MCS) and *Feature Recovery Rate* (FRR), which are used to measure the quality of the alignment between the learned neurons and the ground-truth features, or the consistency of the learned features across different runs. We also introduce the neuron percentage, constructed from the MCS, which is used to generate Figure 5.

We introduce maximum activation and neuron Z-score of a neuron $m$ as follows.

**Maximum activation.** Unless specified, we define the maximum activation of a neuron $m$ as the maximum of its pre-activations over the validation set:

$$\textbf{Maximum Activation}(m) = \max_{x \in \text{Validation Set}} y_m(x), \quad \text{where} \quad y_m(x) = w_m^\top(x - b_{\text{pre}}) + b_m.$$
$$(\text{C.1})$$

Note that the maximum activation is computed based on the tokens in the validation set, which is a held-out dataset separate from the training data used for evaluation purposes. It maps each neuron to a scalar, characterizing the maximum pre-activation of the neuron across all validation tokens.

**Neuron Z-score.** Let $\phi(\cdot)$ denote the neuron's activation function (e.g., ReLU, or JumpReLU). For each neuron $m$ and a minibatch $\{x_i\}_{i=1}^B$, we define its post-activation responses as

$$\phi_{m,i} = \phi\big(w_m^\top(x_i - b_{\text{pre}}) + b_m\big), \qquad i = 1, \ldots, B,$$

where $w_m \in \mathbb{R}^d$ is the neuron's weight vector and $b_m \in \mathbb{R}$ is its bias. We can compute the mean and standard deviation of these activations in the minibatch as

$$\mu_m = \frac{1}{B}\sum_{i=1}^B \phi_{m,i}, \qquad s_m = \sqrt{\frac{1}{B}\sum_{i=1}^B (\phi_{m,i} - \mu_m)^2}.$$

The Z-score of neuron $m$ on data point $x_i$ is defined as

$$Z_{m,i} = (\phi_{m,i} - \mu_m)/s_m \in \mathbb{R}.$$

We can also take the maximum of the Z-scores over the batch:

$$Z_m^{\max} = (\phi_{m,\max} - \mu_m)/s_m, \quad \text{where} \quad \phi_{m,\max} = \max_{1 \le i \le B} \phi_{m,i}. \qquad (\text{C.2})$$

A large value of $Z_{m,i}$ (or $Z_m^{\max} \gg 0$) indicates that on some input $x_i$, the neuron's activation $\phi_{m,i}$ lies multiple standard deviations above its mean. Thus, when $Z_m^{\max}$ is large, neuron $m$ is *well-learned* to sensitively detect certain data points within the batch. More specifically, when $Z_m^{\max}$ is large, the two following conditions hold:

- *Strong Selectivity:* There exists some $x_i$ within the batch such that $\phi_{m,i} \gg \mu_m$, i.e., the neuron's activation $\phi_{m,i}$ "spikes" for input $x_i$.
- *Low Baseline Variability:* Within the whole batch, the neuron's activation $\phi_{m,i}$ is relatively stable, i.e., the standard deviation $s_m$ is moderate.

As a result, $Z_m^{\max}$ serves as a quantitative measure of the neuron's specificity on the batch of data. When generating Figure 5, we use the maximum Z-score of each neuron across the whole validation set to select a subset of neurons.

Next, we introduce the *Max Cosine Similarity* (MCS) and *Feature Recovery Rate* (FRR) metrics, which are used to measure the quality of the alignment between the learned neurons and the ground-truth features, or the consistency of the learned features across different runs.

**Max Cosine Similarity (MCS) for synthetic data.** For each neuron $m$ with weight vector $w_m \in \mathbb{R}^d$, we define

$$\text{MCS}(m) = \max_{i \in [n]} \frac{\langle w_m, v_i \rangle}{\|w_m\|_2 \|v_i\|_2} \in [-1, 1].$$

By definition, $\text{MCS}(m) = 1$ if and only if $w_m$ coincides with one of the true features $v_i$.

**Max Cosine Similarity (MCS) for real data.** For real data, as we do not have access to the ground-truth features, we define the MCS as the maximum cosine similarity between neurons across different runs. This definition is used in Figure 5. Specifically, consider the trained neurons weights $W^{(j)} \in \mathbb{R}^{M \times d}$ for $j = 1, \ldots, J$ where $J$ is the number of runs with different random seeds. We fix the first run as the *host* run and compute the MCS for the $m$-th neuron in the host run with respect to the $j$-th run with $j \ge 2$ as follows:

$$\text{MCS}(m, j) = \max\big\{\cos(W^{(j)}, w_m^{(1)})\big\}.$$

Here, the term inside the max is the cosine similarity between the $m$-th neuron in the host run and all neurons in the $j$-th run, which is an $M$-dimensional vector. The maximum taken outside can be interpreted as finding the best match for the $m$-th neuron in the host run. Now, given a threshold $\tau$ for the MCS value, i.e., the x-axis in Figure 5, we define neuron $m$ to *have an MCS above the threshold if* $\mathrm{MCS}(m, j) \geq \tau$ *for all* $j \geq 2$. We require this condition to hold for all runs $j \geq 2$ because if the algorithm learns a consistent feature, it should be present no matter which random seed is used. When this is the case, neuron $m$ in the host run can find a corresponding neuron in each of the other runs that has a cosine similarity above the threshold $\tau$. Thus, by computing MCS for all the neurons in the host run, we evaluate the consistency of the learned features across different runs.

**Neuron percentage in Figure 5.** Recall that we call the first run of the algorithm the *host run*. Under the definition of MCS, in Figure 5 we plot the **neuron percentage** as a function of the MCS threshold $\tau$. In particular, for any threshold $\tau$ (x-axis in Figure 5), we compute the fraction of neurons in the host run that have an MCS above the threshold across all runs. That is, we define

$$\textbf{Neuron Percentage}(\tau) = \frac{1}{M} \sum_{m=1}^{M} \mathbb{1}\big(\mathrm{MCS}(m, j) \geq \tau, \forall j \geq 2\big). \tag{C.3}$$

By definition, this quantity computes the fraction of neurons in the host run that have an MCS above the threshold $\tau$ across all runs $j \geq 2$. If this quantity is large, the algorithm is able to produce consistent results across different runs with different random seeds. Moreover, because a considerable portion of the neurons of SAE are rarely activated, instead of enumerating over all neurons as in (C.3), we can also consider the neuron percentage over a subset of neurons, denoted by $\mathcal{M} \subseteq [M]$. Then, focusing on $\mathcal{M}$, we define the neuron percentage as

$$\textbf{Neuron Percentage}(\tau, \mathcal{M}) = \frac{1}{|\mathcal{M}|} \sum_{m \in \mathcal{M}} \mathbb{1}\big(\mathrm{MCS}(m, j) \geq \tau, \forall j \geq 2\big). \tag{C.4}$$

In particular, in Figure 5, we choose $\mathcal{M}$ to be the top-$\alpha$ subset of neurons in terms of the maximum activations or neuron Z-score in the host run, which are defined in (C.1) and (C.2), respectively. Note that these two metrics are computed based on the validation dataset. The y-axis in Figure 5 is computed as in (C.4) with these two versions of $\mathcal{M}$.

The notion of Feature Recovery Rate (FRR) is only used for synthetic data, where we have access to the ground-truth features.

**Feature Recovery Rate (FRR).** For one monosemantic feature $v_i$, we say it is *recovered* if there exists a neuron $m \in [M]$ such that the cosine similarity between the neuron and the feature is above a certain threshold $\tau_{\mathrm{align}}$:

$$\mathbb{1}_i = \begin{cases} 1 & \text{if } \exists\, m \in [M] \text{ such that } \big|\langle \widehat{w}_m, v_i \rangle\big|/\|v_i\|_2 \geq \tau_{\mathrm{align}}, \\ 0 & \text{otherwise.} \end{cases}$$

Then the *Feature Recovery Rate* is

$$\mathrm{FRR} = \frac{1}{n} \sum_{i=1}^{n} \mathbb{1}_i \in [0, 1].$$

In words, FRR is the fraction of ground-truth features $v_i$ that have been recovered, i.e., aligned to at least one learned neuron. Here, we find the following way to define the threshold $\tau_{\mathrm{align}}$ useful:

$$\tau_{\mathrm{align}} = \cos\Big(\frac{1}{3}\arccos\Big(\max_{i \neq j} \frac{\langle v_i, v_j \rangle}{\|v_i\|_2 \, \|v_j\|_2}\Big)\Big). \tag{C.5}$$

Intuitively, the angle given by $\arccos$ in (C.5) is the smallest angle among all pairs of features $v_i$ and $v_j$ in $V$, which is denoted by $\theta$ in Figure 11. Then, if a neuron exhibits a cosine similarity above

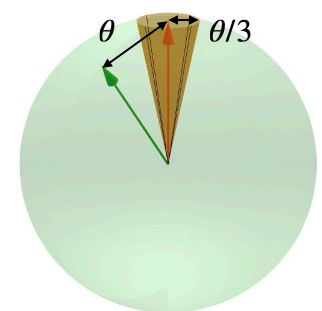

Figure 11: An illustration of the learnable region surrounding the feature. Any neuron weight within the cone has cosine similarity above the threshold with the feature.

the threshold $\tau_{\mathrm{align}}$ with a feature $v_i$, then it lies within the cone centered at $v_i$ with angle $\theta/3$. See Figure 11 for an illustration. By our choice of $\tau_{\mathrm{align}}$, these cones associated to all monosemantic features lie in the $d-1$-dimensional sphere without overlapping, ensuring that each neuron exceeding the threshold is *uniquely* aligned with a single feature.

# D ADDITIONAL EXPERIMENTS DETAILS

We provide additional experimental results and implementation details that complement the main findings presented in the paper.

## D.1 SYNTHETIC EXPERIMENTAL SETUP

We generate synthetic data $X = HV$ satisfying decomposable conditions outlined in Definition B.1. In the default setting, each row of $H$ contains exactly $s$ nonzero entries, each with value $1/\sqrt{s}$, and the support of each row is chosen independently at random. We implement the BA algorithm with a fixed TAF $p$, where the SAE adopts the ReLU activation. We fix the output scale $a_m = 1$ for all $m \in [M]$ and the pre-bias $b_{\text{pre}} = 0$, and initialize the weights $w_m^{(0)}$ uniformly on the unit sphere $\mathbb{S}^{d-1}$ with bias $b_m^{(0)} = 0$.

In synthetic experiments, we use Spherical Gaussian features. For each sample $x_j$ ($j \in [N]$), we randomly sample $s$ indices (with replacement) from $[n]$ to form a multi-set $S_j$. The corresponding features are then combined with a weight $1/\sqrt{s}$ to construct the reconstruction target:

$$x_j = \sum_{i \in S_j} v_i / \sqrt{s}.$$

To evaluate feature learning of neuron $m$, we use the Max Cosine Similarity (MCS) metric. For any neuron $m$, MCS is defined as $\max_{i \in [n]} |\langle w_m / \|w_m\|_2, v_i / \|v_i\|_2 \rangle|$. Thus, MCS measures how well a neuron aligns with the most aligned feature in $V$. We say a neuron is *aligned with some feature* if the MCS for that neuron exceeds a certain threshold. To evaluate overall feature recovery, we use the Feature Recovery Rate (FRR) metric, defined as the proportion of features that are aligned with at least one neuron. See §C.4 for more details on these metrics and the choice of thresholds.

## D.2 ADDITIONAL DETAILS FOR §5

**Data and model details.** We choose the subsets of `Github` and `Wikipedia_en` of `Pile` (Gao et al., 2020) without copyright as our datasets. The `Github` dataset is a collection of 1.2 billion tokens from public `GitHub` repositories, while the `Wikipedia_en` subset contains 1.5 billion tokens from English `Wikipedia` articles. We use the first 99.8k rows from each dataset for training and the next 0.2k rows for validation. Each row in the dataset is truncated to the first 1024 tokens after tokenization. Therefore, the total number of tokens is roughly $N = 100m$. We use the `Qwen2.5-1.5B` base model (Yang et al., 2024) as our LLM, which has 1.5 billion parameters and MLP output dimension 1536. We attach an SAE to the output of the LLM's MLP output at layer 2, 13, and 26 with $M = 66k$ neurons, resulting in three different SAEs for each dataset. The dimension $d$ of the input data points is equal to $d = 1536$. We use the JumpReLU activation (Erichson et al., 2019; Rajamanoharan et al., 2024b) for all training methods.

**Training details.** We train the SAEs using methods such as GBA, TopK, L1, and BA, where BA is simply GBA with one group. For all these methods, we use the AdamW optimizer with a learning rate of $10^{-4}$ and a weight decay of $10^{-2}$. Since the sentences are truncated to 1024 with padding token removed, we set the batch size to $L = 8192$ tokens and a buffer size of $B = 40k$ tokens. Each run can be completed using a single NVIDIA A100 GPU with 80GB memory, and we train 8 epochs for each method. The hyperparameters of each method are set as follows:

- For GBA, we set $K \in \{3, 10, 20\}$, $p_1 \in \{0.05, 0.1, 0.3, 0.5\}$, and $p_K \in \{10^{-4}, 10^{-3}, 5 \times 10^{-3}\}$, where $K$ is the number of groups, $p_1$ is the target frequency of the first group, and $p_K$ is the target frequency of the last group. In addition, we have $\{p_k\}_{k \in [K]}$ form a geometric sequence.
- For the BA method, we set the HTF to be from $\{10^{-1}, 3 \times 10^{-2}, 10^{-2}, 3 \times 10^{-3}\}$ and vary the choice. The other parameters are the same as GBA.
- For TopK method, we implement two versions — the pre-activation TopK and the post-activation TopK. See §C.2 for details. We vary the value of $K$ in $\{50, 100, 200, 300, 400, 500, 600\}$.
- For L1 method, we vary the penalty parameter $\lambda$ in $\{10^{-1}, 3 \times 10^{-2}, 10^{-2}, 3 \times 10^{-3}, 10^{-3}\}$.

### D.3 COMPARISON BETWEEN JUMPRELU AND RELU ACTIVATION

For the SAE trained on the Github dataset at layer 26, we compare the performance between JumpReLU and the standard ReLU activations across all methods considered in this paper. As shown in Figure 12, the sparsity-loss frontiers for TopK and L1 methods are nearly identical under both activations. However, the GBA method demonstrates a marked improvement when using JumpReLU activation. With ReLU, decreasing the neuron bias also reduces the output magnitude. Thus more neurons are needed to compensate for the loss of output magnitude, which leads to a less sparse model, which degrades the sparsity-loss frontier. In contrast, JumpReLU decouples the neuron output magnitude from its bias—only the activation frequency is influenced—yielding a more robust sparsity-loss performance.

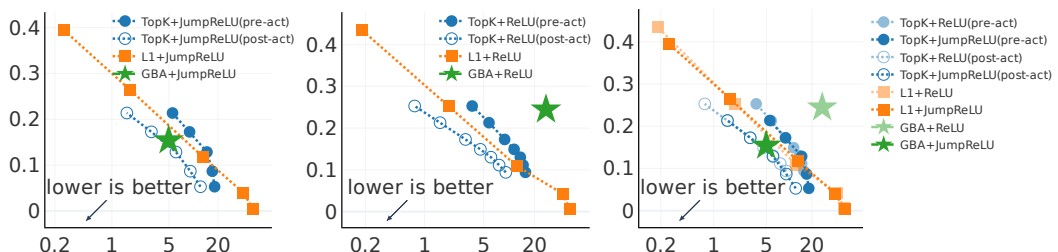

Figure 12: Comparison of sparsity-loss frontier between JumpReLU and ReLU activations. The **left** and **middle** plots show the sparsity-loss frontier with JumpReLU and ReLU activations, respectively. The **right** plot is a combination of the two, where the faded plots represent the sparsity-loss frontier of the ReLU activation.

**Bias clamping to prevent over-sparsification.** During the bias scheduling subroutine of the GBA algorithm (Algorithm 1), we enforce a clamp on the bias values, restricting them to the range $[-1, 0]$. This constraint serves two primary purposes. The upper bound of $0$ ensures that a neuron is only activated when the input data exhibits a sufficient alignment with the neuron's weight vector. Consequently, allowing negative bias values ($b_m < 0$) effectively prevents excessive or premature activation of neurons.

The lower bound of $-1$ is implemented to avoid over-deactivation and the emergence of a reinforcing loop. We have observed experimentally that when the pre-bias ($b_{\text{pre}}$) significantly deviates from zero, certain neurons may develop weights that are in opposition to the pre-bias to compensate for this drift. As these compensatory neurons are more likely to be activated by the initial pre-bias, the GBA algorithm might inadvertently continue to deactivate them by further reducing their bias ($b_m$). This deactivation would then necessitate an increase in the neuron's weight to maintain its influence, leading to a counterproductive cycle of deactivation and weight growth.

By limiting the bias to be no less than $-1$, we effectively interrupt this reinforcing loop and promote training stability. The rationale behind choosing $-1$ as the lower bound stems from the fact that our input data is normalized. This normalization typically results in pre-activation values that are significantly smaller than $1$, with values approaching $1$ only when the data strongly activates specific neurons. Therefore, a lower bias bound of $-1$ provides sufficient range for deactivation without causing the problematic feedback loop. This clamping strategy has been shown to significantly enhance the stability of the training process.

## E    GOOD INITIALIZATION AND GAUSSIAN CONDITIONING

In this section, we provide proofs for two important lemmas: Theorem E.1 on the initialization properties and Theorem E.2 on the Gaussian conditioning. These lemmas provide the necessary foundation for analyzing the SAE training dynamics, enabling us to isolate and control the relevant sources of randomness throughout the analysis.

### E.1 INITIALIZATION PROPERTIES

If we initialize the network with a sufficiently large number of neurons $M$, then for each neuron, there must exist a feature that aligns well with it. However, the question is how many neurons we need to achieve a *sufficiently large alignment* and with *all features* of interest simultaneously. Theorem E.1 provides an answer to this question. In particular, we prove that when $M$ is sufficiently large, for each feature $v_i$, we can find a neuron $m_i$ that aligns well with it (***InitCond-1***) while maintaining a small alignment with all other features (***InitCond-2***).

**Lemma E.1** (Good initialization). *Given $n$ i.i.d. features $\{v_i\}_{i=1}^n$ with $v_i \sim \mathcal{N}(0, I_d)$ and weights $\{w_m^{(0)}\}_{m=1}^M$ independently initialized from the uniform distribution on the unit sphere, then for any constants $\varepsilon \in (0,1)$ and $c > 0$ such that $n^{-c}$ upper bound $\exp(-n^{O(\varepsilon)})$, with probability at least $1 - n^{-c}$ over the randomness of both $\{v_i\}_{i=1}^n$ and $\{w_m^{(0)}\}_{m=1}^M$, one can select a sequence of neurons $\{m_i\}_{i=1}^n$ satisfying the following properties:*

1. *For any $i \in [n]$, we have*

$$\textbf{\textit{InitCond-1}}: \quad \langle v_i, w_{m_i}^{(0)} \rangle \geq (1 - \varepsilon)\sqrt{2\log(M/n)}.$$

2. *For any $i \in [n]$, when conditioned on the selection of neuron $m_i$, which **aligns well** with feature $v_i$ in the sense of **InitCond-1**, the distribution of the remaining features $\{v_j\}_{j \neq i}$ remains unchanged, i.e., they are independently drawn from $\mathcal{N}(0, I_d)$.*

3. *For any $i \in [n]$, when conditioned on selecting neuron $m_i$, with probability at least $1 - n^{-1-4\varepsilon}$ over the randomness of $\{v_j\}_{j \neq i}$, we have*

$$\textbf{\textit{InitCond-2}}: \quad \langle v_j, w_{m_i}^{(0)} \rangle \leq \sqrt{2}(1 + \varepsilon) \cdot \sqrt{2\log n}, \quad \forall j \neq i$$

*Proof of Theorem E.1.* We present the proof by constructing such $m_1, m_2, \ldots, m_n$ explicitly. Suppose we are provided with $n$ features $v_1, v_2, \ldots, v_n$ and $M$ neurons with initial weights $w_1^{(0)}, w_2^{(0)}, \ldots, w_M^{(0)}$. We first put all the pair-wise alignments $\langle v_i, w_m^{(0)} \rangle$ into a matrix $A \in \mathbb{R}^{n \times M}$, where $A_{im} = \langle v_i, w_m^{(0)} \rangle$ for $i \in [n]$ and $m \in [M]$. The algorithm execute as follows for $i$ going from 1 to $n$:

1. Randomly divide the $M$ neurons into $n$ disjoint groups $\mathcal{M}_1, \mathcal{M}_2, \ldots, \mathcal{M}_n$ such that each group $\mathcal{M}_i$ contains $M/n$ neurons.

2. For each $\mathcal{M}_i$, find the neuron $m_i$ as the one that maximizes the alignment with feature $v_i$, i.e.,

$$m_i = \underset{m \in \mathcal{M}_i}{\operatorname{argmax}} A_{i,m} = \underset{m \in \mathcal{M}_i}{\operatorname{argmax}} \langle v_i, w_m^{(0)} \rangle.$$

By construction, we know that the selection of $m_i$ is independent of the selection of $m_j$ for $i \neq j$. It is not hard to see that the distribution of $\langle v_i, w_m^{(0)} \rangle$ is the same (up to scaling) as the distribution of the first coordinate of a random vector uniformly distributed on the unit sphere. Therefore, for each $i \in [n]$, each group $\{A_{i,m}\}_{\mathcal{M}_i}$ is iid sampled from the following distribution:

$$A_{i,m}\big|_{m \in \mathcal{M}_i} \overset{d}{=} \frac{Z_1 \|v_i\|_2}{\sqrt{Z_1^2 + \ldots + Z_d^2}}, \quad \text{where} \quad Z_k \sim \mathcal{N}(0,1), \forall k \in [d].$$

By the concentration for Chi-square distribution, Theorem J.1, we know that the denominator and also the norm of $\|v_i\|_2$ satisfies

$$\mathbb{P}\left( \left| \sum_{k=1}^d Z_k^2 - d \right| \geq 2\sqrt{d \log \delta^{-1}} + 2\log \delta^{-1} \right) \leq \delta,$$

$$\mathbb{P}\left( \left| \|v_i\|_2^2 - d \right| \geq 2\sqrt{d \log \delta^{-1}} + 2\log \delta^{-1} \right) \leq \delta.$$

To proceed, we label each $d$-dimensional random vector as $Z^{(i,m)} = (Z_1^{(i,m)}, \ldots, Z_d^{(i,m)})$, where the superscript $(i,m)$ corresponds to feature $i$ and neuron $m$. Applying a union bound over all $n \times M/n$ pairs of $(i,m)$ and choosing $\delta = n^{-c}/M$ for some universal constant $c$, we deduce that with probability at least $1 - n^{-c}$, the following holds for all $i \in [n]$ and $m \in [M]$:

$$A_{i,m} \geq \frac{Z_1^{(i,m)} \big(d - C\sqrt{d\log(nM)} - C\log(nM)\big)^{1/2}}{\big(d + C\sqrt{d\log(nM)} + C\log(nM)\big)^{1/2}},$$

where $C$ is a universal constant. Moreover, by property of the maximum of Gaussian random variables in Theorem J.4, it holds that

$$\mathbb{P}\Big(\max_{m \in \mathcal{M}_i} Z_1^{(i,m)} \geq (1 - \varepsilon/2)\sqrt{2\log(M/n)}\Big) \geq 1 - \exp\Big(-\frac{(M/n)^{\varepsilon - \varepsilon^2/4}}{3\sqrt{\pi \log(M/n)}}\Big). \tag{E.1}$$

Here, we divide $\varepsilon$ by 2 because

$$A_{i,m} \geq Z_1^{(i,m)} \cdot \frac{\big(d - C\sqrt{d\log(nM)} - C\log(nM)\big)^{1/2}}{\big(d + C\sqrt{d\log(nM)} + C\log(nM)\big)^{1/2}} \geq \frac{1 - \varepsilon}{1 - \varepsilon/2} Z_1^{(i,m)}$$

for small constant $\varepsilon$. Consequently, by multiplying both sides of the inequality inside $\mathbb{P}(\cdot)$ in (E.1) by $\frac{1-\varepsilon}{1-\varepsilon/2}$, we can recast the probability statement so that the maximum of $A_{i,m}$ over all $m \in \mathcal{M}_i$ exceeds $(1-\varepsilon)\sqrt{2\log(M/n)}$. By taking a union bound for $i \in [n]$, the probability of successfully finding a sequence of neurons $m_1, m_2, \ldots, m_n$ satisfying $A_{i,m} > (1-\varepsilon)\sqrt{2\log(M/n)}$ for all $i \in [n]$ and $m \in [M]$ is at least

$$\mathbb{P}\Big(\forall i \in [n] : \max_{m \in \mathcal{M}_i} A_{i,m} > (1-\varepsilon)\sqrt{2\log(M/n)}\Big) \geq 1 - n \cdot \exp\Big(-\frac{(M/n)^{\varepsilon - \varepsilon^2/4}}{3\sqrt{\pi \log(M/n)}}\Big) \geq 1 - n^{-c}.$$

where we can safely take $c$ to some constant as the failure probability is exponentially small in $n$ given that $M \geq n^2$. To this end, we conclude that with probability at least $1 - n^{-c}$, we can find a sequence of non-overlapping neurons $m_1, m_2, \ldots, m_n$ such that $A_{i,m_i} > (1-\varepsilon)\sqrt{2\log(M/n)}$ for all $i \in [n]$.

Observe that the selection of each neuron $m_i$ is done independently for each feature. Consequently, when we condition on the selection of $m_i$, the distribution for the remaining features $\{v_j\}_{j \neq i}$ remains unchanged. This proves the second statement.

It remains to analyze the probability that $A_{j,m_i} < \sqrt{2}(1+\varepsilon) \cdot \sqrt{2\log n}$ for all $j \in [n]$ and $i \neq j$. By the second statement, we know that when conditioned on neuron $m_i$, the collection $\{A_{j,m_j}\}_{j \neq i}$ (for any fixed $i$) consists of $(n-1)$ independent and identically distributed random variables with distribution $\mathcal{N}(0,1)$. Thus, we can apply the tail probability for the maximum of Gaussian random variables in Theorem J.2 to obtain

$$\mathbb{P}\Big(\max_{j \in [n]: j \neq i} A_{j,m_i} > \sqrt{2}(1+\varepsilon) \cdot \sqrt{2\log n}\Big) \leq n^{1-2(1+\varepsilon)^2} \leq n^{-1-4\varepsilon}.$$

Thus, we prove the last argument for Theorem E.1. $\qquad\square$

A direct corollary of Theorem E.1 is that **InitCond-1** and **InitCond-2** hold simultaneously for all $i \in [n]$ and $j \neq i$ with probability at least $1 - n^{-c} - n^{-4\varepsilon} \leq 1 - n^{-\varepsilon}$ after taking a union bound over the success of InitCond-2 for all $i \in [n]$. These two conditions together imply that the neuron $m_i$ exclusively focuses on feature $v_i$ at initialization, which is crucial for developing a $1 - o(1)$ alignment with feature $v_i$ during training.

## E.2 Rewriting the Gradient Descent Iteration

**Single neuron analysis.** In the previous Theorem E.1, we have shown a correspondence between each feature $v_i$ and a neuron $m_i$ such that the initial weight of neuron $m_i$ aligns well with feature $v_i$ while maintaining small alignments with all other features. In other words, $m_i$ is the neuron that is most likely to learn feature $v_i$ during training. As the neuron dynamics are decoupled under the small output scale assumption, we only need to analyze the dynamics of neuron $m_i$ to understand how feature $v_i$ is learned.

**Notation.** In the following, we denote by $v$ the feature of interest and by $w_t$ the weight of the corresponding neuron at iteration $t$. Let $T$ be the maximum number of steps considered and the time step $t$ ranges from $0$ to $T$. For the sake of notational convenience, we also denote the feature of interest by $w_{-1} = v$ and the normalization $\bar{w}_{-1} = v/\|v\|_2$. Meanwhile, $w_0 = \bar{w}_0$ is the initialization that is already normalized to unit length. Here, the bar notation indicates that the vector is normalized to unit length throughout the whole proof.

**Reformulating the iteration.** In this section, we reformulate the gradient descent update (B.1) to isolate the contribution of a specific feature $v$ from the remaining features. Recall that the data matrix is given by $X = HV$, where $H \in \mathbb{R}^{N \times n}$ is the weight matrix and $V \in \mathbb{R}^{n \times d}$ is the feature matrix. The gradient descent update (B.1) with gradient explicit in (B.2) is

$$\textbf{Modified BA:} \quad w_t = \frac{w_{t-1} + \eta\, g_t}{\|w_{t-1} + \eta\, g_t\|_2}, \qquad \text{where} \qquad g_t = \sum_{\ell=1}^{N} \varphi(w_{t-1}^\top x_\ell; b_t) x_\ell,$$

which can be written in terms of $H$ and $V$ as:

$$
\begin{aligned}
y_t &= V \bar{w}_{t-1}, \quad b_t = \mathcal{A}_t(H y_t), \quad u_t = H^\top \varphi(H y_t; b_t), \\
w_t &= V^\top u_t + \eta^{-1} \bar{w}_{t-1}, \quad \bar{w}_t = w_t/\|w_t\|_2.
\end{aligned}
\tag{E.2}
$$

Here, the meaning of these quantities are given as follows:

- $y_t \in \mathbb{R}^d$ is the projection of the normalized weight vector onto all the features, which we refer to as the *feature pre-activation*.

- $b_t \in \mathbb{R}$ is the bias term updated by a bias adaptation algorithm $\mathcal{A}_t(\cdot)$ that depends on the feature preactivation and time $t$.

- $u_t \in \mathbb{R}^n$ is the *feature post-activation* that aggregates the post-activation information from all the data points back to the feature space.

- $w_t \in \mathbb{R}^d$ is the unnormalized weight vector after one step of gradient descent update, and $\bar{w}_t \in \mathbb{R}^d$ is the normalized weight vector.

In our analysis, as the bias is fixed, $\mathcal{A}_t(\cdot)$ always returns the same bias value. However, we keep this general form which can be useful for adapting the current proof framework to handle more complex bias adaptation algorithms. Note that $\varphi(H y_t; b_t) \in \mathbb{R}^N$ obtained from the gradient calculation in (B.2) is not exactly the post-activation (recall definition $\varphi(x; b) = \phi(x + b) + \phi'(x + b)x$, where $\phi$ is the actual activation function. ) However, in the following proof, we will abuse the notation and refer to $\varphi(H y_t; b_t)$ as the post-activation for brevity.

(a) The weight matrix $H$ is splitted into matrices $E$ and $F$ by row according to whether the corresponding entries in the $i$-th column are zero or not. The nonzero entries in the $i$-th column of $H$ are collected as vector $\theta$.

(b) Isolating the $i$-th feature from feature matrix $V$.

Figure 13: Illustration of the split of matrices $H$ and $V$.

Without loss of generality, suppose $v$ is the $i$-th feature. To *isolate the contribution from feature of interest $v$ from the remaining features*, we decompose the weight matrix $H$ into three parts: (i) $\theta$: the non-zero entries of the $i$-th column, (ii) $F$: the rows with non-zero entries in the $i$-th column,

and (iii) $E$: the remaining rows with zero entries in the $i$-th column. Formally, suppose $v$ is the $i$-th feature, then we decompose $H$ as follows:

$$\theta = \left(H_{ki} : H_{ki} \neq 0\right)_{k \in [N]}, \ F = \left(H_{kj} : H_{ki} \neq 0\right)_{k \in [N], j \in [n] \setminus \{i\}}, \ E = \left(H_{kj} : H_{ki} = 0\right)_{k \in [N], j \in [n] \setminus \{i\}}.$$
(E.3)

Notably, the rows of $E$ and $F$ do not include the $i$-th column of $H$, as it is already isolated as vector $\theta$. See Figure 13a for an illustration of this decomposition.

Using the above decomposition, we can rewrite the actual projection of the weights $\bar{w}_{t-1}$ on each data point as

$$HV\bar{w}_{t-1} = \text{Interleave}\left([F; E] \cdot V_{-i}\bar{w}_{t-1} + [\theta; \mathbf{0}] \cdot v^\top \bar{w}_{t-1}\right)$$
$$= \text{Interleave}\left([F; E] \cdot y_{t,-i} + [\theta; \mathbf{0}] \cdot v^\top \bar{w}_{t-1}\right),$$

where $[E; F]$ is the vertical concatenation of $E$ and $F$, $V_{-i}$ is the feature matrix $V$ with the $i$-th row removed, and $y_{t,-i} = V_{-i}\bar{w}_{t-1}$ is the vector $y_t$ with the $i$-th entry removed. The interleave operation simply restores the original order of the rows in $H$. Therefore, we can rewrite the original $u_t$ in (E.2) as

$$u_t = H^\top \varphi(Hy_t; b_t) = E^\top \varphi(Ey_{t,-i}; b_t) + F^\top \varphi(Fy_{t,-i} + \theta \cdot v^\top \bar{w}_{t-1}; b_t). \qquad \text{(E.4)}$$

In order to avoid overcomplicated subscripts, we let $V$ denote the feature matrix $V_{-i}$ with the $i$-th row removed, and let $v$ refer to the original $i$-th row of $V$. See Figure 13b for an illustration of this decomposition. We also rewrite $y_{t,-i}$ as $y_t$, and following the above notation, we still have $y_t = V\bar{w}_{t-1}$. Now with (E.4), we can explicitly separate the contribution of feature $v$ from the remaining features in the gradient descent iteration (E.2) and obtain the following equivalent iteration:

> **Gradient Descent Iteration**
>
> $$\textbf{feature pre-activation:} \quad y_t = V\bar{w}_{t-1}, \quad \bar{w}_{t-1} = w_{t-1}/\|w_{t-1}\|_2,$$
> $$\textbf{bias scheduling:} \quad b_t = \mathcal{A}_t(b_{t-1}, Ey_t, Fy_t + \theta \cdot v^\top \bar{w}_{t-1}),$$
> $$\textbf{feature post-activation:} \quad u_t = E^\top \varphi(Ey_t; b_t) + F^\top \varphi(Fy_t + \theta \cdot v^\top \bar{w}_{t-1}; b_t),$$
> $$\textbf{weight update:} \quad w_t = V^\top u_t + v\theta^\top \varphi(Fy_t + \theta \cdot v^\top \bar{w}_{t-1}; b_t) + \eta^{-1}\bar{w}_{t-1},$$
>
> (E.5)

Note that the notation in (E.5) is self-consistent with $E, F, \theta$ defined in (E.3) and $V, v$ defined below (E.4). We will keep using this notation throughout the rest of the proof.

### E.3 GAUSSIAN CONDITIONING

Since both the feature of interest $v$ and each row of the feature matrix $V$ follow Gaussian distributions, we can leverage the properties of Gaussian distributions to simplify the dynamics. However, the coupling between different iterations prohibits a direct application of Gaussian properties. This challenge motivates us to explicitly split the intermediate variables in (E.5) into two components: (i) a *coupling component* that lies in the subspace spanned by the previous intermediate variables, and (ii) an *independent component* that is orthogonal to this subspace. We can then apply some Gaussian concentration arguments to the orthogonal component to simplify the dynamics.

**Additional notation.** To achieve this, we introduce some additional notations. Let us define $P_{w_{-1:t-1}}x$ as the projection of $x$ onto the subspace spanned by $\{w_{-1}, \ldots, w_{t-1}\}$, and $P^\perp_{w_{-1:t-1}}x = x - P_{w_{-1:t-1}}x$ as the orthogonal projection. In the following, we use the notations $w_t^\perp = P^\perp_{w_{-1:t-1}}w_t$ to denote the new direction induced by $w_t$, and we define $u_t^\perp = P^\perp_{u_{1:t-1}}u_t$ in a similar manner (note that $u_t$ starts from $t = 1$). Note that when $t < 2$, $u_{1:t-1}$ is empty and $P^\perp_{u_{1:t-1}}$ becomes the identity mapping. Also, we enforce $w_{-1} = w_{-1}^\perp = v$.

In the following, we use the trick of Gaussian conditioning (Wu & Zhou, 2023; Bayati & Montanari, 2011; Montanari & Wu, 2023) to simplify the dynamics in (E.5). Specifically, we will define an alternative dynamics that is distributionally equivalent to the original one, where for each iteration, two

new independent Gaussian vectors are introduced to replace the original Gaussian components coming from the $V$ matrix. To make the presentation clearer, we will denote the variables in the original dynamics in (E.5) by $(y_t, w_t, u_t, b_t)$ and the variables in the alternative dynamics by $(\widetilde{y}_t, \widetilde{w}_t, \widetilde{u}_t, \widetilde{b}_t)$ in the following proofs.

**Lemma E.2** (Alternative dynamics). *For any $t \in \mathbb{N}$, let $z_{-1}, z_0, \ldots, z_t$ and $\widetilde{z}_1, \ldots, \widetilde{z}_t$ be sequences of i.i.d. random vectors from $\mathcal{N}(0, I_{n-1})$ and $\mathcal{N}(0, I_{d-1})$, respectively, with mutual independence. In addition $z_{-1:t}$ and $\widetilde{z}_{1:t}$ are also independent of the initialization $\bar{w}_0$ and the feature of interest $v$. Consider the following alternative iteration for $(\widetilde{y}_t, \widetilde{w}_t)$:*

$$\widetilde{y}_t = \sum_{\tau=-1}^{t-1} \widetilde{\alpha}_{\tau,t-1} \cdot P_{\widetilde{u}_{1:\tau}}^{\perp} z_\tau + \sum_{\tau=1}^{t-1} \widetilde{\alpha}_{\tau,t-1} \cdot \frac{\|\widetilde{w}_\tau^{\perp}\|_2}{\|\widetilde{u}_\tau^{\perp}\|_2} \cdot \frac{\widetilde{u}_\tau^{\perp}}{\|\widetilde{u}_\tau^{\perp}\|_2}, \tag{E.6}$$

$$\widetilde{w}_t = \sum_{\tau=-1}^{t-1} \langle P_{\widetilde{u}_{1:\tau}}^{\perp} z_\tau, \widetilde{u}_t \rangle \cdot \frac{\widetilde{w}_\tau^{\perp}}{\|\widetilde{w}_\tau^{\perp}\|_2} + \sum_{\tau=1}^{t-1} \frac{\langle \widetilde{u}_\tau^{\perp}, \widetilde{u}_t \rangle}{\|\widetilde{u}_\tau^{\perp}\|_2} \cdot \frac{\|\widetilde{w}_\tau^{\perp}\|_2}{\|\widetilde{u}_\tau^{\perp}\|_2} \cdot \frac{\widetilde{w}_\tau^{\perp}}{\|\widetilde{w}_\tau^{\perp}\|_2}$$

$$+ P_{\widetilde{w}_{-1:t-1}}^{\perp} \widetilde{z}_t \cdot \|\widetilde{u}_t^{\perp}\|_2 + v\,\theta^\top \varphi(F\widetilde{y}_t + \theta \cdot v^\top \widetilde{w}_{t-1}; b_t) + \eta^{-1}\widetilde{w}_{t-1},$$

*where we define the alignment*

$$\widetilde{\alpha}_{\tau,t} = \frac{\langle \widetilde{w}_\tau^{\perp}, \widetilde{\bar{w}}_t \rangle}{\|\widetilde{w}_\tau^{\perp}\|_2} \quad with \quad \widetilde{\bar{w}}_t = \frac{\widetilde{w}_t}{\|\widetilde{w}_t\|_2}.$$

*In addition, $(b_t, \widetilde{u}_t)$ in the alternative dynamics are updated by the same formula as in (E.5):*

$$b_t = \mathcal{A}_t(b_{t-1}, E\widetilde{y}_t, F\widetilde{y}_t + \theta \cdot v^\top \widetilde{w}_{t-1}), \quad \widetilde{u}_t = E^\top \varphi(E\widetilde{y}_t; b_t) + F^\top \varphi(F\widetilde{y}_t + \theta \cdot v^\top \widetilde{w}_{t-1}; b_t). \tag{E.7}$$

*Then, conditioned on $\widetilde{w}_{-1} = v$ (the same as our previous definition of $w_{-1} = v$) and $\widetilde{w}_0 = w_0$ being the initialization of the neuron weight, the alternative dynamics $(\widetilde{y}_\tau, \widetilde{w}_\tau, \widetilde{u}_\tau, \widetilde{b}_\tau)_{\tau=1}^t$ from (E.6) and (E.7) and the original dynamics $(y_\tau, w_\tau, u_\tau, b_\tau)_{\tau=1}^t$ from (E.5) follow the same distribution.*

*Proof of Theorem E.2.* To show that the trajectory from (E.6) and (E.7) follow the same distribution as the trajectory from (E.5), we first decompose the iteration in (E.5) in the following lemma.

**Lemma E.3** (Decomposition). *For the iteration in (E.5), define the alignment between the weight vector $\bar{w}_t$ and the weight direction $w_t^{\perp}$ as $\alpha_{\tau,t} = \langle \bar{w}_t, w_\tau^{\perp} \rangle / \|w_\tau^{\perp}\|_2$, Then, we have the following decomposition for the preactivation vector $y_t \in \mathbb{R}^{n-1}$:*

$$y_t = \sum_{\tau=-1}^{t-1} \alpha_{\tau,t-1} \cdot P_{u_{1:\tau}}^{\perp} V \frac{w_\tau^{\perp}}{\|w_\tau^{\perp}\|_2} + \sum_{\tau=1}^{t-1} \alpha_{\tau,t-1} \cdot \frac{\|w_\tau^{\perp}\|_2}{\|u_\tau^{\perp}\|_2} \cdot \frac{u_\tau^{\perp}}{\|u_\tau^{\perp}\|_2},$$

*and the following decomposition for the unnormalized weight vector $w_t \in \mathbb{R}^d$:*

$$w_t = \sum_{\tau=-1}^{t-1} \left\langle P_{u_{1:\tau}}^{\perp} V \frac{w_\tau^{\perp}}{\|w_\tau^{\perp}\|_2}, u_t \right\rangle \cdot \frac{w_\tau^{\perp}}{\|w_\tau^{\perp}\|_2} + \sum_{\tau=1}^{t-1} \frac{\langle u_\tau^{\perp}, u_t \rangle}{\|u_\tau^{\perp}\|_2} \cdot \frac{\|w_\tau^{\perp}\|_2}{\|u_\tau^{\perp}\|_2} \cdot \frac{w_\tau^{\perp}}{\|w_\tau^{\perp}\|_2}$$

$$+ P_{w_{-1:t-1}}^{\perp} V^\top \frac{u_\tau^{\perp}}{\|u_\tau^{\perp}\|_2} \cdot \|u_t^{\perp}\|_2 + v\theta^\top \varphi(Fy_t + \theta \cdot v^\top \bar{w}_{t-1}; b_t) + \eta^{-1}\bar{w}_{t-1},$$

*Proof.* See §E.4 for the proof of Theorem E.3. □

With the above decomposition, if we do the following substitution for $y_t$ and $w_t$ in the above lemma:

$$P_{u_{1:t}}^{\perp} z_t \leftarrow P_{u_{1:t}}^{\perp} V \frac{w_t^{\perp}}{\|w_t^{\perp}\|_2}, \qquad P_{w_{-1:t-1}}^{\perp} \widetilde{z}_t \leftarrow P_{w_{-1:t-1}}^{\perp} V^\top \frac{u_t^{\perp}}{\|u_t^{\perp}\|_2},$$

the assertion in Theorem E.2 follows immediately. The following proof is devoted to showing that the substitution does not change the joint distribution of the whole dynamics. To show that, we just need to verify that for each iteration $t$, when *conditioned on all the history up to iteration $t - 1$*, the

two newly introduced vectors $P_{u_{1:t}}^\perp V w_t^\perp / \|w_t^\perp\|_2$ and $P_{w_{-1:t-1}}^\perp V^\top u_t^\perp / \|u_t^\perp\|_2$ still follow a standard Gaussian distribution and are independent of all the history.

To proceed, we denote the original iteration in (E.5) by $(y_t, w_t, u_t, b_t)$ and the alternative iteration in (E.6) and (E.7) by $(\widetilde{y}_t, \widetilde{w}_t, \widetilde{u}_t, \widetilde{b}_t)$. Following explicitly from the decomposition in Theorem E.3 and the construction in (E.6), we can further derive the following dependency between the variables in both iterations.

**Lemma E.4.** *For each iteration $(u_t, w_t)$ in (E.5), it holds for any $t \geq 1$ that*

$$u_t \in \sigma\left( w_{-1:0}, \left\{ P_{u_{1:\tau}}^\perp V \frac{w_\tau^\perp}{\|w_\tau^\perp\|_2} \right\}_{\tau=-1}^{t-1}, \left\{ P_{w_{-1:\tau-1}}^\perp V^\top \frac{u_\tau^\perp}{\|u_\tau^\perp\|_2} \right\}_{\tau=1}^{t-1} \right),$$

$$w_t \in \sigma\left( w_{-1:0}, \left\{ P_{u_{1:\tau}}^\perp V \frac{w_\tau^\perp}{\|w_\tau^\perp\|_2} \right\}_{\tau=-1}^{t-1}, \left\{ P_{w_{-1:\tau-1}}^\perp V^\top \frac{u_\tau^\perp}{\|u_\tau^\perp\|_2} \right\}_{\tau=1}^{t} \right).$$

*where $\sigma(X)$ denotes the $\sigma$-algebra generated by the random variable $X$. For the Gaussian conditioning iteration $(\widetilde{u}_t, \widetilde{w}_t)$ in (E.6) and (E.7), it holds for any $t \geq 1$ that*

$$\widetilde{u}_t \in \sigma(\widetilde{w}_{-1:0}, \{P_{\widetilde{u}_{1:\tau}}^\perp z_\tau\}_{\tau=-1}^{t-1}, \{\widetilde{z}_\tau\}_{\tau=1}^{t-1}), \quad \widetilde{w}_t \in \sigma(\widetilde{w}_{-1:0}, \{P_{\widetilde{u}_{1:\tau}}^\perp z_\tau\}_{\tau=-1}^{t-1}, \{\widetilde{z}_\tau\}_{\tau=1}^{t}).$$

*Proof.* See §E.4 for a proof of Theorem E.4. $\square$

The message of the above lemma is intuitive: each iteration only inserts new randomness coming from

$$P_{u_{1:t-1}}^\perp V \frac{w_{t-1}^\perp}{\|w_{t-1}^\perp\|_2} \quad \text{and} \quad P_{w_{-1:t-1}}^\perp V^\top \frac{u_t^\perp}{\|u_t^\perp\|_2}$$

for the original iteration, and from

$$P_{\widetilde{u}_{1:t-1}}^\perp z_{t-1} \quad \text{and} \quad P_{\widetilde{w}_{-1:t-1}}^\perp \widetilde{z}_t$$

for the alternative iteration. Using the dependency results, we next prove the equivalence between the trajectory $\{\widetilde{w}_{-1}, \widetilde{w}_0, (\widetilde{y}_\tau, \widetilde{w}_\tau, \widetilde{u}_\tau, \widetilde{b}_\tau)_{\tau=1}^t\}$ from the Gaussian conditioning and the trajectory $\{w_{-1}, w_0, (y_\tau, w_\tau, u_\tau, b_\tau)_{\tau=1}^t\}$ from the original iteration by considering the conditional distribution of the newly introduced randomness at each iteration. Let us define $A_t$ as a realization of the random variables $(\widetilde{w}_{-1:0}, z_{-1:t-1}, \widetilde{z}_{1:t})$ or

$$\left( w_{-1:t}, \left\{ P_{u_{1:\tau}}^\perp V \frac{w_\tau^\perp}{\|w_\tau^\perp\|_2} \right\}_{\tau=-1}^{t-1}, \left\{ P_{w_{-1:\tau-1}}^\perp V^\top \frac{u_\tau^\perp}{\|u_\tau^\perp\|_2} \right\}_{\tau=1}^{t} \right).$$

By property of the Gaussian ensembles, it holds that

$$P_{u_{1:t}}^\perp V \frac{w_t^\perp}{\|w_t^\perp\|_2} \left| \left\{ \left( w_{-1:0}, \left\{ P_{u_{1:\tau}}^\perp V \frac{w_\tau^\perp}{\|w_\tau^\perp\|_2} \right\}_{\tau=-1}^{t-1}, \left\{ P_{w_{-1:\tau-1}}^\perp V^\top \frac{u_\tau^\perp}{\|u_\tau^\perp\|_2} \right\}_{\tau=1}^{t} \right) = A_t \right\} \right.$$

$$\overset{d}{=} P_{u_{1:t}}^\perp V_t \frac{w_t^\perp}{\|w_t^\perp\|_2} \left| \left\{ \left( w_{-1:0}, \left\{ P_{u_{1:\tau}}^\perp V \frac{w_\tau^\perp}{\|w_\tau^\perp\|_2} \right\}_{\tau=-1}^{t-1}, \left\{ P_{w_{-1:\tau-1}}^\perp V^\top \frac{u_\tau^\perp}{\|u_\tau^\perp\|_2} \right\}_{\tau=1}^{t} \right) = A_t \right\} \right.$$

$$\overset{d}{=} P_{\widetilde{u}_{1:t}}^\perp z_t \,\big|\, \left\{ (\widetilde{w}_{-1:t}, \widetilde{u}_{1:t}, z_{-1:t-1}, \widetilde{z}_{1:t}) = A_t \right\}. \tag{E.8}$$

where $V_t \overset{d}{=} V$ is an independent copy of $V$ and is independent of all the histories. Here, the first equality holds because $P_{u_{1:t}}^\perp V w_t^\perp / \|w_t^\perp\|_2$ is orthogonal to any of the previous row/column space that we have conditioned on. In particular,

- $P_{u_{1:t}}^\perp V \frac{w_t^\perp}{\|w_t^\perp\|_2}$ is orthogonal to $\{P_{u_{1:\tau}}^\perp V \frac{w_\tau^\perp}{\|w_\tau^\perp\|_2}\}_{\tau=-1}^{t-1}$ in the column space of $V$ since $w_t^\perp$ is orthogonal to $w_\tau^\perp$ for any $\tau < t$.

- $P_{u_{1:t}}^\perp V \frac{w_t^\perp}{\|w_t^\perp\|_2}$ is orthogonal to $\{P_{w_{-1:\tau-1}}^\perp V^\top \frac{u_\tau^\perp}{\|u_\tau^\perp\|_2}\}_{\tau=1}^{t}$ in the row space of $V$ since $P_{u_{1:t}}^\perp$ is projecting to the row space orthogonal to $u_\tau^\perp$ for any $\tau < t$.

Moreover, $V$ is also independent of $w_{-1} = v$ and the initialization $w_0$. See Figure 14 for a more intuitive explanation. Therefore, the conditional distribution of $P^\perp_{u_{1:t}} V w_t^\perp / \|w_t^\perp\|_2$ is the same as that of an $(n-t)$-dimensional Gaussian vectors. Hence, we are able to replace $V$ by an independent copy $V_t$. For the second equality, we can set $z_t = V_t w_t^\perp / \|w_t^\perp\|_2$, which is again a Gaussian vector independent of all the histories. Similarly, let $B_t$ be a realization of $(\widetilde{w}_{-1:0}, z_{-1:t-1}, \widetilde{z}_{1:t-1})$ or

$$\left( w_{-1:0}, \left\{ P^\perp_{u_{1:\tau}} V \frac{w_\tau^\perp}{\|w_\tau^\perp\|_2} \right\}^{t-1}_{\tau=-1}, \left\{ P^\perp_{w_{-1:\tau-1}} V^\top \frac{u_\tau^\perp}{\|u_\tau^\perp\|_2} \right\}^{t-1}_{\tau=1} \right)$$

we similarly have for $P^\perp_{w_{-1:t-1}} V^\top u_t^\perp / \|u_t^\perp\|_2$ that

$$P^\perp_{w_{-1:t-1}} V^\top \frac{u_t^\perp}{\|u_t^\perp\|_2} \left| \left\{ \left( w_{-1:0}, \left\{ P^\perp_{u_{1:\tau}} V \frac{w_\tau^\perp}{\|w_\tau^\perp\|_2} \right\}^{t-1}_{\tau=-1}, \left\{ P^\perp_{w_{-1:\tau-1}} V^\top \frac{u_\tau^\perp}{\|u_\tau^\perp\|_2} \right\}^{t-1}_{\tau=1} \right) = B_t \right\} \right.$$

$$\stackrel{d}{=} P^\perp_{\widetilde{w}_{-1:t-1}} \widetilde{z}_t \mid \left\{ (\widetilde{w}_{-1:0}, z_{-1:t-1}, \widetilde{z}_{1:t-1}) = B_t \right\}. \tag{E.9}$$

To this end, it can be concluded that

1. The initializations $(w_{-1}, w_0)$ and $(\widetilde{w}_{-1}, \widetilde{w}_0)$ are the same.

2. By (E.8) and (E.9), we have the same conditional distributions for the updates of $(P^\perp_{u_{1:t}} V w_t^\perp / \|w_t^\perp\|_2, P^\perp_{w_{-1:t-1}} V^\top u_t^\perp / \|u_t^\perp\|_2)$ and those of $(P^\perp_{\widetilde{u}_{1:t}} z_t, P^\perp_{\widetilde{w}_{-1:t-1}} \widetilde{z}_t)$, which means the conditional distributions of $(y_t, w_t)$ and $(\widetilde{y}_t, \widetilde{w}_t)$ given the past are the same.

3. The updates of $(b_t, u_t)$ and those of $(\widetilde{b}_t, \widetilde{u}_t)$ are also the same.

We hence conclude that the joint distribution for the two iterations are the same for any time $t$. Consequently, we obtain that

$$\left\{ \widetilde{w}_{-1}, \widetilde{w}_0, (\widetilde{y}_\tau, \widetilde{w}_\tau, \widetilde{u}_\tau, \widetilde{b}_\tau)^t_{\tau=1} \right\} \stackrel{d}{=} \left\{ w_{-1}, w_0, (y_\tau, w_\tau, u_\tau, b_\tau)^t_{\tau=1} \right\}.$$

This completes the proof. □

Since the alternative dynamics in Theorem E.2 are distributionally equivalent to the original dynamics, we work exclusively with the alternative formulation below. We emphasize the following key point when running the alternative dynamics for $T$ steps:

> The randomness in the alternative dynamics comes from the initialization $\bar{w}_0$, the feature of interest $v$, and the random vectors $z_{-1:T}$ and $\widetilde{z}_{1:T}$.

Since the system is rotation-invariant, without loss of generality, we fix the direction of the initialization $\bar{w}_0$ in the following analysis, and only consider the randomness over $v$, $z_{-1:T}$, and $\widetilde{z}_{1:T}$.

**Remark.** In fact, the iteration in (E.6) is a reformulation of (E.5) obtained by decomposing the random matrix $V$ into its projections along the row spaces $u_1^\perp, u_2^\perp, \ldots$ and column spaces $w_1^\perp, w_2^\perp, \ldots$, and then replacing the corresponding components by the following rules:

$$P^\perp_{u_{1:t}} z_t \leftarrow P^\perp_{\widetilde{u}_{1:t}} V \frac{\widetilde{w}_t^\perp}{\|\widetilde{w}_t^\perp\|_2},$$

$$P^\perp_{w_{-1:t-1}} \widetilde{z}_t \leftarrow P^\perp_{\widetilde{w}_{-1:t-1}} V^\top \frac{\widetilde{u}_t^\perp}{\|\widetilde{u}_t^\perp\|_2}.$$

For a detailed explanation, we refer interested readers to Theorem E.3 and its following discussions. In essence, the terms on the right-hand side combine to reconstruct the matrix $V$, as illustrated in Figure 14. A crucial property is that these terms are orthogonal in direction; within a Gaussian ensemble, such orthogonality implies their mutual independence. This decoupling of randomness across iterations considerably simplifies the subsequent analysis.

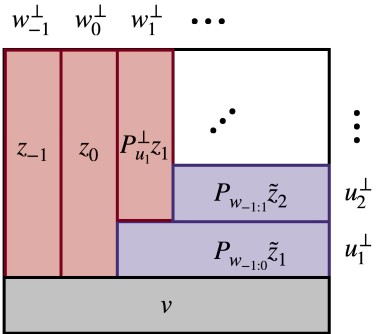

Figure 14: Illustration of the Gaussian conditioning. After removing the feature of interest $v$, the remaining part of $V$ are sliced into $P^\perp_{u_{1:t}} z_t$ and $P^\perp_{w_{-1:t-1}} \widetilde{z}_t$ that are orthogonal to each other.

**Rewriting the initial conditions under the alternative dynamics.** Let us now specify the randomness in equation (E.5) by describing the distributions of the vector $v$ and the matrix $V$. In the absence of any conditioning on the initialization, $v$ and $V$ have i.i.d. standard normal entries. However, the neuron selected for analysis is not arbitrary; it must satisfy the initialization conditions detailed in Theorem E.1. We first restate these conditions in the following more concise form:

$$\langle v, \widetilde{\overline{w}}_0 \rangle \geq (1 - \varepsilon)\sqrt{2\log(M/n)} =: \zeta_0, \quad \widetilde{y}_1 = V\widetilde{\overline{w}}_0 \preceq \sqrt{2}(1 + \varepsilon) \cdot \sqrt{2\log n} \cdot \mathbf{1} =: \zeta_1 \cdot \mathbf{1},$$

where $a \preceq b$ indicates that every element of $a$ is no greater than the corresponding element of $b$. In fact, these two conditions induce a correlation among $v$, $V$, and the initialization $\overline{w}_0$. Under the alternative dynamics in (E.6) and (E.7), we can reformulate these conditions without involving $V$ as follows:

$$\textbf{InitCond-1:} \quad \alpha_{-1,0}\|v\|_2 \geq \zeta_0, \quad \textbf{InitCond-2:} \quad y_1 = \alpha_{-1,0}z_{-1} + \alpha_{0,0}z_0 \preceq \zeta_1 \cdot \mathbf{1}, \quad \text{(E.10)}$$

where

$$\zeta_0 := (1 - \varepsilon)\sqrt{2\log(M/n)}, \quad \zeta_1 := \sqrt{2}(1 + \varepsilon)\sqrt{2\log n}. \tag{E.11}$$

Here, we recall that $\alpha_{-1,0} = \langle v, \overline{w}_0 \rangle / \|v\|_2$ and $\alpha_{0,0} = \langle w_0^\perp, \overline{w}_0 \rangle / \|w_0^\perp\|_2$.

**Decoupling the randomness.** In the following analysis, we can safely decouple the randomness in $v$ and $w_0$ from the randomness in $z_{-1:T}$ and $\widetilde{z}_{1:T}$ by definition of the alternative dynamics. Notably, the second initial condition in (E.10) only couples $z_{-1}$ and $z_0$ if we treat $\alpha_{-1,0}$ and $\alpha_{0,0}$ as deterministic quantities when conditioning on $v$ and $w_0$. In fact, if we condition on $v$ and $w_0$, the second condition can be satisfied with probability at least $1 - n^{-\varepsilon}$ by Theorem E.1.

**Rewriting the alignment recurrence under the alternative dynamics.** Under the reformulation (E.6), the alignment we are interested in is $\alpha_{-1,t} = \langle v, w_t \rangle / (\|v\|_2\|w_t\|_2)$. Note that in the decomposition of $w_t$, only the terms in the direction of $w_{-1}^\perp = w_{-1} = v$ contribute to the inner product $\langle v, w_t \rangle$. Therefore, the alignment can be expressed as

$$\alpha_{-1,t} = \frac{\langle z_{-1}, u_t \rangle + \|v\|_2 \cdot \theta^\top \varphi(Fy_t + \theta \cdot v^\top \overline{w}_{t-1}; b_t) + \eta^{-1}\alpha_{-1,t-1}}{\|w_t\|_2}. \tag{E.12}$$

This formula will be useful in the later proof.

### E.4 ADDITIONAL PROOFS

*Proof of Theorem E.3.* The proof follows from a direct decomposition of the preactivation vector $y_t$ and the unnormalized weight vector $w_t$. By a direct decomposition of $V\overline{w}_t^\perp$, we have

$$V\overline{w}_t^\perp = P_{u_{1:t}}^\perp V\overline{w}_t^\perp + u_t^\perp \cdot \frac{\langle u_t^\perp, V\overline{w}_t^\perp \rangle}{\|u_t^\perp\|_2^2} \cdot \mathbb{1}(t \geq 1) + P_{u_{1:t-1}} V\overline{w}_t^\perp$$

$$\overset{(i)}{=} P_{u_{1:t}}^\perp V\overline{w}_t^\perp + u_t^\perp \cdot \frac{\langle V^\top u_t^\perp, \overline{w}_t^\perp \rangle}{\|u_t^\perp\|_2^2} \cdot \mathbb{1}(t \geq 1)$$

$$\overset{(ii)}{=} P_{u_{1:t}}^\perp V\overline{w}_t^\perp + \frac{u_t^\perp}{\|u_t^\perp\|_2} \cdot \frac{\|w_t^\perp\|_2}{\|u_t^\perp\|_2} \cdot \|\overline{w}_t^\perp\|_2 \cdot \mathbb{1}(t \geq 1).$$

Here, (i) follows from the fact that for any $\tau = 1, \ldots, t - 1$,

$$V^\top u_\tau = w_\tau - v\theta^\top \varphi(Fy_\tau + \theta \cdot v^\top \overline{w}_{\tau-1}; b_\tau) - \eta^{-1}\overline{w}_{\tau-1} \in \text{span}(w_{-1:\tau}),$$

which is orthogonal to $\overline{w}_t^\perp$. In (ii), we use the fact that

$$V^\top u_t^\perp - w_t = V^\top u_t - w_t - V^\top P_{u_{1:t-1}} u_t$$
$$= -v\theta^\top \varphi(Fy_t + \theta \cdot v^\top \overline{w}_{t-1}; b_t) - \eta^{-1}\overline{w}_{t-1} - V^\top P_{u_{1:t-1}} u_t \in \text{span}(w_{-1:t-1}).$$

Therefore, $\langle V^\top u_t^\perp, \bar{w}_t^\perp \rangle = \langle w_t, \bar{w}_t^\perp \rangle = \langle w_t^\perp, \bar{w}_t^\perp \rangle = \|w_t^\perp\|_2 \cdot \|\bar{w}_t^\perp\|_2$.

Using the above result, we derive for the preactivation vector $y_t$ that

$$y_t = V\bar{w}_{t-1} = \sum_{\tau=-1}^{t-1} \frac{\langle \bar{w}_\tau^\perp, \bar{w}_{t-1} \rangle}{\|\bar{w}_\tau^\perp\|_2^2} \cdot V\bar{w}_\tau^\perp$$

$$= \sum_{\tau=-1}^{t-1} \frac{\alpha_{\tau,t-1}}{\|\bar{w}_\tau^\perp\|_2} \cdot \left( P_{u_{1:\tau}}^\perp V\bar{w}_\tau^\perp + \frac{u_\tau^\perp}{\|u_\tau^\perp\|_2} \cdot \frac{\|w_\tau^\perp\|_2}{\|u_\tau^\perp\|_2} \cdot \|\bar{w}_\tau^\perp\|_2 \cdot \mathbb{1}(\tau \geq 1) \right)$$

$$= \sum_{\tau=-1}^{t-1} \alpha_{\tau,t-1} \cdot P_{u_{1:\tau}}^\perp V \frac{w_\tau^\perp}{\|w_\tau^\perp\|} + \sum_{\tau=1}^{t-1} \alpha_{\tau,t-1} \cdot \frac{\|w_\tau^\perp\|_2}{\|u_\tau^\perp\|_2} \cdot \frac{u_\tau^\perp}{\|u_\tau^\perp\|_2}.$$

And also for the unnormalized weight vector $w_t$, we have

$$w_t - v\theta^\top \varphi(Fy_t + \theta \cdot v^\top \bar{w}_{t-1}; b_t) - \eta^{-1}\bar{w}_{t-1}$$

$$= P_{w_{-1:t-1}}^\perp V^\top u_t + \sum_{\tau=-1}^{t-1} \frac{\bar{w}_\tau^\perp}{\|\bar{w}_\tau^\perp\|_2^2} \cdot \langle V\bar{w}_\tau^\perp, u_t \rangle$$

$$= P_{w_{-1:t-1}}^\perp V^\top \frac{u_t^\perp}{\|u_t^\perp\|} \cdot \|u_t^\perp\|_2 + \sum_{\tau=-1}^{t-1} \langle P_{u_{1:\tau}}^\perp V \frac{w_\tau^\perp}{\|w_\tau^\perp\|}, u_t \rangle \cdot \frac{\bar{w}_\tau^\perp}{\|\bar{w}_\tau^\perp\|_2} + \sum_{\tau=1}^{t-1} \frac{\langle u_\tau^\perp, u_t \rangle}{\|u_\tau^\perp\|_2} \cdot \frac{\|w_\tau^\perp\|_2}{\|u_\tau^\perp\|_2} \cdot \frac{\bar{w}_\tau^\perp}{\|\bar{w}_\tau^\perp\|_2}.$$

Therefore, we complete the proof of Theorem E.3. □

*Proof of Theorem E.4.* Recall that

$$u_t = E^\top \varphi(Ey_t; b_t) + F^\top \varphi(Fy_t + \theta \cdot v^\top \bar{w}_{t-1}; b_t).$$

This implies that $u_t$ can be expressed as a function of $y_t$ only. This also holds for $\tilde{u}_t$. For each iteration $(u_t, w_t)$ in (E.5), it holds by the explicit decomposition in Theorem E.3 that

$$u_t \in \sigma\left( w_{-1:t-1}, u_{1:t-1}, \left\{ P_{u_{1:\tau}}^\perp V \frac{w_\tau^\perp}{\|w_\tau^\perp\|_2} \right\}_{\tau=-1}^{t-1} \right),$$

$$w_t \in \sigma\left( w_{-1:t-1}, u_{1:t}, \left\{ P_{u_{1:\tau}}^\perp V \frac{w_\tau^\perp}{\|w_\tau^\perp\|_2} \right\}_{\tau=-1}^{t-1}, P_{w_{-1:t-1}}^\perp V^\top \frac{u_t^\perp}{\|u_t^\perp\|_2} \right), \quad \text{(E.13)}$$

where $\sigma(X)$ denotes the $\sigma$-algebra generated by the random variable $X$. For the Gaussian conditioning iteration $(\tilde{u}_t, \tilde{w}_t)$ in (E.6) and (E.7), it also holds that

$$\tilde{u}_t \in \sigma\left( \tilde{w}_{-1:t-1}, \tilde{u}_{1:t-1}, \{P_{\tilde{u}_{1:\tau}}^\perp z_\tau\}_{\tau=-1}^{t-1} \right), \quad \tilde{w}_t \in \sigma\left( \tilde{w}_{-1:t-1}, \tilde{u}_{1:t}, \{P_{\tilde{u}_{1:\tau}}^\perp z_\tau\}_{\tau=-1}^{t-1}, P_{\tilde{w}_{-1:t-1}}^\perp \tilde{z}_t \right).$$

Notably, for $u_1$ (only depending on $y_1$) we have

$$y_1 = \alpha_{-1,0} \cdot V \frac{w_{-1}}{\|w_{-1}\|_2} = \frac{\langle w_{-1}, \bar{w}_0 \rangle}{\|w_{-1}\|_2} \cdot V \frac{w_{-1}}{\|w_{-1}\|_2} \in \sigma\left( w_{-1:0}, P_{u_{1:-1}}^\perp V \frac{w_{-1}^\perp}{\|w_{-1}^\perp\|_2} \right)$$

by the definition that $P_{u_{1:-1}}^\perp$ is the identity mapping and $w_{-1}^\perp = w_{-1}$. Similarly, $w_1$ is also measurable by

$$w_1 \in \sigma\left( w_{-1:0}, P_{u_{1:-1}}^\perp V \frac{w_{-1}^\perp}{\|w_{-1}^\perp\|_2}, P_{w_{-1:0}}^\perp V^\top \frac{u_1^\perp}{\|u_1^\perp\|_2} \right).$$

This verifies the base case for $t = 1$. Now we can recursively apply the dependency results in (E.13) for $t = 2, 3, \ldots$ and obtain the desired conclusion. This completes the proof of Theorem E.4. □

## F  CONCENTRATIONS RESULTS FOR THE SAE DYNAMICS

**Notation.** In the following proofs, we use the blue color box to highlight the definitions that are used in the proofs for readers' convenience, and use the olive color box to highlight different versions

of the conditions in (B.5) and (B.6) to inform the readers how the conditions evolve throughout the proof. We use $N_1$ to denote the number of rows in matrix $E$ and $N_2$ to denote the number of rows in matrix $F$. In the statement of a lemma, we use $c > 4, C > 0$ to denote some universal constants that may change from line to line. We redefine

$$
\begin{aligned}
\rho_1 &:= \max\left\{ \max_{i \in [n]} \frac{\|H_{:,i}\|_0}{N}, \ \max_{i \neq j} \frac{\sum_{l=1}^{N} \mathbb{1}(H_{l,j} \neq 0)\, \mathbb{1}(H_{l,i} = 0)}{\sum_{l=1}^{N} \mathbb{1}(H_{l,i} = 0)} \right\}, \\
\rho_2 &:= \max_{i \neq j} \frac{\sum_{l=1}^{N} \mathbb{1}(H_{l,i} \neq 0)\, \mathbb{1}(H_{l,j} \neq 0)}{\sum_{l=1}^{N} \mathbb{1}(H_{l,i} \neq 0)}.
\end{aligned}
\tag{F.1}
$$

Compared to the original definition in the main text, we add an additional term in the definition of $\rho_1$. We remark that this is not an issue as

$$
\max_{i \neq j} \frac{\sum_{l=1}^{N} \mathbb{1}(H_{l,j} \neq 0)\, \mathbb{1}(H_{l,i} = 0)}{\sum_{l=1}^{N} \mathbb{1}(H_{l,i} = 0)} \leq \max_{i \neq j} \frac{\|H_{:,j}\|_0}{N - \|H_{:,i}\|_0} \leq \frac{\max_{j \in [n]} \|H_{:,j}\|_0 / N}{1 - \max_{i \in [n]} \|H_{:,i}\|_0 / N}.
$$

Since we assume in the main theorem that $\max_{i \in [n]} \|H_{:,i}\|_0 / N \ll 1$, we have

$$
\max_{i \neq j} \frac{\sum_{l=1}^{N} \mathbb{1}(H_{l,j} \neq 0)\, \mathbb{1}(H_{l,i} = 0)}{\sum_{l=1}^{N} \mathbb{1}(H_{l,i} = 0)} \leq (1 + o(1)) \cdot \max_{i \in [n]} \frac{\|H_{:,i}\|_0}{N}.
$$

The two terms in the definition of $\rho_1$ are only different up to a factor of $1 + o(1)$, and hence we can safely stick to the new definition of $\rho_1$ in the proof. Consequently, $\rho_1 \geq \max_{i \in [n-1]} \|E_{:,i}\|_0 / N_1$, $\rho_2 \geq \max_{i \in [n-1]} \|F_{:,i}\|_0 / N_2$. In addition, $N_1 \geq (1 - \rho_1) N$. By assuming $\rho_1 \leq 1/2$, we have $N_1 \geq N/2$. We use notation $z = x \pm y$ to indicate $z \in [x - y, x + y]$.

**Initialization conditions.** In the following analysis, we focus on a single neuron whose initialization satisfies the conditions in (E.10) for a given feature of interest, $v$. For clarity, we restate the initialization conditions:

**InitCond-1:** $\alpha_{-1,0} \|v\|_2 \geq \zeta_0$, **InitCond-2:** $y_1 = \alpha_{-1,0} z_{-1} + \alpha_{0,0} z_0 \preceq \zeta_1 \cdot \mathbf{1}$,

where

$$
\zeta_0 := (1 - \varepsilon)\sqrt{2 \log(M/n)}, \quad \zeta_1 := \sqrt{2}(1 + \varepsilon)\sqrt{2 \log n}.
$$

Once **InitCond-1** is satisfied for fixed $w_0$ and $v$, it remains to ensure that the Gaussian vectors $z_{-1}$ and $z_0$ satisfy **InitCond-2**. In the subsequent analysis, we sometimes relax **InitCond-2** so as to leverage the standard Gaussian properties of $z_{-1}$ and $z_0$. In fact, if an event $\mathcal{E}$ holds with probabiity $1 - p$ without enforcing **InitCond-2**, then the joint event that both **InitCond-2** and $\mathcal{E}$ hold occurs with probability at least $1 - p - n^{-\varepsilon}$ by a union bound. *For this reason, unless otherwise specified, we*

**Roadmap.** In §F.1, we decompose the pre-activation $y_t$ into two parts: the Gaussian component $y_t^\star$, which aggregates independent Gaussian contributions and captures the nominal dynamics, and the non-Gaussian component $\Delta y_t$, which accounts for deviations induced by cross-iteration coupling that is typically non-Gaussian. Using this decomposition, in §F.2 we demonstrate that only a small fraction of the training examples activate the neuron—a phenomenon we refer to as sparse activation.

## F.1 ISOLATION OF GAUSSIAN COMPONENT

As is discussed in §E.3, the key step in our analysis is to isolate the Gaussian component from the non-Gaussian component. In the following, we decompose $y_t$, which is the alignments between the weight and all features, into the Gaussian component that contains weighted sum of i.i.d. Gaussian vectors, and a non-Gaussian part whose $\ell_2$-norm can be bounded by tracking the evolution of the

dynamics. Recall the definition of $y_t$ in (E.6), we use the fact that $P^\perp_{u_{1:\tau}} z_\tau = z_\tau - P_{u_{1:\tau}} z_\tau$ to decompose $y_t$ as

$$y_t = \sum_{\tau=-1}^{t-1} \alpha_{\tau,t-1} \cdot P^\perp_{u_{1:\tau}} z_\tau + \sum_{\tau=1}^{t-1} \alpha_{\tau,t-1} \cdot \frac{\|w^\perp_\tau\|_2}{\|u^\perp_\tau\|_2} \cdot \frac{u^\perp_\tau}{\|u^\perp_\tau\|_2}$$

$$= \sum_{\tau=-1}^{t-1} \alpha_{\tau,t-1} \cdot z_\tau + \left( \sum_{\tau=1}^{t-1} \alpha_{\tau,t-1} \cdot \frac{\|w^\perp_\tau\|_2}{\|u^\perp_\tau\|_2} \cdot \frac{u^\perp_\tau}{\|u^\perp_\tau\|_2} - \sum_{\tau=1}^{t-1} \alpha_{\tau,t-1} \cdot P_{u_{1:\tau}} z_\tau \right).$$

We can thus define the Gaussian component $y^\star_t$ and the non-Gaussian component $\Delta y_t$ as

$$y^\star_t := \sum_{\tau=-1}^{t-1} \alpha_{\tau,t-1} \cdot z_\tau, \quad \Delta y_t := \sum_{\tau=1}^{t-1} \alpha_{\tau,t-1} \cdot \frac{\|w^\perp_\tau\|_2}{\|u^\perp_\tau\|_2} \cdot \frac{u^\perp_\tau}{\|u^\perp_\tau\|_2} - \sum_{\tau=1}^{t-1} \alpha_{\tau,t-1} \cdot P_{u_{1:\tau}} z_\tau. \quad \text{(F.2)}$$

In the above, the Gaussian component $y^\star_t = \sum_{\tau=-1}^{t-1} \alpha_{\tau,t-1} z_\tau$ is obtained by summing independent Gaussian vectors $z_{-1}, z_0, \ldots, z_{t-1}$ with weights $\alpha_{\tau,t-1}$. Conditional on these coefficients, $y^\star_t$ is simply a standard Gaussian vector independent of the learned directions $w_{1:t-1}$ and $u_{1:t-1}$. In contrast, the non-Gaussian component $\Delta y_t$ quantifies the deviation of the true feature pre-activation $y_t$ from $y^\star_t$ due to cross-iteration coupling.

In the sequel, let us recall the form of $\alpha_{\tau,t-1}$ in (E.12) and define $\beta_{t-1}$ as

$$\alpha_{-1,t} = \frac{\langle z_{-1}, u_t \rangle + \|v\|_2 \cdot \theta^\top \varphi(Fy_t + \theta \cdot v^\top \bar{w}_{t-1}; b_t) + \eta^{-1} \alpha_{-1,t-1}}{\|w_t\|_2},$$

$$\beta_{t-1} := \sqrt{\sum_{\tau=1}^{t-1} \alpha^2_{\tau,t-1}} = \|P^\perp_{w_{-1:0}} \bar{w}_{t-1}\|_2. \quad \text{(F.3)}$$

Here, $\alpha_{-1,t}$ is the alignment between $\bar{w}_t$ and the feature of interest $v = w_{-1}$, and $\beta_t$ is the norm of the projection of $\bar{w}_t$ onto the subspace orthogonal to both $\bar{w}_{-1}$ and $\bar{w}_0$. Tracking $\alpha_{-1,t}$ quantifies how far the neuron has progressed from its initialization $\bar{w}_0$ toward the feature direction $\bar{w}_{-1}$. Ideally, we want $\alpha_{-1,t} \to 1$, indicating strong alignment with the feature while remaining confined to the plane spanned by $\bar{w}_{-1}$ and $\bar{w}_0$. In contrast, $\beta_t$ measures the extent to which the neuron drifts away from that plane due to the influence of irrelevant features. We can build an interesting connection between the non-Gaussian component $\Delta y_t$ and $\beta_{t-1}$ as stated in the following lemma.

**Lemma F.1** (Upper bound the non-Gaussian component $\Delta y_t$). *Suppose $T \leq \sqrt{d}$ and $d \in (n^{1/c_1}, n^{c_1})$ for some universal constant $c_1 > 1$. For all $t = 1, \ldots, T$, it holds with probability at least $1 - n^{-c}$ for some universal constants $c, C > 0$ that*

$$\|\Delta y_t\|^2_2 \leq Cd \cdot \beta^2_{t-1}.$$

*Proof.* See §H.1.1 for a detailed proof. □

## F.2 SPARSE ACTIVATION

Before we move on to studying the evolution of $\alpha_{-1,t}$ and $\beta_t$ defined in (F.3), we first present concentration results for the neuron's activation frequency. To leverage the benefits of sparse activation, we analyze how the scheduled bias $b_t$ induces sparsity in the neuron.

**Concentration for ideal activation.** We will first study the ideal case where $\Delta y_t = 0$, and then move on to the real case in Theorem F.2 where we replace $y^\star_t$ with $y_t$ in Theorem F.3. For more generality, we present a full version in Theorem H.4 and derive Theorem F.2 as a direct corollary. In the following, recall that $e_l$ is the $l$-th row of matrix $E$, which is a submatrix of $H$ defined in (E.3). We study the activation frequency of the neuron on the set of data that does not contain the feature $v$ (i.e., the rows contained in $E$).

**Corollary F.2** (Concentration for ideal activation). *Let $e_l$ be the $l$-th row of matrix $E$. For $\kappa_0$ as the threshold defined in Definition B.3, we denote by $\bar{b}_t = b_t + \kappa_0$. Let $y_t^\star = \sum_{\tau=-1}^{t-1} \alpha_{\tau,t-1} z_\tau$ with $z_\tau$ being the i.i.d. standard Gaussian vectors. It holds for all $t \leq T \leq n^c$, $\alpha_{t-1} = (\alpha_{-1,t-1}, \ldots, \alpha_{t-1,t-1})^\top \in \mathbb{S}^t$, $b_t \in \mathbb{R}$ and any $\delta \in (\exp(-n/4), 1)$ that with probability at least $1 - \delta$ over the randomness of $z_{-1:T}$, the following holds:*

$$\frac{1}{N_1} \sum_{l=1}^{N_1} \mathbb{1}(e_l^\top y_t^\star + \bar{b}_t > 0) \leq C \cdot \left( \Phi(-\bar{b}_t) + \rho_1 s t \log(n) + \rho_1 s \log(\delta^{-1}) \right). \tag{F.4}$$

*Proof.* This is a direct corollary of Theorem H.4. $\qquad\qquad\square$

Here, a neuron is considered active when its ideal pre-activation $e_l^\top y_t^\star + b_t$ exceeds the threshold $-\kappa_0$. In the idealized setting (i.e., as $N_1 \to \infty$, and $y_t^\star \sim \mathcal{N}(0, I_{n-1})$), the expected activation frequency is exactly $\Phi(-\bar{b}_t)$, making the $\Phi(-\bar{b}_t)$ term tight. The additional terms in the bound capture the empirical fluctuations in the activation frequency due to data coupling. In particular, the parameter $\rho_1$ quantifies the maximum fraction of data coupled through a single feature, thereby governing the fluctuation term. A key point to note is that $\alpha_{t-1} \in \mathbb{S}^t$ also depends on the randomness of $z_{-1:T}$, hence how to approximate $y_t^\star$ with random Gaussian vector is not straightforward. In the proof, we decouple the dependence of $y_t^\star$ on $\alpha_{t-1}$ by proving a concentration result for all $\alpha_{t-1}$ that form a covering net of $\mathbb{S}^t$, and then take a union bound over the covering net of size $n^{O(t)}$. This gives rise to the $t \log n$ factor in the bound when taking the logarithm of the covering number.

**Efron-Stein inequality for handling data correlation.** In proving the lemma, we use a refined version of the Efron-Stein inequality (Boucheron et al., 2003) to overcome challenges caused by data correlation. In our setting, two data points may be correlated if they share the same feature, which violates the independence assumption required by classical concentration results such as Bernstein's inequality.

Traditional techniques based on the bounded-differences property—for example, McDiarmid's inequality (McDiarmid et al., 1989)—would treat the left-hand side (LHS) of (F.4) as a function

$$f\left(y_t^\star(1), \ldots, y_t^\star(n-1)\right)$$

of $(n-1)$ variables, where $y_t^\star(i)$ is the $i$-th coordinate of $y_t^\star$. Since altering a single coordinate of $y_t^\star$ has the same effect as modifying the projection of $\bar{w}_t$ onto a single feature, and because each feature influences at most a $\rho_1 N_1$ fraction of the terms in the sum on the LHS, we obtain the bounded-differences property

$$|f(y_t^\star(1), \ldots, y_t^\star(i), \ldots, y_t^\star(n-1)) - f(y_t^\star(1), \ldots, y_t^\star(i)', \ldots, y_t^\star(n-1))| \leq \rho_1.$$

Consequently, McDiarmid's inequality would yield a fluctuation bound of order

$$\sqrt{\sum_{i=1}^{n-1} \rho_1^2} \approx \rho_1 \sqrt{n},$$

which is clearly suboptimal. Unlike McDiarmid's bounded-differences inequality, which requires each individual input change to have a uniformly small impact on $f$, Efron-Stein only demands a weaker bound on the variance incurred by altering one coordinate. We defer interested readers to §H.2.1 for a detailed proof.

**Concentration for original activation.** To fully characterize the behavior of the activation, we also need to take into account the non-Gaussian component $\Delta y$. This gives rise to the following lemma.

**Lemma F.3** (Activation with non-Gaussian component). *Following the setup of Theorem F.2, suppose $\bar{b}_t < -2$. Then for all $t \leq T \leq n^c$, $\alpha_{t-1} \in \mathbb{S}^t$ and $b_t \in \mathbb{R}$, it holds with probability at least $1 - n^{-c}$ over the randomness of $z_{-1:T}$ that*

$$\frac{1}{N_1} \sum_{l=1}^{N_1} \mathbb{1}(e_l^\top y_t + \bar{b}_t > 0) \leq C \cdot \left( \Phi(-\bar{b}_t) + \rho_1 s t \log(n) + \rho_1 |\bar{b}_t|^2 \|\Delta y_t\|_2^2 \right).$$

*Proof.* See §H.2.2 for a detailed proof. □

The fluctuation term in the upper bound now depends on both $\rho_1$ and the $\ell_2$ norm of the non-Gaussian $\Delta y_t$. This is because a larger $\|\Delta y_t\|_2$ can shift the pre-activations further away from the ideal Gaussian case, thereby in the worst case, increasing the activation frequency.

**Concentration for $\alpha_{-1,t}$ and $\beta_t$.** We next aim to characterize the evolution of the parameters $\alpha_{-1,t}$ and $\beta_t$ defined in (F.3). Note that in the formula of $\alpha_{-1,t}$

$$\alpha_{-1,t-1} = \frac{\langle z_{-1}, u_t \rangle + \|v\|_2 \cdot \theta^\top \varphi(Fy_t + \theta \cdot v^\top \bar{w}_{t-1}; b_t) + \eta^{-1}\alpha_{-1,t-1}}{\|w_t\|_2},$$

we can decompose the first term in the numerator as follows:

$$\langle z_{-1}, u_t \rangle = \langle z_{-1}, E^\top \varphi(Ey_t; b_t) \rangle + \langle z_{-1}, F^\top \varphi(Fy_t + \theta \cdot v^\top \bar{w}_{t-1}; b_t) \rangle$$

according to the defintion of $u_t$ in (E.5). Here, $E$ and $F$ are the submatrices of $H$ defined in (E.3), where $E$ corresponds to the rows not containing the feature of interest $v$, and $F$ corresponds to the rows containing $v$. To this end, we just need to control

$$\langle z_\tau, E^\top \varphi(Ey_t; b_t) \rangle, \quad \langle z_\tau, F^\top \varphi(Fy_t + \theta \cdot v^\top \bar{w}_{t-1}; b_t) \rangle, \tag{F.5}$$

for general $\tau \in [-1:T]$ and then specialize to $\tau = -1$. Note that the above two terms for general $\tau$ will also be used in computing the norm of $\|w_t\|_2$ later. Let us just consider a simplfied case where $z_\tau$ is independent of $y_t$ (which does not hold in general). To control the fluctuation of the above terms, it is important to compute the second-order moments with respect to the randomness of $z_\tau$. As a concrete example, for the first term, we have the second-order moment computed as

$$\mathbb{E}_{z_\tau \sim \mathcal{N}(0, I_{n-1})}\big[\langle z_\tau, E^\top \varphi(Ey_t; b_t) \rangle^2\big] = \|E^\top \varphi(Ey_t; b_t)\|_2^2.$$

The second-order moment of the second term can be computed similarly. Therefore, as a first step, we will focus on the follwoing two terms:

$$\|E^\top \varphi(Ey_t; b_t)\|_2^2, \quad \|F^\top \varphi(Fy_t + \theta \cdot v^\top \bar{w}_{t-1}; b_t)\|_2^2. \tag{F.6}$$

In §F.3, we will first present concentration results for the second-order terms in (F.6) and then use them to derive the concentration results for the two first-order terms in (F.5). In addition, we will also derive the concentration result for the term $\theta^\top \varphi(Fy_t + \theta \cdot v^\top \bar{w}_{t-1}; b_t)$ as in the numerator of $\alpha_{-1,t-1}$.

### F.3 SECOND ORDER CONCENTRATION

In this subsection, we present concentration results for the second-order terms with respect to the Gaussian component $y_t^\star$ defined in (F.2):

$$\|E^\top \varphi(Ey_t^\star; b_t)\|_2^2 \quad \text{and} \quad \|F^\top \varphi(Fy_t^\star + \theta \cdot v^\top \bar{w}_{t-1}; b_t)\|_2^2. \tag{F.7}$$

We will bridge the gap between these two terms and the original terms in (F.5) by using the analysis of the non-Gaussian component $\Delta y_t$ in §F.5. For now, let us focus on the two terms in (F.7). We now present our concentration result formally in the following lemma.

**Lemma F.4** (Second-order concentration for $E$-related term)**.** *Under Definition B.3, let $\bar{b}_t = b_t + \kappa_0 < 0$, and assume further that $-\bar{b}_t = \Theta\big(\sqrt{\log n}\big)$ and $-\bar{b}_t < \zeta_1$, with $\zeta_1$ defined in (E.11) as required by* **InitCond-2**. *Suppose $\rho_1 < 1 - 1/C_1$ for some universal constant $C_1 > 0$. Then with probability at least $1 - n^{-c}$ over the randomness of standard Gaussian vectors $z_{-1:T}$, it holds for all $t \leq T$ with $T \leq n^c$ that*

$$\frac{1}{N_1^2}\|E^\top \varphi(Ey_t^\star; b_t)\|_2^2 \cdot \mathbb{1}(\mathcal{E}_0) \leq CL^2 \cdot \rho_1^2 st^2 (\log n)^2 \cdot \mathcal{K}_t^2$$

$$+ CL^2 \cdot \Phi(|\bar{b}_t|) \cdot \widehat{\mathbb{E}}_{l,l'} \left[ \Phi\Big(|\bar{b}_t|\sqrt{\frac{1 - \langle h_l, h_{l'} \rangle}{1 + \langle h_l, h_{l'} \rangle}}\Big) \langle h_l, h_{l'} \rangle \right]. \tag{F.8}$$

*where $\widehat{\mathbb{E}}_{l,l'}$ denotes the empirical average over $l, l' \in [N]$, $h_l$ denotes the $l$-th row of $H$, $L = \gamma_2 + |b_t|\gamma_1$, and $\mathcal{E}_0$ is the event such that $z_{-1}$ and $z_0$ satisfy **InitCond-2**. Here we define $\mathcal{K}_t$ as*

$$
\mathcal{K}_t := \left( n\,|\bar{b}_t|\,\Phi\left( \frac{-\bar{b}_t}{\sqrt{\frac{3}{4}\hbar_{4,\star}^2 + \frac{1}{4}}} \right) \right)^{1/4} + \left( \rho_2 s n |\bar{b}_t| \Phi\left( \frac{-\bar{b}_t}{\sqrt{\frac{2}{3}\hbar_{3,\star}^2 + \frac{1}{3}}} \right) \right)^{1/4}
$$
$$
+ \left( \Phi\left( -\frac{\bar{b}_t + \hbar_{4,t}\zeta_t}{\sqrt{1 - \hbar_{4,t}^2}} \right) + (\rho_2 s)^{1/4} \right) \cdot \left( t \log(n) \right)^{1/4} + n^{1/4} \rho_2 \, s \, t \log(n),
$$

(F.9)

*In the above definition, we let $\hbar_{q,\star}$ and $\hbar_{q,t}$ for any positive $q > 1$ and time $t \geq 1$ be the smallest real values in $[0,1]$ such that the following inequalities hold:*

$$
\max_{j \in [n]} \frac{1}{|\mathcal{D}_j|} \sum_{l \in \mathcal{D}_j} \Phi\left( \frac{-\bar{b}_t}{\sqrt{\frac{q-1}{q}H_{l,j}^2 + \frac{1}{q}}} \right) \leq \Phi\left( \frac{-\bar{b}_t}{\sqrt{\frac{q-1}{q}\hbar_{q,\star}^2 + \frac{1}{q}}} \right),
$$

(F.10)

$$
\max_{j \in [n]} \frac{1}{|\mathcal{D}_j|} \sum_{l \in \mathcal{D}_j} \Phi\left( -\frac{\bar{b}_t + H_{l,j}\zeta_t}{\sqrt{1 - H_{l,j}^2}} \right)^q \leq \Phi\left( -\frac{\bar{b}_t + \hbar_{q,t}\zeta_t}{\sqrt{1 - \hbar_{q,t}^2}} \right)^q.
$$

(F.11)

*Here $\mathcal{D}_j = \{ l \in [N] : h_{l,j} \neq 0 \}$ is the set of row indices in matrix $H$ that has non-zero entries in the $j$-th column, and $\zeta_t = \zeta_1 + \mathbb{1}(t \geq 2) \cdot C(\beta_{t-1} + |\alpha_{-1,t-1}| + |\alpha_{-1,0}|)\sqrt{t \log(nt)}$ with the value $\zeta_1$ in **InitCond-2** and $\beta_{t-1} = \sqrt{\sum_{\tau=1}^{t-1} \alpha_{\tau,t-1}^2}$.*

*Proof.* See §H.2.3 for a detailed proof. $\qquad\square$

**Validity of the definition of $\hbar_{q,t}$ and $\hbar_{q,\star}$.** The definitions of $\hbar_{q,\star}$ and $\hbar_{q,1}$ are valid as the right-hand sides (RHSs) of the above two inequalities are strictly increasing in terms of $\hbar_{q,\star}$ and $\hbar_{q,1}$, respectively, under the condition $-\bar{b}_t < \zeta_1$.

- To see this for $\hbar_{q,\star}$, we note that $\Phi(\cdot)$ is a strictly decreasing function, while $\frac{-\bar{b}_t}{\sqrt{\frac{q-1}{q}H_{l,j}^2 + \frac{1}{q}}}$ is also strictly decreasing in terms of $H_{l,j}$. Therefore, the composition of the two functions is strictly increasing in terms of $\hbar_{q,\star}$.

- To see this for $\hbar_{q,t}$, observe that $\zeta_t \geq \zeta_1 > -c_1 = -\bar{b}_t$, since the bias is fixed at $b_t = b$ in the current algorithm. Moreover, the derivative of the right-hand side of the inequality in (F.11) with respect to $\hbar_{q,t}$ is

$$
\frac{\mathrm{d}}{\mathrm{d}x} \Phi\left( -\frac{\bar{b}_t + x\zeta_1}{\sqrt{1 - x^2}} \right)^q = q\Phi\left( -\frac{\bar{b}_t + x\zeta_1}{\sqrt{1 - x^2}} \right)^{q-1} \cdot p\left( -\frac{\bar{b}_t + x\zeta_1}{\sqrt{1 - x^2}} \right) \cdot \frac{\zeta_1 - (-\bar{b}_t)x}{(1 - x^2)^{3/2}} > 0. \quad \text{(F.12)}
$$

Therefore, the definitions of $\hbar_{q,\star}$ and $\hbar_{q,t}$ as the smallest real values satisfying the inequalities in (F.10) and (F.11) are valid.

**Heuristic derivation for $\| E^\top \varphi(Ey_t^\star; b_t) \|_2^2$.** The first term involves the submatrix $E$. Before we present the concentration result, let us derive heuristically what the concentration result should look like. Let us denote by $e_l$ the $l$-th row of matrix $E$. We can compute the expectation of the squared norm as

$$
\frac{1}{N_1^2} \cdot \mathbb{E}\left[ \| E^\top \varphi(Ey_t^\star; b_t) \|_2^2 \right] = \frac{1}{N_1^2} \sum_{l,l'=1}^{N_1} \mathbb{E}\left[ |\varphi(e_l^\top y_t^\star; b_t) \cdot \varphi(e_{l'}^\top y_t^\star; b_t)| \right] \cdot \langle e_l, e_{l'} \rangle.
$$

If we assume $\alpha_{:,t-1}$ are fixed, then $y_t^\star$ is just a standard Gaussian vector, and

$$
(e_l^\top y_t^\star, e_{l'}^\top y_t^\star) \sim \mathcal{N}\left( \begin{bmatrix} 0 \\ 0 \end{bmatrix}, \begin{bmatrix} 1 & \langle e_l, e_{l'} \rangle \\ \langle e_l, e_{l'} \rangle & 1 \end{bmatrix} \right).
$$

This fact enables a direct upper bound on the expectation, as detailed in Theorem F.5.

**Lemma F.5.** *Let $\bar{b} = b + \kappa_0 < 0$. Suppose $|\varphi(x; b)| \leq (n \vee d)^{-c_0} + L(x + \bar{b}) \cdot \mathbb{1}(x > -\bar{b})$ for some $L > 0$ and $c_0 > 0$ under Definition B.3. For two independent $x, z \sim \mathcal{N}(0, 1)$ and $\iota \in (0, 1)$, it holds that*

$$\mathbb{E}[\varphi(x; b)\varphi(\iota x + \sqrt{1 - \iota^2} \cdot z; b)] \leq CL(n \vee d)^{-c_0} + C(L^2 + 1) \cdot \Phi(|\bar{b}|) \cdot \Phi\Big(|\bar{b}|\sqrt{\frac{1 - \iota}{1 + \iota}}\Big).$$

*Proof.* See §H.4.1 for a detailed proof. □

By relaxing the rows $e_l, e_{l'}$ of $E$ to the corresponding rows $h_l, h_{l'}$ of $H$, we derive the second term in the concentration result (F.8). The first fluctuation term is obtained again via the Efron-Stein inequality, which needs a careful analysis up to the 4-th moment. In particular, we also apply a uniform bound over the sphere $\mathbb{S}^t$ for $\alpha_{t-1}$, which gives rise to the dependency on $t$ in the definition of $\mathcal{K}_t$ in (F.9).

We now turn to the second term in (F.7), which is $\|F^\top \varphi(Fy_t^\star + \theta \cdot v^\top \bar{w}_{t-1}; b_t)\|_2^2$.

**Lemma F.6** (Second-order concentration for $F$-related term). *Under Definition B.3, suppose $b_t \leq -\kappa_0$ and let $L = \gamma_2 + |b_t|\gamma_1$. For all $t \leq T \leq n^c$, it holds with probability at least $1 - n^{-c}$ over the randomness of standard Gaussian vectors $z_{-1:T}$ that*

$$\frac{1}{N_2^2}\|F^\top \varphi(Fy_t^\star + \theta \cdot v^\top \bar{w}_{t-1}; b_t)\|_2^2 \leq CL^2\rho_2 \cdot \big(\overline{\theta^2}\|v\|_2^2\alpha_{-1,t-1}^2 + \rho_2 n + \rho_2 t \log n\big),$$

*where $\overline{\theta^2} = \|\theta\|_2^2/N_2$.*

*Proof.* See §H.2.4 for a detailed proof. □

### F.4 FIRST ORDER CONCENTRATION

In this subsection, we continue to present the concentration results on the first order terms specified in (F.5). Let's first consider the concentration for $\langle z_\tau, E^\top \varphi(Ey_t^\star; b_t)\rangle$.

**Heuristic derivation for $\langle z_\tau, E^\top \varphi(Ey_t^\star; b_t)\rangle$.** Let us recall that $y_t^\star = \sum_{\tau=-1}^{t-1} \alpha_{\tau,t-1}z_\tau$, and we can rewrite the term as

$$\langle z_\tau, E^\top \varphi(Ey_t^\star; b_t)\rangle = \sum_{l=1}^{N_1} e_l^\top z_\tau \cdot \varphi(e_l^\top y_t^\star; b_t)$$

for $e_l$ being the $l$-th row of matrix $E$. Moreover, we have for any fixed $\alpha_{t-1} = (\alpha_{-1,t-1}, \ldots, \alpha_{t-1,t-1})^\top \in \mathbb{S}^t$ and by the fact that $\|e_l\|_2 = 1$ for all $l \in [N_1]$, we have

$$(e_l^\top z_\tau, e_l^\top y_t^\star) \sim \mathcal{N}\left(\begin{bmatrix} 0 \\ 0 \end{bmatrix}, \begin{bmatrix} 1 & \alpha_{\tau,t-1} \\ \alpha_{\tau,t-1} & 1 \end{bmatrix}\right) \tag{F.13}$$

where $j \in [n-1]$ is the entry index of the vectors. Hence, the term we are interested in should be close to

$$\sum_{l=1}^{N_1} \mathbb{E}_{\zeta,\xi \overset{\text{i.i.d.}}{\sim} \mathcal{N}(0,1)}\big[\big(\alpha_{\tau,t-1}\zeta + \sqrt{1 - \alpha_{\tau,t-1}^2} \cdot \xi\big) \cdot \varphi(\zeta; b_t)\big] = N_1 \cdot \alpha_{\tau,t-1} \cdot \widehat{\varphi}_1(b_t),$$

where we define

$$\widehat{\varphi}_1(b) = \mathbb{E}_{u\sim\mathcal{N}(0,1)}[\varphi(u; b)u].$$

Building on this intuition, the following lemma provides the concentration result in more detail.

**Lemma F.7** (First-order concentration for $E$-related term). *Under the condition of Theorem F.4, let $L = \gamma_2 + |b_t|\gamma_1$. For all $t \le T \le n^c$, it holds with probability at least $1 - n^{-c}$ over the randomness of standard Gaussian vectors $z_{-1:T}$ that*

$$\left| \frac{1}{N_1} \langle z_\tau, E^\top \varphi(Ey_t^\star; b_t) \rangle - \alpha_{\tau,t-1} \cdot \widehat{\varphi}_1(b_t) \right|$$

$$\le CL\alpha_{\tau,t-1} t \log(n) \cdot \left( \sqrt{s\rho_1 \Phi(|\bar{b}_t|) t \log(n)} + s\rho_1 t \log(n) \right)$$

$$+ \frac{C}{N_1} \sqrt{1 - \alpha_{\tau,t-1}^2} \cdot \sqrt{\|E^\top \varphi(Ey_t^\star; b_t)\|_2^2 \cdot t \log(n)}.$$

*Proof.* See §H.2.5 for a detailed proof. □

In the above lemma, we bound the deviation of the first-order term $\langle z_\tau, E^\top \varphi(Ey_t^\star; b_t) \rangle$ from its expectation $\alpha_{\tau,t-1} \cdot \widehat{\varphi}_1(b_t)$ by some $\rho_1$ and $\Phi(|\bar{b}_t|)$-dependent fluctuation terms. The dependence on $\Phi(|\bar{b}_t|)$ is consistent with the intuition that sparser activation which avoids unnecessary activations on other features except the one of interest, often leads to less fluctuation. The following lemma provides upper and lower bound for $\widehat{\varphi}_1(b_t)$.

**Lemma F.8** (Upper and lower bounds for $\widehat{\varphi}_1(b_t)$). *Suppose Definition B.3 holds and let $\bar{b}_t = b_t + \kappa_0 < 0$, $L = \gamma_2 + |b_t|\gamma_1$. If $|\bar{b}_t| = \omega(1)$, and $\kappa_0|\bar{b}_t| = O(1)$, then*

$$\frac{C_0}{4} \cdot \Phi(|\bar{b}_t|) \le \widehat{\varphi}_1(b_t) \le 2 \cdot C_0 L \Phi(|\bar{b}_t|).$$

*Proof.* See §H.4.2 for a detailed proof. □

The message from Theorem F.8 is quite straightforward: the expectation term $\widehat{\varphi}_1(b_t)$ is on the same order as the activation sparsity level $\Phi(|\bar{b}_t|)$.

**Heuristic derivation for** $\langle z_\tau, F^\top \varphi(Fy_t^\star + \theta \cdot v^\top \bar{w}_{t-1}; b_t) \rangle$**.** Similar to the previous case, we still use the approximation in (F.13) except that this time each row $f_l$ of $F$ has norm $\sqrt{1 - \theta_l^2}$, and have

$$(f_l^\top z_\tau, f_l^\top y_t^\star) \sim \mathcal{N}\left( \begin{bmatrix} 0 \\ 0 \end{bmatrix}, (1 - \theta_l^2) \cdot \begin{bmatrix} 1 & \alpha_{\tau,t-1} \\ \alpha_{\tau,t-1} & 1 \end{bmatrix} \right).$$

This leads to the following approximation:

$$\langle z_\tau, F^\top \varphi(Fy_t^\star + \theta \cdot v^\top \bar{w}_{t-1}; b_t) \rangle \approx \sum_{l=1}^{N_2} \alpha_{\tau,t-1} \sqrt{1 - \theta_l^2} \cdot \mathbb{E}_{x \sim \mathcal{N}(0,1)} \left[ x\varphi\left( \sqrt{1 - \theta_l^2} x + \theta_l v^\top \bar{w}_{t-1}; b_t \right) \right].$$

We now present the formal concentration result for $\langle z_\tau, F^\top \varphi(Fy_t^\star + \theta \cdot v^\top \bar{w}_{t-1}; b_t) \rangle$ in the following lemma.

**Lemma F.9** (First-order concentration for $F$-related term). *Under Definition B.3, suppose $\bar{b}_t = b_t + \kappa_0 \le 0$ and let $L = \gamma_2 + |b_t|\gamma_1$. For all $\tau < t \le T$ with $T \le n^c$, it holds with probability at least $1 - n^{-c}$ over the randomness of standard Gaussian vectors $z_{-1:T}$ that*

$$\frac{1}{N_2} \left| \langle z_\tau, F^\top \varphi(Fy_t^\star + \theta \cdot v^\top \bar{w}_{t-1}; b_t) \rangle - \sum_{l=1}^{N_2} \alpha_{\tau,t-1} \sqrt{1 - \theta_l^2} \cdot \mathbb{E}_{x \sim \mathcal{N}(0,1)} \left[ x\varphi\left( \sqrt{1 - \theta_l^2} x + \theta_l v^\top \bar{w}_{t-1}; b_t \right) \right] \right|$$

$$\le CL\alpha_{\tau,t-1} \cdot \left( \sqrt{t \log(n)} + \|v\|_2 \alpha_{-1,t-1} \right) \cdot \sqrt{\rho_2 s} \cdot (t \log(n))^{3/2}$$

$$+ \frac{C}{N_2} \sqrt{1 - \alpha_{\tau,t-1}^2} \cdot \sqrt{\|F^\top \varphi(Fy + \theta \cdot v^\top \bar{w}_{t-1}; b_t)\|_2^2 \cdot t \log(n)}.$$

*Proof.* See §H.2.6 for a detailed proof. □

**Heuristic derivation for** $\theta^\top \varphi(Fy_t + \theta \cdot v^\top \bar{w}_{t-1}; b_t)$. The last term we need to control is $\theta^\top \varphi(Fy_t^\star + \theta \cdot v^\top \bar{w}_{t-1}; b_t)$. Using the Gaussian approximation $f_l^\top y_t^\star \sim \mathcal{N}(0, 1 - \theta_l^2)$ as in the previous case, we have

$$\theta^\top \varphi(Fy_t^\star + \theta \cdot v^\top \bar{w}_{t-1}; b_t) \approx \sum_{l=1}^{N_2} \theta_l \cdot \mathbb{E}_{x \sim \mathcal{N}(0,1)}\left[\varphi(\sqrt{1-\theta_l^2}x + \theta_l v^\top \bar{w}_{t-1}; b_t)\right].$$

For our convenience, let us define

$$\psi_t := \frac{\sqrt{d}}{N} \sum_{l=1}^{N_2} \mathbb{E}_{x \sim \mathcal{N}(0,1)}\left[\theta_l \cdot \varphi(\sqrt{1-\theta_l^2} \cdot x + \theta_l \cdot v^\top \bar{w}_{t-1}; b_t)\right], \tag{F.14}$$

and it follows that $\theta^\top \varphi(Fy_t^\star + \theta \cdot v^\top \bar{w}_{t-1}; b_t) \approx N \cdot \psi_t / \sqrt{d}$. Lastly, we present the concentration for $\theta^\top \varphi(Fy_t^\star + \theta \cdot v^\top \bar{w}_{t-1}; b_t)$.

**Lemma F.10** (First-order concentration for signal term). *Under Definition B.3, suppose $\bar{b}_t = b_t + \kappa_0 \leq 0$ and let $L = \gamma_2 + |b_t|\gamma_1$. For all $t \leq T \leq n^c$, it holds with probability at least $1 - n^{-c}$ over the randomness of standard Gaussian vectors $z_{-1:T}$ that*

$$\left| \frac{1}{N_2} \theta^\top \varphi(Fy_t^\star + \theta \cdot v^\top \bar{w}_{t-1}; b_t) - \frac{\psi_t N}{\sqrt{d} N_2} \right| \leq CL\left(\sqrt{t \log(n)} + \|v\|_2 \alpha_{-1, t-1}\right) \cdot \sqrt{\rho_2 s \overline{\theta^2}} \cdot t \log(n).$$

*Proof.* See §H.2.7 for a detailed proof. □

Lastly, we provide a useful bound for the term $\psi_t$ defined in (F.14) in the following lemma, which is related to the *strength* of the weight vector $\theta$ for the feature of interest. To quantify the strength, we make the following definition

$$Q_t := \frac{1}{N_2} \sum_{l=1}^{N_2} \mathbb{1}\left(\theta_l > \frac{-b_t}{\sqrt{d}\alpha_{-1,t-1}}\right), \quad \overline{\theta^2} := \frac{\|\theta\|_2^2}{N_2}. \tag{F.15}$$

**Lemma F.11** (Bounds for the signal term). *Under Definition B.3, it holds for $\psi_t$ defined in Theorem F.10 that*

$$C^{-1}\overline{\theta^2}Q_t \cdot N_2 d\alpha_{-1,t-1} \leq N\psi_t \leq CL\overline{\theta^2} \cdot N_2 d\alpha_{-1,t-1}.$$

*Proof.* See §H.4.3 for a detailed proof. □

### F.5 NON-GAUSSIAN ERROR PROPOGATION

In the following, let us define the following error terms

$$\Delta E_t = E^\top \varphi(Ey_t; b_t) - E^\top \varphi(Ey_t^\star; b_t),$$
$$\Delta F_t = F^\top \varphi(Fy_t + \theta \cdot v^\top \bar{w}_{t-1}; b_t) - F^\top \varphi(Fy_t^\star + \theta \cdot v^\top \bar{w}_{t-1}; b_t),$$
$$\Delta \varphi_{F,t} = \varphi(Fy_t + \theta \cdot v^\top \bar{w}_{t-1}; b_t) - \varphi(Fy_t^\star + \theta \cdot v^\top \bar{w}_{t-1}; b_t).$$

The last piece of the puzzle is to control the error propagation in the dynamics due to the non-Gaussian component $\Delta y_t$ in the pre-activation. Let us recall the error terms

$$\Delta E_t = E^\top \varphi(Ey_t; b_t) - E^\top \varphi(Ey_t^\star; b_t)$$
$$\Delta F_t = F^\top \varphi(Fy_t + \theta \cdot v^\top \bar{w}_{t-1}; b_t) - F^\top \varphi(Fy_t^\star + \theta \cdot v^\top \bar{w}_{t-1}; b_t).$$

We are interested in how the error $\Delta y_t$ propagates through the nonlinear function $\varphi$ in the update.

**Lemma F.12** (Error propogation for $\Delta E_t$). *Under* Definition B.3 *on the activation function, let* $\overline{b}_t = b_t + \kappa_0$, $L = \gamma_2 + |b_t|\gamma_1$ *and suppose* $\overline{b}_t < -2$. *For all* $t \leq T \leq n^c$, *it holds with probability at least* $1 - n^{-c}$ *over the randomness of standard Gaussian vectors* $z_{-1:T}$ *that*

$$\|\Delta E_t\|_1 \leq CLN_1 \cdot \left( \left( \sqrt{s\rho_1 \Phi(-\overline{b}_t)} + s\rho_1 \sqrt{t \log n} \right) \cdot \|\Delta y_t\|_2 + \sqrt{s}\rho_1 |\overline{b}_t| \cdot \|\Delta y_t\|_2^2 \right)$$
$$+ CN_1 \sqrt{s}(2 + |b_t|) \cdot (n \vee d)^{-c_0},$$

*and the* $\ell_2$ *norm of* $\Delta E_t$ *are bounded as* $\|\Delta E_t\|_2 \leq (\gamma_2 + |b_t|\gamma_1) \cdot \rho_1 N_1 \|\Delta y_t\|_2$.

*Proof.* See §H.3.1 for a detailed proof. $\qquad\qquad\square$

In the above lemma, we incorporate the sparsity in the activation to obtain a more refined bound for $\|\Delta E_t\|_1$. Next, we also present the error bound for $\Delta F_t$.

**Lemma F.13** (Error propogation for $\Delta F_t$). *Define* $\Delta \varphi_{F,t} = \varphi(Fy_t + \theta \cdot v^\top \overline{w}_{t-1}; b_t) - \varphi(Fy_t^\star + \theta \cdot v^\top \overline{w}_{t-1}; b_t)$. *The following bounds hold:*

1. $\|\Delta F_t\|_1 \leq \sqrt{s}N_2 L \cdot \|\Delta y_t\|_2$.

2. $\|\Delta F_t\|_2 \leq \rho_2 N_2 L \cdot \|\Delta y_t\|_2$.

3. $\|\Delta \varphi_{F,t}\|_2 \leq \sqrt{\rho_2 N_2} L \cdot \|\Delta y_t\|_2$.

*Proof.* See §H.3.2 for a detailed proof. $\qquad\qquad\square$

# G   SAE DYNAMICS ANALYSIS: PROOF OF THEOREM B.2

In the sequel, we will first state a more general version of Theorem B.2, accompanied by the full details on the related definitions and assumptions that are mentioned in the main text. Then we will present the proof of the theorem.

## G.1   A GENERAL VERSION OF THE THEOREM

In the follwoing, we first state the definition of the *concentration coefficient* $h_\star$ and a general version of the main theorem. Then, we present the rigorous definition of the ReLU-like activation function.

**Details on concentration parameters** $h_\star$. To measure the magnitude of coefficients associated with each feature, we recall in the definition of the *cut-off* level for feature $i$ in (B.3) as

$$h_i := \max\left\{ h \leq 1 : \frac{1}{|\mathcal{D}_i|} \sum_{l \in \mathcal{D}_i} \mathbb{1}\{H_{l,i} \geq h\} \geq \text{polylog}(n)^{-1} \right\}.$$

To measure the concentration level of the global coefficients across all features, we define the *concentration coefficient* $h_\star$ as follows. We first recall the definitions of $\hbar_{q,\star}$ and $\hbar_{q,t}$ from Theorem F.4 (with $t = 1$ for any $q > 1$). In particular, $\hbar_{q,\star}$ and $\hbar_{q,1}$ are defined as the *smallest* numbers satisfying the following inequalities:

$$\max_{j \in [n]} \frac{1}{|\mathcal{D}_j|} \sum_{l \in \mathcal{D}_j} \Phi\left( \frac{-\overline{b}_t}{\sqrt{\frac{q-1}{q}H_{l,j}^2 + \frac{1}{q}}} \right) \leq \Phi\left( \frac{-\overline{b}_t}{\sqrt{\frac{q-1}{q}\hbar_{q,\star}^2 + \frac{1}{q}}} \right),$$

$$\max_{j \in [n]} \frac{1}{|\mathcal{D}_j|} \sum_{l \in \mathcal{D}_j} \Phi\left( \frac{-\overline{b}_t + H_{l,j}\zeta_1}{\sqrt{1 - H_{l,j}^2}} \right)^q \leq \Phi\left( \frac{-\overline{b}_t + \hbar_{q,1}\zeta_1}{\sqrt{1 - \hbar_{q,1}^2}} \right)^q.$$

Here, $\mathcal{D}_j = \{l \in [N] : H_{l,j} \neq 0\}$ is the set of row indices in matrix $H$ that has non-zero entries in the $j$-th column, and $\zeta_1 = 2(1 + \varepsilon)\sqrt{\log n}$ that is formally defined in (E.11). Here, $\Phi(\cdot)$ is the tail probability function of the standard Gaussian distribution, i.e., $\Phi(x) = \int_x^\infty e^{-u^2/2}/\sqrt{2\pi} \cdot du$. The definitions of $\hbar_{q,\star}$ and $\hbar_{q,1}$ are valid as the right-hand sides (RHSs) of the above two inequalities are strictly increasing in terms of $\hbar_{q,\star}$ and $\hbar_{q,1}$, respectively. We defer readers to the discussion under

**Theorem F.4.** We define the *concentration coefficient* for the weight matrix $H$, denoted by $h_\star$, as the *smallest* number such that

$$\max\{\hbar_{4,\star}^2, \hbar_{3,\star}^2, \hbar_{4,1}^2\} \le h_\star, \quad \sum_{j=1}^{n} \frac{1}{|\mathcal{D}_j|^2} \sum_{l,l' \in \mathcal{D}_j} \Phi\left(|\bar{b}|\sqrt{\frac{1 - H_{l,j} H_{l',j}}{1 + H_{l,j} H_{l',j}}}\right) \le n\Phi\left(|\bar{b}|\sqrt{\frac{1 - h_\star^2}{1 + h_\star^2}}\right), \tag{G.1}$$

In fact, the RHS of the last inequality in (G.1) is also strictly increasing in terms of $h_\star$, and hence the definition is valid. In the extreme case where $H$ does not have any diversity in its nonzero entries, we have the following simple relationship between $s_\star, s_i$ and $s$:

**Proposition G.1** (Concentrated coefficient $H$). *If $H_{lj} \in \{0, 1/\sqrt{s}\}$ for all $l \in [N]$ and $j \in [n]$, then $h_\star = h_i = 1/\sqrt{s}$.*

In this extreme case, every row of $H$ has exactly $s$ non-zero entries, and the non-zero entries are all equal to $1/\sqrt{s}$. In the following, let us define $\overline{\theta_i^2} = \|\theta_i\|_2^2/N_2$ and $\widehat{\mathbb{Q}}_i(x) = |\mathcal{D}_i|^{-1} \sum_{l \in \mathcal{D}_i} \mathbb{1}(H_{l,i} \ge x)$ for $x \in [0, 1]$. The following proposition relates $s_i$ and $s_\star$ to the sparsity $s$ through inequalities that must be satisfied.

**Proposition G.2** (General coefficient). *Recall the definitions of $h_i$ in (B.3) and $h_\star$ in (G.1). Suppose the bias $b < -\sqrt{3}$, then for any feature $i \in [n]$ satisfying the conditions in (B.5) and (B.6) and that $\overline{\theta_i^2} > \widehat{\mathbb{Q}}_i(h_i)$, we have the following inequalities:*

$$h_\star \ge 1/\sqrt{s}, \quad h_i \ge \sqrt{\overline{\theta_i^2} - \widehat{\mathbb{Q}}_i(h_i)}.$$

*Proof.* See §I.1 for a detailed proof. $\square$

**General version of Theorem B.2.** In the following, we will let $s_\star = 1/h_\star^2$. To ensure consistency in the notation, we will also define $s_i = 1/h_i^2$ for $h_i$ defined in (B.3). We give a more general version of Theorem B.2 in the following theorem, which will be formally proved in the remaining part of this section.

**Theorem G.3.** *For feature $i \in [n]$, let us take some small constant $\varepsilon \in (0, 1)$ and define $Q^{(i)}$ as*

$$Q^{(i)} = \widehat{\mathbb{Q}}^{(i)}\left(\frac{-b/\sqrt{\log n}}{(1 - \varepsilon)\sqrt{2(\log_n M - 1)}}\right).$$

*Suppose*

$$\frac{\log \eta}{\log n} \gtrsim \frac{b^2/2 - \log N}{\log n}.$$

*For any feature $i \in [i]$, consider the following joint conditions for $\rho_2$, $d$, $Q^{(i)}$ and bias $b < 0$ with respect to constant parameter $\varsigma \in (0, 1)$:*

> ***Individual Feature Occurrence:*** $\quad \dfrac{\|H_{:,i}\|_0}{\rho_1 N} \ge \text{polylog}(n)^{-1},$
>
> ***Limited Feature Co-ocurrence:*** $\quad \log_n(\rho_2^{-1}) \gtrsim \max\left\{-4\log_n Q^{(i)}, \frac{1}{2} - \log_n Q^{(i)}\right\},$
>
> ***Bias Range:*** $1 \gtrsim \dfrac{b^2}{2 \log n} \gtrsim \max\left\{\frac{1}{2} + \frac{h_\star^2}{2} - (1 + h_\star^2)\log_n Q^{(i)}, \frac{1}{4} + \frac{h_\star^2}{4} - (3h_\star^2 + 1)\log_n Q^{(i)}, \right.$
> $\left. \left(\sqrt{2}h_\star(1 + \varepsilon) + \sqrt{-(1 - h_\star^2)\log_n Q_1}\right)^2, 1 - (1 - \varsigma)\log_n d - \log_n Q^{(i)}\right\}.$

*Here $x \gtrsim y$ means $x \ge y + O(\log \log(n)/\log(n))$. Then with probability at least $1 - n^{-4\varepsilon}$ over the randomness of the features $V$, for any feature $i$ such that there exists some constant $\varsigma_i$ satisfying the above conditions, there exists at least one unique neuron $m_i$ and after at most $T_i = \max\{(2\varsigma_i)^{-1}, 1\}$ steps of training, we have $\langle w_{m_i}^{T_i}, v_i\rangle/\|v_i\|_2 \ge 1 - o(1)$.*

**Relationship between Theorem B.2 and Theorem G.3.** The main difference between Theorem B.2 and Theorem G.3 is that the latter allows $Q^{(i)}$ to have a larger range of values, while the former requires $Q^{(i)} = \widehat{\mathbb{Q}}^{(i)}(h_i)$ to be strictly larger than $\mathrm{polylog}(n)^{-1}$. A direct consequence of this restriction in Theorem B.2 is that the range of $M$ is smaller compared to that in Theorem G.3. However, the conditions in Theorem G.3 have $Q^{(i)}$ and $\rho_2, b$ coupled together, which makes it difficult to gain a clear understanding, while in Theorem B.2, we decouple the conditions by enforcing the range of $Q^{(i)}$. Specifically,

1. The condition $Q^{(i)} \geq \mathrm{polylog}(n)^{-1}$ is equivalent to

$$\frac{-b}{(1-\varepsilon)\sqrt{2(\log_n M - 1)}} \leq h_i$$

   by recalling the definition of $h_i$. This gives the range of $M$ as in (B.4) if we require all the features to be learned simultaneously. In fact, if the condition is satisfied for only a subset of features, our theorem still holds on that subset of features.

2. The individual feature occurrence condition is the same in both theorems, and the limited feature co-occurrence condition in Theorem G.3 will reduce to $\rho_2 \ll n^{-1/2-o(1)}$, which is already implied by the data condition in Definition B.1.

3. The bias range condition in Theorem G.3 will reduce to the version in Theorem B.2 by removing the terms that involve $Q^{(i)}$ as $\log\log(n)/\log(n)$ gap is already enforced by the $\gtrsim$ notation.

Moreover, we assume that $s \geq 3$ as mandated in Theorem B.2. Since $s_\star \leq s$ by Theorem G.2, if $s_\star \leq s \leq 2$ the following inequality

$$1 \gtrsim \frac{b^2}{2\log n} \gtrsim \frac{1}{s_\star}\left(\sqrt{2}(1+\varepsilon) + \sqrt{-(s_\star-1)\log_n Q^{(i)}}\right)^2$$

cannot hold, because the right-hand side would exceed 1.

**Roadmap for the proof of Theorem G.3.** The remaining part of this section is organized as follows:

- **Concentration simplification:** In §G.2, we will combine the concentration results derived in §F to derive explicitly the simplified concentration results for the atomic terms in (F.3) for the evolution of $\alpha_{-1,t}$ and $\beta_t$.

- **Conditions for strong alignment:** In §G.3, we formulate a set of conditions Cond.(i) to Cond.(iii), Cond.(I) and Cond.(II) that will yield a simple two-state recursion. Building upon these conditions, we further identify Cond.(iv) to Cond.(vi) that will guarantee a strong alignment $\alpha_{-1,T} = 1 - o(1)$ with only $T = O(1)$ steps of training.

- **Conditions simplification:** In §G.4, we further simplified the series of conditions into a more concise form as in (G.12), which yields the full list of conditions in Theorem G.3.

**Notation.** Following the convention in §F, we let $\overline{b}_t = b_t + \kappa_0$ where $\kappa_0 = O((\log n)^{-1/2})$ is defined in Definition B.3. Recall the definition $\zeta_1 = 2(1+\varepsilon)\sqrt{\log n}$ and $\zeta_0 = (1-\varepsilon)\sqrt{2\log n}$ in (E.11) for some small constant $\varepsilon \in (0,1)$. We let $C$ be a universal constant that may vary from line to line.

## G.2   CONCENTRATION RESULTS COMBINED

We now combine the concentration results for the second-order terms in Theorem F.4 and Theorem F.6 under the assumption that $t \log n \ll n$. In particular, by taking the square root of the upper bounds in these lemmas and noting that $\|v\|_2^2 = O(d)$ holds with probability at least $1 - n^{-c}$ (see Theorem J.1), we can express the combined square-root upper bound as

$$\xi_t = \sqrt{s}\, t \log n\, \mathcal{K}_t + \rho_1^{-1} \sqrt{\Phi(|\bar{b}_t|) \cdot \widehat{\mathbb{E}}_{l,l'}\left[\Phi\left(|\bar{b}_t|\sqrt{\frac{1 - \langle h_l, h_{l'}\rangle}{1 + \langle h_l, h_{l'}\rangle}}\right)\langle h_l, h_{l'}\rangle\right]}$$

$$+ \sqrt{\rho_2 d}\,|\alpha_{-1,t-1}| + \rho_2\sqrt{n}\,.$$

We formally state the combination of the above two lemmas in the following corollary.

**Corollary G.4** (Second-order concentration combined). *Then under the conditions $t \log n \ll n$, $-\bar{b}_t = \Theta(\sqrt{\log n}) < \zeta_1$, $\rho_1 \ll 1$, it holds for all $t \le T \le n^c$ with probability at least $1 - n^{-c}$ over the randomness of standard Gaussian vectors $z_{-1:T}$ and $v$ that*

$$\sqrt{\|E^\top \varphi(Ey_t^\star; b_t)\|_2^2 + \|F^\top \varphi(Fy_t + \theta \cdot v^\top \bar{w}_{t-1}; b_t)\|_2^2} \le CLN\rho_1\xi_t.$$

Here, the constant $C$ hides some factors from using the inequality $\sqrt{a} + \sqrt{b} \le \sqrt{2(a+b)}$. We refrain from a detailed proof here. With the second order concentration results in Theorem G.4, we can now derive the first-order concentration results for the terms $\langle z_\tau, u_t \rangle$ based on Theorem F.7 and Theorem F.8. To further simplify the concentration bound, we impose the additional condition $\Phi(|\bar{b}_t|) \gg L\, s\, \rho_1\, (t \log(n))^3$, which in particular holds if $\Phi(|\bar{b}_t|) \gg n^{-1} \mathrm{polylog}(n)$. This requirement is reasonable because it ensures that the neuron is not activated too rarely compared to the average occurrence frequency ($s/n$) of the features.

**Lemma G.5** (First-order concentration combined). *If $\Phi(|\bar{b}_t|) \gg Ls\rho_1(t\log(n))^3$, $-\bar{b}_t = \Theta(\sqrt{\log n}) < \zeta_1$, $\kappa_0|\bar{b}_t| = O(1)$, for all $t \le T \le n^c$, it holds with probability at least $1 - n^{-c}$ over the randomness of standard Gaussian vectors $z_{-1:T}$ that*

$$\langle z_\tau, u_t \rangle = N\alpha_{\tau,t-1}\widehat{\varphi}_1(b_t) \cdot (1 \pm o(1)) \pm CNL\rho_1\sqrt{\rho_2 s}(t\log n)^{3/2} \cdot d\,|\alpha_{\tau,t-1}\alpha_{-1,t-1}|$$

$$\pm CN\rho_1 L\sqrt{t\log n} \cdot \xi_t \pm CLN\sqrt{\log n} \cdot \left(\sqrt{s\rho_1 d\Phi(|\bar{b}_t|)} + \sqrt{s}\rho_1|\bar{b}_t|d\,\beta_{t-1}\right) \cdot \beta_{t-1},$$

*where $\xi_t$ is defined in Theorem G.4.*

*Proof.* See §I.2.1 for a detailed proof. □

In order to derive the recursion for $\alpha_{-1,t}$ in (F.3), we need to control the numerator

$$\alpha_{-1,t}\|w_t\|_2 = \frac{\langle v, w_t \rangle}{\|v\|_2} = \langle z_{-1}, u_t \rangle + \|v\|_2 \cdot \theta^\top \varphi(Fy_t + \theta \cdot v^\top \bar{w}_{t-1}; b_t) + \eta^{-1}\alpha_{-1,t-1}.$$

Using Theorem G.5 and the concentration for the second term in Theorem F.10, we derive the following lemma for $\langle v, w_t\rangle/\|v\|_2$.

**Lemma G.6** (Concentration for numerator in $\alpha$-recursion). *Suppose $\rho_1 d(st\log n)^{-1} \gg \Phi(|\bar{b}_t|) \gg Ls\rho_1(t\log(n))^3$, $-\bar{b}_t = \Theta(\sqrt{\log n}) < \zeta_1$, $\kappa_0|\bar{b}_t| = O(1)$, $\sqrt{ts\log n}|\bar{b}_t|\beta_{t-1} \ll 1$, and $\sqrt{d}\alpha_{-1,t-1} \gg 1$. Furthermore, assume that*

$$\frac{N_2}{N}C_0\overline{\theta^2}Q_t \gg \max\left\{L\rho_1\sqrt{\rho_2 s}(t\log n)^{3/2}, \, Ld^{-1}\Phi(|\bar{b}_t|), \, L\sqrt{t\log n}\rho_1\frac{\xi_t}{d\alpha_{-1,t-1}}, \, L\rho_1\frac{\beta_{t-1}}{\alpha_{-1,t-1}}\right\}.$$

*If $\eta^{-1} \ll N_2 dC_0\overline{\theta^2}Q_t$ Then it holds with probability at least $1 - n^{-c}$ over the randomness of standard Gaussian vectors $z_{-1:T}$ and $v$ that*

$$\frac{\langle v, w_t\rangle}{\|v\|_2} = (1 \pm o(1))N\psi_t.$$

*Proof.* See §I.2.2 for a detailed proof. □

Now that we have characterized the "numerator" for $\alpha$-recursion. It remains to control the "denumerator" $\|w_t\|_2$. In what follows, we will decompose the norm $\|w_t\|_2$ into two parts: the projection onto the subspace spanned by $w_{-1:0}$ and the projection onto the orthogonal compliment of this subspace. For $P^\perp_{w_{-1:0}}w_t$ being the projection onto the orthogonal complement of the subspace spanned by $w_{-1:0}$, we have the following bound.

**Lemma G.7.** *Suppose $\rho_1 d(st \log n)^{-1} \gg \Phi(|\bar{b}_t|) \gg L s \rho_1 (t \log(n))^3$, $-\bar{b}_t = \Theta(\sqrt{\log n}) < \zeta_1$, $\kappa_0 |\bar{b}_t| = O(1)$, $\sqrt{ts \log n}|\bar{b}_t|\beta_{-1} \ll 1$, $\sqrt{d}\alpha_{-1,t-1} \gg 1$, $\sqrt{\rho_2 s}(t \log n)^{3/2} \ll 1$, and $\eta^{-1} \ll N\Phi(|\bar{b}_t|)$. Then, for all $t \leq T \leq \sqrt{d}$, it holds with probability at least $1 - n^{-c}$ over the randomness of standard Gaussian vectors $z_{-1:T}$ and $v$ that*

$$\|P_{w_{-1:0}}^\perp w_t\|_2 \leq CNL\rho_1 \sqrt{d}(\xi_t + \sqrt{d}\beta_{t-1}).$$

*Proof.* See §I.2.3 for a detailed proof. $\qquad\square$

For $P_{w_{-1:0}} w_t$ being the projection onto the subspace spanned by $w_{-1:0}$, we have the following bound.

**Lemma G.8.** *Suppose $\rho_1 d(st \log n)^{-1} \gg \Phi(|\bar{b}_t|) \gg L s \rho_1 (t \log(n))^3$, $-\bar{b}_t = \Theta(\sqrt{\log n}) < \zeta_1$, $\kappa_0 |\bar{b}_t| = O(1)$, $\sqrt{ts \log n}|\bar{b}_t|\beta_{-1} \ll 1$, and $\sqrt{d}\alpha_{-1,t-1} \gg 1$. Furthermore, assume for some constant $C_0 > 0$ that*

$$\frac{N_2}{N} C_0 \overline{\theta^2} Q_t \gg \max\left\{ L\rho_1 \sqrt{\rho_2 s}(t \log n)^{3/2}, \; L d^{-1} \Phi(|\bar{b}_t|), \; L\sqrt{t \log n}\rho_1 \frac{\xi_t}{d\alpha_{-1,t-1}}, \; L\rho_1 \frac{\beta_{t-1}}{\alpha_{-1,t-1}} \right\}.$$

*If $\eta^{-1} \ll N_2 d C_0 \overline{\theta^2} Q_t \wedge N\Phi(|\bar{b}_t|)$, then it holds with probability at least $1 - n^{-c}$ over the randomness of standard Gaussian vectors $z_{-1:T}$ and $v$ that*

$$\left\| P_{w_{-1:0}} w_t \right\|_2 = (1 \pm o(1)) \cdot \sqrt{(N\psi_t)^2 + \left(N\alpha_{0,t-1}\widehat{\varphi}_1(b_t)\right)^2}.$$

*Proof.* See §I.2.4 for a detailed proof. $\qquad\square$

Combining the results from Theorems G.7 and G.8, we obtain the upper bound for $\|w_t\|_2$.

**Lemma G.9.** *Suppose $\rho_1 d(st \log n)^{-1} \gg \Phi(|\bar{b}_t|) \gg L s \rho_1 (t \log(n))^3$, $-\bar{b}_t = \Theta(\sqrt{\log n})$, $\kappa_0 |\bar{b}_t| = O(1)$, $\sqrt{ts \log n}|\bar{b}_t|\beta_{-1} \ll 1$, $\sqrt{d}\alpha_{-1,t-1} \gg 1$ and $\sqrt{\rho_2 s}(t \log n)^{3/2} \ll 1$. Furthermore, assume for some constant $C_0 > 0$ that*

$$\frac{N_2}{N} C_0 \overline{\theta^2} Q_t \gg \max\left\{ L\rho_1 \sqrt{\rho_2 s}(t \log n)^{3/2}, \; L d^{-1} \Phi(|\bar{b}_t|), \; L\sqrt{t \log n}\rho_1 \frac{\xi_t}{d\alpha_{-1,t-1}}, \; L\rho_1 \frac{\beta_{t-1}}{\alpha_{-1,t-1}} \right\}.$$

*If $\eta^{-1} \ll N_2 d C_0 \overline{\theta^2} Q_t \wedge N\Phi(|\bar{b}_t|)$, then it holds for all $t \leq T \leq \sqrt{d}$ with probability at least $1 - n^{-c}$ over the randomness of standard Gaussian vectors $z_{-1:T}$ and $v$ that*

$$\|w_t\|_2 \leq (1 \pm o(1)) \cdot \sqrt{(N\psi_t)^2 + \left(N\widehat{\varphi}_1(b_t)\right)^2} + CNL\rho_1 \sqrt{d}\xi_t.$$

*Proof of Theorem G.9.* By the triangle inequality, it holds that

$$\|w_t\|_2 \leq \|P_{w_{-1:0}} w_t\|_2 + \|P_{w_{-1:0}}^\perp w_t\|_2$$
$$\leq (1 + o(1))\sqrt{(N\psi_t)^2 + \left(N\alpha_{0,t-1}\widehat{\varphi}_1(b_t)\right)^2} + CNL\rho_1 \sqrt{d}(\xi_t + \sqrt{d}\beta_{t-1}).$$

By condition $\frac{N_2}{N} C_0 \overline{\theta^2} Q_t \gg L\rho_1 \frac{\beta_{t-1}}{\alpha_{-1,t-1}}$ and the lower bound $N\psi_t \geq C\overline{\theta^2} Q_t N_2 d\alpha_{-1,t-1}$ shown in Theorem F.11, we have $N\psi_t \gg CNL\rho_1 d\beta_{t-1}$ satisfied and can be absorbed into the upper bound of $\|P_{w_{-1:0}} w_t\|_2$. Hence, we conclude the proof. $\qquad\square$

When we derive the above lemmas step by step, we collect all the conditions used in the final Theorem G.9. In the following proof, we will be focusing on the conditions listed in the statement of this lemma.

### G.3 A TWO-STATE ALIGNMENT RECURSION

From now on, we adhere to the fact that the bias remains fixed throughout the dynamics. Thus, we drop the time index in $b_t$ (writing it simply as $b$) and define $\bar{b} = b + \kappa_0$. To further simplify the conditions in the previous section, we have the following lemma.

**Lemma G.10.** *Consider fixing the bias to be $b < 0$ and $\bar{b} = b + \kappa_0 < 0$. Suppose **InitCond-1** and **InitCond-2** hold. With the following conditions at initialization:*

(i) $-\bar{b} = \Theta(\sqrt{\log n}) < \zeta_1$, $\kappa_0|\bar{b}| = O(1)$, $\sqrt{\rho_2 s}(T \log n)^{3/2} \ll 1$, $\eta^{-1} \ll N_2 d C_0 \overline{\theta^2} Q_1 \wedge N\Phi(|\bar{b}|)$.

(ii) $\rho_1 d (sT \log n)^{-1} \gg \Phi(|\bar{b}|) \gg L s \rho_1 (T \log(n))^3$.

(iii) $\frac{N_2}{N} C_0 \overline{\theta^2} Q_1 \gg \max\left\{ L\rho_1 \sqrt{\rho_2 s}(T \log n)^{3/2},\ L d^{-1}\Phi(|\bar{b}|),\ L\sqrt{T \log n}\rho_1 \cdot \frac{\xi_1}{d\alpha_{-1,0}} \right\}$.

*If for some time step $t \leq T \leq \sqrt{d}$ we have*

(I) $\alpha_{-1,t-1} \geq t^2 \alpha_{-1,0}$, $\sqrt{Ts \log n}|\bar{b}|\beta_{t-1} \ll 1$,

(II) $\frac{N_2}{N} C_0 \overline{\theta^2} Q_t \gg L\rho_1 \frac{\beta_{t-1}}{\alpha_{-1,t-1}}$.

*Let us define*

$$\lambda_0 = \frac{CL\rho_1}{C_0\overline{\theta^2} \cdot N_2/N}, \quad \lambda_t = \frac{\lambda_0}{Q_t}, \quad \widetilde{\xi}_t = \frac{1}{\sqrt{d}}\left( \frac{\xi_1}{\alpha_{-1,0}} + C\sqrt{s}t^2 (\log n)^{3/2} \cdot \mathbb{1}(t \geq 2) \right)$$

*for some sufficiently large constant $C > 0$. Under the above conditions, we have the following conclusions:*

(1). *All the conditions in Theorem G.9 hold for $t \leq T$;*

(2). *Then with probability at least $1 - n^{-c}$ over the randomness of standard Gaussian vectors $z_{-1:T}$ and $v$, we have the following two-state alignment recursion:*

> **Two-State Alignment Recursion**
>
> $$\frac{\beta_t}{\alpha_{-1,t}} \leq \lambda_t \cdot \left( \widetilde{\xi}_t + \frac{\beta_{t-1}}{\alpha_{-1,t-1}} \right), \quad \frac{1}{\alpha_{-1,t}} \leq (1 + o(1)) + \lambda_t \cdot \left( \frac{\Phi(|\bar{b}|)}{\rho_1 d} \cdot \frac{1}{\alpha_{-1,t-1}} + \widetilde{\xi}_t \right).$$

*Proof.* See §I.3.1 for a detailed proof. $\qquad\square$

From the above lemma, we can obtain the following observations:

- The ratio $\lambda_t \Phi(|\bar{b}|)/\rho_1 d$ controls the growth of the alignment $\alpha_{-1,t}$. In order for the alignment to grow faster, we need a smaller activation frequency $\Phi(|\bar{b}|)$, i.e., a larger bias $|\bar{b}|$ in the absolute value.
- The term $\lambda_t$ controls the growth of the ratio $\beta_t/\alpha_{-1,t}$. By definition, we know that $\lambda_t \geq 1$.
- The maximum alignment achievable is $1 - o(1)$.

Therefore, the best we can do is to set $\lambda_t$ as close to 1 as possible while exploiting a small ratio $\Phi(|\bar{b}|)/\rho_1 d$ to ensure that the alignment $\alpha_{-1,t}$ goes to 1 before the ratio $\beta_t/\alpha_{-1,t}$ blows up. Since $\beta_0 = 0$, we have $\beta_1/\alpha_{-1,1} = \lambda_1 \cdot \widetilde{\xi}_1$. This means we also need a small initial value $\widetilde{\xi}_1$ to avoid a large ratio $\beta_1/\alpha_{-1,1}$ at the beginning. In the sequel, we quantitatively analyze the evolution of the above recursions. Before we proceed, by definition $Q_t = \frac{1}{N_2}\sum_{l=1}^{N_2} \mathbb{1}\left(\theta_l > \frac{-b}{\sqrt{d}\alpha_{-1,t-1}}\right)$, we note that $Q_t$ is nondecreasing in $\alpha_{-1,t-1}$. Therefore, we have the following fact:

**Fact G.11.** *If $\alpha_{-1,t-1} \geq \alpha_{-1,1}$, then $Q_t \geq Q_2$ and $\lambda_t \leq \lambda_2$.*

**Expanding the recursions.** Let us define $T_0 + 1$ as the minimum of $t$ such that either of the following conditions fails:

$T_0$-**Cond.(1).** Cond.(I) or Cond.(II);

$T_0$-**Cond.(2).** $\alpha_{-1,t-1} \geq \alpha_{-1,1}$;

$T_0$-**Cond.(3).** $t < \log(n)$.

In other word, $T_0$ is the *stopping time* up to which all the conditions above hold. We have $\lambda_t \leq \lambda_2$ by Theorem G.11 and the definition $\lambda_t = \lambda_0/Q_t$. To obtain a simple recursion for $\alpha_{-1,t}$, we take

$$C_1 = \left(1 + o(1) + \frac{\lambda_2 \xi_1}{\sqrt{d}\alpha_{-1,0}} + \frac{C\lambda_2\sqrt{s}T_0^2(\log n)^{3/2}}{\sqrt{d}}\right) \cdot \frac{1}{1 - \lambda_2\Phi(|\bar{b}|)/\rho_1 d}. \tag{G.2}$$

Here, we take the $o(1)$ term above to be the maximum of all the $o(1)$ terms in the recursion for $\alpha_{-1,t}$ for any $t \leq T_0$. For $2 \leq t \leq T_0$, we have from substracting $C_1$ from both sides of the recursion for $\alpha_{-1,t}$ that

$$\frac{1}{\alpha_{-1,t}} - C_1 \leq (1 + o(1)) + \lambda_t \cdot \left(\frac{\Phi(|\bar{b}|)}{\rho_1 d} \cdot \frac{1}{\alpha_{-1,t-1}} + \frac{1}{\sqrt{d}}\left(\frac{\xi_1}{\alpha_{-1,0}} + C\sqrt{s}t^2(\log n)^{3/2} \cdot \mathbb{1}(t \geq 2)\right)\right)$$

$$\underbrace{-\left(1 + o(1) + \frac{\lambda_2\xi_1}{\sqrt{d}\alpha_{-1,0}} + \frac{C\lambda_2\sqrt{s}T_0^2(\log n)^{3/2}}{\sqrt{d}}\right) - \frac{C_1\lambda_2\Phi(|\bar{b}|)}{\rho_1 d}}_{-C_1}$$

$$\leq \frac{\lambda_2\Phi(|\bar{b}|)}{\rho_1 d} \cdot \left(\frac{1}{\alpha_{-1,t-1}} - C_1\right), \quad \forall 2 \leq t \leq T_0. \tag{G.3}$$

Using the fact that

$$\frac{1}{\alpha_{-1,1}} - C_1 \leq 1 + o(1) + \left(\frac{\lambda_1\Phi(|\bar{b}|)}{\rho_1 d} + \frac{\lambda_1\xi_1}{\sqrt{d}}\right) \cdot \frac{1}{\alpha_{-1,0}} - C_1$$

$$\leq \left(\frac{\lambda_1\Phi(|\bar{b}|)}{\rho_1 d} + \frac{\lambda_1\xi_1}{\sqrt{d}}\right) \cdot \frac{1}{\alpha_{-1,0}}, \tag{G.4}$$

we obtain that

$$\frac{1}{\alpha_{-1,t}} \leq \left(\frac{\lambda_2\Phi(|\bar{b}|)}{\rho_1 d}\right)^{t-1} \cdot \left(\frac{\lambda_1\Phi(|\bar{b}|)}{\rho_1 d} + \frac{\lambda_1\xi_1}{\sqrt{d}}\right) \cdot \frac{1}{\alpha_{-1,0}} + C_1, \quad \forall 1 \leq t \leq T_0. \tag{G.5}$$

In the above formula, we can extend $t$ to allow $t = 1$ as

$$\frac{1}{\alpha_{-1,1}} \leq 1 + o(1) + \left(\frac{\lambda_1\Phi(|\bar{b}|)}{\rho_1 d} + \frac{\lambda_1\xi_1}{\sqrt{d}}\right) \cdot \frac{1}{\alpha_{-1,0}} \leq C_1 + \left(\frac{\lambda_1\Phi(|\bar{b}|)}{\rho_1 d} + \frac{\lambda_1\xi_1}{\sqrt{d}}\right) \cdot \frac{1}{\alpha_{-1,0}}.$$

For the ratio $\beta_t/\alpha_{-1,t}$, we use the fact that $\lambda_t \leq \lambda_2$ for $2 \leq t \leq T_0$ and also that

$$\widetilde{\xi}_t \leq \frac{1}{\sqrt{d}}\left(\frac{\xi_1}{\alpha_{-1,0}} + C\sqrt{s}T_0^2(\log n)^{3/2}\right), \quad 2 \leq t \leq T_0$$

to expand the recursion for $\beta_t/\alpha_{-1,t}$ as follows:

$$\frac{\beta_t}{\alpha_{-1,t}} \leq \frac{1}{\sqrt{d}}\left(\frac{\xi_1}{\alpha_{-1,0}} + C\sqrt{s}T_0^2(\log n)^{3/2}\right) \cdot \sum_{\tau=2}^{t} \lambda_2^{t-\tau+1} + \lambda_2^{t-1} \cdot \frac{\beta_1}{\alpha_{-1,1}}$$

$$\leq \frac{T_0}{\sqrt{d}}\left(\frac{\xi_1}{\alpha_{-1,0}} + C\sqrt{s}T_0^2\log(n)^{3/2}\right) \cdot \lambda_2^{t-1} + \lambda_2^{t-1} \cdot \frac{\lambda_1\xi_1}{\sqrt{d}\alpha_{-1,0}}$$

$$= \frac{\lambda_2^{t-1}}{\sqrt{d}} \cdot \left((T_0 + \lambda_1) \cdot \frac{\xi_1}{\alpha_{-1,0}} + C\sqrt{s}T_0^3\log(n)^{3/2}\right), \quad \forall 1 \leq t \leq T_0, \tag{G.6}$$

where in the second inequality, we use the fact that $\lambda_2 \geq 1$ and the recursion for the ratio that

$$\frac{\beta_1}{\alpha_{-1,1}} \leq \lambda_1\widetilde{\xi}_1 = \frac{\lambda_1\xi_1}{\sqrt{d}\alpha_{-1,0}}. \tag{G.7}$$

Also in the last equality of (G.6), we can relax the condition to allow $t = 1$ as the right-hand side for $t = 1$ clearly upper bounds the right-hand side of (G.7). Using the results derived in (G.5) and (G.6), we now have the following statement. Now, building upon the results derived in (G.5) and (G.6), we have the following lemma, which summarizes the additional conditions needed to ensure that the alignment $\alpha_{-1,t}$ can be driven to $1 - o(1)$.

**Lemma G.12.** *Let $\varsigma \in (0, 1)$ be a constant. Take $\epsilon = C' \log \log n / (\varsigma \log d)$ for some sufficiently large constant $C' > 0$. Suppose **InitCond-1** and **InitCond-2**, Cond.(i) to Cond.(iii) hold. Under the following conditions*

*(iv)* $\lambda_0 = \Theta(\text{polylog}(n))$.

*(v)* $\lambda_0^{-1} Q_1 \cdot d^{-\varsigma} = \Phi(|\bar{b}|)/\rho_1 d$.

*(vi)* $\xi_1/Q_1 \ll d^{-\epsilon}/(\lambda_0\sqrt{s}\log n)$.

*there exists a time $t^\star \leq ((2\varsigma)^{-1} \vee 1) \wedge T_0$ such that $\alpha_{-1,t} = 1 - o(1)$, where $T_0$ is the stopping time before and at which $T_0$-Cond.(1) to $T_0$-Cond.(3) hold.*

*Proof.* See §I.3.2 for a detailed proof of the lemma. □

Since $t^\star \leq T_0$, Cond.(I) and Cond.(II) hold for all $t \leq t^\star$ *automatically*. In summary, in Theorem G.12, we have shown that under Cond.(i) to Cond.(vi), the alignment $\alpha_{-1,t}$ can be driven to $1 - o(1)$ in constant time steps.

## G.4  SIMPLIFYING THE CONDITIONS OF THEOREM G.13

To finish the proof of Theorem B.2, it remains to simplify the conditions in Theorem G.12. As a first step, we have the following lemma.

**Lemma G.13.** *Under **InitCond-1**, **InitCond-2**, and Definition B.3, Cond.(i) to Cond.(vi) hold upon the following conditions for some constant $\varsigma \in (0, 1)$ and $\epsilon = C' \log \log n / (\varsigma \log d)$ for some sufficiently large constant $C' > 0$:*

$$\frac{Q_1}{\lambda_0} \cdot d^{-\varsigma} = \frac{\Phi(|\bar{b}|)}{\rho_1 d} \gg \max\left\{d^{\epsilon-\varsigma}\sqrt{s}\log n \cdot \xi_1, \frac{Ls\log(n)^3}{d}\right\}, \tag{G.8}$$

$$\lambda_0 = O(\text{polylog}(n)), \quad \eta^{-1} \ll N\Phi(|\bar{b}|).$$

*Proof.* See §I.4.1 for a detailed proof. □

Next, we will plug in the definition of $\xi_1$ into the above condition to obtain the statement in Theorem B.2. In what follows, let us define $h_\star$ as the smallest number such that

$$\max\{\hbar_{4,\star}^2, \hbar_{3,\star}^2, \hbar_{4,1}^2\} \leq h_\star^2, \quad \sum_{j=1}^n \frac{1}{|\mathcal{D}_j|^2} \sum_{l,l'\in\mathcal{D}_j} \Phi\left(|\bar{b}|\sqrt{\frac{1 - H_{l,j}H_{l',j}}{1 + H_{l,j}H_{l',j}}}\right) \leq n\Phi\left(|\bar{b}|\sqrt{\frac{1 - h_\star^2}{1 + h_\star^2}}\right), \tag{G.9}$$

where $\hbar_{4,\star}^2, \hbar_{3,\star}^2$ and $\hbar_{4,1}^2$ are defined in Theorem F.4. The definition is valid as the right-hand sides of both inequalities are increasing in $h_\star$. In addition, we notice that $h_\star \leq 1$ always holds, as $h_\star = 1$ gives the trival upper bounds for all the inequalities in (G.9). In fact, the quantity $h_\star$ characterize the concentration level for the empirical distribution of $\{H_{l,j}\}_{l\in\mathcal{D}_j}$.

**Lemma G.14.** *If $(1 - h_\star^2)/(1 + h_\star^2) = \Theta(1)$ for $h_\star$ defined in (G.9), it holds that*

$$\widehat{\mathbb{E}}_{l,l'}\left[\Phi\left(|\bar{b}|\sqrt{\frac{1 - \langle h_l, h_{l'}\rangle}{1 + \langle h_l, h_{l'}\rangle}}\right)\langle h_l, h_{l'}\rangle\right] \leq Cn\rho_1^2 \cdot \Phi(|\bar{b}|)^{\frac{1-h_\star^2}{1+h_\star^2}} + \rho_1\rho_2 s^2. \tag{G.10}$$

*Proof.* See §I.4.2 for a detailed proof. □

Next, we also upper bound $\mathcal{K}_1$ in terms of $h_\star$.

**Lemma G.15.** *Under the conditions that $\zeta_1 h_\star / |\bar{b}| < 1 - \nu$ for some small constant $\nu \in (0,1)$ and $\Phi(|\bar{b}|) \geq \rho_1$, it holds for some sufficiently large constant $C > 0$ that*

$$C^{-1} \mathcal{K}_1 \leq \left(n |\bar{b}|\right)^{1/4} \Phi(|\bar{b}|)^{\frac{1}{3h_\star^2 + 1}} + (\rho_2 s n |\bar{b}|)^{1/4} \cdot \Phi(|\bar{b}|)^{\frac{3}{8h_\star^2 + 4}}$$

$$+ \left(\Phi(|\bar{b}|)^{\frac{(1 - h_\star \zeta_1 / |\bar{b}|)^2}{1 - h_\star^2}} + (\rho_2 s)^{1/4}\right) \cdot (\log n)^{1/4} + n^{1/4} \rho_2 s \log n.$$

*Proof.* See §I.4.3 for a detailed proof. $\qquad\square$

In the following, let us take $s = O(\mathrm{polylog}(n))$, $L = O(\mathrm{polylog}(n))$ and

$$d = n^{x_0}, \quad \rho_1 = n^{-x_1}, \quad \rho_2 = n^{-x_2}, \quad \Phi(|\bar{b}|) = n^{-1+x_3}. \tag{G.11}$$

Using the above configurations, we have by the Mill's ratio that

$$|\bar{b}| = \sqrt{2(1 - x_3) \log n} \pm O(\log \log(n)).$$

In the following, we use the notation $x \lesssim y$ to denote that $x \leq y + O(\log \log(n) / \log n)$, and $x \simeq y$ to denote that $x \lesssim y$ and $y \lesssim x$. Consequently, we have $|\bar{b}| / \sqrt{\log n} \simeq \sqrt{2(1 - x_3)}$, and

$$\frac{\zeta_1}{|\bar{b}|} \simeq \frac{2(1 + \varepsilon)\sqrt{\log n}}{\sqrt{2(1 - x_3) \log n}} = (1 + \varepsilon) \cdot \sqrt{\frac{2}{1 - x_3}}$$

With Theorems G.14 and G.15, we can now upper bound $\xi_1$ as

$$\log_n \mathcal{K}_1 \lesssim \max\left\{\frac{1}{4} + \frac{x_3 - 1}{3h_\star^2 + 1}, \frac{1 - x_2}{4} + \frac{3(x_3 - 1)}{8h_\star^2 + 4}, -\frac{(\sqrt{1 - x_3} - \sqrt{2}h_\star(1 + \varepsilon))^2}{1 - h_\star^2}, -\frac{x_2}{4}, \frac{1}{4} - x_2\right\}.$$

In addition, using Theorem G.14, the second term in the definition of $\xi_1$ is upper bounded as

$$\log_n \left(\rho_1^{-1} \sqrt{\Phi(|\bar{b}|) \widehat{\mathbb{E}}_{l,l'} \left[\Phi\left(|\bar{b}| \sqrt{\frac{1 - \langle h_l, h_{l'} \rangle}{1 + \langle h_l, h_{l'} \rangle}}\right) \langle h_l, h_{l'} \rangle\right]}\right) \lesssim \max\left\{\frac{x_3 + h_\star^2}{1 + h_\star^2} - \frac{1}{2}, \frac{x_3 - x_2 + x_1 - 1}{2}\right\}.$$

Therefore, $\xi_1$ is upper bounded as

$$\log_n \xi_1 \lesssim \max\left\{\frac{1}{4} + \frac{x_3 - 1}{3h_\star^2 + 1}, \frac{1 - x_2}{4} + \frac{3(x_3 - 1)}{8h_\star^2 + 4}, -\frac{(\sqrt{1 - x_3} - \sqrt{2}h_\star(1 + \varepsilon))^2}{1 - h_\star^2}, \right.$$

$$\left. -\frac{x_2}{4}, \frac{1}{4} - x_2, \frac{x_3}{2} + \frac{(1 - h_\star^2)(x_3 - 1)}{2(1 + h_\star^2)}, \frac{x_3 - x_2 + x_1 - 1}{2}, -\frac{x_2}{2}, \frac{1}{2} - x_2\right\}.$$

Plugging this bound into the first inequality in (G.8), we have the following reformulation:

$$\log_n Q_1 \simeq x_3 - (1 - \varsigma)x_0 \gtrsim \log_n \xi_1, \quad x_3 \gtrsim 0.$$

where we note that $\epsilon = O(\log \log(n) / \log n)$ and can be ignored in the context of $\simeq$ notation. Therefore, we just need to solve the following inequality system:

$$\log_n Q_1 \gtrsim \max\left\{\frac{1}{4} + \frac{x_3 - 1}{3h_\star^2 + 1}, \frac{1 - x_2}{4} + \frac{3(x_3 - 1)}{8h_\star^2 + 4}, -\frac{(\sqrt{1 - x_3} - \sqrt{2}h_\star(1 + \varepsilon))^2}{1 - h_\star^2}, \right.$$

$$\left. -\frac{x_2}{4}, \frac{1}{4} - x_2, \frac{x_3}{2} + \frac{(1 - h_\star^2)(x_3 - 1)}{2(1 + h_\star^2)}, \frac{x_3 - x_2 + x_1 - 1}{2}, -\frac{x_2}{2}, \frac{1}{2} - x_2\right\}$$

$$0 \lesssim x_3 \simeq (1 - \varsigma)x_0 + \log_n Q_1, \quad 0 \lesssim x_2 \lesssim 1, \quad x_0 \gtrsim 0, \quad 0 \lesssim x_1 \lesssim 1,$$

Solving this inequality system, we arrive at the following conditions that ensures (G.8):

$$1 \gtrsim x_2 \gtrsim \max\Big\{-4\log_n Q_1, \; \frac{1}{2} - \log_n Q_1\Big\}, \quad \log_n\Big(\frac{N_2}{\rho_1 N}\Big) \gtrsim 0, \quad \eta^{-1} \ll N\Phi(|\bar{b}|)$$

$$0 \lesssim x_3 \lesssim \min\Big\{\frac{1}{2} - \frac{h_\star^2}{2} + (1+h_\star^2)\log_n Q_1, \; \frac{3}{4} - \frac{h_\star^2}{4} + (3h_\star^2+1)\log_n Q_1, \quad \text{(G.12)}$$

$$1 - \big(\sqrt{2}h_\star(1+\varepsilon) + \sqrt{-(1-h_\star^2)\log_n Q_1}\big)^2, \; (1-\varsigma)x_0 + \log_n Q_1\Big\},$$

Now, the first condition involving $x_2 = \log_n(\rho_2^{-1})$ can be transformed into the **Limited Feature Co-occurrence** condition in Theorem G.3. The second condition $\log_n(N_2/\rho_1 N) \gtrsim 0$ can be transformed into the **Individual Feature Occurrence** condition in Theorem G.3 by noting that $N_2 = \|H_{:,i}\|_0$ for feature $i$ of interest. The third condition $\eta^{-1} \ll N\Phi(|\bar{b}|)$ can be transformed into

$$\log \eta \geq -\log N - \log \Phi(|\bar{b}|) + O(\log\log n),$$

where the second term on the right-hand side can be further upper bounded as

$$-\log \Phi(|\bar{b}|) \leq \frac{\bar{b}^2}{2} + O(\log\log n) \leq \frac{b^2}{2} + O(\log\log n + |\bar{b}|\kappa_0 + \kappa_0^2) \simeq \frac{b^2}{2} + O(\log\log n),$$

where we use the Mill's ration in the first inequality and the fact that $\kappa_0 = O((\log n)^{-1/2})$ and $|\bar{b}| < \sqrt{2\log n}$ in the second inequality. Therefore, a sufficient condition will be

$$\frac{\log \eta}{\log n} \gtrsim \frac{b^2/2 - \log N}{\log n}.$$

The last condition involving

$$x_3 = 1 - \frac{|b|^2 \pm O(\log\log(n))}{2\log n} = 1 - \frac{|b|^2 \pm O\big(\log\log(n) + |b|\kappa_0 + \kappa_0^2\big)}{2\log n} \simeq 1 - \frac{b^2}{2\log n}$$

can be transformed into the **Limited Feature Co-occurrence** condition in Theorem G.3. Lastly, we remind the readers that $Q_1$ is also lower bounded as a function of $x_3$, which is shown in the following proposition.

**Proposition G.16.** *Under **InitCond-1** and the reparameterization in* (G.11), *we have*

$$Q_1 \geq \widehat{\mathbb{Q}}\Big(\frac{-b/\sqrt{\log n}}{(1-\varepsilon)\sqrt{\log_n M - 1}}\Big), \quad where \quad \widehat{\mathbb{Q}}(x) := \frac{1}{N_2}\sum_{l=1}^{N_2} \mathbb{1}(\theta_l \geq x)$$

*is the tail function for the empirical distribution of $\theta_l$.*

*Proof of Theorem G.16.* By **InitCond-1**, we have $\sqrt{d}\alpha_{-1,0} \geq (1-\varepsilon)\sqrt{2\log(M/n)}$. Recall the definition of $Q_t$ in (F.15), we have by the non-increasing property of $\widehat{\mathbb{Q}}(\cdot)$ that

$$Q_1 = \widehat{\mathbb{Q}}\Big(\frac{|b|}{\sqrt{d}\alpha_{-1,0}}\Big) \geq \widehat{\mathbb{Q}}\Big(\frac{-b}{(1-\varepsilon)\sqrt{2\log(M/n)}}\Big).$$

This completes the proof of Theorem G.16. $\qquad\square$

Note that using the lower bound on $Q_1$ only strengthens the conditions in (G.12). Hence, we can directly plug in the lower bound of $Q_1$ into all the conditions in (G.12), and this gives us the final statement of Theorem G.3.

## H  PROOFS FOR CONCENTRATION RESULTS

In this section, we provide proof for the concentration results presented in the previous section. We first provide proofs for Theorem F.1 that controls the norm of the non-Gaussian component $\Delta y_t$. Then we give the proof for the concentrations of the second-order and first-order terms in the decomposition of the alignment recursion. Finally, we provide the proof for the error propagation in the dynamics due to the non-Gaussian component $\Delta y_t$.

## H.1 PROOFS FOR NON-GAUSSIAN COMPONENTS

In this subsection, we provide the proofs that are related to the Gaussian & non-Gaussian components. In particular, we provide the proof for Theorem F.1 that controls the norm of the non-Gaussian component $\Delta y_t$.

### H.1.1 PROOF OF THEOREM F.1

By definition of $\Delta y_t$ in (F.2), we further define

$$\Delta y_t^{(1)} = \sum_{\tau=1}^{t-1} \alpha_{\tau,t-1} \cdot \frac{\|w_\tau^\perp\|_2}{\|u_\tau^\perp\|_2} \cdot \frac{u_\tau^\perp}{\|u_\tau^\perp\|_2}, \quad \Delta y_t^{(2)} = -\sum_{\tau=1}^{t-1} \alpha_{\tau,t-1} \cdot P_{u_{1:\tau}} z_\tau,$$

and thus $\Delta y_t = \Delta y_t^{(1)} + \Delta y_t^{(2)}$. The proof of Theorem F.1 is then based on the bounding the $\ell_2$ norm of $\Delta y_t^{(1)}$ and $\Delta y_t^{(2)}$ respectively. To proceed with controlling the norm $\|\Delta y_t^{(1)}\|_2$, we first control the ratio $\|w_\tau^\perp\|_2/\|u_\tau^\perp\|_2$ via the following lemma.

**Lemma H.1** (Ratio $\|w_\tau^\perp\|_2/\|u_\tau^\perp\|_2$). *Take some total step $T \le \sqrt{d}$ and suppose $d \in (n^{1/c_1}, n^{c_1})$ for some universal constant $c_1 \in (0,1)$. For all $t = 1, \ldots, T$, it holds with probability at least $1 - n^{-c}$ for some universal constant $c, C > 0$ that*

$$\left| \frac{\|w_t^\perp\|_2}{\|u_t^\perp\|_2} - \sqrt{d} \right| \le C(\log n)^{1/2}, \quad \left| \frac{\|w_t^\perp\|_2^2}{\|u_t^\perp\|_2^2} - d \right| \le C\sqrt{d \log n}.$$

*Proof.* See §H.1.2 for a detailed proof. $\square$

With Theorem H.1, we can now control the $\ell_2$ norm of $\Delta y_t^{(1)}$ and $\Delta y_t^{(2)}$ respectively with the following two lemmas.

**Lemma H.2** ($\ell_2$ norm of $\Delta y_t^{(1)}$). *Under the conditions in Theorem H.1, for all $t = 1, \ldots, T$, it holds with probability at least $1 - n^{-c}$ for some universal constant $c, C > 0$ that*

$$(d - C\sqrt{d \log n}) \cdot \|P_{w_{-1:0}}^\perp \bar{w}_{t-1}\|_2^2 \le \|\Delta y_t^{(1)}\|_2^2 \le (d + C\sqrt{d \log n}) \cdot \|P_{w_{-1:0}}^\perp \bar{w}_{t-1}\|_2^2.$$

*Proof.* See §H.1.2 for a detailed proof. $\square$

**Lemma H.3** ($\ell_2$ norm of $\Delta y_t^{(2)}$). *Under the conditions in Theorem H.1, for all $t = 1, \ldots, T$, it holds with probability at least $1 - n^{-c}$ for some universal constants $c, C > 0$ that*

$$\|\Delta y_t^{(2)}\|_2^2 \le C(t + \log n) \cdot \|P_{w_{-1:0}}^\perp \bar{w}_{t-1}\|_2^2.$$

*Proof.* See §H.1.2 for a detailed proof. $\square$

Combining Theorem H.2 and Theorem H.3, we complete the proof of Theorem F.1 by additionally noting that

$$\|\Delta y_t\|_2^2 \le 2\|\Delta y_t^{(1)}\|_2^2 + 2\|\Delta y_t^{(2)}\|_2^2 \le 2(d + C\sqrt{d \log n} + C(t + \log n)) \cdot \|P_{w_{-1:0}}^\perp \bar{w}_{t-1}\|_2^2.$$

As the first term $d\|P_{w_{-1:0}}^\perp \bar{w}_{t-1}\|_2^2 = d\beta_{t-1}^2$ is the leading term, we conclude the proof of Theorem F.1.

### H.1.2 ADDITIONAL PROOFS FOR THEOREM F.1

*Proof of Theorem H.1.* Recall from (E.6) that

$$w_t = \sum_{\tau=-1}^{t-1} \langle P_{u_{1:\tau}}^\perp z_\tau, u_t \rangle \cdot \frac{w_\tau^\perp}{\|w_\tau^\perp\|_2} + \sum_{\tau=1}^{t-1} \langle u_\tau^\perp, u_t \rangle \cdot \frac{\|w_\tau^\perp\|_2}{\|u_\tau^\perp\|_2} \cdot \frac{w_\tau^\perp}{\|w_\tau^\perp\|_2} + P_{w_{-1:t-1}}^\perp \tilde{z}_t \cdot \|u_t^\perp\|_2$$

$$+ v\theta^\top \varphi(Fy_t + \theta \cdot v^\top \bar{w}_{t-1}; b_t) + \eta^{-1} \bar{w}_{t-1}.$$

Applying projection $P^\perp_{w_{-1:t-1}}$ to both sides, we have

$$w^\perp_t = P^\perp_{w_{-1:t-1}} w_t = P^\perp_{w_{-1:t-1}} \widetilde{z}_t \cdot \|u^\perp_t\|_2,$$

which implies that $\|w^\perp_t\|_2 / \|u^\perp_t\|_2 = \|P^\perp_{w_{-1:t-1}} \widetilde{z}_t\|_2$. Note that $\widetilde{z}_t$ is independent of $\sigma(w_{-1:t-1})$ and follows standard Gaussian distribution. Therefore, $\|w^\perp_t\|^2_2 / \|u^\perp_t\|^2_2 \sim \chi^2(d - t - 1)$ and we have by the concentration in Theorem J.1 that with probability at least $1 - n^{-c}$ for all $t \in [T]$,

$$\left| \frac{\|w^\perp_t\|^2_2}{\|u^\perp_t\|^2_2} - d \right| \le T + 2\sqrt{d \log(Tn^c)} + 2\log(Tn^c) \le C\sqrt{d \log(n)},$$

where the last inequality holds by conditions $T \le \sqrt{d}$ and $d \in (n^{1/c_1}, n^{c_1})$. Therefore, we conclude that with probability at least $1 - n^{-c}$ for all $t \in [T]$,

$$\left| \frac{\|w^\perp_t\|_2}{\|u^\perp_t\|_2} - \sqrt{d} \right| \le C(\log n)^{1/2}.$$

This completes the proof of Theorem H.1. $\qquad\qquad\square$

*Proof of Theorem H.2.* By definition of $\Delta y^{(1)}_t$, we have

$$\left| \|\Delta y^{(1)}_t\|^2_2 - d\|P^\perp_{w_{-1:0}} \bar{w}_{t-1}\|^2_2 \right| = \left| \sum_{\tau=1}^{t-1} \alpha^2_{\tau,t-1} \cdot \left( \frac{\|w^\perp_\tau\|^2_2}{\|u^\perp_\tau\|^2_2} - d \right) \right| \le \sup_{\tau=1,\ldots,t-1} \left| \frac{\|w^\perp_\tau\|^2_2}{\|u^\perp_\tau\|^2_2} - d \right| \cdot \sum_{\tau=1}^{t-1} \alpha^2_{\tau,t-1}$$

$$\le C\sqrt{d \log n} \cdot \|P^\perp_{w_{-1:0}} \bar{w}_{t-1}\|^2_2,$$

where the first equality holds by $\sum_{\tau=-1}^{t-1} \alpha^2_{\tau,t-1} = 1$ according to the definition of $\alpha_{\tau,t}$, and the second inequality holds by Theorem H.1 with probability at least $1 - n^{-c}$. $\qquad\square$

*Proof of Theorem H.3.* By rewriting the definition of $\Delta y^{(2)}_t$, we have

$$\Delta y^{(2)}_t = \sum_{\tau=1}^{t-1} \sum_{j=\tau}^{t-1} \alpha_{j,t-1} \frac{u^\perp_\tau}{\|u^\perp_\tau\|_2} z_j.$$

We note that when conditioned on $\{\alpha_{\tau,T-1}\}^{T-1}_{\tau=-1}$ and $u_{1:T-1}$, the random variables $\{\frac{u^\perp_\tau}{\|u^\perp_\tau\|_2} z_j\}_{j,\tau}$ for any $1 \le \tau \le j \le t - 1$ are i.i.d. standard Gaussian. Let us denote the filtration $\mathcal{F} = \sigma(\{\alpha_{\tau,T-1}\}^{T-1}_{\tau=-1}, u_{1:T-1})$. Therefore, we have

$$\Delta y^{(2)}_t \,|\, \mathcal{F} \overset{d}{=} \sum_{\tau=1}^{t-1} \sqrt{\sum_{j=\tau}^{t-1} \alpha^2_{j,t-1}} \cdot \frac{u^\perp_\tau}{\|u^\perp_\tau\|_2} \cdot z'_\tau,$$

where $\{z'_\tau\}^{t-1}_{\tau=1}$ are i.i.d. standard Gaussian independent of the filtration $\mathcal{F}$. Hence,

$$\|\Delta y^{(2)}_t\|^2_2 \,|\, \mathcal{F} \overset{d}{=} \sum_{\tau=1}^{t-1} \sum_{j=\tau}^{t-1} \alpha^2_{j,t-1} \cdot (z'_\tau)^2.$$

Using the concentration of $\chi^2$ distribution in Theorem J.1 gives us

$$\mathbb{P}\left( \left| \|\Delta y^{(2)}_t\|^2_2 - \sum_{\tau=1}^{t-1} \sum_{j=\tau}^{t-1} \alpha^2_{j,t-1} \right| \ge C\sqrt{\sum_{\tau=1}^{t-1} \left( \sum_{j=\tau}^{t-1} \alpha^2_{j,t-1} \right)^2} \cdot \sqrt{\log(n)} + C\sum_{\tau=1}^{t} \alpha^2_{\tau,t-1} \log(nT) \,\Big|\, \mathcal{F} \right) \le \frac{n^{-c}}{T}.$$

Each term inside the probability can be upper bounded by

$$\sqrt{\sum_{\tau=1}^{t-1} \left( \sum_{j=\tau}^{t-1} \alpha^2_{j,t-1} \right)^2} \le \sqrt{t} \cdot (1 - \alpha^2_{-1,t-1} - \alpha^2_{0,t-1}) = \sqrt{t} \cdot \|P^\perp_{w_{-1:0}} \bar{w}_{t-1}\|^2_2,$$

$$\sum_{\tau=1}^{t} \alpha^2_{\tau,t-1} = 1 - \alpha^2_{-1,t-1} - \alpha^2_{0,t-1} = \|P^\perp_{w_{-1:0}} \bar{w}_{t-1}\|^2_2, \qquad \sum_{\tau=1}^{t-1} \sum_{j=\tau}^{t-1} \alpha^2_{j,t-1} \le t\|P^\perp_{w_{-1:0}} \bar{w}_{t-1}\|^2_2.$$

Therefore, we conclude that when conditioning on $\mathcal{F}$, it holds with probability at least $1 - n^{-c}$ and for all $t = 1, \ldots, T$ that

$$\|\Delta y_t^{(2)}\|_2^2 \leq C(t + \sqrt{t \log n} + \log n) \cdot \|P_{w_{-1:0}}^{\perp} \bar{w}_{t-1}\|_2^2 \leq C(t + \log n) \cdot \|P_{w_{-1:0}}^{\perp} \bar{w}_{t-1}\|_2^2,$$

where $C$ is a universal constant that changes from line to line. Here, we also use the condition that $T \leq n$. Now, since for any event in the filtration $\mathcal{F}$, the failure probability is at most $n^{-c}$, we can safely remove the conditioning and conclude the proof of Theorem H.3. $\qquad\square$

## H.2 PROOFS FOR CONCENTRATION LEMMAS

In this subsection, we first provide a formal lemma that characterizes the sparsity of the activations when tuning the bias $b_t$ to be some negative value. Building upon this result, we then provide the proofs for the concentration results concerning the recursion of the alignment.

### H.2.1 CONCENTRATION FOR IDEAL ACTIVATIONS

The statement of the following lemma slightly generalize beyond the settings in (F.2) for technical convenience. Specifically, we want to understand how the neuron's activation frequency concentrates around $\Phi(-b_t)$. As we have the coefficient matrix $H$ decomposed into $E$ and $F$, we want to have a general result that can be applied to all of them. Therefore, we consider a general sparse weight matrix $G$ in the following lemma.

**Lemma H.4** (Concentration for Activations). *Let $G \in \mathbb{R}_+^{L \times n}$ be a nonnegative weight matrix whose rows $(g_l)_{l \in [L]}$ satisfy $\|g_l\|_2 = 1$, and assume that $G$ is sparse in both rows and columns:*

- *For every coordinate $i \in [n]$, the $i$th column satisfies $\|G_{:,i}\|_0 \leq \rho L$ for some $\rho \in [n^{-1}, 1]$.*

- *For every row $l \in [L]$, we have $\|g_l\|_0 \leq s$.*

*For any integer $t \leq n^c$ (with some fixed constant $c > 0$), define*

$$y_t = \sum_{\tau = -1}^{t-1} \alpha_{\tau, t-1} z_\tau,$$

*where the vectors $z_\tau \in \mathbb{R}^n$ (for $\tau = -1, 0, \ldots, t-1$) are independent standard Gaussian random vectors, and the coefficients $\alpha_{t-1} = (\alpha_{\tau, t-1})_{\tau = -1}^{t-1} \in \mathbb{S}^t$ belong to the unit sphere in $\mathbb{R}^{t+1}$. Next, let $b_t \in \mathbb{R}$ be an arbitrary bias and let $\vartheta_t \in \mathbb{R}^{t+1}$ and $\varsigma = (\varsigma_l)_{l \in [L]} \in \mathbb{R}_+^L$ be fixed vectors. For each neuron $l \in [L]$, define its shifted bias by*

$$b_{t,l} = b_t - \varsigma_l \, \alpha_{t-1}^{\top} \vartheta_t.$$

*Then, for any failure probability $\delta \in \big(\exp(-n/4), 1\big)$, there exists a universal constant $C > 0$ such that with probability at least $1 - \delta$ (over the randomness of the Gaussian vectors $\{z_\tau\}_{\tau = -1}^{t-1}$) the following holds simultaneously for all choices of $\alpha_{t-1} \in \mathbb{S}^t$ and $b_t \in \mathbb{R}$:*

$$\frac{1}{L} \sum_{l=1}^{L} \mathbb{1}\big\{g_l^{\top} y_t > b_{t,l}\big\} \leq C\left(\frac{1}{L} \sum_{l=1}^{L} \Phi(b_{t,l}) + \rho \, s \, t \, \log\big(n(1 + \|\varsigma\|_\infty \|\vartheta_t\|_\infty)\big) + \rho \, s \, \log(\delta^{-1})\right),$$

*where $\Phi(\cdot)$ denotes the standard Gaussian tail probability. In particular, if $t$, $\alpha_{t-1}$ and $b_t$ are also fixed, then with probability at least $1 - \delta$ it holds that*

$$\frac{1}{L} \sum_{l=1}^{L} \mathbb{1}\big\{g_l^{\top} y_t > b_{t,l}\big\} \leq C\left(\frac{1}{L} \sum_{l=1}^{L} \Phi(b_{t,l}) + \rho \, s \, \log(\delta^{-1})\right).$$

**Reduction to Theorem F.2.** We remark that when take $G$ to be the weight matrix $E$, $L$ to be $N_1$, $n$ to be $n-1$, $\rho$ to be $\rho_1$, $b_t$ to be $\bar{b}_t$, and letting $\vartheta_t = 0$, we directly obtain Theorem F.2 as a special case. In the remaining of this subsection, we will present the proof of this lemma.

*Proof of Theorem H.4.* In the following proof, we will use $C$ to denote universal constants that change from line to line.

**Step I: Concentration for fixed $\alpha_{t-1}$, $b_t$ and $\vartheta_t$.** When fixing $\alpha_{t-1}$, $b_t$ and $\vartheta_t$, note that

$$b_{t,l} = b_t - \varsigma_l \alpha_{t-1}^\top \vartheta_t$$

is also fixed and the only randomness comes from the Gaussian vectors $z_{-1}, z_0, \ldots, z_{t-1}$. In particular, $y_t \sim \mathcal{N}(0, I_n)$ since $\|\alpha_{t-1}\|_2 = 1$ by assumption. In the sequel, the discussion will be focused on one time step $t$ and we omit the subscript $t$ for simplicity. The following is a table of the notations we will use in the proof:

| | |
|---|---|
| $y \leftarrow y_t$ | $y_t = \sum_{\tau=-1}^{t-1} \alpha_{\tau,t-1} z_\tau$ |
| $b_l \leftarrow b_{t,l}$ | $b_{t,l} = b_t - \varsigma_l \alpha_{t-1}^\top \vartheta_t$ |
| $\alpha \leftarrow \alpha_{t-1}$ | $\alpha_{t-1} = (\alpha_{\tau,t-1})_{\tau=-1}^{t-1} \in \mathbb{S}^t$ |
| $y^{(i)}$ | $y^{(i)}$ is the vector $y$ with the $i$-th coordinate $y_i$ |
| | replaced by an independent copy $y'_i \sim \mathcal{N}(0, 1)$ |
| $Z$ | $Z = L^{-1} \sum_{l=1}^{L} \mathbb{1}(g_l^\top y > b_{t,l})$ |
| $Z^{(i)}$ | $Z^{(i)} = L^{-1} \sum_{l=1}^{L} \mathbb{1}(g_l^\top y^{(i)} > b_{t,l})$ |

Table 4: Summary of notations used in the proof of Theorem H.4.

Define $Z = L^{-1} \sum_{l=1}^{L} \mathbb{1}(g_l^\top y > b_l)$. To study the concentration of $Z$, we need to analyze the fluctuations when we change one coordinate of $y$. This leads us to the definition of $y^{(i)}$ in Table 4 with the corresponding $Z^{(i)} = L^{-1} \sum_{l=1}^{L} \mathbb{1}(g_l^\top y^{(i)} > b_{t,l})$. Let us also define the Exceedance-Perturbed Variance (EPV) as follows:

$$V_+ = \mathbb{E}\left[\sum_{i=1}^{n} (Z^{(i)} - Z)^2 \mathbb{1}(Z > Z^{(i)}) \,\Big|\, y\right].$$

In the definition of EPV, we only count the contribution from the $i$-th coordinate of $y$ when $Z$ exceeds its perturbed counterpart $Z^{(i)}$. Next, we show that $V_+$ is actually controlled by $Z$ itself up to a small factor. In particular, for the term inside the expectation in the definition of $V_+$, we have

$$\sum_{i=1}^{n} (Z^{(i)} - Z)^2 \mathbb{1}(Z > Z^{(i)}) = \sum_{i=1}^{n} (Z^{(i)} - Z)^2 \mathbb{1}(y_i > y'_i)$$

$$\leq \frac{1}{L^2} \sum_{i=1}^{n} \left(\sum_{l=1}^{L} \mathbb{1}(g_{l,i} \neq 0) \cdot \mathbb{1}(g_l^\top y > b_{t,l})\right)^2$$

$$\leq \frac{\rho}{L} \cdot \sum_{i=1}^{n} \sum_{l=1}^{L} \mathbb{1}(g_{l,i} \neq 0) \cdot \mathbb{1}(g_l^\top y > b_{t,l}) = \rho s Z.$$

where

- in the first identity, we use the fact that $Z$ is monotone in the $i$-th coordinate $y_i$ due to the nonnegativity of the weight matrix $G$.

- In the first inequality, we use the fact that $0 \leq \mathbb{1}(g_l^\top y > b_{t,l}) - \mathbb{1}(g_l^\top y^{(i)} > b_{t,l}) \leq \mathbb{1}(g_l^\top y > b_{t,l})$ thanks to the condition $y_i > y'_i$ which is guaranteed by the condition $Z > Z^{(i)}$.

- In the last line, we use the Cauchy-Schwarz inequality with the fact that $\sum_{l=1}^{L} \mathbb{1}(g_{l,i} \neq 0) \mathbb{1}(g_l^\top y > b_{t,l}) \leq \sum_{l=1}^{L} \mathbb{1}(g_{l,i} \neq 0) \leq \rho L$. Then, by also noting that each $g_l$ is also $s$-sparse, we obtain the last equality.

Meanwhile, the mean of $Z$ is simply $\mathbb{E}[Z] = L^{-1} \sum_{l=1}^{L} \Phi(b_{t,l})$, where we use the fact that $\|g_l\|_2 = 1$ by assumption and $g^{\top} y \sim \mathcal{N}(0,1)$. Invoking Theorem J.5, we conclude that for fixed $\alpha$ and $b_t$, we have with probability at least $1 - \delta$,

$$Z \leq \mathbb{E}[Z] + C\sqrt{\rho s \mathbb{E}[Z] \log \delta^{-1}} + C\rho s \log \delta^{-1} \leq C \cdot \left( \frac{1}{L} \sum_{l=1}^{L} \Phi(b_{t,l}) + \rho s \log(\delta^{-1}) \right) \quad \text{(H.1)}$$

for some universal constant $C > 0$. Here, we directly apply the inequality $\sqrt{ab} \leq a + b$ for $a, b > 0$ in the last inequality. In the following, we will apply a union bound on $\alpha_{t-1}, b_t$ to extend the above bound to arbitrary choices of $\alpha_{t-1}$ and $b_t$.

**Step II: Union bound over $\alpha_{t-1}$ and $b_t$.** In the following argument, we will also drop the subscript $t$. Since $Z$ is a function of $\alpha$ and $b$, we use the following notation:

$$Z(\alpha, b) = \frac{1}{L} \sum_{l=1}^{L} \mathbb{1} \left( \sum_{\tau=-1}^{t-1} \alpha_\tau g_l^{\top} z_\tau > b - \varsigma_l \alpha^{\top} \vartheta \right).$$

It is sufficient to construct a covering net for the pair $(\alpha, b)$. Since the Gaussian vectors $z_\tau$ are unbounded, we first introduce a truncation step in our covering argument. By applying the Chernoff bound for Gaussian tails and then taking a union bound over all indices $\tau = -1, 0, \ldots, t-1$, we deduce that with probability at least

$$1 - (t+1)n \cdot \exp(-n/2) \geq 1 - \exp(-n/4)/2,$$

we have

$$\max_{\tau=-1,0,\ldots,t-1} \|z_\tau\|_{\infty} \leq \sqrt{n}.$$

In what follows we condition on this high-probability event.

For $\alpha \in \mathbb{S}^t$, we take a uniform covering net on the sphere, denoted by $\mathcal{N}_\alpha$, such that for any $\alpha$, there exists $\alpha' \in \mathcal{N}_\alpha$ satisfying $\|\alpha - \alpha'\|_{\infty} \leq \epsilon$. The covering number is upper bounded by $|\mathcal{N}_\alpha| \leq \epsilon^{-t}$. See for example Example 5.8 in Wainwright (2019). To proceed, let us define

$$\mu = (t+1) \cdot (\sqrt{stn} + \|\varsigma\|_{\infty} \|\vartheta\|_{\infty}).$$

The intuition for this definition is that $\mu$ represents the Lipschitz constant of $\sum_{\tau=-1}^{t-1} \alpha_\tau g_l^{\top} z_\tau + \varsigma_l \alpha^{\top} \vartheta$ with respect to any perturbation on $\alpha$ in the $\ell_{\infty}$-norm. For $b$, leveraging the Gaussian tail property, we define the following covering net with size at most $4\mu\epsilon^{-1} + 4$:

$$\mathcal{N}_b = \left\{ k \cdot \mu \cdot \epsilon \mid k \in \mathbb{Z}, k \in \left[ -\lceil 2\epsilon^{-1} \rceil, \lfloor 2\epsilon^{-1} \rfloor \right] \right\} \cup \{-\infty\}.$$

There are three special points in $\mathcal{N}_b$: $-\infty$, the minimal finite point $b_{\min} = -\lceil 2\epsilon^{-1} \rceil \cdot \mu \cdot \epsilon$, and the maximal point $b_{\max} = \lfloor 2\epsilon^{-1} \rfloor \cdot \mu \cdot \epsilon$. For any $\alpha \in \mathbb{S}^t$ and $b \in \mathbb{R}$, we pick $\widehat{\alpha} = \operatorname{argmin}_{\alpha' \in \mathcal{N}_\alpha} \|\alpha - \alpha'\|_{\infty}$ and $\widehat{b} = \operatorname{argmax}\{b' \in \mathcal{N}_b : b' < b - \mu \cdot \epsilon\}$. Therefore, we have by the monotonicity of the indicator function that

$$Z(\alpha, b) \leq \frac{1}{L} \sum_{l=1}^{L} \mathbb{1} \left( \sum_{\tau=-1}^{t-1} \widehat{\alpha}_\tau g_l^{\top} z_t > b - \varsigma_l \widehat{\alpha}^{\top} \vartheta - \mu \cdot \epsilon \right) \leq Z(\widehat{\alpha}, \widehat{b}). \quad \text{(H.2)}$$

On the other hand, for $\widehat{b}_l = \widehat{b} - \varsigma_l \widehat{\alpha}^{\top} \vartheta$, using the definition of $\widehat{\alpha}$ and $\widehat{b}$, it holds that

$$\Phi(\widehat{b}_l) \leq \Phi(b_l - 3\mu \cdot \epsilon) \cdot \mathbb{1}(-b_{\min} \leq \widehat{b}_l < b_{\max}) + \mathbb{1}(\widehat{b}_l = -\infty)$$

$$+ \Phi(2\mu - \varsigma_l \alpha^{\top} \vartheta) \cdot \mathbb{1}(\widehat{b}_l = b_{\max}).$$

The above inequality holds by considering three cases:

- When $\widehat{b}_l \in [-b_{\min}, b_{\max})$, we have $b_l$ close to $\widehat{b}_l$ up to an approximation error of $3\mu \cdot \epsilon$, where one $\mu \cdot \epsilon$ comes from the approximation between $\alpha$ and $\widehat{\alpha}$ and the other $2\mu \cdot \epsilon$ comes from the approximation between $b_l$ and $\widehat{b}_l$.

- When $\widehat{b}_l = -\infty$, we simply upper bound the tail probability by 1.

- When $\widehat{b}_l = b_{\max}$, we have $\Phi(\widehat{b}_l) = \Phi(b_{\max} - \varsigma_l \alpha^\top \vartheta) \geq \Phi(2\mu - \varsigma_l \alpha^\top \vartheta)$.

Next, we characterize in each case the approximation error between $\Phi(b_l)$ and the bound given above, which are $\Phi(b_l - 3\mu \cdot \epsilon)$, 1, and $\Phi(2\mu - \varsigma_l \alpha^\top \vartheta)$ respectively. In particular,

- For the first case $\widehat{b}_t \in [-b_{\min}, b_{\max})$, we have the approximation error $\Phi(b_l - 3\mu \cdot \epsilon) - \Phi(b_l)$ directly bounded by $3\mu\epsilon$ by Lipschitz continuity of the Gaussian tail function.

- For the second case $\widehat{b}_t = -\infty$, it must hold that $b_t < b_{\min} + \mu\epsilon$, and the approximation error is thus upper bounded by $1 - \Phi(b_{t,l}) = 1 - \Phi(b_t - \varsigma_l \alpha^\top \vartheta) \leq \exp(-(|b_{\min}| - (1 + \epsilon)\mu)^2/2) \leq \exp(-\mu^2/4)$.

- For the third case $\widehat{b}_t = b_{\max}$, it must hold that $b_t > b_{\max} > 2\mu$. Hence, the approximation error is upper bounded by $\Phi(2\mu - \varsigma_l \alpha^\top \vartheta) - \Phi(b_{t,l}) \leq \Phi(2\mu - \varsigma_l \alpha^\top \vartheta) \leq \exp(-(2\mu - \mu)^2/2) \leq \exp(-\mu^2/4)$.

Combining these three cases, we conclude that

$$\Phi(\widehat{b}_l) \leq \Phi(b_l) + \exp(-\mu^2/4) + 3\mu \cdot \epsilon. \tag{H.3}$$

If we choose the covering net parameter $\epsilon = \rho\mu^{-1}$, then the upper bound can be simplified as $\Phi(\widehat{b}_l) \leq \Phi(b_l) + \exp(-n/4) + 3\rho$. Since $\rho$ is at least $1/n$, we can further conclude that $\Phi(\widehat{b}_l) \leq \Phi(b_l) + 4\rho$ given that $\exp(-n/4) \ll 1/n \leq \rho$. Lastly, note that the log cardinality of the joint covering net is upper bounded by

$$\log(|\mathcal{N}_\alpha|) + \log(|\mathcal{N}_b|) \leq t \log(\epsilon^{-1}) + \log(4\mu\epsilon^{-1}) \leq Ct \log(n(1 + \|\varsigma\|_\infty \|\vartheta\|_\infty)) \tag{H.4}$$

given that $\epsilon = \rho\mu^{-1} > (n\mu)^{-1}$. Here, for the last inequality, we use the fact that $\log(\mu) = \log((t+1)(\sqrt{stn} + \|\varsigma\|_\infty \|\vartheta\|_\infty)) \leq C \log(n(1 + \|\varsigma\|_\infty \|\vartheta\|_\infty))$ since $t \leq n^c$ for some constant $c > 0$ and $s \leq n$. We can also apply a similar argument for every $t < n^c$. This only increases the size of the covering net by a factor $n^c$. Combining (H.1), (H.2) and (H.3) with the log cardinality (H.4), we conclude that with probability at least $1 - \delta$ for all $\alpha, b_t$ and $\delta > \exp(-n/4)$ that

$$Z(\alpha_{t-1}, b_t) \leq Z(\widehat{\alpha}_{t-1}, \widehat{b}_t) \leq C \cdot \left( \frac{1}{L} \sum_{l=1}^{L} \Phi(\widehat{b}_{l,t}) + \rho st \log(n(1 + \|\varsigma\|_\infty \|\vartheta\|_\infty)) + \rho s \log(\delta^{-1}) \right)$$

$$\leq C \cdot \left( \frac{1}{L} \sum_{l=1}^{L} \Phi(b_{l,t}) + \rho st \log(n(1 + \|\varsigma\|_\infty \|\vartheta\|_\infty)) + \rho s \log(\delta^{-1}) \right),$$

where in the second inequality, we apply a union bound on the joint covering net for $\alpha$ and $b$ and also for all $t \leq n^c$. In the last inequality, we just need a change in the constant factor $C$ to absorb the approximation error $4\rho$ for the approximation error $\Phi(b_{t,l}) - \Phi(\widehat{b}_{t,l})$. Here, the lower bound $\delta > \exp(-n/4)$ is to ensure that the good event $\max_{\tau=-1,0,\ldots,t-1} \|z_\tau\|_\infty \leq \sqrt{tn}$ holds true. This concludes the proof of Theorem H.4. $\qquad\square$

## H.2.2 Activations with Non-Gaussian Component: Proof of Theorem F.3

In the following proof, we will use $C$ to denote universal constants that change from line to line. Let us denote by $\bar{b}_t = b_t + \kappa_0$ as the shifted bias. Let us pick $\varrho_t > 0$ to be specified later. For any $l \in [N_1]$, the neuron is activated only if either of the following two conditions hold:

1. $e_l^\top y_t^\star + \bar{b}_t > -\varrho_t$;

2. $e_l^\top y_t^\star + \bar{b}_t \leq -\varrho_t$ and $e_l^\top \Delta y_t > \varrho_t$.

For the first case, by Theorem F.2, we have with probability at least $1 - \delta$ that

$$\frac{1}{N_1} \sum_{l=1}^{N_1} \mathbb{1}(e_l^\top y_t^\star + \bar{b}_t > -\varrho_t) \leq C \cdot \big( \Phi(-\bar{b}_t - \varrho_t) + \rho_1 st \log(n) + \rho_1 s \log(\delta^{-1}) \big). \tag{H.5}$$

For the second case, we only need to control $N_1^{-1} \sum_{l=1}^{N_1} \mathbb{1}(e_l^\top \Delta y_t > \varrho_t)$. We have the following upper bound

$$
\begin{aligned}
\frac{1}{N_1} \sum_{l=1}^{N_1} \mathbb{1}(e_l^\top \Delta y_t > \varrho_t) &\leq \frac{1}{N_1} \sum_{l=1}^{N_1} \mathbb{1}\Big( \|e_l\|_2^2 \cdot \sum_{i=1}^{n-1} \Delta y_{t,i}^2 \, \mathbb{1}(E_{l,i} \neq 0) > \varrho_t^2 \Big) \\
&= \frac{1}{N_1} \sum_{l=1}^{N_1} \mathbb{1}\Big( \sum_{i=1}^{n-1} \Delta y_{t,i}^2 \, \mathbb{1}(E_{l,i} \neq 0) > \varrho_t^2 \Big) \\
&\leq \frac{1}{N_1 \varrho_t^2} \sum_{l=1}^{N_1} \sum_{i=1}^{n-1} \Delta y_{t,i}^2 \, \mathbb{1}(E_{l,i} \neq 0) \leq \frac{\rho_1}{\varrho_t^2} \cdot \|\Delta y_t\|_2^2, 
\end{aligned}
\tag{H.6}
$$

where the first inequality holds by the Cauchy-Schwarz inequality and the following equality holds by the fact that $\|e_l\|_2 = 1$. The second inequality follows from the fact that $\mathbb{1}(x > a) \leq x/a$ for any $a > 0$ and $x > 0$. The last inequality holds by noting that $\|E_{:,i}\|_0 \leq \rho_1 N_1$. Combining (H.5) and (H.6), we conclude that with probability at least $1 - n^{-c}$,

$$\frac{1}{N_1} \sum_{l=1}^{N_1} \mathbb{1}(e_l^\top y_t > \bar{b}_t) \leq C \cdot \big( \Phi(-\bar{b}_t - \varrho_t) + \rho_1 st \log(n) \big) + \frac{\rho_1}{\varrho_t^2} \cdot \|\Delta y_t\|_2^2. \tag{H.7}$$

Let us pick $\varrho_t = |\bar{b}_t|^{-1}$. Note that by assumption $\bar{b}_t < -2$, we have $-\bar{b}_t - \varrho_t > 3/2$ and by the Mills ratio inequality $(x^{-1} - x^{-3}) < \Phi(x)/p(x) < x^{-1} - x^{-3} + 3x^{-5}$ for $x > 0$, where $p(x) = \exp(-x^2/2)/\sqrt{2\pi}$ is the density for standard Gaussian distribution, we have

$$
\begin{aligned}
\Phi(-\bar{b}_t - \varrho_t) &\leq \frac{1 + 3(|\bar{b}_t| - \varrho_t)^{-4}}{\sqrt{2\pi} \cdot (|\bar{b}_t| - \varrho_t)} \cdot \exp\Big( -\frac{(|\bar{b}_t| - \varrho_t)^2}{2} \Big) \tag{H.8} \\
&\leq \frac{1 - |\bar{b}_t|^{-2}}{\sqrt{2\pi}|\bar{b}_t|} \cdot \exp\Big( -\frac{|\bar{b}_t|^2}{2} \Big) \cdot \frac{(1 + 3(|\bar{b}_t| - |\bar{b}_t|^{-1})^{-4})|\bar{b}_t|}{(|\bar{b}_t| - |\bar{b}_t|^{-1})(1 - |\bar{b}_t|^{-2})} \cdot \exp\Big( \frac{2 - |\bar{b}_t|^{-2}}{2} \Big) \leq C\Phi(-\bar{b}_t),
\end{aligned}
$$

where in the last inequality, we note that the highlighted ratios are bounded by a universal constant. Combining (H.7) and (H.8), we conclude the proof of Theorem F.3.

### H.2.3  Concentration for $\|E^\top \varphi(Ey_t^\star; b_t)\|_2^2$: Proof of Theorem F.4

When treating $\{\alpha_{\tau,t-1}\}_{\tau=-1}^{t-1}$ and $b_t$ to be deterministic, it follows that $y_t^\star \sim \mathcal{N}(0, 1)$. When conditioned on the good event $\mathcal{E}$, we always have $\|y_t^\star\|_\infty \leq (1 + c)\sqrt{2(t+1) \log(nt)}$. In the following, we use $y$ to replace $y_t^\star$ for notation simplicity. We use $y_j$ to denote the $j$-th coordinate of $y$. Let $\bar{b}_t = b_t + \kappa_0$.

**Good event on bounded Gaussian vectors.** Let $\mathcal{E}_0$ denote the event that **InitCond-2** is satisfied by the vectors $z_{-1:0}$. Throughout the proof, $C$ will denote a universal constant whose value may change from line to line. Fix a time step $t \geq 1$ (we omit the subscript t for notational simplicity). Define the "good event"

$$\mathcal{E}_1 = \Big\{ \max_{\tau=-1,0,\dots,t-1} \|z_\tau\|_\infty \leq (1 + c)\sqrt{2 \log(nt)} \Big\}.$$

Then, by Theorem J.2 (applied to the i.i.d. standard Gaussian vectors $z_{-1:t-1}$), we have

$$\mathbb{P}(\mathcal{E}_1) \geq 1 - (nt)^{-c} \geq 1 - n^{-c}.$$

**Good event on the activation sparsity.** Let us define $\mathcal{S}_j = \{l \in [N_1] : E_{l,j} \neq 0\}$. It holds that $|\mathcal{S}_j| \leq N_1 \rho_1$. In addition, we define event $\mathcal{E}_2$ as

$$\mathcal{E}_2 = \left\{ \begin{array}{c} \forall j \in [n-1] \\ \forall \alpha_{t-1} \in \mathbb{S}^t \\ \forall b_t \in \mathbb{R} \end{array}, \quad \sum_{l \in \mathcal{S}_j} \mathbb{1}(e_l^\top y + \bar{b}_t > 0) \leq C \cdot \left( \sum_{l \in \mathcal{S}_j} \Phi\left( -\frac{\bar{b}_t + E_{l,j} y_j}{\sqrt{1 - E_{l,j}^2}} \right) + |\mathcal{S}_j| \rho_2 st \log(n) \right) \right\}.$$

To show that $\mathcal{E}_2$ holds with high probability, let us define $\widetilde{E}$ as the submatrix of $E$ by keeping the rows indexed by $\mathcal{S}_j$ while removing the $j$-th column. We also normalize each row of $\widetilde{E}$ to have $\ell_2$-norm equal to one. We then have

1. $\|\widetilde{E}_{l,:}\|_2 = 1$, $\|\widetilde{E}_{l,:}\|_0 \leq s$ and $\|\widetilde{E}_{:,k}\|_0 \leq \sum_{l=1}^N \mathbb{1}(H_{l,j} \neq 0)\,\mathbb{1}(H_{l,k} \neq 0) \leq |\mathcal{S}_j| \rho_2$, where the last inequality holds by definition of $\rho_2$.

2. It holds that

$$|\mathcal{S}_j|^{-1} \sum_{l \in \mathcal{S}_j} \mathbb{1}(e_l^\top y + \bar{b}_t > 0) = |\mathcal{S}_j|^{-1} \sum_{l \in \mathcal{S}_j} \mathbb{1}\left( \widetilde{e}_l^\top y_{-j} + \frac{\bar{b}_t + E_{l,j} y_j}{\sqrt{1 - E_{l,j}^2}} > 0 \right),$$

where $\widetilde{e}_l$ is the $l$-th row of $\widetilde{E}$ and $y_{-j}$ is the vector $y$ with the $j$-th coordinate removed.

In the following, we use $z_{\tau,j}$ to denote the $j$-th coordinate of $z_\tau$, and $y_j = \sum_{\tau=-1}^{t-1} \alpha_{\tau,t-1} z_{\tau,j}$. We denote by $z_{\tau,-j}$ the vector $z_\tau$ with the $j$-th coordinate removed. Therefore, we can invoke Theorem H.4 with the configurations

$$G \leftarrow \widetilde{E}, \rho \leftarrow \rho_2, \vartheta \leftarrow (z_{-1,j}, z_{0,j}, \ldots, z_{t-1,j}),$$
$$\varsigma_l \leftarrow E_{l,j}/\sqrt{1 - E_{l,j}^2}, b_t \leftarrow -\bar{b}_t/\sqrt{1 - E_{l,j}^2} \text{ and } z_\tau \leftarrow z_{\tau,-j}$$

to obtain that with probability at least $1 - \delta/n$ over the randomness of standard Gaussian vectors $z_{-1:t-1,-j}$, and for fixed $t$, $\alpha_{t-1}$, $b_t$ and $\vartheta = (z_{-1,j}, z_{0,j}, \ldots, z_{t-1,j})$,

$$\sum_{l \in \mathcal{S}_j} \mathbb{1}(e_l^\top y + \bar{b}_t > 0) \leq C \cdot \left( \sum_{l \in \mathcal{S}_j} \Phi\left( -\frac{\bar{b}_t + E_{l,j} y_j}{\sqrt{1 - E_{l,j}^2}} \right) + |\mathcal{S}_j| \rho_2 s \log(n\delta^{-1}) \right)$$

$$\leq C \cdot \left( \sum_{l \in \mathcal{D}_j} \Phi\left( -\frac{\bar{b}_t + H_{l,j} y_j}{\sqrt{1 - H_{l,j}^2}} \right) + |\mathcal{S}_j| \rho_2 s \log(n\delta^{-1}) \right), \tag{H.9}$$

where $C$ is a universal constant independent of $t$, $\alpha_{t-1}$, $b_t$ and $\vartheta$. Here, in the last inequality, we define $\mathcal{D}_j = \{l \in [N] : H_{l,j} \neq 0\}$ as the set of rows in matrix $H$ that have nonzero $j$-th coordinate. Since $E$ is just a submatrix of $H$, adding more rows to the summation does not decrease the target value in the second inequality. Note that $z_{-1:t-1,-j}$ are independent of $z_{-1:t-1,j}$. We thus conclude that the above bound holds with probability at least $1 - \delta/n$ over the randomness of $z_{-1:t-1}$. Further applying the union bound for all $j \in [n-1]$, we conclude that (H.9) holds with probability at least $1 - \delta$ for all $j \in [n-1]$.

Note that the randomness discussed above is only over $z_{-1:t-1}$. We invoke a covering argument over $\alpha_{t-1} \in \mathbb{S}^t$ and $b_t \in \mathbb{R}$ similar to the proof of Theorem H.4. Since the argument is largely the same, we will not repeat it here. The size of the covering net is $n^{O(t+1)}$, and we can pick $\delta = n^{-c - O(t+1)}$ in (H.9), which gives us the upper bound in the definition of $\mathcal{E}_2$ with probability at least $1 - n^{-c}$.

**Refined upper bound on $y$.** We work with a fixed time step $t$ and aim to bound every coordinate $y_j$ for $j \in [n-1]$. Here, we recall definitions

$$y_j = \sum_{\tau=-1}^{t-1} \alpha_{\tau,t-1} z_{\tau,j}, \quad \beta_{t-1} = \sqrt{\sum_{\tau=1}^{t-1} \alpha_{\tau,t-1}^2}$$

where $\beta_{t-1}$ represents the $\ell_2$-norm of the component of $\bar{w}_{t-1}$ in the subspace orthogonal to $w_{-1:0}$. (Recall that the coefficients $\{\alpha_{\tau,t-1}\}_{\tau=1}^{t-1}$ arise when projecting $\bar{w}_{t-1}$ onto the orthonormal basis

$$\left\{ \bar{w}_{-1}, \frac{w_0^\perp}{\|w_0^\perp\|}, \frac{w_1^\perp}{\|w_1^\perp\|}, \ldots, \frac{w_{t-1}^\perp}{\|w_{t-1}^\perp\|} \right\}.$$

To leverage **InitCond-2**, we make a change of basis for the first two directions, namely, we replace

$$\left\{\bar{w}_{-1}, \frac{w_0^\perp}{\|w_0^\perp\|}\right\} \quad \text{with} \quad \{\bar{w}_0, \tilde{w}\}, \quad \text{where} \quad \tilde{w} = \alpha_{0,0}\,\bar{w}_{-1} - \alpha_{-1,0}\frac{w_0^\perp}{\|w_0^\perp\|}\,.$$

Note that $\tilde{w}$ is orthogonal to $\bar{w}_0$. The projection of $\bar{w}_{t-1}$ onto the direction $\tilde{w}$ satisfies

$$\left|\langle \bar{w}_{t-1},\, \tilde{w}\rangle\right| = \left|\alpha_{0,0}\,\alpha_{-1,t-1} - \alpha_{-1,0}\,\alpha_{0,t-1}\right| \leq |\alpha_{-1,t-1}| + |\alpha_{-1,0}|.$$

Since $\bar{w}_0$, $\tilde{w}$, and $\{w_\tau^\perp/\|w_\tau^\perp\|\}_{\tau=1}^{t-1}$ form an orthonormal basis, the component of $\bar{w}_{t-1}$ orthogonal to $\bar{w}_0$ is bounded by $\beta_{t-1} + |\alpha_{-1,t-1}| + |\alpha_{-1,0}|$. Moreover, we can also decompose $y_t$ into the new basis as follows:

$$y_t = \langle \bar{w}_0,\, \bar{w}_{t-1}\rangle\left(\alpha_{-1,0}\,z_{-1} + \alpha_{0,0}\,z_0\right) + \langle \tilde{w},\, \bar{w}_{t-1}\rangle\left(\alpha_{0,0}\,z_{-1} - \alpha_{-1,0}\,z_0\right) + \sum_{\tau=1}^{t-1}\alpha_{\tau,t-1}\,z_\tau$$

$$= \langle \bar{w}_0,\, \bar{w}_{t-1}\rangle\,y_1 + \langle \tilde{w},\, \bar{w}_{t-1}\rangle\left(\alpha_{0,0}\,z_{-1} - \alpha_{-1,0}\,z_0\right) + \sum_{\tau=1}^{t-1}\alpha_{\tau,t-1}\,z_\tau$$

Under **InitCond-2** the first term, $\langle \bar{w}_0,\, \bar{w}_{t-1}\rangle\,y_1$, is bounded by $\zeta_1$. Moreover, since both

$$\alpha_{0,0}\,z_{-1} - \alpha_{-1,0}\,z_0 \quad \text{and} \quad \{z_\tau\}_{\tau=1}^{t-1}$$

have their entries bounded by $2(1 + c)\sqrt{\log(nt)}$ on the good event $\mathcal{E}_1$, the contribution from the subspace orthogonal to $\bar{w}_0$ is bounded by

$$C\left(\beta_{t-1} + |\alpha_{-1,t-1}| + |\alpha_{-1,0}|\right)\sqrt{t\log(nt)}.$$

Thus, by the triangle inequality, for every coordinate $j$ we have under event $\mathcal{E}_0$ and $\mathcal{E}_1$ that

$$y_j \leq \zeta_1 + C\left(\beta_{t-1} + |\alpha_{-1,t-1}| + |\alpha_{-1,0}|\right)\sqrt{t\log(nt)} =: \zeta_t. \tag{H.10}$$

**Good event on the Bernstein concentration.** In the following, we will use another good event to control the upper bound in the definition of $\mathcal{E}_2$. Consider the function $\Phi(-(\bar{b}_t + xy_j)/\sqrt{1-x^2})^q$ for $q \geq 1$. We demonstrate that this function is Lipschitz continuous and monotonically increasing on the interval $x \in [0,1]$ if $y_j > -\bar{b}_t$ by taking the derivative with respect to $x$:

$$\frac{\mathrm{d}}{\mathrm{d}x}\Phi\left(-\frac{\bar{b}_t + xy_j}{\sqrt{1-x^2}}\right)^q = q\Phi\left(-\frac{\bar{b}_t + xy_j}{\sqrt{1-x^2}}\right)^{q-1} \cdot p\left(-\frac{\bar{b}_t + xy_j}{\sqrt{1-x^2}}\right) \cdot \frac{y_j - (-\bar{b}_t)x}{(1-x^2)^{3/2}} > 0.$$

Using the upper bound for $y$ specified in (H.10), we can define the *critical value* $\hbar_{q,t}$ as the smallest real number such that the following inequality holds:

$$\frac{1}{|\mathcal{D}_j|}\sum_{l\in\mathcal{D}_j}\Phi\left(-\frac{\bar{b}_t + H_{l,j}y_j}{\sqrt{1-H_{l,j}^2}}\right)^q \mathbb{1}(\mathcal{E}_0 \cap \mathcal{E}_1) \leq \max_{j\in[n]}\frac{1}{|\mathcal{D}_j|}\sum_{l\in\mathcal{D}_j}\Phi\left(-\frac{\bar{b}_t + H_{l,j}\zeta_t}{\sqrt{1-H_{l,j}^2}}\right)^q \leq \Phi\left(-\frac{\bar{b}_t + \hbar_{q,t}\zeta_t}{\sqrt{1-\hbar_{q,t}^2}}\right)^q.$$

$$\tag{H.11}$$

As we will only be using $q \in \{3,4\}$ in the following proof, we define the event $\mathcal{E}_3$ as the event such that for all $q \in \{3,4\}$, $\alpha_{t-1} \in \mathbb{S}^t$, $b_t \in \mathbb{R}$ and $j \in [n-1]$,

$$\mathcal{E}_3: \quad \sum_{j=1}^{n-1}\frac{1}{|\mathcal{D}_j|}\sum_{l\in\mathcal{D}_j}\Phi\left(-\frac{\bar{b}_t + H_{l,j}y_j}{\sqrt{1-H_{l,j}^2}}\right)^q \mathbb{1}(\mathcal{E}_0)\,\mathbb{1}(\mathcal{E}_1)$$

$$\leq C\cdot\left(\sum_{j=1}^{n-1}\frac{1}{|\mathcal{D}_j|}\sum_{l\in\mathcal{D}_j}\mathbb{E}\left[\Phi\left(-\frac{\bar{b}_t + H_{l,j}y_j}{\sqrt{1-H_{l,j}^2}}\right)^q\right] + \Phi\left(-\frac{\bar{b}_t + \hbar_{q,t}\zeta_t}{\sqrt{1-\hbar_{q,t}^2}}\right)^q t\log(n)\right),$$

where $C$ is a universal constant independent of $t$, $\alpha_{t-1}$, $b_t$ and $\zeta_t$. To show the event $\mathcal{E}_3$ holds with high probability, we can apply the Bernstein concentration inequality in Theorem J.3 for the bounded random variables

$$\frac{1}{|\mathcal{D}_j|}\sum_{l\in\mathcal{D}_j}\Phi\left(-\frac{\bar{b}_t + H_{l,j}y_j}{\sqrt{1-H_{l,j}^2}}\right)^q.$$

That is, for fixed $\alpha_{t-1}$, $b_t$ and with probability at least $1 - \delta$ over the randomness of $z_{-1:t-1}$, we have

$$\sum_{j=1}^{n-1} \frac{1}{|\mathcal{D}_j|} \sum_{l \in \mathcal{D}_j} \Phi\Big(-\frac{\bar{b}_t + H_{l,j}y_j}{\sqrt{1 - H_{l,j}^2}}\Big)^q \mathbb{1}(\mathcal{E}_0)\,\mathbb{1}(\mathcal{E}_1)$$

$$\leq \sqrt{2 \log \delta^{-1} \cdot \sum_{j=1}^{n-1} \mathbb{E}\Big[\Big(\frac{1}{|\mathcal{D}_j|} \sum_{l \in \mathcal{D}_j} \Phi\Big(-\frac{\bar{b}_t + H_{l,j}y_j}{\sqrt{1 - H_{l,j}^2}}\Big)^q\Big)^2 \mathbb{1}(\mathcal{E}_0 \cap \mathcal{E}_1)\Big]}$$

$$+ \sum_{j=1}^{n-1} \frac{1}{|\mathcal{D}_j|} \sum_{l \in \mathcal{D}_j} \mathbb{E}\Big[\Phi\Big(-\frac{\bar{b}_t + H_{l,j}y_j}{\sqrt{1 - H_{l,j}^2}}\Big)^q\Big] + \frac{1}{3}\Phi\Big(-\frac{\bar{b}_t + \hbar_{q,t}\zeta_t}{\sqrt{1 - \hbar_{q,t}^2}}\Big)^q \log(\delta^{-1}).$$

Moreover, we have for the second moment term that

$$\mathbb{E}\Big[\Big(\frac{1}{|\mathcal{D}_j|} \sum_{l \in \mathcal{D}_j} \Phi\Big(-\frac{\bar{b}_t + H_{l,j}y_j}{\sqrt{1 - H_{l,j}^2}}\Big)^q\Big)^2 \mathbb{1}(\mathcal{E}_0 \cap \mathcal{E}_1)\Big]$$

$$\leq \sum_{j=1}^{n-1} \frac{1}{|\mathcal{D}_j|^2} \cdot \sum_{l \in \mathcal{D}_j} \mathbb{E}\Big[\Phi\Big(-\frac{\bar{b}_t + H_{l,j}y_j}{\sqrt{1 - H_{l,j}^2}}\Big)^q \cdot \mathbb{1}(\mathcal{E}_0 \cap \mathcal{E}_1)\Big] \cdot \sum_{l' \in \mathcal{D}_j} \Phi\Big(-\frac{\bar{b}_t + H_{l',j}\zeta_t}{\sqrt{1 - H_{l',j'}^2}}\Big)^q$$

$$\leq \sum_{j=1}^{n-1} \Big(\frac{1}{|\mathcal{D}_j|} \sum_{l \in \mathcal{D}_j} \mathbb{E}\Big[\Phi\Big(-\frac{\bar{b}_t + H_{l,j}y_j}{\sqrt{1 - H_{l,j}^2}}\Big)^q\Big]\Big) \cdot \Phi\Big(-\frac{\bar{b}_t + \hbar_{q,t}\zeta_t}{\sqrt{1 - \hbar_{q,t}^2}}\Big)^q,$$

where in the first inequality, we invoke the upper bound in (H.11). Using the fact that $\sqrt{a \cdot b} \leq a + b$ for $a, b \geq 0$, we derive that

$$\sum_{j=1}^{n-1} \frac{1}{|\mathcal{D}_j|} \sum_{l \in \mathcal{D}_j} \Phi\Big(-\frac{\bar{b}_t + H_{l,j}y_j}{\sqrt{1 - H_{l,j}^2}}\Big)^q \mathbb{1}(\mathcal{E}_0)\,\mathbb{1}(\mathcal{E}_1)$$

$$\leq C \cdot \Big(\sum_{j=1}^{n-1} \frac{1}{|\mathcal{D}_j|} \sum_{l \in \mathcal{D}_j} \mathbb{E}\Big[\Phi\Big(-\frac{\bar{b}_t + H_{l,j}y_j}{\sqrt{1 - H_{l,j}^2}}\Big)^q\Big] + \frac{1}{3}\Phi\Big(-\frac{\bar{b}_t + \hbar_{q,t}\zeta_t}{\sqrt{1 - \hbar_{q,t}^2}}\Big)^q \log(\delta^{-1})\Big) \cdot$$

$$\tag{H.12}$$

Now, we apply the covering argument over $\alpha_{t-1} \in \mathbb{S}^t$ and $b_t \in \mathbb{R}$ similar to the proof of Theorem H.4. The size of the covering net is $n^{O(t+1)}$, and we can pick $\delta = n^{-c-O(t+1)}$ in (H.12), which gives us the upper bound in the definition of $\mathcal{E}_3$ with $\mathbb{P}(\mathcal{E}_3) \geq 1 - n^{-c}$.

**The Perturbed Variance.** Given the good events $\mathcal{E}_0, \mathcal{E}_1, \mathcal{E}_2$, and $\mathcal{E}_3$, we define

$$Z = \frac{1}{N_1^2} \sum_{l,l'=1}^{N_1} Z_{l,l'}, \quad \text{where} \quad Z_{l,l'} = \varphi(e_l^\top y; b_t) \cdot \varphi(e_{l'}^\top y; b_t) \cdot \langle e_l, e_{l'} \rangle \cdot \mathbb{1}(\mathcal{E}_0 \cap \mathcal{E}_1 \cap \mathcal{E}_2 \cap \mathcal{E}_3).$$

$$\tag{H.13}$$

For concentration of $Z$, we consider the following Perturbed Variance (PV) defined as

$$V := \mathbb{E}\Big[\sum_{i=1}^{n-1} (Z - Z^{(i)})^2 \,\Big|\, y\Big],$$

where the perturbed term $Z^{(i)}$ is defined as follows:

$$Z^{(i)} = \frac{1}{N_1^2} \sum_{l,l'=1}^{N_1} Z_{l,l'}^{(i)}, \quad \text{where} \quad Z_{l,l'}^{(i)} = \varphi(e_l^\top y^{(i)}; b_t) \cdot \varphi(e_{l'}^\top y^{(i)}; b_t) \cdot \langle e_l, e_{l'} \rangle \cdot \mathbb{1}(\cap_{\iota=0}^3 \mathcal{E}_\iota^{(i)}).$$

Here, $y^{(i)} = \sum_{\tau=-1}^{t-1} \alpha_{\tau,t-1} z_\tau^{(i)}$ and $z_\tau^{(i)}$ is given by replacing the $i$-th coordinate of $z_\tau$ by an independent $\mathcal{N}(0,1)$ random variable. In addition, the good events $\{\mathcal{E}_\iota^{(i)}\}_{\iota=0}^3$ are defined similarly to $\mathcal{E}_\iota$, but using $z_{-1:t-1}^{(i)}$ instead of $z_{-1:t-1}$. We begin by noting the elementary inequality $(a-b)^2 \le 2a^2 + 2b^2$. Thus, we obtain

$$V \le \frac{2}{N_1^4} \underbrace{\mathbb{E}\left[\sum_{i=1}^{n-1}\left(\sum_{l,l'=1}^{N_1} Z_{l,l'} \, \mathbb{1}\left\{E_{l,i} \neq 0 \vee E_{l',i} \neq 0\right\}\right)^2 \Big| y\right]}_{(I)}$$

$$+ \frac{2}{N_1^4} \underbrace{\mathbb{E}\left[\sum_{i=1}^{n-1}\left(\sum_{l,l'=1}^{N_1} Z_{l,l'}^{(i)} \, \mathbb{1}\left\{E_{l,i} \neq 0 \vee E_{l',i} \neq 0\right\}\right)^2 \Big| y\right]}_{(II)},$$

where the upper bound is obtained by the following reasoning. For each perturbed quantity $Z^{(i)}$, we have

$$Z - Z^{(i)} = \frac{1}{N_1^2} \sum_{l,l'=1}^{N_1} \left(Z_{l,l'} - Z_{l,l'}^{(i)}\right) \cdot \mathbb{1}\left\{E_{l,i} \neq 0 \vee E_{l',i} \neq 0\right\}.$$

Note that the difference $Z_{l,l'} - Z_{l,l'}^{(i)}$ is nonzero only when at least one of the vectors $e_l$ or $e_{l'}$ has a nonzero $i$th coordinate. The two terms (I) and (II) correspond to the contributions from the original and the perturbed parts, respectively. In what follows we focus on an upper bound for the term (I); the term (II) can be estimated by a completely analogous argument.

**Controlling Term (I).** Due to the $L$-Lipschitz continuity of $\varphi$ with $L = \gamma_2 + |b_t|\gamma_1$, on the good event $\mathcal{E}_1$, the absolute value of $\varphi(e_l^\top y; b_t)$ is bounded by $|\varphi(e_l^\top y; b_t)| \le |\varphi(0; b_t)| + L \cdot |e_l^\top y|$, which can be further bounded as

$$|\varphi(e_l^\top y; b_t)| \le (d \vee n)^{-c_0} + L\sqrt{s} \cdot \|y\|_\infty \le C L \sqrt{t s \log(n)} := B_t,$$

where we used that $t \le n^c$, $\|e_l\|_1 \le \sqrt{s}$, and that $(d \vee n)^{-c_0} \le 1 \le L\sqrt{t s \log(n)}$. Note that the same bound holds for $\varphi(e_l^\top y^{(i)}; b_t)$ on the corresponding good event $\mathcal{E}_1^{(i)}$. For $Z_{l,l'}$ defined in (H.13), we first upper bound $\varphi(e_l^\top y; b_t) \cdot \varphi(e_{l'}^\top y; b_t)$ by

$$\varphi(e_l^\top y; b_t) \cdot \varphi(e_{l'}^\top y; b_t) \le B_t^2 \mathbb{1}(e_l^\top y + \bar{b}_t > 0) \mathbb{1}(e_{l'}^\top y + \bar{b}_t > 0) + 2B_t(d \vee n)^{-c_0} + (d \vee n)^{-2c_0},$$

where we recall that if $e_l^\top y + \bar{b}_t > 0$, the neuron is deemed activated and its output is bounded above by $B_t$. Otherwise, by Definition B.3, the activation is bounded by $(d \vee n)^{-c_0}$. Note that the term $(d \vee n)^{-c_0} B_t^{-1}$ can be made arbitrarily small as $c_0$ is some large constant no less than 4. Therefore, we just keep the first term above. Secondly, the inner product $\langle e_l, e_{l'} \rangle$ is upper bounded by $\sum_{j=1}^{n-1} \mathbb{1}(E_{l,j} \neq 0) \cdot \mathbb{1}(E_{l',j} \neq 0)$ as $\|E\|_\infty \le 1$. Lastly, the indicator $\mathbb{1}(E_{l,i} \neq 0 \vee E_{l',i} \neq 0)$ can be upper bounded by $\mathbb{1}(E_{l,i} \neq 0) + \mathbb{1}(E_{l',i} \neq 0)$. For (I), we then have

$$(I) \le \frac{C B_t^4}{N_1^4} \cdot \mathbb{E}\left[\sum_{i=1}^{n-1}\left(\sum_{j=1}^{n-1}\sum_{l,l'=1}^{N_1} \mathbb{1}(e_l^\top y + \bar{b}_t > 0) \, \mathbb{1}(e_{l'}^\top y + \bar{b}_t > 0)\right.\right.$$

$$\left.\left. \cdot \left(\mathbb{1}\{E_{l,i} \neq 0\} + \mathbb{1}\{E_{l',i} \neq 0\}\right) \mathbb{1}\{E_{l,j} \neq 0\} \mathbb{1}\{E_{l',j} \neq 0\}\right)^2 \cdot \mathbb{1}\left(\cap_{\iota=0}^3 \mathcal{E}_\iota\right) \Big| y\right].$$

Due to symmetry in the indices $l$ and $l'$, we can multiply the constant factor $C$ by 2 and obtain

$$
\text{(I)} \leq \frac{CB_t^4}{N_1^4} \cdot \mathbb{E}\Bigg[\sum_{i=1}^{n-1}\Big(\sum_{j=1}^{n-1}\sum_{l=1}^{N_1} \mathbb{1}(e_l^\top y + \bar{b}_t > 0) \cdot \mathbb{1}(E_{l,i} \neq 0) \cdot \mathbb{1}(E_{l,j} \neq 0)
$$
$$
\cdot \sum_{l'=1}^{N_1} \mathbb{1}(E_{l',j} \neq 0) \cdot \mathbb{1}(e_{l'}^\top y + \bar{b}_t > 0)\Big)^2 \cdot \mathbb{1}(\cap_{\iota=0}^3 \mathcal{E}_\iota)\,\Big|\, y\Bigg]
$$
$$
\leq \frac{CB_t^4}{N_1^4} \cdot \mathbb{E}\Bigg[\sum_{i=1}^{n-1}\sum_{j=1}^{n-1}\Big(\sum_{l=1}^{N_1}\mathbb{1}(e_l^\top y + \bar{b}_t > 0)\cdot\mathbb{1}(E_{l,i}\neq 0)\cdot\mathbb{1}(E_{l,j}\neq 0)\Big)^2
$$
$$
\cdot \Big(\sum_{l'=1}^{N_1}\mathbb{1}(E_{l',j}\neq 0)\cdot\mathbb{1}(e_{l'}^\top y + \bar{b}_t > 0)\Big)^2 \cdot \mathbb{1}(\cap_{\iota=0}^3 \mathcal{E}_\iota)\,\Big|\, y\Bigg],
$$

where the last inequality holds by the Cauchy-Schwarz inequality. Note that for $i \neq j$:

$$
\sum_{l=1}^{N_1} \mathbb{1}(E_{l,i}\neq 0)\cdot\mathbb{1}(E_{l,j}\neq 0) \leq \sum_{l=1}^{N}\mathbb{1}(H_{l,i}\neq 0)\cdot\mathbb{1}(H_{l,j}\neq 0) \leq \rho_1\rho_2 N.
$$

Using $\rho_1\rho_2 N$ to substitue one $\sum_{l=1}^{N_1}\mathbb{1}(e_l^\top y + \bar{b}_t > 0)\cdot\mathbb{1}(E_{l,i}\neq 0)\cdot\mathbb{1}(E_{l,j}\neq 0)$ for $i \neq j$, we obtain

$$
\text{(I)} \leq \frac{B_t^4 N \rho_1\rho_2}{N_1^4} \cdot \mathbb{E}\Bigg[\sum_{j=1}^{n-1}\sum_{i\neq j}\sum_{l=1}^{N_1}\mathbb{1}(e_l^\top y + \bar{b}_t > 0)\cdot\mathbb{1}(E_{l,i}\neq 0)\cdot\mathbb{1}(E_{l,j}\neq 0)
$$
$$
\cdot\Big(\sum_{l'=1}^{N_1}\mathbb{1}(E_{l',j}\neq 0)\cdot\mathbb{1}(e_l^\top y + \bar{b}_t > 0)\Big)^2 \cdot \mathbb{1}(\cap_{\iota=0}^3 \mathcal{E}_\iota)\,\Big|\, y\Bigg]
$$
$$
+ \frac{B_t^4}{N_1^4}\cdot\mathbb{E}\Bigg[\sum_{j=1}^{n-1}\Big(\sum_{l=1}^{N_1}\mathbb{1}(e_l^\top y + \bar{b}_t > 0)\cdot\mathbb{1}(E_{l,j}\neq 0)\Big)^2
$$
$$
\cdot\Big(\sum_{l'=1}^{N_1}\mathbb{1}(E_{l',j}\neq 0)\cdot\mathbb{1}(e_{l'}^\top y + \bar{b}_t > 0)\Big)^2 \cdot \mathbb{1}(\cap_{\iota=0}^3 \mathcal{E}_\iota)\,\Big|\, y\Bigg].
$$

Rearranging the order of summation and using the fact that $\sum_{i\neq j}\mathbb{1}(E_{l,i}\neq 0) \leq s$ for any fixed $j$, we can further simplify the terms as

$$
\text{(I)} \leq \frac{2B_t^4\rho_1\rho_2 s}{N_1^3}\cdot\mathbb{E}\Bigg[\sum_{j=1}^{n-1}\Big(\sum_{l=1}^{N_1}\mathbb{1}(e_l^\top y + \bar{b}_t > 0)\cdot\mathbb{1}(E_{l,j}\neq 0)\Big)^3 \cdot\mathbb{1}(\cap_{\iota=0}^3 \mathcal{E}_\iota)\,\Big|\, y\Bigg]
$$
$$
+ \frac{2B_t^4}{N_1^4}\cdot\mathbb{E}\Bigg[\sum_{j=1}^{n-1}\Big(\sum_{l=1}^{N_1}\mathbb{1}(e_l^\top y + \bar{b}_t > 0)\cdot\mathbb{1}(E_{l,j}\neq 0)\Big)^4 \cdot\mathbb{1}(\cap_{\iota=0}^3 \mathcal{E}_\iota)\,\Big|\, y\Bigg]. \qquad \text{(H.14)}
$$

Observe that the above two terms share a common structure. We define the common structure as

$$
\text{(III)} := \frac{1}{N_1^q}\cdot\mathbb{E}\Bigg[\sum_{j=1}^{n-1}\Big(\sum_{l=1}^{N_1}\mathbb{1}(e_l^\top y + \bar{b}_t > 0)\cdot\mathbb{1}(E_{l,j}\neq 0)\Big)^q \cdot\mathbb{1}(\cap_{\iota=0}^3 \mathcal{E}_\iota)\,\Big|\, y\Bigg],
$$

where $q \in \{3, 4\}$. Recall the definition $\mathcal{S}_j = \{l \in [N_1] : E_{l,j} \neq 0\}$. It holds that $|\mathcal{S}_j| \leq N_1\rho_1$. We aim to control

$$
\sum_{l=1}^{N_1}\mathbb{1}(e_l^\top y + \bar{b}_t > 0)\cdot\mathbb{1}(E_{l,j}\neq 0) = |\mathcal{S}_j|^{-1}\sum_{l\in\mathcal{S}_j}\mathbb{1}(e_l^\top y + \bar{b}_t > 0)
$$

in the following. By the definition of the good event $\mathcal{E}_2$, we have

$$
\text{(III)} \leq \frac{C}{N_1^q} \cdot \sum_{j=1}^{n-1} \left( \left( \sum_{l \in \mathcal{D}_j} \Phi\left(-\frac{\overline{b}_t + H_{l,j} y_j}{\sqrt{1 - H_{l,j}^2}}\right) + |\mathcal{S}_j| \rho_2 st \log(n) \right)^q \mathbb{1}(\cap_{\iota=0}^3 \mathcal{E}_\iota) \right)
$$

$$
\leq \frac{2^{q-1} C}{N_1^q} \cdot \sum_{j=1}^{n-1} |\mathcal{D}_j|^q \cdot \left( \frac{1}{|\mathcal{D}_j|} \sum_{l \in \mathcal{D}_j} \Phi\left(-\frac{\overline{b}_t + H_{l,j} y_j}{\sqrt{1 - H_{l,j}^2}}\right)^q \mathbb{1}(\cap_{\iota=0}^3 \mathcal{E}_\iota) + \left(\rho_2 st \log(n)\right)^q \right)
$$

$$
\leq C \rho_1^q \cdot \left( \sum_{j=1}^{n-1} \frac{1}{|\mathcal{D}_j|} \sum_{l \in \mathcal{D}_j} \Phi\left(-\frac{\overline{b}_t + H_{l,j} y_j}{\sqrt{1 - H_{l,j}^2}}\right)^q \mathbb{1}(\cap_{\iota=0}^3 \mathcal{E}_\iota) + n\left(\rho_2 st \log(n)\right)^q \right). \tag{H.15}
$$

where we use the Hölder's inequality for the second line, and in the last line, we absorb the constant factor $2^{q-1}$ into the universal constant $C$ and use the fact that $|\mathcal{S}_j| \leq |\mathcal{D}_j| \leq N\rho_1 \leq N_1\rho_1/(1 - \rho_1) \leq C_1 N_1 \rho_1$ for all $j \in [n-1]$, where we also absorb the factor $C_1^q$ into the universal constant $C$. By the definition of the good event $\mathcal{E}_3$, it holds that

$$
\sum_{j=1}^{n-1} \frac{1}{|\mathcal{D}_j|} \sum_{l \in \mathcal{D}_j} \Phi\left(-\frac{\overline{b}_t + H_{l,j} y_j}{\sqrt{1 - H_{l,j}^2}}\right)^q \mathbb{1}(\cap_{\iota=0}^3 \mathcal{E}_\iota)
$$

$$
\leq C \cdot \left( \sum_{j=1}^{n-1} \frac{1}{|\mathcal{D}_j|} \sum_{l \in \mathcal{D}_j} \mathbb{E}\left[\Phi\left(-\frac{\overline{b}_t + H_{l,j} y_j}{\sqrt{1 - H_{l,j}^2}}\right)^q\right] + \Phi\left(-\frac{\overline{b}_t + \hbar_{q,t}\zeta_t}{\sqrt{1 - \hbar_{q,t}^2}}\right)^q t \log(n) \right). \tag{H.16}
$$

To evaluate the expectation term, we use the Mills ratio $\Phi(x) \leq C p(x)$ for some universal constant $C > 0$, $x > 0$ and $p(x) = \exp(-x^2/2)/\sqrt{2\pi}$ to obtain

$$
\mathbb{E}\left[\Phi\left(-\frac{\overline{b}_t + H_{l,j} y_j}{\sqrt{1 - H_{l,j}^2}}\right)^q\right] \leq C \cdot \mathbb{E}\left[\exp\left(-\frac{q(\overline{b}_t + H_{l,j} y_j)^2}{2(1 - H_{l,j}^2)}\right) \mathbb{1}(\overline{b}_t + H_{l,j} y_j \leq 0)\right] + \mathbb{P}(\overline{b}_t + H_{l,j} y_j > 0)
$$

$$
\leq C \cdot \mathbb{E}\left[\exp\left(-\frac{q(\overline{b}_t + H_{l,j} y_j)^2}{2(1 - H_{l,j}^2)}\right)\right] + \Phi\left(-\frac{\overline{b}_t}{H_{l,j}}\right)
$$

$$
= C \sqrt{\frac{1 - H_{l,j}^2}{1 + (q-1)H_{l,j}^2}} \cdot \exp\left(-\frac{\overline{b}_t^2}{2(\frac{q-1}{q} H_{l,j}^2 + \frac{1}{q})}\right) + \Phi\left(-\frac{\overline{b}_t}{H_{l,j}}\right), \tag{H.17}
$$

where the third equality holds by direct algebraic calculation for Gaussian integral. By the Mills ratio $\Phi(x)/p(x) \geq x^{-1} - x^{-3} = C x^{-1}$ for $x \gg 1$, and also the fact that $H_{l,j} \in [0,1]$, we conclude that the right-hand side of (H.17) is bounded by

$$
\mathbb{E}\left[\Phi\left(-\frac{\overline{b}_t + H_{l,j} y_j}{\sqrt{1 - H_{l,j}^2}}\right)^q\right] \leq C |\overline{b}_t| \Phi\left(\frac{-\overline{b}_t}{\sqrt{\frac{q-1}{q} H_{l,j}^2 + \frac{1}{q}}}\right). \tag{H.18}
$$

Similar to the previous argument, we also have $\Phi\left(-\frac{\overline{b}_t}{\sqrt{\frac{q-1}{q} x^2 + \frac{1}{q}}}\right)$ as a non-decreasing function of $x$ for $x \in [0,1]$ by checking the derivative. We define $\hbar_{q,\star}$ as the smallest real number such that the following inequality holds:

$$
\sum_{j=1}^{n} \frac{1}{|\mathcal{D}_j|} \sum_{l \in \mathcal{D}_j} \Phi\left(\frac{-\overline{b}_t}{\sqrt{\frac{q-1}{q} H_{l,j}^2 + \frac{1}{q}}}\right) \leq n \cdot \Phi\left(\frac{-\overline{b}_t}{\sqrt{\frac{q-1}{q} \hbar_{q,\star}^2 + \frac{1}{q}}}\right). \tag{H.19}
$$

Plugging (H.18) and (H.19) into (H.16), we have that

$$
\sum_{j=1}^{n-1} \frac{1}{|\mathcal{D}_j|} \sum_{l \in \mathcal{D}_j} \Phi\left(-\frac{\overline{b}_t + H_{l,j} y_j}{\sqrt{1 - H_{l,j}^2}}\right)^q \mathbb{1}(\cap_{\iota=0}^3 \mathcal{E}_\iota)
$$

$$
\leq C \cdot \left( n|\overline{b}_t| \Phi\left(\frac{-\overline{b}_t}{\sqrt{\frac{q-1}{q} \hbar_{q,\star}^2 + \frac{1}{q}}}\right) + \Phi\left(-\frac{\overline{b}_t + \hbar_{q,t}\zeta_t}{\sqrt{1 - \hbar_{q,t}^2}}\right)^q t \log(n) \right). \tag{H.20}
$$

Combining (H.15) and (H.20), we obtain

$$\text{(III)} \le C\rho_1^q \cdot \left( n\,|\bar{b}_t|\,\Phi\left( \frac{-\bar{b}_t}{\sqrt{\frac{q-1}{q}\,\hbar_{q,\star}^2 + \frac{1}{q}}} \right) + \Phi\left( -\frac{\bar{b}_t + \hbar_{q,t}\zeta_t}{\sqrt{1 - \hbar_{q,t}^2}} \right)^q t\log(n) + n\big(\rho_2\,s\,t\log(n)\big)^q \right).$$

Note that we always have $\hbar_{q,\star} \le 1$ and $\hbar_{q,t} \le 1$ for $t \ge 1$. As both $\Phi\left( \frac{-\bar{b}_t}{\sqrt{\frac{q-1}{q}x^2 + \frac{1}{q}}} \right)$ and $\Phi\left( -\frac{\bar{b}_t + x\zeta_t}{\sqrt{1-x^2}} \right)^q$ (when $\zeta_t > -\bar{b}_t$) are non-decreasing functions with respect to $x$, for the first term in the right-hand side of (H.14), we take $q = 3$ and $\hbar_{q,t} = 1$ to have the following upper bound:

$$\frac{2B_t^4\rho_1\rho_2 s}{N_1^3} \cdot \mathbb{E}\left[ \sum_{j=1}^{n-1}\left( \sum_{l=1}^{N_1} \mathbb{1}(e_l^\top y + \bar{b}_t > 0)\cdot\mathbb{1}(E_{l,j}\neq 0) \right)^3 \cdot \mathbb{1}(\cap_{\iota=0}^3 \mathcal{E}_\iota)\,\Big|\,y \right]$$

$$\le CB_t^4\rho_1^4\rho_2 s \cdot \left( n|\bar{b}_t|\Phi\left( \frac{-\bar{b}_t}{\sqrt{\frac{2}{3}\hbar_{3,\star}^2 + \frac{1}{3}}} \right) + t\log(n) + n(\rho_2 st\log(n))^3 \right). \qquad \text{(H.21)}$$

For the second term on the right-hand side of (H.14), we take $q = 4$ and obtain

$$\frac{2B_t^4}{N_1^4} \cdot \mathbb{E}\left[ \sum_{j=1}^{n-1}\left( \sum_{l=1}^{N_1} \mathbb{1}(e_l^\top y + \bar{b}_t > 0)\cdot\mathbb{1}(E_{l,j}\neq 0) \right)^4 \cdot \mathbb{1}(\cap_{\iota=0}^3 \mathcal{E}_\iota)\,\Big|\,y \right]$$

$$\le CB_t^4\rho_1^4 \cdot \left( n\,|\bar{b}_t|\,\Phi\left( \frac{-\bar{b}_t}{\sqrt{\frac{3}{4}\,\hbar_{4,\star}^2 + \frac{1}{4}}} \right) + \Phi\left( -\frac{\bar{b}_t + \hbar_{4,t}\zeta_t}{\sqrt{1 - \hbar_{4,t}^2}} \right)^4 t\log(n) + n\big(\rho_2\,s\,t\log(n)\big)^4 \right).$$

$$\text{(H.22)}$$

We conclude by combining (H.21) and (H.22) that

$$\text{(I)} \le CB_t^4\rho_1^4 \cdot \left( n\,|\bar{b}_t|\,\Phi\left( \frac{-\bar{b}_t}{\sqrt{\frac{3}{4}\,\hbar_{4,\star}^2 + \frac{1}{4}}} \right) + \rho_2 sn|\bar{b}_t|\Phi\left( \frac{-\bar{b}_t}{\sqrt{\frac{2}{3}\hbar_{3,\star}^2 + \frac{1}{3}}} \right) \right.$$

$$\left. + \left( \Phi\left( -\frac{\bar{b}_t + \hbar_{4,t}\zeta_t}{\sqrt{1 - \hbar_{4,t}^2}} \right)^4 + \rho_2 s \right)t\log(n) + n\big(\rho_2\,s\,t\log(n)\big)^4 \right) =: V_0.$$

Similarly, (II) can be bounded by $V_0$. We are now ready to invoke Theorem J.9. Since $V \le 2V_0$ with probability 1, the final bound for $|Z - \mathbb{E}[Z]|$ is then given by

$$|Z - \mathbb{E}[Z]| \le C\sqrt{V_0 \log(\delta^{-1})},$$

where the inequality holds with probability at least $1 - \delta$ over the randomness of standard Gaussian vectors $z_{-1:T}$. Plugging in the formula for $V_0$, we obtain the following upper bound

$$|Z - \mathbb{E}[Z]| \le CB_t^2\rho_1^2 \cdot \left( n\,|\bar{b}_t|\,\Phi\left( \frac{-\bar{b}_t}{\sqrt{\frac{3}{4}\,\hbar_{4,\star}^2 + \frac{1}{4}}} \right) + \rho_2 sn|\bar{b}_t|\Phi\left( \frac{-\bar{b}_t}{\sqrt{\frac{2}{3}\hbar_{3,\star}^2 + \frac{1}{3}}} \right) \right.$$

$$\left. + \left( \Phi\left( -\frac{\bar{b}_t + \hbar_{4,t}\zeta_t}{\sqrt{1 - \hbar_{4,t}^2}} \right)^4 + \rho_2 s \right)t\log(n) + n\big(\rho_2\,s\,t\log(n)\big)^4 \right)^{1/2} \cdot \log\delta^{-1}$$

with probability $1 - \delta$. For notational convenience, we define $\mathcal{K}_t$ as the $1/4$ power of each term inside the bracket in the above equation (see (F.9) for the definition). The fluctuation of $Z$ is controlled by

$$|Z - \mathbb{E}[Z]| \le CL^2\rho_1^2 ts\log n \cdot \mathcal{K}_t^2 \cdot \log\delta^{-1},$$

where we plug in the definition $B_t = L\sqrt{ts\log n}$ and $L = \gamma_2 + |b_t|\gamma_1$ is the Lipschitz constant for the activation function $\varphi$.

**Expectation $\mathbb{E}[Z]$.** For $\mathbb{E}[\|E^\top \varphi(Ey_t^\star; b_t)\|_2^2]$, we have

$$\frac{1}{N_1^2} \cdot \mathbb{E}[\|E^\top \varphi(Ey_t^\star; b_t)\|_2^2] \leq \frac{1}{N_1^2} \sum_{l,l'=1}^N \mathbb{E}\big[|\varphi(\widetilde{h}_l^\top y; b_t) \cdot \varphi(\widetilde{h}_{l'}^\top y; b_t)|\big] \cdot \langle \widetilde{h}_l, \widetilde{h}_{l'} \rangle$$

$$\leq C_1^2 \cdot \widehat{\mathbb{E}}_{l,l'}\Big[\mathbb{E}\big[|\varphi(\widetilde{h}_l^\top y; b_t) \cdot \varphi(\widetilde{h}_{l'}^\top y; b_t)|\big] \cdot \langle \widetilde{h}_l, \widetilde{h}_{l'} \rangle\Big],$$

where in the first inequality, we obtain the upper bound by also adding the rows of $F$ that are not contained in the submatrix $E$ to the sum. Here, we use the notation

$$\widetilde{h}_l = (H_{l,1}, \ldots, H_{l,i-1}, H_{l,i+1}, \ldots, H_{l,n-1})^\top$$

to denote the $l$-th row of $H$ with the $i$-th entry removed. This structure comes from the definition (E.3) where we decompose the matrix $H$ into submatrices $E$, $F$ and the column vector $\theta$ as the non-zero entries in $H_{.,i}$ if the feature of interest is the $i$-th feature. In the second inequality, we use the fact that $N/N_1 \leq C_1$, and define $\widehat{\mathbb{E}}_{l,l'}$ as the empirical expectation over $l, l' \in [N]^2$. Invoking Theorem F.5 with $L = \gamma_2 + |b_t|\gamma_1$, $\overline{b} = \overline{b}_t = b_t + \kappa_0$, we conclude that

$$\widehat{\mathbb{E}}_{l,l'}\Big[\mathbb{E}\big[|\varphi(\widetilde{h}_l^\top y; b_t) \cdot \varphi(\widetilde{h}_{l'}^\top y; b_t)|\big] \cdot \langle \widetilde{h}_l, \widetilde{h}_{l'} \rangle\Big]$$

$$\leq CL \cdot (n \vee d)^{-c_0} + CL^2 \cdot \Phi(|\overline{b}_t|) \cdot \widehat{\mathbb{E}}_{l,l'}\left[\Phi\Big(|\overline{b}_t|\sqrt{\frac{1 - \langle \widetilde{h}_l, \widetilde{h}_{l'} \rangle}{1 + \langle \widetilde{h}_l, \widetilde{h}_{l'} \rangle}}\Big)\langle \widetilde{h}_l, \widetilde{h}_{l'} \rangle\right]$$

$$\leq CL \cdot (n \vee d)^{-c_0} + CL^2 \cdot \Phi(|\overline{b}_t|) \cdot \widehat{\mathbb{E}}_{l,l'}\left[\Phi\Big(|\overline{b}_t|\sqrt{\frac{1 - \langle h_l, h_{l'} \rangle}{1 + \langle h_l, h_{l'} \rangle}}\Big)\langle h_l, h_{l'} \rangle\right],$$

where in the first inequality, we directly apply Theorem F.5 to the expectation term, and in the second inequality, we use the fact that $\langle \widetilde{h}_l, \widetilde{h}_{l'} \rangle \leq \langle h_l, h_{l'} \rangle$ for $l, l' \in [N_1]$ and the fact that the term inside the expectation is non-decreasing when increasing the value of $\langle \widetilde{h}_l, \widetilde{h}_{l'} \rangle$. Just as before, since $c_0 > 4$ is large enough, the first term is negligible, and we can absorb it into the constant $C$ and focus on the second term:

$$\frac{1}{N_1^2} \cdot \mathbb{E}\big[\|E^\top \varphi(Ey_t^\star; b_t)\|_2^2\big] \leq CL^2 \cdot \Phi(|\overline{b}_t|) \cdot \widehat{\mathbb{E}}_{l,l'}\left[\Phi\Big(|\overline{b}_t|\sqrt{\frac{1 - \langle h_l, h_{l'} \rangle}{1 + \langle h_l, h_{l'} \rangle}}\Big)\langle h_l, h_{l'} \rangle\right].$$

Since $\|E^\top \varphi(Ey_t^\star; b_t)\|_2^2$ is non-negative, the same upper bound applies to $\mathbb{E}[Z]$, where $Z$ includes the indicator condition $\mathbb{1}(\cap_{\iota=0}^3 \mathcal{E}_\iota)$.

Finally, we plug in $\delta = n^{-c}$ to conclude that with probability at least $1 - n^{-c}$ it holds that

$$\frac{1}{N_1^2}\|E^\top \varphi(Ey_t^\star; b_t)\|_2^2 \cdot \mathbb{1}(\mathcal{E}_0) \cdot \mathbb{1}(\cap_{\iota=0}^3 \mathcal{E}_\iota) \leq CL^2 \cdot \rho_1^2 st^2 (\log n)^2 \cdot \mathcal{K}_t^2$$

$$+ CL^2 \cdot \Phi(|\overline{b}_t|) \cdot \widehat{\mathbb{E}}_{l,l'}\left[\Phi\Big(|\overline{b}_t|\sqrt{\frac{1 - \langle h_l, h_{l'} \rangle}{1 + \langle h_l, h_{l'} \rangle}}\Big)\langle h_l, h_{l'} \rangle\right].$$

Note that the joint event $\mathbb{1}(\cap_{\iota=1}^3 \mathcal{E}_\iota)$ holds with probability at least $1 - n^{-c}$ as we discussed earlier. Therefore, we can safely drop the indicator $\mathbb{1}(\cap_{\iota=1}^3 \mathcal{E}_\iota)$ in the above inequality. This completes the proof of Theorem F.4.

### H.2.4 CONCENTRATION FOR $\|F^\top \varphi(Fy_t + \theta \cdot v^\top \overline{w}_{t-1}; b_t)\|_2^2$: PROOF OF THEOREM F.6

In the following proof, we will use $C$ to denote universal constants that change from line to line. Let us fix $\{\alpha_{\tau,t-1}\}_{\tau=-1}^{t-1}$ and $b_t$. Then $y_t^\star \sim \mathcal{N}(0, I_{n-1})$. For simplicity, we will denote $y_t^\star$ by $y$ in the following. Let us define the good event

$$\mathcal{E} = \big\{\max_{\tau=-1,0,\ldots,t-1}\|z_\tau\|_\infty \leq (1 + \sqrt{c})\sqrt{2\log(nt)}\big\}.$$

It then follows from Theorem J.2 that $\mathbb{P}(\overline{\mathcal{E}}) \leq (nt)^{-c} \leq n^{-c}$, and also $\|y\|_\infty \leq (1 + \sqrt{c})\sqrt{2t\log(nt)}$ on $\mathcal{E}$. In particular,

$$|\varphi(f_l^\top y + \theta_l v^\top \bar{w}_{t-1}; b_t)| \mathbb{1}(\mathcal{E}) \leq (\gamma_2 + |b_t|\gamma_1)((1 + \sqrt{c})\sqrt{2t\log(nt)} + \theta_l \|v\|_2 \alpha_{-1,t-1}) + (n \vee d)^{-c_0} := B_t,$$

where the inequality holds by the Lipschitz continuity of $\varphi$ in Definition B.3 and also the fact that $b_t + \kappa_0 \leq 0$ for the bias. Define

$$Z = \frac{1}{N_2^2} \sum_{l,l'=1}^{N_2} \langle f_l, f_{l'} \rangle \cdot \varphi(f_l^\top y + \theta_l v^\top \bar{w}_{t-1}; b_t) \cdot \varphi(f_{l'}^\top y + \theta_{l'} v^\top \bar{w}_{t-1}; b_t) \mathbb{1}(\mathcal{E}).$$

Using the Cauchy-Schwarz inequality, we have

$$Z \leq \frac{1}{N_2^2} \sum_{l,l'=1}^{N_2} \left( \varphi(f_l^\top y + \theta_l v^\top \bar{w}_{t-1}; b_t)^2 + \varphi(f_{l'}^\top y + \theta_{l'} v^\top \bar{w}_{t-1}; b_t)^2 \right) \cdot \mathbb{1}(\langle f_l, f_{l'} \rangle \neq 0) \cdot \mathbb{1}(\mathcal{E})$$

$$\leq \frac{2\rho_2}{N_2} \sum_{l=1}^{N_2} \varphi(f_l^\top y + \theta_l v^\top \bar{w}_{t-1}; b_t)^2 \mathbb{1}(\mathcal{E}), \tag{H.23}$$

where the first inequality follows from $ab \leq a^2 + b^2$, and the second inequality follows from the fact that $\langle f_l, f_{l'} \rangle^2$ is nonzero for at most $N_2\rho_2$ terms when going over $l'$ by definition (F.1). Next, we concentrate the right-hand side of (H.23). Note that by the Lipschitz continuity of $\varphi$, we have

$$|\varphi(f_l^\top y + \theta_l v^\top \bar{w}_{t-1}; b_t)| \leq (\gamma_2 + |b_t|\gamma_1)(|f_l^\top y| + \theta_l \|v\|_2 \alpha_{-1,t-1}) + (n \vee d)^{-c_0}.$$

By the Cauchy-Schwarz inequality, we further obtain

$$\varphi(f_l^\top y + \theta_l v^\top \bar{w}_{t-1}; b_t)^2 \leq C(\gamma_2 + |b_t|\gamma_1)^2 \left( (f_l^\top y)^2 + (\theta_l \|v\|_2 \alpha_{-1,t-1})^2 \right) + C(n \vee d)^{-2c_0}. \tag{H.24}$$

To this end, we apply the Cauchy-Schwarz inequality again to obtain that

$$\frac{1}{N_2} \sum_{l=1}^{N_2} (f_l^\top y)^2 \leq \frac{1}{N_2} \sum_{l=1}^{N_2} \left( \sum_{j=1}^{n-1} y(j)^2 \mathbb{1}(f_l(j) \neq 0) \right) \cdot \|f_l\|_2^2 \leq \rho_2 \cdot \|y\|_2^2.$$

Under the good event $\mathcal{E}$, we have $\|y\|_2 \leq (1 + \sqrt{c})\sqrt{2t\log(nt)}$. In fact, $\|y\|_2^2 \sim \chi_{n-1}^2$, and we can apply the concentration inequality for the chi-squared distribution in Theorem J.1 to obtain that with probability at least $1 - \delta$, it holds over the randomness of $y$ that

$$\frac{1}{N_2} \sum_{l=1}^{N_2} (f_l^\top y)^2 \mathbb{1}(\mathcal{E}) \leq \frac{1}{N_2} \sum_{l=1}^{N_2} (f_l^\top y)^2 \leq C\rho_2 \cdot (n + \log \delta^{-1}).$$

Applying a union bound over $\{\alpha_{\tau,t-1}\}_{\tau=-1}^{t-1}$ and $b_t$ similar to Theorem H.4, and since $Z$ is uniformly bounded, we conclude that with probability at least $1 - n^{-c}$, it holds for all $t \leq n^c$ that

$$\frac{1}{N_2} \sum_{l=1}^{N_2} (f_l^\top y)^2 \mathbb{1}(\mathcal{E}) \leq C\rho_2 \cdot (n + t\log(n)). \tag{H.25}$$

Combining (H.23), (H.24), and (H.25), we conclude that with probability at least $1 - n^{-c}$, it holds for all $t \leq n^c$ that

$$Z \leq C(\gamma_2 + |b_t|\gamma_1)^2 \rho_2 \cdot \left( N_2^{-1} \|\theta\|_2^2 \|v\|_2^2 \alpha_{-1,t-1}^2 + \rho_2 n + \rho_2 t \log n \right).$$

As the good event $\mathcal{E}$ holds with sufficiently high probability if we choose $c$ large enough in the definition of $\mathcal{E}$, A similar bound holds for the original quantity $\|F^\top \varphi(Fy_t + \theta \cdot v^\top \bar{w}_{t-1}; b_t)\|_2^2$. This completes the proof of Theorem F.6.

### H.2.5 Concentration for $\langle z_\tau, E^\top \varphi(E y_t^\star; b_t)\rangle$: Proof of Theorem F.7

In the following proof, we will use $C$ to denote universal constants that change from line to line. When treating $\{\alpha_{\tau,t-1}\}_{\tau=-1}^{t-1}$ and $b_t$ to be deterministic, we have $z_\tau \stackrel{d}{=} \alpha_{\tau,t-1} y_t^\star + \sqrt{1 - \alpha_{\tau,t-1}^2} \cdot z$, where $z \sim \mathcal{N}(0, I_{n-1})$ and is independent of $y_t^\star$. In the following, we use $y$ to replace $y_t^\star$, and $\alpha$ to replace $\alpha_{\tau,t-1}$ for notational simplicity. Therefore, the concentration we consider can be reduced to the concentration of

$$\alpha \cdot \frac{1}{N_1}\langle y, E^\top \varphi(Ey; b_t)\rangle + \sqrt{1 - \alpha^2} \cdot \frac{1}{N_1}\langle z, E^\top \varphi(Ey; b_t)\rangle,$$

Firstly, note that when conditioned on $y$, $\langle z, E^\top \varphi(Ey; b_t)\rangle$ is a gaussian random variable with mean zero and variance $\|E^\top \varphi(Ey; b_t)\|_2^2$, it holds with probability at least $1 - \delta$ over the randomness of $y$ that

$$\frac{1}{N_1}|\langle z, E^\top \varphi(Ey; b_t)\rangle| \leq \frac{1}{N_1}\sqrt{2\|E^\top \varphi(Ey; b_t)\|_2^2 \log \delta^{-1}},$$

where the second order term has already been handled in Theorem F.4. Similar to the proof of Theorem H.4, we can use a covering argument over $\{\alpha_{\tau,t-1}\}_{\tau=-1}^{t-1} \in \mathbb{S}^{t+1}, b_t \in \mathbb{R}, \tau = -1, 0, \ldots, t-1$ and $t \leq n^c$ to obtain that with probability at least $1 - n^{-c}$, it holds for all $(\tau, t)$ that

$$\frac{1}{N_1}|\langle z, E^\top \varphi(Ey; b_t)\rangle| \leq \frac{C}{N_1}\sqrt{\|E^\top \varphi(Ey; b_t)\|_2^2 \cdot t \log(n)}.$$

Now it remains to control the first term. Define good event

$$\mathcal{E} = \left\{\|y\|_\infty \leq (1 + \sqrt{c})\sqrt{2t \log(nt)}\right\}.$$

In fact, the above good event can be directly implied by the following good event:

$$\mathcal{E} = \left\{\max_{\tau=-1,0,\ldots,t-1}\|z_\tau\|_\infty \leq (1 + \sqrt{c})\sqrt{2 \log(nt)}\right\}.$$

For notational simplicity, we will just focus on the latter definition of the good event. It follows from Theorem J.2 that $\mathbb{P}(\mathcal{E}) \geq 1 - (tn)^{-c} \geq 1 - n^{-c}$. Let us define

$$Z = \frac{1}{N_1}\langle y, E^\top \varphi(Ey; b_t)\rangle \cdot \mathbb{1}(\mathcal{E}), \quad \text{and} \quad V := \mathbb{E}\left[\sum_{i=1}^{n-1}(Z - Z^{(i)})^2 \,\Big|\, y\right],$$

where $Z^{(i)} = \langle y^{(i)}, E^\top \varphi(Ey^{(i)}; b_t)\rangle \cdot \mathbb{1}(\mathcal{E}^{(i)})$ and $y^{(i)}$ is given by replacing the $i$-th coordinate $y_i$ with an independent copy $y_i' \sim \mathcal{N}(0, 1)$. Note that this is equivalent to replacing the $i$-th coordinate of each $z_\tau$ with an independent copy $z_\tau^{(i)}$. Thus, the good event $\mathcal{E}$ can be also changed to $\mathcal{E}^{(i)}$ accordingly. Next, we show how to control the variance $V$. Let us define

$$Z_l = e_l^\top y \cdot \varphi(e_l^\top y; b_t) \cdot \mathbb{1}(\mathcal{E}) \quad \text{and} \quad Z_l^{(i)} = e_l^\top y^{(i)} \cdot \varphi(e_l^\top y^{(i)}; b_t) \cdot \mathbb{1}(\mathcal{E}^{(i)})$$

for any $l \in [N_1]$. On the joint event $\mathcal{E} \cup \mathcal{E}^{(1)} \cup \ldots \cup \mathcal{E}^{(n-1)}$, we have by the Lipschitzness of $\varphi$ in Definition B.3 that

$$|Z_l| \leq C(\gamma_2 + |b_t|\gamma_1)t \log(nt) =: B_t, \quad \forall l \in [N_1]. \tag{H.26}$$

This bounds also holds for all $Z_l^{(i)}$ for $i \in [n-1]$. By a reformulation, we obtain for the joint event $\mathcal{E} \cup \mathcal{E}_1 \cup \ldots \cup \mathcal{E}_{n-1}$ that

$$(Z - Z^{(i)})^2 = \frac{1}{N_1^2} \cdot \left(\sum_{l=1}^{N_1}(Z_l - Z_l^{(i)})\right)^2 = \frac{1}{N_1^2} \cdot \left(\sum_{l=1}^{N_1}(Z_l - Z_l^{(i)}) \cdot \mathbb{1}(E_{l,i} \neq 0)\right)^2$$

$$\leq \frac{\rho_1}{N_1} \cdot \sum_{l=1}^{N_1}(Z_l - Z_l^{(i)})^2 \cdot \mathbb{1}(E_{l,i} \neq 0) \leq \frac{2\rho_1}{N_1} \cdot \sum_{l=1}^{N_1}(Z_l^2 + (Z_l^{(i)})^2) \cdot \mathbb{1}(E_{l,i} \neq 0)$$

$$\leq \frac{2\rho_1}{N_1}B_t^2 \cdot \sum_{l=1}^{N_1}\left(\mathbb{1}(e_l^\top y + \bar{b}_t > 0) + \mathbb{1}(e_l^\top y^{(i)} + \bar{b}_t > 0) + 2B_t^{-1}(n \vee d)^{-c_0}\right) \cdot \mathbb{1}(E_{l,i} \neq 0),$$

where the first inequality holds by the Cauchy-Schwarz inequality, the second one holds by $(a - b)^2 \leq 2(a^2 + b^2)$, and the last line holds by Definition B.3 and the upper bound in (H.26). Since $c_0$ is some sufficiently large constant, we can safely ignore the term involving $(n \vee d)^{-c_0}$ in the sequel (when invoking a constant factor $C$). Taking a summation over $i = 1, \ldots, n - 1$ on both sides and taking the conditional expectation, we obtain that

$$V \leq \frac{C\rho_1}{N_1} B_t^2 \cdot \sum_{i=1}^{n-1} \sum_{l=1}^{N_1} \left( \mathbb{1}(e_l^\top y + \bar{b}_t > 0) + \mathbb{E}\left[ \mathbb{1}(e_l^\top y^{(i)} + \bar{b}_t > 0) \mid y \right] \right) \cdot \mathbb{1}(E_{l,i} \neq 0).$$

Let us define

$$g(y) = \frac{2\rho_1 B_t^2}{N_1} \cdot \sum_{i=1}^{n-1} \sum_{l=1}^{N_1} \mathbb{1}(e_l^\top y + \bar{b}_t > 0) \cdot \mathbb{1}(E_{l,i} \neq 0).$$

Therefore, the moment generating function of $V$ is controlled by

$$\mathbb{E}[\exp(\lambda V)] \leq \mathbb{E}\left[ \exp(\lambda g(y)) \cdot \exp\left( \lambda \mathbb{E}[g(y^{(i)}) \mid y] \right) \right] \leq \mathbb{E}\left[ \exp(\lambda g(y)) \cdot \exp\left( \lambda g(y^{(i)}) \right) \right]$$

for $\lambda > 0$. Here, the last inequality follows from the Jensen's inequality. To this end, we notice that $g$ is a non-decreasing functions of $y$. Then by Theorem J.10, we have that $\mathbb{E}[\exp(\lambda g(y)) \cdot \exp(\lambda g(y^{(i)}))] \leq \mathbb{E}[\exp(2\lambda g(y))]$. Therefore, we just need to focus on the moment generating function of $g(y)$. Note that since $e_l$ is $s$-sparse, with probability at least $1 - \delta$ over the randomness of $y$, we have

$$g(y) \leq \frac{2s\rho_1 B_t^2}{N_1} \cdot \sum_{l=1}^{N_1} \mathbb{1}(e_l^\top y + \bar{b}_t > 0) \leq Cs\rho_1 B_t^2 \cdot \left( \Phi(|\bar{b}_t|) + \rho_1 s \log \delta^{-1} \right).$$

where in the last inequality, we invoke Theorem H.4. This can be transformed into the following tail bound

$$\mathbb{E}\left[ \exp(\lambda V) \right] \leq \mathbb{E}\left[ \exp(2\lambda g(y)) \right], \quad \text{where} \quad \mathbb{P}\left( g(y) > Cs\rho_1 B_t^2 \Phi(|\bar{b}_t|) + v \right) \leq \exp\left( -\frac{v}{C\rho_1^2 s^2 B_t^2} \right),$$

and any $v > 0$. In particular, for $V_+$ and $V_-$ defined in (J.2), we always have $0 \leq V_+ \leq V$ and $0 \leq V_- \leq V$. With the sub-exponential tail bound, we now invoke Condition 1 of Theorem J.8 to conclude that with probability at least $1 - \delta$ over the randomness of $y$,

$$|Z - \mathbb{E}[Z]| \leq CB_t\left( \sqrt{s\rho_1 \Phi(|\bar{b}_t|) \log \delta^{-1}} + \rho_1 s \log \delta^{-1} \right). \tag{H.27}$$

Since $Z$ is Lipschitz over $\{\alpha_{\tau,t-1}\}_{\tau=-1}^{t-1}$ and $\{z_\tau\}_{\tau=-1}^{t-1}$, we follow a similar covering argument over the balls $\{\mathbb{S}^{t-1}\}_{t=1}^T$ with $T \leq n^c$. Note that the failure probability of the joint event $\mathcal{E} \cup \mathcal{E}^{(1)} \cup \ldots \cup \mathcal{E}^{(n-1)}$ is at most $n^{1-c}$. In addition, we can set $\delta = n^{-c}(n^{-c}\varepsilon^{n^c})$ in (H.27), where $\epsilon$ is the approximation error in the covering argument in the infinity norm. By a union bound of the covering net of size $n^c \varepsilon^{-n^c}$, we will obtain a failure probability at most $n^{-c}$ as well. By decreasing the constant $c$ slightly (up to 2), we can combine the two failure probabilities to obtain that for all $t \leq n^c$, it holds with probability at least $1 - n^{-c}$ that

$$|Z - \mathbb{E}[Z]| \leq CB_t\left( \sqrt{s\rho_1 \Phi(|\bar{b}_t|) \cdot t \log(n)} + s\rho_1 \cdot t \log(n) \right).$$

Next, let us evaluate the expectation $\mathbb{E}[Z]$. By definition,

$$\left| \mathbb{E}[Z] - \frac{1}{N_1} \mathbb{E}\left[ \langle y, E^\top \varphi(Ey; b_t) \rangle \right] \right| = \frac{1}{N_1} \mathbb{E}\left[ \langle y, E^\top \varphi(Ey; b_t) \rangle \cdot \mathbb{1}(\overline{\mathcal{E}}) \right]$$

$$\leq \frac{1}{N_1} \sqrt{ \mathbb{E}\left[ \langle y, E^\top \varphi(Ey; b_t) \rangle^2 \right] \cdot \mathbb{P}(\overline{\mathcal{E}}) }.$$

Since $\mathbb{P}(\overline{\mathcal{E}}) \leq n^{-c}$, while $\mathbb{E}\left[ \langle y, E^\top \varphi(Ey; b_t) \rangle^2 \right]$ is at most $C(\bar{b}_t^2 + (\gamma_1 + |\bar{b}_t|\gamma_2)^2)$ for some universal constant $C$ by the Lipschitzness of $\varphi$ given by Definition B.3. We can pick $c$ in the definition of $\mathcal{E}$ to be sufficiently large, Thereby, the approximation error in the expectation is negligible. We thus just need to evaluate

$$\frac{1}{N_1} \mathbb{E}\left[ \langle y, E^\top \varphi(Ey; b_t) \rangle \right] = \frac{1}{N_1} \sum_{l=1}^{N_1} \mathbb{E}\left[ e_l^\top y \cdot \varphi(e_l^\top y; b_t) \right] = \mathbb{E}_{x \sim \mathcal{N}(0,1)}[x\varphi(x; b_t)] =: \widehat{\varphi}_1(b_t).$$

Hence, we conclude that for all $\tau \leq t - 1$ and $t \leq n^c$, it holds with probability at least $1 - n^{-c}$ that

$$\left| \frac{1}{N_1} \langle z_\tau, E^\top \varphi(Ey_t; b_t) \rangle - \alpha_{\tau,t-1} \cdot \widehat{\varphi}_1(b_t) \right| \leq \alpha_{\tau,t-1} \cdot CB_t \left( \sqrt{s\rho_1 \Phi(|\bar{b}_t|) \cdot t \log(n)} + s\rho_1 \cdot t \log(n) \right)$$
$$+ \sqrt{1 - \alpha_{\tau,t-1}^2} \cdot \frac{C}{N_1} \sqrt{2\|E^\top \varphi(Ey; b_t)\|_2^2 \cdot t \log(n)}.$$

Plugging in the definition of $B_t = C(\gamma_2 + |b_t|\gamma_1)t \log(nt)$, we complete the proof of Theorem F.7.

## H.2.6 CONCENTRATION FOR $\langle z_\tau, F^\top \varphi(Fy_t + \theta \cdot v^\top \bar{w}_{t-1}; b_t) \rangle$: PROOF OF THEOREM F.9

In this proof, we will show the concentration for the term $N_2^{-1}\langle z_\tau, F^\top \varphi(Fy_t + \theta \cdot v^\top \bar{w}_{t-1}; b_t) \rangle$. Similar to the proof of Theorem F.7, when fixing $\{\alpha_{\tau,t-1}\}_{\tau=-1}^{t-1}$ and $\{b_{t,l}\}_{l=1}$, we have $y_t^\star \sim \mathcal{N}(0, I_{n-1})$. For simplicity, we will denote $y_t^\star$ by $y$ in the following. Note that $z_\tau \overset{d}{=} \alpha_{\tau,t-1}y + \sqrt{1 - \alpha_{\tau,t-1}^2} \cdot z$ where $z \sim \mathcal{N}(0, I_{n-1})$ is independent of $y$. In the sequel, we also simplify $\alpha_{\tau,t-1}$ to $\alpha$. Therefore, the concentration we consider can be reduced to

$$\alpha \cdot \frac{1}{N_2} \langle y, F^\top \varphi(Fy + \theta \cdot v^\top \bar{w}_{t-1}; b_t) \rangle + \sqrt{1 - \alpha^2} \cdot \frac{1}{N_2} \langle z, F^\top \varphi(Fy + \theta \cdot v^\top \bar{w}_{t-1}; b_t) \rangle.$$

The concentration for the second part follows directly from the Gaussian tail bound. That said, with probability at least $1 - \delta$, it holds that

$$\frac{1}{N_2} \left| \langle z, F^\top \varphi(Fy + \theta \cdot v^\top \bar{w}_{t-1}; b_t) \rangle \right| \leq \frac{1}{N_2} \cdot \sqrt{2\|F^\top \varphi(Fy + \theta \cdot v^\top \bar{w}_{t-1}; b_t)\|_2^2 \cdot \log \delta^{-1}},$$

where the right-hand side can be controlled by Theorem F.6. Then by a covering argument over $\{\alpha_{\tau,t-1}\}_{\tau=-1}^{t-1}$ and $b_t$ similar to Theorem H.4 (with proper truncation of the random variables that yields a sufficiently small error probability), we conclude that with probability at least $1 - n^{-c}$, it holds for all $t = 1, \ldots, T$ and $\tau = -1, 0, \ldots, t-1$ that

$$\frac{1}{N_2} \left| \langle z, F^\top \varphi(Fy + \theta \cdot v^\top \bar{w}_{t-1}; b_t) \rangle \right| \leq \frac{C}{N_2} \cdot \sqrt{\|F^\top \varphi(Fy + \theta \cdot v^\top \bar{w}_{t-1}; b_t)\|_2^2 \cdot t \log(n)}.$$

To control the first term, define good event

$$\mathcal{E} = \left\{ \max_{\tau=-1,0,\ldots,t-1} \|z_\tau\|_\infty \leq (1 + \sqrt{c})\sqrt{2 \log(nt)} \right\}.$$

On this good event, $\|y\|_\infty \leq (1 + \sqrt{c})\sqrt{2t \log(nt)}$ and this good event holds with probability at least $1 - (tn)^{-c} \geq 1 - n^{-c}$. We define

$$Z = \frac{1}{N_2} \langle y, F^\top \varphi(Fy + \theta \cdot v^\top \bar{w}_{t-1}; b_t) \rangle \mathbb{1}(\mathcal{E}), \quad \text{and} \quad V = \mathbb{E}\left[ \sum_{i=1}^{n-1} (Z - Z^{(i)})^2 \,\Big|\, y \right],$$

where $Z^{(i)} = N_2^{-1}\langle y^{(i)}, F^\top \varphi(Fy^{(i)} + \theta \cdot v^\top \bar{w}_{t-1}; b_t) \rangle \mathbb{1}(\mathcal{E}^{(i)})$. Here, we define $y^{(i)} = \sum_{j=-1}^{t-1} \alpha_{j,t-1} z_j^{(i)}$ with $z_\tau^{(i)}$ given by replacing the $i$-th coordinate of $z_j$ with an independent copy, and $\mathcal{E}^{(i)}$ is the event defined with respect to $z_\tau^{(i)}$. Let us define

$$Z_l = f_l^\top y \cdot \varphi(f_l^\top y + \theta_l v^\top \bar{w}_{t-1}; b_t) \mathbb{1}(\mathcal{E}), \quad Z_l^{(i)} = f_l^\top y^{(i)} \cdot \varphi(f_l^\top y^{(i)} + \theta_l v^\top \bar{w}_{t-1}; b_t) \mathbb{1}(\mathcal{E}^{(i)}),$$

where $f_l$ is the $l$-th row of $F$. On the joint event $\mathcal{E} \cup \mathcal{E}^{(1)} \cup \cdots \cup \mathcal{E}^{(n-1)}$, we have by the Lipschitz continuity of $\varphi$ in Definition B.3 that

$$|Z_l| \leq C\big((\gamma_2 + |b_t|\gamma_1) \cdot (\sqrt{t \log(n)} + \|v\|_2 \alpha_{-1,t-1}) + (n \vee d)^{-c_0}\big) \cdot \sqrt{t \log(n)} := B_t,$$

where we also use the fact that $b_t + \kappa_0 \leq 0$ for the bias. Note that the $(n \vee d)^{-c_0}$ term is negligible when $c_0$ is sufficiently large. For notation simplicity, we define $\widetilde{b}_{t,l} = b_t + \kappa_0 + \theta_l v^\top \bar{w}_{t-1}$. This

bound also holds for $Z_l^{(i)}$. On the joint event $\mathcal{E} \cup \mathcal{E}^{(1)} \cup \cdots \cup \mathcal{E}^{(n-1)}$, we have

$$
(Z - Z^{(i)})^2 \leq \frac{1}{N_2^2} \sum_{l=1}^{N_2} (Z_l - Z_l^{(i)})^2 \leq \frac{1}{N_2^2} \cdot \left( \sum_{l=1}^{N_2} (Z_l - Z_l^{(i)})^2 \, \mathbb{1}(F_{l,i} \neq 0) \right)^2
$$

$$
\leq \frac{\rho_2}{N_2} \sum_{l=1}^{N_2} (Z_l - Z_l^{(i)})^2 \, \mathbb{1}(F_{l,i} \neq 0) \leq \frac{2\rho_2}{N_2} \sum_{l=1}^{N_2} \big( Z_l^2 + (Z_l^{(i)})^2 \big) \, \mathbb{1}(F_{l,i} \neq 0)
$$

$$
\leq \frac{2\rho_2 B_t^2}{N_2} \sum_{l=1}^{N_2} \big( \mathbb{1}(f_l^\top y + \widetilde{b}_{t,l} > 0) + \mathbb{1}(f_l^\top y^{(i)} + \widetilde{b}_{t,l} > 0) + 2B_t^{-1}(n \vee d)^{-c_0} \big) \, \mathbb{1}(F_{l,i} \neq 0),
$$

where the first inequality holds by the Cauchy-Schwarz inequality, the second one holds by $(a - b)^2 \leq 2(a^2 + b^2)$, and the last line holds by Definition B.3 and the upper bound for $Z_l$ and $Z_l^{(i)}$. We can also ignore the $2B_t^{-1}(n \vee d)^{-c_0}$ term by multiplying some universal constant. Taking a summation over $i = 1, \ldots, n-1$ on both sides with the conditional expectation, we obtain

$$
V \leq \frac{C\rho_2 B_t^2}{N_2} \sum_{i=1}^{n-1} \sum_{l=1}^{N_2} \big( \mathbb{1}(f_l^\top y + \widetilde{b}_{t,l} > 0) + \mathbb{E}\big[ \mathbb{1}(f_l^\top y^{(i)} + \widetilde{b}_{t,l} > 0) \big] \big) \cdot \mathbb{1}(F_{l,i} \neq 0).
$$

Let us take

$$
g(y) := \frac{C\rho_2 B_t^2}{N_2} \sum_{i=1}^{n-1} \sum_{l=1}^{N_2} \mathbb{1}(f_l^\top y + \widetilde{b}_{t,l} > 0) \, \mathbb{1}(F_{l,i} \neq 0)
$$

$$
= \frac{C\rho_2 s B_t^2}{N_2} \sum_{l=1}^{N_2} \mathbb{1}(f_l^\top y + \widetilde{b}_{t,l} > 0) \leq C\rho_2 s B_t^2.
$$

Then we have by the monotonicity of $g$ and Theorem J.10 that $\mathbb{E}[\exp(\lambda V)] \leq \mathbb{E}[\exp(2\lambda g(y))]$ for all $\lambda > 0$. Invoking Theorem J.9 for this bounded variance, we obtain that with probability at least $1 - \delta$ over the randomness of $y$, it holds that

$$
|Z - \mathbb{E}[Z]| \leq CB_t \sqrt{\rho_2 s} \cdot \log \delta^{-1}.
$$

By a covering argument over $\{\alpha_{\tau, t-1}\}_{\tau=-1}^{t-1}$ and $b_t$ similar to Theorem H.4, we conclude that $|Z - \mathbb{E}[Z]| \leq CB_t \sqrt{\rho_2 s} \cdot t \log(n)$ with probability at least $1 - n^{-c}$ for all $t = 1, \ldots, T$ and $\tau = -1, 0, \ldots, t-1$. In addition, the approximation error

$$
\left| \mathbb{E}[Z] - \frac{1}{N_2} \mathbb{E}[\langle y, F^\top \varphi(Fy + \theta \cdot v^\top \bar{w}_{t-1}; b_t) \rangle] \right| \propto \sqrt{\mathbb{P}(\overline{\mathcal{E}})}
$$

by the Cauchy-Schwarz inequality and the fact that $f_l^\top y \varphi(f_l^\top y + \theta_l v^\top \bar{w}_{t-1}; b_t)$ has bounded second moment. Therefore, by taking a sufficiently large $c$ in the definition of the good event $\mathcal{E}$, we can make this approximation error negligible. Moreover, we also have

$$
\mathbb{E}[f_l^\top y \varphi(f_l^\top y + \theta_l v^\top \bar{w}_{t-1}; b_t)] = \sqrt{1 - \theta_l^2} \cdot \mathbb{E}_{x \sim \mathcal{N}(0,1)} \Big[ x\varphi\big(\sqrt{1 - \theta_l^2} x + \theta_l v^\top \bar{w}_{t-1}; b_t\big) \Big].
$$

Combining everything, we conclude that with probability at least $1 - n^{-c}$, it holds for all $t = 1, \ldots, T$ and $\tau = -1, 0, \ldots, t-1$ that

$$
\frac{1}{N_2} \Big| \langle z_\tau, F^\top \varphi(Fy_t^\star + \theta \cdot v^\top \bar{w}_{t-1}; b_t) \rangle - \sum_{l=1}^{N_2} \alpha_{\tau, t-1} \sqrt{1 - \theta_l^2} \cdot \mathbb{E}_{x \sim \mathcal{N}(0,1)} \Big[ x\varphi\big(\sqrt{1 - \theta_l^2} x + \theta_l v^\top \bar{w}_{t-1}; b_t\big) \Big] \Big|
$$

$$
\leq \frac{C}{N_2} \sqrt{1 - \alpha_{\tau, t-1}^2} \cdot \sqrt{\|F^\top \varphi(Fy + \theta \cdot v^\top \bar{w}_{t-1}; b_t)\|_2^2 \cdot t \log(n)}
$$

$$
+ C\alpha_{\tau, t-1} \cdot (\gamma_2 + |b_t|\gamma_1) \cdot (\sqrt{t \log(n)} + \|v\|_2 \alpha_{-1, t-1}) \cdot \sqrt{\rho_2 s} \cdot (t \log(n))^{3/2}
$$

This completes the proof of Theorem F.9.

### H.2.7 CONCENTRATION FOR $\theta^\top \varphi(Fy_t^\star + \theta v^\top \overline{w}_{t-1}; b_t)$: PROOF OF THEOREM F.10

In the following, we will use $C$ to denote universal constants that change from line to line. Let $f_l$ denote the $l$-th row of $F$. Let us first fix $\{\alpha_{\tau,t-1}\}_{\tau=-1}^{t-1}$ and $b_t$. Then $y_t^\star \sim \mathcal{N}(0, I_{n-1})$. In the sequel, we will simplify $y \leftarrow y_t^\star$. Let us define the good event

$$\mathcal{E} = \big\{ \max_{\tau=-1,0,\dots,t-1} \|z_\tau\|_\infty \le (1+\sqrt{c})\sqrt{2\log(nt)} \big\}.$$

It then follows from Theorem J.2 that $\mathbb{P}(\overline{\mathcal{E}}) \le (nt)^{-c} \le n^{-c}$, and also $\|y\|_\infty \le (1+\sqrt{c})\sqrt{2t\log(nt)}$ on $\mathcal{E}$. In particular,

$$|\varphi(f_l^\top y + \theta_l v^\top \overline{w}_{t-1}; b_t)| \, \mathbb{1}(\mathcal{E}) \le (\gamma_2 + |b_t|\gamma_1)((1+\sqrt{c})\sqrt{2t\log(nt)} + \|v\|_2\alpha_{-1,t-1}) + (n \vee d)^{-c_0} := B_t.$$

where the last inequality holds by noting that $\varphi(\cdot; b_t)$ is $\gamma_2 + |b_t|\gamma_1$-Lipschitz by Definition B.3, and also the fact that $\overline{b}_t = b_t + \kappa_0 \le 0$. The target function to study is

$$Z = \frac{1}{N_2} \sum_{l=1}^{N_2} \theta_l \varphi(f_l^\top y; \widetilde{b}_{t,l}) \, \mathbb{1}(\mathcal{E}), \quad \text{where} \quad \widetilde{b}_{t,l} = b_{t,l} + \theta_l \|v\|_2 \alpha_{-1,t-1}.$$

Let $y^{(i)}$ be the vector obtained by replacing the $i$-th element of $y_t$ with an independent standard Gaussian random variable $y_t'(i)$. The good event $\mathcal{E}^{(i)}$ is defined similarly. Define $Z^{(i)}$ as the correspondence of $Z$ with $y^{(i)}$ and $\mathcal{E}^{(i)}$. Let us define variance $V = \mathbb{E}[\sum_{i=1}^{n-1}(Z - Z^{(i)})^2]$. Notice that this $V$ upper bounds both $V_+ = \mathbb{E}[\sum_{i=1}^{n-1}(Z - Z^{(i)})^2 \, \mathbb{1}(Z > Z^{(i)})]$ and $V_- = \mathbb{E}[\sum_{i=1}^{n-1}(Z - Z^{(i)})^2 \, \mathbb{1}(Z < Z^{(i)})]$. Note that when changing one coordinate in $y$, the total number of terms affected in $Z$ is at most $N_2\rho_2$ by definition (F.1). It then holds by the Cauchy-Schwarz inequality that

$$V \le \frac{C\rho_2}{N_2} \sum_{i=1}^{n-1} \sum_{l=1}^{N_2} \theta_l^2 \cdot \mathbb{E}\big[\big(\varphi(f_l^\top y; \widetilde{b}_{t,l})\,\mathbb{1}(\mathcal{E}) - \varphi(f_l^\top y^{(i)}; \widetilde{b}_{t,l})\,\mathbb{1}(\overline{\mathcal{E}})\big)^2 \,\big|\, y\big]$$

$$\le \frac{CB_t^2\rho_2}{N_2} \sum_{i=1}^{n-1} \sum_{l=1}^{N_2} \theta_l^2 \, \mathbb{1}(f_l(i) \neq 0),$$

where in the second inequality, the indicator is included since the term will be zero if $f_l(i) = 0$. Additionally, we invoke the bound $B_t$ to upper bound the $\varphi(\cdot)$ term. Let us define

$$g(y) := \frac{CB_t^2\rho_2}{N_2} \sum_{i=1}^{n-1} \sum_{l=1}^{N_2} \theta_l^2 \cdot \mathbb{1}(f_l(i) \neq 0) \le \frac{CB_t^2\rho_2 s}{N_2} \|\theta\|_2^2.$$

By Theorem J.10, we know that the MGF of $V$ can be upper bounded by $\mathbb{E}[\exp(\lambda V)] \le \mathbb{E}[\exp(2\lambda g(y))]$. Thanks to the bounded variance, invoking Theorem J.9, we conclude that with probability at least $1 - \delta$ over the randomness of $y$, it holds that

$$|Z - \mathbb{E}[Z]| \le CB_t\|\theta\|_2 \sqrt{\frac{\rho_2 s}{N_2}} \log(\delta^{-1}).$$

Next, we invoke a union covering argument over the ball $\mathbb{S}^{t+1}$ for $\alpha_{\tau,t-1}$ and also for $b_t$. Since $Z$ is Lipschitz and bounded, the approximation error can be made sufficiently small. Therefore, we conclude that with probability at least $1 - n^{-c}$, it holds for all $t \le n^c$ that

$$|Z - \mathbb{E}[Z]| \le CB_t\|\theta\|_2 \sqrt{\frac{\rho_2 s}{N_2}} \cdot t\log(n).$$

Similar to previous proof, the error in $\mathbb{E}[Z]$ and $N_2^{-1}\mathbb{E}[\theta^\top \varphi(Fy_t + \theta \cdot v^\top \overline{w}_{t-1}; b_t)]$ can be made sufficiently small if we choose a large $c$ in the definition of the good event $\mathcal{E}$. Consequently we just need to plug in the expectatin

$$\frac{1}{N_2}\mathbb{E}[\theta^\top \varphi(Fy_t + \theta \cdot v^\top \overline{w}_{t-1}; b_t)] = \frac{1}{N_2} \sum_{l=1}^{N_2} \mathbb{E}_{x\sim\mathcal{N}(0,1)}\big[\theta_l \cdot \varphi(\sqrt{1-\theta_l^2}\cdot x + \theta_l \cdot v^\top \overline{w}_{t-1}; b_t)\big].$$

This completes the proof of Theorem F.10.

### H.3 Propagation of the Non-Gaussian Error

In this subsection, we analyze how to Non-Gaussian error $\Delta y_t$ propagates through the nonlinear activation.

#### H.3.1 Error Analysis for $\Delta E_t$: Proof of Theorem F.12

In the following proof, we will use $C$ to denote universal constants that change from line to line.

**Bounding $\|\Delta E_t\|_1$.** By definition of $\Delta E_t$, we have

$$\|\Delta E_t\|_1 = \|E^\top \varphi(E(y_t^\star + \Delta y_t); b_t) - E^\top \varphi(E y_t^\star; b_t)\|_1 \leq \sqrt{s} \cdot \|\varphi(E(y_t^\star + \Delta y_t); b_t) - \varphi(E y_t^\star; b_t)\|_1$$

$$\leq \sqrt{s}(\gamma_2 + |b_t|\gamma_1) \cdot \sum_{l=1}^{N_1} |e_l^\top \Delta y_t| \cdot \mathbb{1}(e_l^\top y_t + \bar{b}_t > 0 \ \vee \ e_l^\top y_t^\star + \bar{b}_t > 0)$$

$$+ \sqrt{s} \cdot \sum_{l=1}^{N_1} 2(2 + |b_t|) \cdot (n \vee d)^{-c_0} \cdot \mathbb{1}(e_l^\top y_t + \bar{b}_t \leq 0 \ \wedge \ e_l^\top y_t^\star + \bar{b}_t \leq 0).$$

where $\bar{b}_t = b_t + \kappa_0$ is the shifted bias. The first inequality follows from the fact that $\|e_l\|_1 \leq \sqrt{s}$ as each row $e_l$ is $s$-sparse. The second inequality holds by splitting the summation into two parts. For the first part $\{l : e_l^\top y_t + \bar{b}_t > 0 \ \vee \ e_l^\top y_t^\star + \bar{b}_t > 0\}$ where the neuron is activated, we have the term bounded by the Lipschitz continuity of $\varphi$ times the pre-activation difference $|e_l^\top \Delta y_t|$. Here, we recall from Definition B.3 that $\varphi$ is $(\gamma_2 + |b_t|\gamma_1)$-Lipschitz continuous. For the second part $\{l : e_l^\top y_t + \bar{b}_t \leq 0 \wedge e_l^\top y_t^\star + \bar{b}_t \leq 0\}$ where the neuron is inactive, we simply apply the upper bound on $\varphi$ in Definition B.3 as $(2 + |b_t|) \cdot (n \vee d)^{-c_0}$. Note that $c_0$ can be chosen to be a sufficiently large constant. Thus, we just need to focus on the first part. Using the Cauchy-Schwarz inequality twice, we have

$$\sum_{l=1}^{N_1} |e_l^\top \Delta y_t| \cdot \mathbb{1}(e_l^\top y_t + \bar{b}_t > 0) \leq \sum_{l=1}^{N_1} \|e_l\|_2 \cdot \|\Delta y_t \circ \mathbb{1}(e_l \neq 0)\|_2 \cdot \mathbb{1}(e_l^\top y_t + \bar{b}_t > 0)$$

$$\leq \sqrt{\sum_{l=1}^{N_1} \mathbb{1}(e_l^\top y_t + \bar{b}_t > 0) \cdot \sum_{l=1}^{N_1} \|\Delta y_t \circ \mathbb{1}(e_l \neq 0)\|_2^2}, \quad \text{(H.28)}$$

where $x \circ y$ is the Hadamard product between two vectors $x$ and $y$. Note that the second term on the right hand side can be further bounded by

$$\sum_{l=1}^{N_1} \|\Delta y_t \circ \mathbb{1}(e_l \neq 0)\|_2^2 = \sum_{l=1}^{N_1} \sum_{i=1}^{n-1} \Delta y_{t,i}^2 \cdot \mathbb{1}(E_{l,i} \neq 0) \leq \rho_1 N_1 \cdot \|\Delta y_t\|_2^2. \quad \text{(H.29)}$$

Plugging (H.29) back into (H.28), and invoking Theorem F.3, we conclude that with probability at least $1 - n^{-c}$ for all $t \leq n^c$,

$$\sum_{l=1}^{N_1} |e_l^\top \Delta y_t| \cdot \mathbb{1}(e_l^\top y_t + \bar{b}_t > 0) \leq C N_1 \cdot \sqrt{\left(\Phi(-\bar{b}_t) + \rho_1 s t \log(n) + \rho_1 |\bar{b}_t|^2 \|\Delta y_t\|_2^2\right) \rho_1 \|\Delta y_t\|_2^2}$$

$$\leq C N_1 \cdot \left(\left(\sqrt{\rho_1 \Phi(-\bar{b}_t)} + \rho_1 \sqrt{s t \log n}\right) \cdot \|\Delta y_t\|_2 + \rho_1 |\bar{b}_t| \cdot \|\Delta y_t\|_2^2\right).$$

Note that the ideal activation $\sum_{l=1}^{N_1} \mathbb{1}(e_l^\top y_t^\star + \bar{b}_t > 0)$ has an upper bound in Theorem F.2 even tighter than the one we use above. Therefore, we just need to double the above error term. Thereby, we conclude that

$$\|\Delta E_t\|_1 \leq C N_1 (\gamma_2 + |b_t|\gamma_1) \cdot \left(\left(\sqrt{s \rho_1 \Phi(-\bar{b}_t)} + s \rho_1 \sqrt{t \log n}\right) \cdot \|\Delta y_t\|_2 + \sqrt{s} \rho_1 |\bar{b}_t| \cdot \|\Delta y_t\|_2^2\right)$$

$$+ C N_1 \sqrt{s}(2 + |b_t|) \cdot (n \vee d)^{-c_0}.$$

**Bounding $\|\Delta E_t\|_2^2$.** The proof is similar to bounding $\|\Delta E_t\|_1$. Again, we notice that for any test vector $x \in \mathbb{R}^{N_1}$,

$$\|E^\top x\|_2^2 = \sum_{i=1}^{n-1} \Big(\sum_{l=1}^{N_1} E_{l,i}x_l\Big)^2 \leq \sum_{i=1}^{n-1}\Big(\sum_{l=1}^{N_1} \mathbb{1}(E_{l,i} \neq 0)\Big) \cdot \Big(\sum_{l=1}^{N_1} E_{l,i}^2 x_l^2\Big) \leq \rho_1 N_1 \|x\|_2^2.$$

Here, the first inequality holds by the Cauchy-Schwarz inequality while the second inequality holds by the sparsity assumption on the columns of $E$ and also the fact that $\sum_{i=1}^{n-1} E_{l,i}^2 = \|e_l\|_2^2 = 1$. Thereby, it holds for $\|\Delta E_t\|_2^2$ that

$$\|\Delta E_t\|_2^2 \leq \rho_1 N_1 \|\varphi(E(y_t^\star + \Delta y_t); b_t) - \varphi(Ey_t^\star; b_t)\|_2^2 \leq \rho_1 N_1 (\gamma_2 + |b_t|\gamma_1)^2 \cdot \sum_{l=1}^{N_1} |e_l^\top \Delta y_t|^2$$

$$\leq \rho_1 N_1 (\gamma_2 + |b_t|\gamma_1)^2 \cdot \sum_{l=1}^{N_1} \|e_l\|_2^2 \cdot \|\Delta y_t \circ \mathbb{1}(e_l \neq 0)\|_2^2 \leq (\gamma_2 + |b_t|\gamma_1)^2 \cdot (\rho_1 N_1)^2 \|\Delta y_t\|_2^2,$$

where the second inequality holds by the Lipschitz continuity of $\varphi$ and the third inequality follows from the Cauchy-Schwarz inequality. The last inequality holds by invoking (H.29). Hence, we complete the proof of Theorem F.12.

### H.3.2 ERROR ANALYSIS FOR $\Delta F_t$: PROOF OF THEOREM F.13

In the following proof, we will use $C$ to denote universal constants that change from line to line. Let $f_l$ be the $l$-th row of matrix $F$. Note that

$$\|\Delta F_t\|_1 \leq \sqrt{s} \cdot \|\varphi(F(y_t^\star + \Delta y_t) + \theta \cdot v^\top \overline{w}_{t-1}; b_t) - \varphi(Fy_t^\star + \theta \cdot v^\top \overline{w}_{t-1}; b_t)\|_1$$

$$\leq \sqrt{s}(\gamma_2 + |b_t|\gamma_1) \cdot \sum_{l=1}^{N_2} |f_l^\top \Delta y_t| \leq \sqrt{s}(\gamma_2 + |b_t|\gamma_1) \cdot \|\Delta y_t\|_2 \cdot \sum_{l=1}^{N_2} \|f_l\|_2$$

$$\leq \sqrt{s} N_2 (\gamma_2 + |b_t|\gamma_1) \cdot \|\Delta y_t\|_2,$$

where the first inequality follows from the fact that $\|f_l\|_1 \leq \sqrt{s}$ by the Hölder's inequality for $s$-sparse $f_l$ with $\|f_l\|_2 \leq 1$, the second inequality follows from the Lipschitzness of $\varphi$ and the third inequality follows from the Cauchy-Schwarz inequality. In the last inequality, we use the fact that $\|f_l\|_2 \leq 1$. Next, we turn to the bound for $\|\Delta F_t\|_2$. For any test vector $x \in \mathbb{R}^{N_2}$, we have

$$\|F^\top x\|_2^2 = \sum_{i=1}^{n-1} \Big(\sum_{l=1}^{N_2} F_{li}x_l\Big)^2 \leq \sum_{i=1}^{n-1} \|F_{:,i}\|_2^2 \cdot \|x\|_2^2 \leq \rho_2 N_2 \|x\|_2^2, \tag{H.30}$$

where we recall that $\rho_2 = \max_{i\in[n-1]}\|F_{:,i}\|_0/N_2$. Since $\Delta F_t = F^\top \Delta\varphi_{F,t}$, we have $\|\Delta F_t\|_2^2 \leq \rho_2 N_2 \|\Delta\varphi_{F,t}\|_2^2$. Next, we use the same Lipschitzness of $\varphi$ to upper bound $\|\Delta\varphi_{F,t}\|_2^2$ as

$$\|\Delta\varphi_{F,t}\|_2^2 \leq (\gamma_2 + |b_t|\gamma_1)^2 \cdot \sum_{l=1}^{N_2} |f_l^\top \Delta y_t|^2 \leq (\gamma_2 + |b_t|\gamma_1)^2 \cdot \sum_{l=1}^{N_2} \|f_l\|_2^2 \cdot \sum_{i=1}^{n-1} \Delta y_{t,i}^2 \mathbb{1}(F_{l,i} \neq 0)$$

$$\leq (\gamma_2 + |b_t|\gamma_1)^2 \cdot \sum_{i=1}^{n-1}\sum_{l=1}^{N_2} \Delta y_{t,i}^2 \mathbb{1}(F_{l,i} \neq 0) \leq N_2 \rho_2 (\gamma_2 + |b_t|\gamma_1)^2 \cdot \|\Delta y_t\|_2^2, \tag{H.31}$$

where we use the Cauchy-Schwarz inequality in the second inequality, the fact that $\|f_l\|_2 \leq 1$ in the third inequality, and the definition of $\rho_2$ in the last inequality. Combining (H.30) and (H.31), we conclude that $\|\Delta F_t\|_2 \leq \rho_2 N_2 (\gamma_2 + |b_t|\gamma_1) \cdot \|\Delta y_t\|_2$. This completes the proof of Theorem F.13.

## H.4 PROOFS FOR TECHNICAL LEMMAS

### H.4.1 PROOF OF THEOREM F.5

We invoke the upper bound $|\varphi(x; b)| \leq (n \vee d)^{-c_0} + L(x + \bar{b}) \cdot \mathbb{1}(x > -\bar{b})$ to obtain that

$$
\mathbb{E}[\varphi(x; b)\varphi(\iota x + \sqrt{1 - \iota^2} \cdot z; b)]
$$
$$
\leq (n \vee d)^{-2c_0} + 2(n \vee d)^{-c_0} \cdot L \cdot \mathbb{E}[(x + \bar{b}) \cdot \mathbb{1}(x > -\bar{b})]
$$
$$
+ L^2 \cdot \underbrace{\mathbb{E}[(x + \bar{b}) \cdot \mathbb{1}(x > -\bar{b}) \cdot (\iota x + \sqrt{1 - \iota^2}z + \bar{b}) \cdot \mathbb{1}(\iota x + \sqrt{1 - \iota^2}z > -\bar{b})]}_{\text{(I)}}.
$$

Note that $\mathbb{E}[x \mathbb{1}(x > -\bar{b})] = p(|\bar{b}|)$ for any $\bar{b}$ by explicit calculation, where $p(x) = \exp(-x^2/2)/\sqrt{2\pi}$ is the standard Gaussian density function. Therefore, we have

$$
\mathbb{E}[(x + \bar{b}) \cdot \mathbb{1}(x > -\bar{b})] = \mathbb{E}[x \mathbb{1}(x > -\bar{b})] + \bar{b} \cdot \mathbb{P}(x > -\bar{b}) = p(|\bar{b}|) - |\bar{b}|\Phi(|\bar{b}|) = F(|\bar{b}|),
$$

where we define $F(x) = p(x) - x\Phi(x)$. We note that the function $F(x)$ is monotonically decreasing for all $x \in \mathbb{R}$. To see this, we take the derivative of $F(x)$ and using the fact that $p'(x) = -xp(x)$ and $\Phi'(x) = -p(x)$, which gives us

$$
F'(x) = -\Phi(x) - x\Phi'(x) - xp(x) = -\Phi(x) + xp(x) - xp(x) = -\Phi(x) < 0. \tag{H.32}
$$

In particular, function $F(x)$ is always positive for any $x \in \mathbb{R}$ as $\lim_{x \to \infty} F(x) = 0$ by the Mills ratio $\lim_{x \to \infty} x\Phi(x)/p(x) = 1$. Therefore, $F(|\bar{b}|) \leq F(0) = 1/2$ and the first two terms involving $(n \vee d)^{-c_0}$ are negligible. For the last term, by marginalizing $z$, we have

$$
\mathbb{E}[(x + \bar{b}) \cdot \mathbb{1}(x > -\bar{b}) \cdot (\iota x + \sqrt{1 - \iota^2}z + \bar{b}) \cdot \mathbb{1}(\iota x + \sqrt{1 - \iota^2}z > -\bar{b})]
$$
$$
= \mathbb{E}\left[(x + \bar{b}) \cdot \mathbb{1}(x > -\bar{b}) \cdot \sqrt{1 - \iota^2} \cdot \left(\frac{\iota x + \bar{b}}{\sqrt{1 - \iota^2}} \cdot \Phi\left(-\frac{\bar{b} + \iota x}{\sqrt{1 - \iota^2}}\right) + p\left(-\frac{\bar{b} + \iota x}{\sqrt{1 - \iota^2}}\right)\right)\right]
$$
$$
= \mathbb{E}\left[(x + \bar{b}) \cdot \mathbb{1}(x > -\bar{b}) \cdot \sqrt{1 - \iota^2} \cdot F\left(-\frac{\bar{b} + \iota x}{\sqrt{1 - \iota^2}}\right)\right].
$$

Since $F(x)$ is monotonically decreasing, we can upper bound the expectation by just plugging in $x = -\bar{b}$ to obtain that

$$
\text{(I)} \leq \mathbb{E}\left[(x + \bar{b}) \cdot \mathbb{1}(x > -\bar{b})\right] \cdot \sqrt{1 - \iota^2} \cdot F\left(-\bar{b}\sqrt{\frac{1 - \iota}{1 + \iota}}\right) = \sqrt{1 - \iota^2} \cdot F(|\bar{b}|) \cdot F\left(|\bar{b}|\sqrt{\frac{1 - \iota}{1 + \iota}}\right).
$$

Next, we prove that $F(x) \leq 2\Phi(x)$ for all $x > 0$. For any $x > 0$, we have $F'(x) = -\Phi(x)$ by (H.32), and $\Phi'(x) = -p(x)$. Therefore,

$$
\frac{F'(x)}{\Phi'(x)} = \frac{\Phi(x)}{p(x)} \leq \frac{\Phi(0)}{p(0)} = \sqrt{\frac{\pi}{2}} \leq 2,
$$

where we use the fact that $\Phi(x)/p(x)$ is monotonically decreasing. Noting that $\lim_{x \to \infty} F(x) = 0$ and $\lim_{x \to \infty} \Phi(x) = 0$, we thus conclude that $F(x) \leq 2\Phi(x)$ for all $x > 0$. Consequently,

$$
\text{(I)} \leq 2\sqrt{1 - \iota^2} \cdot F(|\bar{b}|) \cdot F\left(|\bar{b}|\sqrt{\frac{1 - \iota}{1 + \iota}}\right) \leq 4\sqrt{1 - \iota^2} \cdot \Phi(|\bar{b}|) \cdot \Phi\left(|\bar{b}|\sqrt{\frac{1 - \iota}{1 + \iota}}\right).
$$

Therefore, we conclude the proof of this proposition.

### H.4.2 PROOF OF THEOREM F.8

*Proof of Theorem F.8.* Note that

$$
\widehat{\varphi}_1(b_t) = \mathbb{E}_{x \sim \mathcal{N}(0,1)}[\varphi(x; b_t)x]
$$
$$
\leq L \cdot \mathbb{E}[\mathbb{1}(x + \bar{b}_t > 0)(x + \bar{b}_t)x] + \mathbb{E}[|x| \mathbb{1}(x + \bar{b}_t \leq 0)] \cdot (d \vee n)^{-c_0}
$$
$$
\leq L \cdot \left(\frac{|\bar{b}_t|}{\sqrt{2\pi}}\exp(-\bar{b}_t^2/2) + \Phi(|\bar{b}_t|) + \frac{\bar{b}_t}{\sqrt{2\pi}}\exp(-\bar{b}_t^2/2)\right) + C(d \vee n)^{-c_0}.
$$

Here, the last inequality holds by the following integral calculation:

$$\int_{\bar{b}}^{\infty} x p(x) \mathrm{d}x = p(\bar{b}), \quad \int_{\bar{b}}^{\infty} x^2 p(x) \mathrm{d}x = \bar{b} p(\bar{b}) + \Phi(\bar{b})$$

for the standard normal distribution $p(x) = \exp(-x^2/2)/\sqrt{2\pi}$. For $\bar{b}_t < 0$, the first and the last term cancel in the bracket, and we conclude that $\widehat{\varphi}_1(b_t) \leq 2C_0 L \Phi(|\bar{b}_t|)$ as $(d \vee n)^{-c_0}$ can be sufficienly small. On the other hand, using the condition $\varphi(x; b_t) \geq x\phi'(x + b) \geq C_0 x(x + b_t)$ for $x \geq -b_t$ by Definition B.3, we have

$$\widehat{\varphi}_1(b_t) \geq C_0 \mathbb{E}[\mathbb{1}(x + b_t > 0)(x + b_t)x] + \mathbb{E}[\varphi(x; b_t)x \, \mathbb{1}(-\bar{b}_t \leq x \leq -b_t)] - (n \vee d)^{-c_0} \mathbb{E}[|x|].$$

Here, we recall definition $\varphi(x; b) = \phi(x + b) + x \cdot \phi'(x + b)$. Therefore, $\varphi(x; b) \geq \phi(x + b)$ for $x > 0$. By Definition B.3, we know that $\phi'(x + b) \geq 0$ for all $x$. Since $-\bar{b}_t > 0$, we have for $x \in [-\bar{b}_t, -b_t]$ that

$$\varphi(x; b_t) \geq \phi(x + b_t) \geq -(n \vee d)^{-c_0},$$

where the last inequality holds by the monotonicity of $\phi$. Therefore, we conclude that

$$\widehat{\varphi}_1(b_t) \geq C_0 \mathbb{E}[\mathbb{1}(x + b_t > 0)(x + b_t)x] - C \cdot (n \vee d)^{-c_0} \geq \frac{C_0}{2} \Phi(|b_t|).$$

Since we can make $\kappa_0 = |b_t| - |\bar{b}_t|$ log-polynomially small, e.g., $\kappa_0 = (\log(n \vee d))^{-C}$, for $|\bar{b}_t| = \Theta(\log(n \vee d)^C)$, we have $2\Phi(|\bar{b}_t|) \geq \Phi(|b_t|) \geq \frac{\Phi(|\bar{b}_t|)}{2}$. This completes the proof. $\square$

### H.4.3 PROOF OF THEOREM F.11

*Proof of Theorem F.11.* **Lower bounding the signal term.** Let us lower bound the signal term. Note that by the monotonicity assumption in Definition B.3,

$$\varphi(x; b_t)|_{x > -b_t} = \phi(x + b_t) + x\phi'(x + b_t)|_{x > -b_t} \geq C_0 x.$$

For $x \in (-\bar{b}_t, b_t)$, we have $\varphi(x; b_t) \geq \varphi(-\bar{b}_t; b_t) \geq -(d \vee n)^{-c_0}$. Together, we conclude that

$$\sum_{l=1}^{N_2} \mathbb{E}_{x \sim \mathcal{N}(0,1)} \left[ \theta_l \cdot \varphi\left(\sqrt{1 - \theta_l^2} x + \theta_l \sqrt{d} \alpha_{-1, t-1}; b_t\right) \right]$$

$$\geq \sum_{l=1}^{N_2} \mathbb{E}_{x \sim \mathcal{N}(0,1)} \left[ \theta_l \mathbb{1}\left(x + \frac{\theta_l \sqrt{d} \alpha_{-1, t-1} + b_t}{\sqrt{1 - \theta_l^2}} > 0\right) \cdot C_0 \left(\sqrt{1 - \theta_l^2} x + \theta_l \sqrt{d} \alpha_{-1, t-1}\right) \right]$$
$$\qquad - N_2 (d \vee n)^{-c_0}$$

$$\geq \sum_{l=1}^{N_2} \Phi\left(\frac{-b_t - \theta_l \sqrt{d} \alpha_{-1, t-1}}{\sqrt{1 - \theta_l^2}}\right) \cdot C_0 \theta_l^2 \sqrt{d} \alpha_{-1, t-1} - N_2 (d \vee n)^{-c_0}$$

$$\geq \frac{1 - o(1)}{2} \sum_{l=1}^{N_2} \mathbb{1}\left(\theta_l > \frac{-b_t}{\sqrt{d} \alpha_{-1, t-1}}\right) \cdot C_0 \theta_l^2 \sqrt{d} \alpha_{-1, t-1},$$

where in the second inequality, it follows from the direct calculation of the integral of the Gaussian that $\mathbb{E}_{x \sim \mathcal{N}(0,1)}[\mathbb{1}(x > a)x] = p(a) > 0$ with $p(a)$ being the density of $\mathcal{N}(0, 1)$ at $a$. The $-(d \vee n)^{-c_0}$ on the right-hand side is negligible. Note that the indicator is selecting the larger half of $\theta_l$, and we can thereby obtain the following lower bound

$$C^{-1} N_2 \cdot C_0 \sqrt{d} \alpha_{-1, t-1} \cdot \overline{\theta^2} Q_t, \quad \text{where} \quad Q_t = \frac{1}{N_2} \sum_{l=1}^{N_2} \mathbb{1}\left(\theta_l > \frac{-b_t}{\sqrt{d} \alpha_{-1, t-1}}\right), \quad \overline{\theta^2} = \frac{\|\theta\|_2^2}{N_2}.$$

**Upper bounding the signal term.** To arrive at an upper bound, we use the fact that $\varphi(x; b_t) \leq (d \vee n)^{-c_0} \mathbb{1}(x < -\bar{b}_t) + Lx \mathbb{1}(x \geq -\bar{b}_t)$ to obtain that

$$\sum_{l=1}^{N_2} \mathbb{E}_{x \sim \mathcal{N}(0,1)} \left[ \theta_l \cdot \varphi\left( \sqrt{1 - \theta_l^2} x + \theta_l \sqrt{d} \alpha_{-1,t-1}; b_t \right) \right]$$

$$\leq L \sum_{l=1}^{N_2} \mathbb{E}_{x \sim \mathcal{N}(0,1)} \left[ \theta_l \mathbb{1}\left( x + \frac{\theta_l \sqrt{d} \alpha_{-1,t-1} + b_t}{\sqrt{1 - \theta_l^2}} > 0 \right) \cdot \left( \sqrt{1 - \theta_l^2} x + \theta_l \sqrt{d} \alpha_{-1,t-1} \right) \right]$$

$$\qquad + N_2 (d \vee n)^{-c_0}$$

$$\leq CL \sum_{l=1}^{N_2} \left( \theta_l \sqrt{1 - \theta_l^2} + \theta_l^2 \sqrt{d} \alpha_{-1,t-1} \right) \leq CLN_2 \overline{\theta^2} \sqrt{d} \alpha_{-1,t-1},$$

where the last second inequality holds by noting that $\mathbb{E}[\mathbb{1}(x > a)x] = p(|a|) \leq 1$, and the last one holds by noting that $\sqrt{d} \alpha_{-1,t-1} \gg 1$. $\qquad \square$

# I  PROOFS FOR SAE DYNAMICS ANALYSIS

In this section, we provide supplementary proofs for the results used in the proof of the main theorem in §G.

## I.1  PROOF OF THEOREM G.2

Let us first prove that there must exists some $i \in [n]$ such that $\overline{\theta_i^2} \geq 1/s$. Since the total sum $\sum_{j \in [n]} \sum_{l \in \mathcal{D}_j} H_{l,j}^2 = \sum_{l=1}^N \|h_l\|_2^2 = N$, and there are at most $Ns$ non-zero entries in the weight matrix $H$, we have the average

$$\overline{H^2} := \frac{\sum_{l=1}^N \sum_{j=1}^n H_{l,j}^2}{\sum_{l=1}^N \sum_{j=1}^n \mathbb{1}(H_{l,j} > 0)} \geq \frac{N}{Ns} = \frac{1}{s}.$$

On the other hand, we also have

$$\overline{H^2} = \frac{\sum_{j=1}^n |\mathcal{D}_j| \cdot \overline{\theta_j^2}}{\sum_{j=1}^n |\mathcal{D}_j|} \leq \max_{j \in [n]} \overline{\theta_j^2}.$$

It thus follows that there exists some $i \in [n]$ such that $\overline{\theta_i^2} \geq 1/s$.

**Proof of the first inequality.** By definition of $h_\star$, we have $h_\star^2 \geq \hbar_{q,\star}^2$ for $q = 4$. To prove the upper bound on $h_\star$, we just need to show that $\hbar_{q,\star}^2 \geq \overline{\theta_j^2}$ for any $j \in [n]$. Let us consider the kernel function in the definition of $\hbar_{q,\star}$:

$$f(x) = \Phi\left( \frac{-\bar{b}}{\sqrt{\frac{q-1}{q} x + \frac{1}{q}}} \right).$$

In particular, we aim to show that $f(\cdot)$ is convex for $x \in [0,1]$. The second derivative of $f(x)$ is given by

$$f''(x) = p\left( \frac{-\bar{b}}{\sqrt{\frac{q-1}{q} x + \frac{1}{q}}} \right) \cdot \frac{\bar{b}\left( \frac{q-1}{q} \right)^2}{4\left( \frac{q-1}{q} x + \frac{1}{q} \right)^{7/2}} \cdot \left[ 3\left( \frac{q-1}{q} x + \frac{1}{q} \right) - \bar{b}^2 \right].$$

Using the property that $\bar{b} < -\sqrt{3}$, we conclude that $f''(x) \geq 0$ for $x \in [0,1]$, and $f$ is convex. Now, by definition of $\hbar_{q,\star}$, we have

$$f(\hbar_{q,\star}^2) \geq \max_{j \in [n]} \frac{1}{|\mathcal{D}_j|} \sum_{l \in \mathcal{D}_j} f(H_{l,j}^2) \geq \max_{j \in [n]} f\left( \frac{1}{|\mathcal{D}_j|} \sum_{l \in \mathcal{D}_j} H_{l,j}^2 \right) = \max_{j \in [n]} f(\overline{\theta_j^2}), \qquad \text{(I.1)}$$

where the second inequality follows from the convexity of $f(x)$ and Jensen's inequality. Moreover, the first derivative of $f(x)$ is given by

$$f'(x) = -p\Big(\frac{-\overline{b}}{\sqrt{\frac{q-1}{q}x + \frac{1}{q}}}\Big)\frac{\overline{b}\big(\frac{q-1}{q}\big)}{2\big(\frac{q-1}{q}x + \frac{1}{q}\big)^{3/2}} > 0.$$

Therefore, we have by (I.1) that $\hbar_{q,\star}^2 \geq \overline{\theta_j^2}$ for any $j \in [n]$ and $q = 4$. Consequently, $h_\star^2 \geq \hbar_{4,\star}^2 \geq \max_{j\in[n]} \overline{\theta_j^2} \geq 1/s$. This proves the first inequality.

**Proof of the second inequality.** Since we have by definition of $\overline{\theta_i^2}$ that

$$\overline{\theta_i^2} = \frac{\|\theta_i\|_2^2}{|\mathcal{D}_i|} \leq (1 - \widehat{\mathbb{Q}}_i(h_i)) \cdot h_i^2 + \widehat{\mathbb{Q}}(h_i) \cdot 1 \leq \widehat{\mathbb{Q}}_i(h_i) + h_i^2,$$

it follows from the condition $\overline{\theta_i^2} > \widehat{\mathbb{Q}}_i(h_i)$ that

$$h_i \geq \sqrt{\overline{\theta_i^2} - \widehat{\mathbb{Q}}(s_i^{-1/2})}.$$

This completes the proof of the third inequality. Hence, we have completed the proof of Theorem G.2.

## I.2 PROOFS FOR CONCENTRATION RESULTS COMBINED

In the following, we present the proofs of the lemmas and propositions used in §G.2.

### I.2.1 PROOF OF THEOREM G.5

From $\Phi(|\overline{b}_t|) \gg Ls\rho_1(t\log n)^3$, we deduce that $t\log n \ll n$, since $Ls\rho_1 n^3 \gg 1 \geq \Phi(|\overline{b}_t|)$ (recalling that $\rho_1 \geq n^{-1}$). Hence, we can directly apply Theorem G.4 in what follows. Using the bound in Theorem F.7 together with Theorem F.8, if we further assume $\Phi(|\overline{b}_t|) \gg Ls\rho_1(t\log n)^3$, then the desired concentration result is obtained as follows:

$$\langle z_\tau, E^\top\varphi(Ey_t^\star; b_t)\rangle = (1 \pm o(1)) \cdot N\alpha_{\tau,t-1}\widehat{\varphi}_1(b_t) \pm C\sqrt{1 - \alpha_{\tau,t-1}^2} \cdot \sqrt{\|E^\top\varphi(Ey_t^\star; b_t)\|_2^2 \cdot t\log(n)}. \tag{I.2}$$

Here, we use the fact that $|N_1/N - 1| \leq \rho_1 \ll 1$, where $\rho_1 \ll 1$ can also be deduced from the condition $\Phi(|\overline{b}_t|) \gg Ls\rho_1(t\log(n))^3$. For the concentration result for $\langle z_\tau, F^\top\varphi(Fy_t + \theta \cdot v^\top\overline{w}_{t-1}; b_t)\rangle$ in Theorem F.9, we use the Stein's lemma to derive that

$$\frac{N_2}{N}\sum_{l=1}^{N_2} |\alpha_{\tau,t-1}|\sqrt{1 - \theta_l^2} \cdot \mathbb{E}_{x\sim\mathcal{N}(0,1)}\Big[x\varphi\big(\sqrt{1 - \theta_l^2}x + \theta_l v^\top\overline{w}_{t-1}; b_t\big)\Big]$$

$$\leq \frac{N_2|\alpha_{\tau,t-1}|}{N}\sum_{l=1}^{N_2}(1 - \theta_l^2) \cdot \mathbb{E}_{x\sim\mathcal{N}(0,1)}\Big[\varphi'\big(\sqrt{1 - \theta_l^2}x + \theta_l v^\top\overline{w}_{t-1}; b_t\big)\Big]$$

$$\leq \rho_1|\alpha_{\tau,t-1}|L = o(\Phi(|\overline{b}_t|) \cdot |\alpha_{\tau,t-1}|) \tag{I.3}$$

where in the second inequality we use the Lipschitzness of $\varphi$ and in the last inequality we use $Ls\rho_1(t\log n)^3 \ll \Phi(|\overline{b}_t|)$. Moreover, we have

$$L|\alpha_{\tau,t-1}| \cdot \frac{N_2}{N} \cdot \big(\sqrt{t\log(n)} + \|v\|_2|\alpha_{-1,t-1}|\big) \cdot \sqrt{\rho_2 s} \cdot (t\log(n))^{3/2}$$

$$\leq L|\alpha_{\tau,t-1}| \cdot \rho_1\sqrt{\rho_2 s}(t\log n)^2 + \rho_1\sqrt{\rho_2 s}(t\log n)^{3/2} \cdot d|\alpha_{-1,t-1}\alpha_{\tau,t-1}|$$

$$\leq o(\Phi(|\overline{b}_t|) \cdot |\alpha_{\tau,t-1}|) + \rho_1\sqrt{\rho_2 s}(t\log n)^{3/2} \cdot d\alpha_{-1,t-1}\alpha_{\tau,t-1}. \tag{I.4}$$

where in the first inequality, we use $N_2/N \leq \rho_1$ by definition and in the second inequality, we use the fact $\rho_1\sqrt{\rho_2 s}(t\log n)^2 \leq \rho_1(t\log n)^2 \ll \Phi(|\overline{b}_t|)$ under the condition $Ls\rho_1(t\log n)^3 \ll \Phi(|\overline{b}_t|)$.

Moreover, by Theorem F.8, we know that $\widehat{\varphi}_1(b_t) = \Omega(\Phi(|\bar{b}_t|))$. Consequently, by combining (I.3) and (I.4) with the upper bound in Theorem F.9, we have

$$
\begin{aligned}
&\left|\langle z_\tau, F^\top \varphi(Fy_t + \theta \cdot v^\top \bar{w}_{t-1}; b_t)\rangle\right| \\
&\quad \le o\left(N|\alpha_{\tau,t-1}|\widehat{\varphi}_1(b_t)\right) + C\sqrt{1-\alpha_{\tau,t-1}^2} \cdot \sqrt{\|F^\top \varphi(Fy + \theta \cdot v^\top \bar{w}_{t-1}; b_t)\|_2^2 \cdot t\log(n)} \\
&\quad\quad + CNL\rho_1\sqrt{\rho_2 s}(t\log n)^{3/2} \cdot d|\alpha_{\tau,t-1}\alpha_{-1,t-1}|.
\end{aligned} \tag{I.5}
$$

Let us consider the good event with respect to some universal constant $C > 0$:

$$
\mathcal{E} : \left\{\|z_\tau\|_\infty \le C\sqrt{\log(tn)}, \quad \forall \tau \le T\right\}.
$$

As we increase the constant $C$, the failure probability of the event $\mathcal{E}$ can be made polynomially small, e.g., $1 - n^{-c}$ for some other constant $c > 0$ (See Theorem J.2). Conditioned on the success of this event, we have for the non-Gaussian components that

$$
\begin{aligned}
|\langle z_\tau, \Delta E_t\rangle + \langle z_\tau, \Delta F_t\rangle| &\le C\sqrt{\log(tn)} \cdot \left(\|\Delta E_t\|_1 + \|\Delta F_t\|_1\right) \\
&\le CLN\sqrt{\log(n)} \cdot \left(\sqrt{s\rho_1}(\sqrt{\Phi(|\bar{b}_t|)} + \sqrt{s\rho_1 t\log n}) \cdot \sqrt{d}\beta_{t-1} + \sqrt{s}\rho_1|\bar{b}_t|d\beta_{t-1}^2\right) \\
&\quad + CLN\sqrt{\log(n)} \cdot \rho_1\sqrt{sd}\beta_{t-1} \\
&\le CLN\sqrt{\log n} \cdot \left(\sqrt{s\rho_1 d\Phi(|\bar{b}_t|)}\beta_{t-1} + \sqrt{s}\rho_1|\bar{b}_t|d\beta_{t-1}^2\right), \tag{I.6}
\end{aligned}
$$

where in the second inequality, we invoke Theorem F.12 and Theorem F.13 to bound the $\ell_1$ norm of the error terms, and also the fact that $t$ is at most polynomial in $n$. In the last inequality, we use the fact that $\|\Delta y_t\|_2 \le \sqrt{d}\beta_{t-1}$ by Theorem F.1. Now, we combine the derived concentration results in (I.2), (I.5) and (I.6) with $1 - \alpha_{\tau,t-1}^2 \le 1$ and the upper bound for $\|E^\top\varphi(Ey_t^\star; b_t)\|_2^2 + \|F^\top\varphi(Fy_t + \theta \cdot v^\top\bar{w}_{t-1}; b_t)\|_2^2$ in Theorem G.4 to obtain that

$$
\begin{aligned}
\langle z_\tau, u_t\rangle &= \langle z_\tau, E^\top\varphi(Ey_t^\star; b_t)\rangle + \langle z_\tau, F^\top\varphi(Fy_t + \theta \cdot v^\top\bar{w}_{t-1}; b_t)\rangle + \langle z_\tau, \Delta E_t\rangle + \langle z_\tau, \Delta F_t\rangle \\
&= N\alpha_{\tau,t-1}\widehat{\varphi}_1(b_t) \cdot (1 \pm o(1)) \pm CNL\rho_1\sqrt{\rho_2 s}(t\log n)^{3/2} \cdot d|\alpha_{\tau,t-1}\alpha_{-1,t-1}| \\
&\quad \pm CN\rho_1 L\sqrt{t\log n} \cdot \xi_t \pm CLN\sqrt{\log n} \cdot \left(\sqrt{s\rho_1 d\Phi(|\bar{b}_t|)}\beta_{t-1} + \sqrt{s}\rho_1|\bar{b}_t|d\beta_{t-1}^2\right).
\end{aligned}
$$

Hence, we complete the proof of the Theorem G.5.

### I.2.2 PROOF OF THEOREM G.6

Recall by definition of $w_t$, $\langle v, w_t\rangle/\|v\|_2$ can be decomposed into

$$
\frac{\langle v, w_t\rangle}{\|v\|_2} = \langle z_{-1}, u_t\rangle + \|v\|_2 \cdot \theta^\top\varphi(Fy_t + \theta \cdot v^\top\bar{w}_{t-1}; b_t) + \eta^{-1}\alpha_{-1,t-1}. \tag{I.7}
$$

Taking $\tau = -1$ in Theorem G.5, we have

$$
\begin{aligned}
\langle z_{-1}, u_t\rangle &= N\alpha_{-1,t-1}\widehat{\varphi}_1(b_t) \cdot (1 \pm o(1)) \pm CNL\rho_1\sqrt{\rho_2 s}(t\log n)^{3/2} \cdot d|\alpha_{-1,t-1}|^2 \\
&\quad \pm CN\rho_1 L\sqrt{t\log n} \cdot \xi_t \pm CLN\sqrt{\log n} \cdot \left(\sqrt{s\rho_1 d\Phi(|\bar{b}_t|)}\beta_{t-1} + \sqrt{s}\rho_1|\bar{b}_t|d\beta_{t-1}^2\right). \tag{I.8}
\end{aligned}
$$

Moreover, by a direct decomposition of the second term, we have

$$
\begin{aligned}
\|v\|_2\theta^\top\varphi(Fy_t + \theta \cdot v^\top\bar{w}_{t-1}; b_t) &= \|v\|_2\theta^\top\varphi(Fy_t^\star + \theta \cdot v^\top\bar{w}_{t-1}; b_t) + \|v\|_2\theta^\top\Delta\varphi_{F,t} \\
&= \|v\|_2\theta^\top\varphi(Fy_t^\star + \theta \cdot v^\top\bar{w}_{t-1}; b_t) \pm \|v\|_2\|\theta\|_2 \cdot \|\Delta\varphi_{F,t}\|_2.
\end{aligned}
$$

Notice that $\|v\|_2 = \sqrt{d} \cdot (1 \pm C\sqrt{\log(n)/d})$ with probability at least $1 - n^{-c}$ by concentration of $\chi^2$ random variables (see Theorem J.1). By Theorem F.13, we have $\|\Delta\varphi_{F,t}\|_2 \le \sqrt{\rho_2 N_2}L \cdot \|\Delta y_t\|_2 \le \sqrt{\rho_2 N_2 d}L\beta_{t-1}$. Therefore,

$$
\|v\|_2\|\theta\|_2 \cdot \|\Delta\varphi_{F,t}\|_2 \le C\sqrt{d} \cdot \sqrt{N_2\overline{\theta^2}} \cdot L\sqrt{\rho_2 N_2 d}L\beta_{t-1} \le CLN\rho_1 d\sqrt{\rho_2}\beta_{t-1}.
$$

Now, combining the concentration results for $\theta^\top \varphi(F^\top y_t^\star + \theta \cdot v^\top \bar{w}_{t-1}; b_t)$ in Theorem F.10, we obtain that

$$
\begin{aligned}
\|v\|_2\, \theta^\top \varphi(Fy_t + \theta \cdot v^\top \bar{w}_{t-1}; b_t) \\
= (1 \pm o(1)) N\psi_t \pm C\,NL\,\rho_1\sqrt{\rho_2\, s}\,(t\log n)^{3/2}\,d\,\alpha_{-1,t-1} \pm C\,NL\,\rho_1 d\sqrt{\rho_2}\beta_{t-1}. \quad \text{(I.9)}
\end{aligned}
$$

Furthermore, we have by Theorem F.11 that $N\psi_t \gtrsim C_0\overline{\theta^2}Q_t \cdot N_2 d\alpha_{-1,t-1}$. Under the conditions

$$
\frac{N_2}{N} C_0\overline{\theta^2}Q_t \gg \max\Big\{ L\rho_1\sqrt{\rho_2 s}(t\log n)^{3/2},\ Ld^{-1}\Phi(|\bar{b}_t|),\ L\sqrt{t\log n}\rho_1\frac{\xi_t}{d\alpha_{-1,t-1}} \Big\},
$$

we conclude by also noting that $\sqrt{d}\alpha_{-1,t-1} \gg 1$ that

$$
N\psi_t \gg \max\Big\{ CNL\rho_1\sqrt{\rho_2 s}(t\log n)^{3/2} \cdot d\alpha_{-1,t-1},\ N\alpha_{-1,t-1}\hat{\varphi}_1(b_t),\ CN\rho_1 L\sqrt{t\log n} \cdot \xi_t \Big\}.
$$

Now we plug (I.9) and (I.8) into (I.7) to obtain

$$
\begin{aligned}
\frac{\langle v, w_t\rangle}{\|v\|_2} = (1 \pm o(1)) N\psi_t + \eta^{-1}\alpha_{-1,t-1} \\
\pm CLN\sqrt{d\rho_1 s\log n} \cdot \big( \sqrt{\Phi(|\bar{b}_t|)} + \sqrt{\rho_1 d\rho_2 s^{-1}} + \sqrt{\rho_1 d}|\bar{b}_t|\beta_{t-1} \big) \cdot \beta_{t-1}. \quad \text{(I.10)}
\end{aligned}
$$

Finally, under the conditions $\sqrt{ts\log n}|\bar{b}_t|\beta_{t-1} \ll 1$, $st\log n \cdot \Phi(|\bar{b}_t|) \ll \rho_1 d$, we have

$$
\sqrt{d\rho_1 s\log n} \cdot \big( \sqrt{\Phi(|\bar{b}_t|)} + \sqrt{\rho_1 d\rho_2 s^{-1}} + \sqrt{\rho_1 d}|\bar{b}_t|\beta_{t-1} \big) \le C\rho_1 d.
$$

Here, we use the fact that $\rho_2\log n \ll 1$, which can be deduced from the following inequality under the condition $\frac{N_2}{N}C_0\overline{\theta^2}Q_t \gg L\rho_1\sqrt{\rho_2 s}(t\log n)^{3/2}$:

$$
\rho_1 \gtrsim \frac{N_2}{N}C_0\overline{\theta^2}Q_t \gg L\rho_1\sqrt{\rho_2 s}(t\log n)^{3/2} \ge \rho_1\sqrt{\rho_2\log n}.
$$

Moreover, under the condition

$$
\frac{N_2}{N}C_0\overline{\theta^2}Q_t \gg CL\rho_1 \cdot \frac{\beta_{t-1}}{\alpha_{-1,t-1}},
$$

we conclude that the second line of (I.10) can be upper bounded by $o(N\psi_t)$. Hence, the proof of Theorem G.6 is completed.

### I.2.3 PROOF OF THEOREM G.7

Recall from the definition of $w_t$ that

$$
\begin{aligned}
\|P^\perp_{w_{-1:0}}(w_t - \eta^{-1}\bar{w}_{t-1})\|_2^2 = \sum_{\tau=1}^{t-1}\Big( \langle z_\tau, u_t\rangle - \langle P_{u_{1:\tau}}z_\tau, u_t\rangle + \frac{\langle u_\tau^\perp, u_t\rangle}{\|u_\tau^\perp\|_2} \cdot \frac{\|w_\tau^\perp\|_2}{\|u_\tau^\perp\|_2} \Big)^2 \\
+ \|P^\perp_{w_{-1:t-1}}\tilde{z}_t\|_2^2 \cdot \|u_t^\perp\|_2^2. \quad \text{(I.11)}
\end{aligned}
$$

**Lemma I.1.** *Assume that $T \le \sqrt{d}$ and $d \in (n^{1/c_1}, n^{c_1})$ for some universal constant $c_1 \in (0,1)$. Then there exist universal constants $c, C > 0$ such that with probability at least $1 - n^{-c}$ over the randomness of i.i.d. standard Gaussian vectors $z_{-1:T}$, for all $t \in [T]$,*

$$
\sum_{\tau=1}^{t-1}\langle P_{u_{1:\tau}}z_\tau, u_t\rangle^2 + \sum_{\tau=1}^{t-1}\Big( \frac{\langle u_\tau^\perp, u_t\rangle}{\|u_\tau^\perp\|_2} \cdot \frac{\|w_\tau^\perp\|_2}{\|u_\tau^\perp\|_2} \Big)^2 + \|P^\perp_{w_{-1:t-1}}\tilde{z}_t\|_2^2 \cdot \|u_t^\perp\|_2^2 \le Cd \cdot \|u_t\|_2^2.
$$

*Proof.* See §I.5.1 for a detailed proof. $\qquad\square$

**Lemma I.2** (Upper Bound for $\|u_t\|_2^2$). *If $t\log n \ll n$, $-\bar{b}_t = \Theta(\sqrt{\log n})$, $\rho_1 \ll 1$, it holds with probability at least $1 - n^{-c}$ for all $t \le T < \sqrt{d}$ that*

$$
\|u\|_2 \le CNL\rho_1(\xi_t + \sqrt{d}\beta_{t-1}).
$$

*Proof.* See §I.5.2 for a detailed proof. $\qquad\square$

Combining Theorems I.1 and I.2, it holds with probability at least $1 - n^{-c}$ for all $t \le \sqrt{d}$,

$$\sqrt{\sum_{\tau=1}^{t-1}\langle P_{u_{1:\tau}}z_\tau, u_t\rangle^2 + \sum_{\tau=1}^{t-1}\Big(\frac{\langle u_\tau^\perp, u_t\rangle}{\|u_\tau^\perp\|_2}\cdot\frac{\|w_\tau^\perp\|_2}{\|u_\tau^\perp\|_2}\Big)^2 + \|P_{w_{-1:t-1}}^\perp \tilde{z}_t\|_2^2\cdot\|u_t^\perp\|_2^2}$$
$$\le C\sqrt{d}\cdot\|u_t\|_2 \le CNL\rho_1\sqrt{d}\big(\xi_t + \sqrt{d}\beta_{t-1}\big). \tag{I.12}$$

It remains to upper bound $\sum_{\tau=1}^{t-1}\langle z_\tau, u_t\rangle^2$. Recall that $\beta_{t-1} = \sqrt{1 - \alpha_{-1,t-1}^2 - \alpha_{0,t-1}^2} = \sqrt{\sum_{\tau=1}^{t-1}\alpha_{\tau,t-1}^2}$. Using Theorem G.5, we conclude that

$$\sqrt{\sum_{\tau=1}^{t-1}\langle z_\tau, u_t\rangle^2} \le CN\beta_{t-1}\widehat{\varphi}_1(b_t)\cdot(1\pm o(1)) + CNL\rho_1\sqrt{\rho_2 s}(t\log n)^{3/2}\cdot d\,|\alpha_{-1,t-1}|\beta_{t-1}$$

$$+ CN\rho_1 Lt\sqrt{\log n}\cdot\xi_t + CLN\sqrt{t\log n}\cdot\big(\sqrt{s\rho_1 d\Phi(|\bar{b}_t|)} + \sqrt{s}\rho_1|\bar{b}_t|d\,\beta_{t-1}\big)\cdot\beta_{t-1}$$
$$\le CN\rho_1 Lt\sqrt{\log n}\cdot\xi_t + CLN\rho_1 d\beta_{t-1}, \tag{I.13}$$

where in the first inequality, the $\beta_{t-1}$ terms in the first line is obtained by the Pythagorean sum with respect to $\alpha_{\tau,t-1}$ for $\tau = 1, \dots, t-1$. In the second line, an additional $\sqrt{t-1}$ factor is added to the upper bound for $|\langle z_\tau, u_t\rangle|$ since $\sqrt{\sum_{\tau=1}^{t-1}x_\tau^2} \le \sqrt{t}\cdot\max_{\tau=1,\dots,t-1}|x_\tau|$. In the last inequality, we use the conditions $\sqrt{\rho_2 s}(t\log n)^{3/2} \ll 1$, $\Phi(|\bar{b}_t|) \ll \rho_1 d(st\log n)^{-1}$, and $\sqrt{st\log n}|\bar{b}_t|\beta_{t-1} \ll 1$ to upper bound all the terms containing $\beta_{t-1}$ by $CLN\rho_1 d\beta_{t-1}$. Plugging (I.12) and (I.13) into (I.11), we obtain

$$\|P_{w_{-1:0}}^\perp w_t\|_2 \le C\sqrt{d}\cdot\|u_t\|_2 + C\sqrt{\sum_{\tau=1}^{t-1}\langle z_\tau, u_t\rangle^2} + \eta^{-1}\beta_{t-1} \le CNL\rho_1\sqrt{d}\big(\xi_t + \sqrt{d}\beta_{t-1}\big) + \eta^{-1}\beta_{t-1}.$$

Here, we use the fact that $t\sqrt{\log n} \le \sqrt{d}$, which is implied by the condition $\rho_1 d(st\log n)^{-1} \gg \Phi(|\bar{b}_t|) \gg Ls\rho_1(t\log(n))^3$. Lastly, by condition $\eta^{-1} \ll N\Phi(|\bar{b}_t|)$ and the fact that $L\rho_1 d \gg \Phi(|\bar{b}_t|)$ by assumption, we can absorb the $\eta^{-1}\beta_{t-1}$ term into the $CNL\rho_1 d\beta_{t-1}$ term. Hence, we complete the proof of Theorem G.7.

### I.2.4 PROOF OF THEOREM G.8

Recall by definition of $w_t$ that

$$\|P_{w_{-1:0}}w_t\|_2 = \sqrt{\frac{\langle v, w_t\rangle^2}{\|v\|_2^2} + \big(\langle z_0, u_t\rangle + \eta^{-1}\alpha_{0,t-1}\big)^2}.$$

By Theorem G.6, we already have $\langle v, w_t\rangle/\|v\|_2 = (1\pm o(1))N\psi_t$. It remains to characterize $\langle z_0, u_t\rangle$. We have by Theorem G.5 that

$$\langle z_0, u_t\rangle = N\alpha_{0,t-1}\widehat{\varphi}_1(b_t)\cdot(1\pm o(1)) \pm CNL\rho_1\sqrt{\rho_2 s}(t\log n)^{3/2}\cdot d\,|\alpha_{0,t-1}\alpha_{-1,t-1}|$$

$$\pm CN\rho_1 L\sqrt{t\log n}\cdot\xi_t \pm CLN\cdot\big(\sqrt{s\log(n)\rho_1 d\Phi(|\bar{b}_t|)} + \sqrt{s\log(n)}\rho_1|\bar{b}_t|d\beta_{t-1}\big)\cdot\beta_{t-1}$$

$$= N\alpha_{0,t-1}\widehat{\varphi}_1(b_t)\cdot(1\pm o(1)) \pm CNL\rho_1\sqrt{\rho_2 s}(t\log n)^{3/2}\cdot d\,|\alpha_{0,t-1}\alpha_{-1,t-1}|$$

$$\pm CN\rho_1 L\sqrt{t\log n}\cdot\xi_t \pm CLN\rho_1 d\beta_{t-1}$$

Here, in the last term we use the condition $\sqrt{ts\log n}|\bar{b}_t|\beta_{t-1} \ll 1$ to upper bound $\sqrt{s\log n}|\bar{b}_t|\beta_{t-1} \ll 1$, and $\rho_1 d(st\log n)^{-1} \gg \Phi(|\bar{b}_t|)$ to upper bound $\sqrt{s\log(n)\rho_1 d\Phi(|\bar{b}_t|)} \le$

$C\rho_1 d$. Note that the fluctuation terms are similar to the one for $\langle z_{-1}, u_t \rangle$ in the proof of Theorem G.6. Specifically, under the same conditions

$$\frac{N_2}{N} C_0 \overline{\theta^2} Q_t \gg \max\left\{ L\rho_1 \sqrt{\rho_2 s} (t \log n)^{3/2}, \ L\sqrt{t \log n} \rho_1 \frac{\xi_t}{d\alpha_{-1,t-1}}, \ L\rho_1 \frac{\beta_{t-1}}{\alpha_{-1,t-1}} \right\}$$

we have

$$N\psi_t \gg CNL \cdot \max\left\{ \rho_1 \sqrt{\rho_2 s} (t \log n)^{3/2} \cdot d|\alpha_{0,t-1}\alpha_{-1,t-1}|, \ \rho_1 \sqrt{t \log n} \cdot \xi_t, \ \rho_1 d\beta_{t-1} \right\}.$$

Thus, we conclude that $\langle z_0, u_t \rangle = N\alpha_{0,t-1}\widehat{\varphi}_1(b_t) \cdot (1 \pm o(1)) \pm o(N\psi_t)$. Thus,

$$\begin{aligned}
\|P_{w_{-1:0}} w_t\|_2 &= \sqrt{\frac{\langle v, w_t \rangle^2}{\|v\|_2^2} + (\langle z_0, u_t \rangle + \eta^{-1}\alpha_{0,t-1})^2} \\
&= \sqrt{\left(N\psi_t \cdot (1 \pm o(1))\right)^2 + \left(N\alpha_{0,t-1}\widehat{\varphi}_1(b_t) \cdot (1 \pm o(1)) \pm o(N\psi_t) + \eta^{-1}\alpha_{0,t-1}\right)^2} \\
&= (1 \pm o(1)) \cdot \sqrt{(N\psi_t)^2 + (N\alpha_{0,t-1}\widehat{\varphi}_1(b_t))^2}.
\end{aligned}$$

Here, the last inequality holds by also noting that $\eta^{-1} \ll N\Phi(|\overline{b}_t|) \leq CN\widehat{\varphi}_1(b_t)$. This completes the proof.

### I.3 PROOFS FOR RECURSION ANALYSIS

#### I.3.1 PROOF OF THEOREM G.10

What we need to prove here is that all the conditions in Theorem G.9 hold for the current time step $t$ if the conditions in Theorem G.10 hold. This is because the conditions in Theorem G.9 are the union of the conditions in Theorems G.4 to G.8. In the following, we check all the listed conditions one by one.

**Step I: Checking all conditions in Theorem G.9.** For the first step, we divide the conditions in Theorem G.9 into three groups.

*Group 1: Implication of Cond.(i) and Cond.(I).* We first notice that since $t \leq T$, conditions

$$-\overline{b}_t = \Theta(\sqrt{\log n}) < \zeta_1, \quad \kappa_0|\overline{b}_t| = O(1), \quad \sqrt{\rho_2 s}(t \log n)^{3/2} \ll 1, \quad \eta^{-1} \ll N\Phi(|\overline{b}_t|) \wedge N_2 dC_0 \overline{\theta^2} Q_t$$

are guaranteed by Cond.(i). Here, we need to be more careful about condition $\eta^{-1} \ll N_2 dC_0 \overline{\theta^2} Q_t$, as $Q_t$ is a function of $t$, and what we directly have in Cond.(i) is for $Q_1$ only. By definition $Q_t = \frac{1}{N_2} \sum_{l=1}^{N_2} \mathbb{1}\left(\theta_l > \frac{-b}{\sqrt{d}\alpha_{-1,t-1}}\right)$, we note that $Q_t$ is nondecreasing in $\alpha_{-1,t-1}$. Therefore, we have the following fact:

**Fact I.3.** *If $\alpha_{-1,t-1} \geq \alpha_{-1,0}$, then $Q_t \geq Q_1$.*

In fact, the condition $\alpha_{-1,t-1} \geq \alpha_{-1,0}$ is automatically guaranteed by Cond.(I). Therefore, the condition $\eta^{-1} \ll N_2 dC_0 \overline{\theta^2} Q_t$ will hold for all successive $t$ as long as it holds for $t = 1$ and $\alpha_{-1,t-1} \geq \alpha_{-1,0}$. Meanwhile, we also have by the same reasoning that

$$\sqrt{d}\alpha_{-1,t-1} \geq \sqrt{d}\alpha_{-1,0} \gg 1$$

where the last inequality is guaranteed by **InitCond-1**. The condition $\sqrt{ts \log n}|\overline{b}_t|\beta_{t-1} \ll 1$ is guaranteed by Cond.(I) as well.

*Group 2: Implication of Cond.(ii) to Cond.(iii).* The direct implication of Cond.(ii) is that

$$\rho_1 d(st \log n)^{-1} \gg \Phi(|\overline{b}_t|) \gg Ls\rho_1(t \log(n))^3.$$

Similarly, the direct implication of Cond.(II) and Cond.(iii) is that

$$\frac{N_2}{N} C_0 \overline{\theta^2} Q_t \gg \max\left\{ L\rho_1 \sqrt{\rho_2 s}(t \log n)^{3/2}, \ Ld^{-1}\Phi(|\overline{b}_t|), \ L\rho_1 \frac{\beta_{t-1}}{\alpha_{-1,t-1}} \right\}.$$

Here, we use the fact that $t \leq T$ and the monotonicity of $Q_t$ in Theorem I.3. It remains to check whether $\frac{N_2}{N} C_0 \overline{\theta^2} Q_t \gg L\sqrt{t \log n} \rho_1 \frac{\xi_t}{d\alpha_{-1,t-1}}$ holds.

***Group 3: Implication of Cond.(ii),Cond.(iii), Cond.(I) and Cond.(II).*** To verify this inequality $\frac{N_2}{N} C_0 \overline{\theta^2} Q_t \gg L\sqrt{t \log n} \rho_1 \frac{\xi_t}{d\alpha_{-1,t-1}}$, we just need to show that $\xi_t/\alpha_{-1,t-1} \leq C\xi_1/\alpha_{-1,0}$ for some universal constant $C > 0$, as the corresponding inequality for the latter is already guaranteed by Cond.(iii). Recall the definition of $\xi_t$ in Theorem G.4, the ratio $\xi_t/\alpha_{-1,t-1}$ is given by

$$\frac{\xi_t}{\alpha_{-1,t}} = \frac{\sqrt{s}\, t \log(n)\, \mathcal{K}_t + \rho_1^{-1}\sqrt{\Phi(|\bar{b}_t|) \cdot \widehat{\mathbb{E}}_{l,l'}\Big[\Phi\Big(|\bar{b}_t|\sqrt{\frac{1-\langle h_l, h_{l'}\rangle}{1+\langle h_l, h_{l'}\rangle}}\Big)\langle h_l, h_{l'}\rangle\Big] + \rho_2\sqrt{n}}}{\alpha_{-1,t-1}} + \sqrt{\rho_2 d}.$$

(I.14)

We obtain the above formula by the nonnegativity of $\alpha_{-1,t-1}$ guaranteed by Cond.(I).

**Proposition I.4.** *If $-\bar{b}_t \leq \sqrt{2 \log n}$ for some universal constant $\kappa > 0$, then for $t \geq 2$,*
$$\mathcal{K}_t \leq t \cdot \big(\mathcal{K}_1 + C\sqrt{\log n} \cdot (\beta_{t-1} + |\alpha_{-1,t-1}| + |\alpha_{-1,0}|)\big).$$

*Proof.* See §I.5.3 for a detailed proof. $\qquad\square$

Combining (I.14), Theorem I.4 and the fact that $\alpha_{-1,t-1} \geq t^2 \alpha_{-1,0} \geq \alpha_{-1,0}$ by Cond.(I), we have

$$\frac{\xi_t}{\alpha_{-1,t-1}} \leq \frac{\sqrt{s}t^2 \log(n)\mathcal{K}_1}{\alpha_{-1,t-1}} + C\sqrt{s}t^2 \log(n)^{3/2}\Big(\frac{\beta_{t-1}}{\alpha_{-1,t-1}} + 2\Big)$$

$$+ \frac{\rho_1^{-1}\sqrt{\Phi(|\bar{b}_t|) \cdot \widehat{\mathbb{E}}_{l,l'}\Big[\Phi\Big(|\bar{b}_t|\sqrt{\frac{1-\langle h_l, h_{l'}\rangle}{1+\langle h_l, h_{l'}\rangle}}\Big)\langle h_l, h_{l'}\rangle\Big] + \rho_2\sqrt{n}}}{\alpha_{-1,t-1}} + \sqrt{\rho_2 d}$$

$$\leq \frac{\xi_1}{\alpha_{-1,0}} + C\sqrt{s}t^2 \log(n)^{3/2}\Big(\frac{\beta_{t-1}}{\alpha_{-1,t-1}} + 2\Big),$$

(I.15)

where in the second inequality, we directly plug in the definition of $\xi_1$ with $t = 1$ in (I.14) and use the fact that $\alpha_{-1,t-1} \geq t^2 \alpha_{-1,0}$ to upper bound the first term in the right-hand side. Furthermore, for each term in Cond.(II), we have the following relationship:

$$\frac{N_2}{N} \leq \rho_1, \quad C_0 \overline{\theta^2} Q_t = O(1), \quad L = \Omega(1),$$

where the first inequality holds by direct definition of $\rho_1$ in (F.1), the second equality holds by noting that $\overline{\theta^2} \leq 1$, $Q_t \leq 1$ and $C_0$ is a universal constant, and the last inequality holds by **??**. Together, we have the following implication:

$$\frac{N_2}{N} C_0 \overline{\theta^2} Q_t \gg L\rho_1 \frac{\beta_{t-1}}{\alpha_{-1,t-1}} \implies \frac{\beta_{t-1}}{\alpha_{-1,t-1}} \ll 1$$

Therefore, we can further simplify the upper bound in (I.15) to

$$\frac{\xi_t}{\alpha_{-1,t-1}} \leq \frac{\xi_1}{\alpha_{-1,0}} + C\sqrt{s}t^2 \log(n)^{3/2} \cdot \mathbb{1}(t \geq 2).$$

(I.16)

Using (I.16), in order for condition $\frac{N_2}{N} C_0 \overline{\theta^2} Q_t \gg L\sqrt{t \log n} \rho_1 \frac{\xi_t}{d\alpha_{-1,t-1}}$ to hold, we just need to ensure

$$\frac{N_2}{N} C_0 \overline{\theta^2} Q_t \gg L\sqrt{t \log n} \rho_1 \frac{\xi_1}{d\alpha_{-1,0}}, \quad \frac{N_2}{N} C_0 \overline{\theta^2} Q_t \gg CLd^{-1}\rho_1 \sqrt{st^5}(\log n)^2.$$

The first one is clearly given by Cond.(iii), and the second one is satisfied because we have by using Cond.(ii) and Cond.(iii) that

$$\frac{N_2}{N} C_0 \overline{\theta^2} Q_t \gg Ld^{-1}\Phi(|\bar{b}|) \gg L^2 d^{-1}\rho_1 s(T \log(n))^3 \gtrsim CLd^{-1}\rho_1 \sqrt{st^5}(\log n)^2.$$

Here, the first inequality holds by the second condition in Cond.(iii), the second inequality holds by Cond.(ii), and the last inequality holds by noting that we are considering any $t \leq T$. The last inequality shows that the last condition $\frac{N_2}{N} C_0 \overline{\theta^2} Q_t \gg L\sqrt{t \log n} \rho_1 \frac{\xi_t}{d\alpha_{-1,t-1}}$ also holds automatically under the conditions in Theorem G.10. To this end, we have shown that all the conditions in Theorem G.9 hold for $t$ if the conditions in Theorem G.10 are satisfied.

**Step II: Deriving the recursion.** As we have shown in the previous step, all the conditions in Theorem G.9 hold for $t$ if the conditions in Theorem G.10 hold. Therefore, we can safely apply all the concentration results derived in §G.2. We next show how to use the previous derived concentration result on $\langle v, w_t \rangle / \|v\|_2$, $\|P_{w_{-1:0}} w_t\|_2$, and $\|P_{w_{-1:0}}^\perp w_t\|_2$ to control the recursion of $\beta_t / \alpha_{-1,t}$ and $1/\alpha_{-1,t}$. Since $\beta_t$ is the projection of $w_t$ onto the $P_{w_{-1:0}}$ direction, and $\alpha_{-1,t}$ is the projection of $w_t$ onto the $v$ direction, we have

$$
\frac{\beta_t}{\alpha_{-1,t}} = \frac{\|v\|_2 \cdot \|P_{w_{-1:0}}^\perp w_t\|_2}{\langle v, w_t \rangle} \le \frac{CL\rho_1 \sqrt{d}(\xi_t + \sqrt{d}\beta_{t-1})}{C_0 \overline{\theta^2} Q_t \cdot N_2/N \cdot d\alpha_{-1,t-1}}
$$

$$
\le \frac{CL\rho_1}{C_0 \overline{\theta^2} Q_t \cdot N_2/N} \cdot \left( \frac{1}{\sqrt{d}} \Big( \frac{\xi_1}{\alpha_{-1,0}} + C\sqrt{s}t^2 \log(n)^{3/2} \cdot \mathbb{1}(t \ge 2) \Big) + \frac{\beta_{t-1}}{\alpha_{-1,t-1}} \right).
$$

where in the first inequality, we use the upper bound for $\|P_{w_{-1:0}}^\perp w_t\|_2$ in Theorem G.7 and the lower bound for $\langle v, w_t \rangle / \|v\|_2$ in Theorem G.6 as $\langle v, w_t \rangle / \|v\|_2 \ge (1 - o(1))N\psi_t \gtrsim NC_0 \overline{\theta^2} Q_t \cdot N_2/N \cdot d\alpha_{-1,t-1}$ by the lower bound of $\psi_t$ in Theorem F.11. The second inequality holds by plugging in the upper bound for $\xi_t / \alpha_{-1,t-1}$ in (I.16). Similarly, we have by definition of $\alpha_{-1,t}$ that

$$
\frac{1}{\alpha_{-1,t}} = \frac{\|v\|_2 \cdot \|w_t\|_2}{\langle v, w_t \rangle} \le \frac{(1 + o(1)) \cdot \sqrt{\psi_t^2 + \widehat{\varphi}_1(b)^2} + CL\rho_1 \sqrt{d}\xi_t}{(1 - o(1)) \cdot \psi_t}
$$

$$
\le \frac{(1 + o(1)) \cdot \sqrt{(C_0 \overline{\theta^2} Q_t \cdot N_2/N \cdot d\alpha_{-1,t-1})^2 + (CL\Phi(|\bar{b}|))^2} + CL\rho_1 \sqrt{d}\xi_t}{C_0 \overline{\theta^2} Q_t \cdot N_2/N \cdot d\alpha_{-1,t-1}}
$$

$$
\le \frac{CL\rho_1}{C_0 \overline{\theta^2} Q_t \cdot N_2/N} \cdot \left( \frac{\Phi(|\bar{b}|)}{\rho_1 d} \cdot \frac{1}{\alpha_{-1,t-1}} + \frac{1}{\sqrt{d}} \Big( \frac{\xi_1}{\alpha_{-1,0}} + C\sqrt{s}t^2 \log(n)^{3/2} \cdot \mathbb{1}(t \ge 2) \Big) \right)
$$

$$
+ (1 + o(1)).
$$

where in the second inequality, we plug in the lower bound for $\psi_t$ and the upper bound for $\widehat{\varphi}_1(b_t)$ in Theorem F.8. The last inequality holds by the triangle inequality and the upper bound for $\xi_t / \alpha_{-1,t-1}$ in (I.16). This completes the proof of Theorem G.10.

### I.3.2 Proof of Theorem G.12

In the following proof, let us take $T_1 = \max\{(2\varsigma)^{-1}, 1\}$. As our goal is to establish that (G.5) and (G.6) holds for all $t \le T_1$, we just need to show that Cond.(I) and Cond.(II) hold for all $t \le T_1$, as they are the only conditions that might be violated over time, and the other conditions only depend on the initial conditions.

**Initial step.** For $t = 1$, we have $\alpha_{-1,t-1} = \alpha_{-1,0}$ and $\beta_{t-1} = \beta_0 = 0$. Hence, Cond.(I) and Cond.(II) hold trivially. Before we start the proof, we first derive some useful inequalities.

**Useful inequalities.** For $\lambda_1$, we have by Cond.(v) and Cond.(vi) that

$$
\lambda_1 = \frac{CL\rho_1}{C_0 \overline{\theta^2} Q_1 \cdot N_2/N} = \frac{\rho_1 d^{1-\varsigma}}{\Phi(|\bar{b}|)}, \quad \lambda_1 \xi_1 = \frac{\lambda_0 \xi_1}{Q_1} \ll \frac{d^{-\epsilon}}{\sqrt{s} \log n} \ll 1. \tag{I.17}
$$

Using the above two inequalities, we have by (G.4) that

$$
\frac{1}{\alpha_{-1,1}} \le 1 + o(1) + \Big( \frac{\lambda_1 \Phi(|\bar{b}|)}{\rho_1 d} + \frac{\lambda_1 \xi_1}{\sqrt{d}} \Big) \cdot \frac{1}{\alpha_{-1,0}} \le 1 + o(1) + (d^{1/2-\varsigma} + 1) \le 3 + d^{1/2-\varsigma}.
$$

In fact, we have the ratio $\alpha_{-1,0}/\alpha_{-1,1}$ as

$$
\frac{\alpha_{-1,0}}{\alpha_{-1,1}} \le (3 + d^{1/2-\varsigma}) \cdot \alpha_{-1,0} \le (3d^{-1/2} + d^{-\varsigma}) \cdot C\sqrt{\log M} \ll 1. \tag{I.18}
$$

Here, we use the fact that $\alpha_{-1,0} = O(\sqrt{\log M})$ with sufficiently high probability $1 - n^{-c}$, and $M = \text{poly}(n)$. The above inequality demonstrates that $\alpha_{-1,1}$ is guaranteed to grow in the first step. Thus, by definition of $Q_t$ in (F.15), we conclude that

$$
Q_2 = \frac{1}{N_2} \sum_{l=1}^{N_2} \mathbb{1}\Big( \theta_l \ge \frac{-b}{\sqrt{d}\alpha_{-1,1}} \Big) \ge \frac{1}{N_2} \sum_{l=1}^{N_2} \mathbb{1}\Big( \theta_l \ge |b|(3d^{-1/2} + d^{-\varsigma}) \Big) =: Q_2^\nu,
$$

where we take $\nu = |b|(3d^{-1/2} + d^{-\varsigma}) = O(\sqrt{\log n} \cdot d^{-\varsigma \wedge 1/2})$ and denote the right-hand side of the above inequality as $Q_2^\nu$. Since $\theta_l \in [0, 1]$, we have

$$\overline{\theta^2} = \frac{1}{N_2} \sum_{l=1}^{N_2} \theta_l^2 \le Q_2^\nu \cdot 1^2 + (1 - Q_2^\nu) \cdot \nu^2 = Q_2^\nu(1 - \nu^2) + \nu^2 \Rightarrow Q_2^\nu \ge \frac{\overline{\theta^2} - \nu^2}{1 - \nu^2} \ge \frac{\overline{\theta^2}}{2},$$

where the last inequality holds from **??** that $\overline{\theta^2} = \Omega(\text{polylog}(n)^{-1}) \gg \nu^2$. In the sequel, we will use $Q_2 \ge \overline{\theta^2}/2$ as the lower bound for $Q_2$. By definition of $T_1$, we have $T_1 = (2\varsigma)^{-1} \vee 1 = \Theta(1)$. In addition, for $\lambda_2$, we have

$$\lambda_2 = \frac{\lambda_0}{Q_2} \le \frac{2\lambda_0}{\overline{\theta^2}} = O(\text{polylog}(n)), \tag{I.19}$$

where in the inequality we use the lower bound for $Q_2$ and in the last equality we use $\overline{\theta^2} = \Omega(\text{polylog}(n)^{-1})$ in **??** and $\lambda_0 = O(\text{polylog}(n))$ in Cond.(iv).

We now have for the coefficient $\lambda_2 \Phi(|\bar{b}|)/(\rho_1 d)$ that

$$\frac{\lambda_2 \Phi(|\bar{b}|)}{\rho_1 d} = \frac{\lambda_0 \Phi(|\bar{b}|)}{Q_2 \rho_1 d} = \frac{Q_1 d^{-\varsigma}}{Q_2} \le d^{-\varsigma},$$

where the second identity holds from Cond.(v) and the last inequality holds by noting that $\alpha_{-1,1} \ge \alpha_{-1,0}$ by the first step's calculation in (I.18) and using the monotonicity of $Q_t$ in Theorem I.3. Next, we upper bound the quantity $C_1$ in (G.2):

$$C_1 = \left(1 + o(1) + \frac{\lambda_2 \xi_1}{\sqrt{d}\alpha_{-1,0}} + \frac{C\lambda_2\sqrt{s}T_0^2(\log n)^{3/2}}{\sqrt{d}}\right) \cdot \frac{1}{1 - \lambda_2\Phi(|\bar{b}|)/\rho_1 d}$$

$$\le \left(1 + o(1) + \frac{\lambda_1 \xi_1}{\sqrt{d}\alpha_{-1,0}} + \frac{C\sqrt{s}\,\text{polylog}(n)}{\sqrt{d}}\right) \cdot \frac{1}{1 - d^{-\epsilon}}$$

$$\le \left(1 + o(1) + \frac{d^{-\epsilon}}{\sqrt{s}\log n}\right) \cdot (1 + o(1)) = 1 + o(1),$$

where in the first inequality, we use the fact that $\lambda_2 \le \lambda_1$ by the fact $Q_2 \ge Q_1$, and we invoke the upper bound $T_0 \le \log n$ and $\lambda_2 = O(\text{polylog}(n))$ in (I.19). In the last inequality, we use the previous bound for $\lambda_1 \xi_1$ in (I.17) together with the fact that $\sqrt{d}\alpha_{-1,0} \ge 1$ by **InitCond-1**.

**Induction step.** Suppose the induction hypothesis holds for $1, 2, \ldots, t$. We will show that Cond.(I) and Cond.(II) hold for $t + 1 \le T_1$ as well. To this end, it is evident that $\alpha_{-1,t}$ is always growing before reaching $C_1$, which is evident from (G.3) by noting that $\lambda_2\Phi(|\bar{b}|)/\rho_1 d \le d^{-\varsigma} < 1$.

We first look at the recursion of $\alpha_{-1,t}$. By (G.5), the ratio $\alpha_{-1,0}/\alpha_{-1,t}$ is bounded by

$$\frac{\alpha_{-1,0}}{\alpha_{-1,t}} \le \left(\frac{\lambda_2\Phi(|\bar{b}|)}{\rho_1 d}\right)^{t-1} \cdot \left(\frac{\lambda_1\Phi(|\bar{b}|)}{\rho_1 d} + \frac{\lambda_1\xi_1}{\sqrt{d}}\right) + C_1\alpha_{-1,0}$$

$$\le d^{-\varsigma(t-1)} \cdot \left(d^{-\varsigma} + \frac{d^{-\epsilon}}{\sqrt{sd}\log n}\right) + (1 + o(1)) \cdot \frac{C\sqrt{\log M}}{\sqrt{d}} \le Cd^{-\varsigma(t-1)-(\varsigma \wedge 1/2)} + d^{-1/2+\epsilon}.$$

The first term on the right-hand side is decaying exponentially fast with respect to $t$. The second term is much smaller than $1/T_0^2$ given that $T_0 \le \log n$ by definition. Therefore, both terms are much smaller than $1/T_0^2$. This implies the first condition in Cond.(I) holds for $t + 1$.

Next, we look at the conditions involving $\beta_t$. By previous analysis on $T_1$ and the upper bound in (I.19), we obtain

$$\lambda_2^{T_1-1} \le (\text{polylog}(n))^{(2\varsigma)^{-1}\vee 1} = O(\text{polylog}(n)).$$

By recursion of $\beta_t/\alpha_{-1,t}$ in (G.6), we have

$$\frac{\beta_t}{\alpha_{-1,t}} \le \frac{\lambda_2^{t-1}}{\sqrt{d}} \cdot \left((T_0 + \lambda_1) \cdot \frac{\xi_1}{\alpha_{-1,0}} + C\sqrt{s}T_0^3\log(n)^{3/2}\right)$$

$$\le \frac{\text{polylog}(n)}{\sqrt{d}} \cdot \left(\frac{\lambda_1\xi_1}{\alpha_{-1,0}} + C\sqrt{s}\,\text{polylog}(n)\right)$$

$$\le \text{polylog}(n) \cdot \left(\frac{d^{-\epsilon}}{\sqrt{s}\log n} + \frac{C\sqrt{s}\,\text{polylog}(n)}{\sqrt{d}}\right) \le \frac{d^{-\epsilon}\,\text{polylog}(n)}{\sqrt{s}}.$$

Here, the second inequality holds by the upper bound for $\lambda_2^{T_1-1}$ and also the fact that $T_0 + \lambda_1 \leq 2\lambda_1 \log n$ since $\lambda_1 \geq 1$ and $T_0 \leq \log n$. In the second inequality, we use the upper bound for $\lambda_1\xi_1$ in (I.17) and the fact that $\sqrt{d}\alpha_{-1,0} \geq 1$ by **InitCond-1**. The last inequality holds because $\epsilon < 1/2$ by definition. Using the above inequality with the fact that $\alpha_{-1,t} \leq 1$, we obtain

$$\beta_t \leq \frac{d^{-\epsilon}\,\text{polylog}(n)}{\sqrt{s}} \ll \frac{1}{\sqrt{T_0 s \log n}|\bar{b}|},$$

where the last inequality holds by noting that both $T_0$ and $|\bar{b}|$ are at most $O(\text{polylog}(n))$. This implies that the second condition in Cond.(I) holds for $t+1$.

Eventually, for Cond.(II), we have

$$\frac{C_0\overline{\theta^2}Q_t \cdot N_2/N}{CL\rho_1} = \frac{Q_t}{\lambda_0} \geq \frac{Q_2}{\lambda_0} \geq \frac{\overline{\theta^2}}{2\lambda_0} = \Omega(\text{polylog}(n)^{-1}).$$

Therefore, the left-hand side of the above inequality is also much larger than $\beta_t/\alpha_{-1,t}$. To this end, we have finished the induction step and proved that Cond.(I) and Cond.(II) hold for all $t \leq T_1$.

**Final step.** According to the recursion in (G.5), let us consider the real value $t^\star$ that satisfies

$$\left(\frac{\lambda_2\Phi(|\bar{b}|)}{\rho_1 d}\right)^{t^\star-1} \cdot \left(\frac{\lambda_1\Phi(|\bar{b}|)}{\rho_1 d} + \frac{\lambda_1\xi_1}{\sqrt{d}}\right) \cdot \frac{1}{\alpha_{-1,0}} = \log(d)^{-c_0} \tag{I.20}$$

for some small constant $c_0 > 0$ to be determined later. We first note that we can obtain the $\varsigma \wedge 1/2$ factor by the inequality for $\lambda_1$ in (I.17) that

$$\frac{\lambda_2\Phi(|\bar{b}|)}{\rho_1 d} \leq \frac{\lambda_1\Phi(|\bar{b}|)}{\rho_1 d} \leq \frac{\lambda_1\Phi(|\bar{b}|)}{\rho_1 d} + \frac{\lambda_1\xi_1}{\sqrt{d}}, \quad \text{and} \quad \frac{\lambda_1\Phi(|\bar{b}|)}{\rho_1 d} + \frac{\lambda_1\xi_1}{\sqrt{d}} \leq d^{-\varsigma} + \frac{d^{-1/2-\epsilon}}{\sqrt{s}\log n} \tag{I.21}$$

Using the above inequality (I.21), and taking a logarithm of both sides with base $d$ for (G.5), we have for $t^\star$ that

$$t^\star \cdot \log_d\left(d^{-\varsigma} + \frac{d^{-1/2-\epsilon}}{\sqrt{s}\log n}\right) + \log_d\left(\frac{1}{\alpha_{-1,0}}\right) \geq -\frac{c_0 \log\log d}{\log d},$$

which implies that

$$t^\star \leq \log_d\left(\frac{1}{d^{-\varsigma} + \frac{d^{-1/2-\epsilon}}{\sqrt{s}\log n}}\right)^{-1} \cdot \left(\log_d\left(\frac{1}{\alpha_{-1,0}}\right) + \frac{c_0\log\log d}{\log d}\right) \leq \frac{1/2}{\varsigma \wedge 1/2} = (2\varsigma)^{-1} \vee 1 = T_1. \tag{I.22}$$

In the second inequality, we use the fact that by **InitCond-1**,

$$\log_d\left(\frac{1}{\alpha_{-1,0}}\right) = \log_d(\|v\|_2) - \log_d\left((1-\varepsilon)\sqrt{2\log(M/n)}\right) \leq \frac{1}{2} - \frac{\log\log(M/n)}{2\log d}.$$

Therefore, we can take $c_0$ to be small enough but still on a constant level such that

$$\frac{1}{2} - \frac{\log\log(M/n)}{2\log d} \leq \frac{1}{2} - \frac{c_0\log\log d}{2\log d}.$$

This justifies the second inequality in (I.22). Thus, there must exists some time $t \leq T_1$ such that (I.20) holds. For this time $t$, we already have

$$\frac{1}{\alpha_{-1,t}} \leq \left(\frac{\lambda_2\Phi(|\bar{b}|)}{\rho_1 d}\right)^{t-1} \cdot \left(\frac{\lambda_1\Phi(|\bar{b}|)}{\rho_1 d} + \frac{\lambda_1\xi_1}{\sqrt{d}}\right) \cdot \frac{1}{\alpha_{-1,0}} + C_1 \leq d^{-\varsigma} + C_1 \leq 1 + o(1).$$

This implies that $\alpha_{-1,t} = 1 - o(1)$.

**Checking $\alpha_{-1,t-1} \geq \alpha_{-1,1}$.** An additional step is needed to show that $\alpha_{-1,t-1} \geq \alpha_{-1,1}$ for all $t \geq 2$ and before $t^\star$ is reached. This is required because we want to ensure that before time $t^\star$, we always have $\alpha_{-1,t-1} \geq \alpha_{-1,1}$, and the stopping time $T_0$ will not prohibit us from reaching $t^\star$. In fact, we have by (G.3) that

$$\frac{1}{\alpha_{-1,t}} \leq \left(\frac{\lambda_2 \Phi(|\bar{b}|)}{\rho_1 d}\right)^{t-2} \cdot \left(\frac{1}{\alpha_{-1,1}} - C_1\right) + C_1.$$

Therefore, the ratio $\alpha_{-1,t-1}/\alpha_{-1,1}$ is bounded by

$$\frac{\alpha_{-1,1}}{\alpha_{-1,t-1}} \leq \left(\frac{\lambda_2 \Phi(|\bar{b}|)}{\rho_1 d}\right)^{t-2} \cdot \left(1 - C_1 \alpha_{-1,1}\right) + C_1 \alpha_{-1,1}.$$

We consider two cases. If $C_1 \alpha_{-1,1} \geq 1$, we can just stop the gradient at $t = 1$ and obtain $\alpha_{-1,1} = 1 - o(1)$ since $C_1 = 1 + o(1)$. In this case, we reach strong alignment in just one step. In another case where $C_1 \alpha_{-1,1} < 1$, since $\lambda_2 \Phi(|\bar{b}|)/(\rho_1 d) \leq d^{-\varsigma}$, we have the above ratio strictly upper bounded by 1. Hence, the condition $\alpha_{-1,t-1} \geq \alpha_{-1,1}$ holds for all $t \geq 2$ and before $t^\star$ is reached.

In both cases, we have shown that $\alpha_{-1,t-1} \geq \alpha_{-1,1}$ hold for $2 \leq t \leq t^\star$. As we have shown that Cond.(I) and Cond.(II) hold for all $t \leq T_1$ from the induction step, $t^\star \leq T_1$ from the final step, and $T_1 \leq \log(n)$ by definition, we conclude that $T_0$-Cond.(1) to $T_0$-Cond.(3) in the definition of the stopping time $T_0$ hold for all $t \leq t^\star$. In other words, we have shown that $T_0 \geq t^\star$.

Thus, we complete the proof of Theorem G.12.

## I.4 PROOFS FOR CONDITION SIMPLIFICATION

### I.4.1 PROOF OF THEOREM G.13

Let us take $t^\star$ as the maximum number of iterations considered. In the following, we first provide a sufficient condition for Cond.(iii), Cond.(v) and Cond.(vi) to hold. Then, we give a reformulation of Cond.(i), Cond.(ii) and Cond.(iv).

A sufficient condition for Cond.(iii) to hold is given by

$$\frac{Q_1}{\lambda_0} \gg \max\left\{\sqrt{\rho_2 s}(\log n)^{3/2}, \frac{\Phi(|\bar{b}|)}{\rho_1 d}, \sqrt{\log n} \cdot \xi_1\right\} \tag{I.23}$$

under the condition $d\alpha_{-1,0} \geq 1$. On the other hand, we note that Cond.(v) and Cond.(vi) can be reformulated as

$$\frac{Q_1}{\lambda_0} \cdot d^{-\varsigma} = \frac{\Phi(|\bar{b}|)}{\rho_1 d} \gg d^{\epsilon-\varsigma}\sqrt{s}\log n \cdot \xi_1. \tag{I.24}$$

Since $d^\epsilon \sqrt{s}\log n \cdot \xi_1 \gg \sqrt{\log n} \cdot \xi_1$, we can safely delete the last term in (I.23). Also by noting that $d^{-\varsigma} \ll 1$, we can safely delete the second term in (I.23). Furthermore, by definition of $\xi_1$, which we recall as follows:

$$\xi_1 = \sqrt{s}\log n\, \mathcal{K}_1 + \rho_1^{-1}\sqrt{\Phi(|\bar{b}|) \cdot \widehat{\mathbb{E}}_{l,l'}\left[\Phi\left(|\bar{b}|\sqrt{\frac{1 - \langle h_l, h_{l'}\rangle}{1 + \langle h_l, h_{l'}\rangle}}\right)\langle h_l, h_{l'}\rangle\right] + \sqrt{\rho_2 d}\,|\alpha_{-1,0}| + \rho_2\sqrt{n}},$$

we conclude that $\xi_1 \geq \sqrt{\rho_2 d}\alpha_{-1,0} \geq \sqrt{\rho_2}$. Therefore,

$$d^\epsilon \sqrt{s}\log n \cdot \xi_1 \geq d^\epsilon \sqrt{\rho_2 s}\log n \geq \sqrt{\rho_2 s}(\log n)^{3/2},$$

where in the last inequality, we use the definition $\epsilon = C'\log\log n/(\varsigma \log d) \geq \log\log n/\log d$. Therefore, the first term in (I.23) can also be deleted. In summary, Cond.(iii) is automatically implied by (I.24).

A reformulation of Cond.(ii) gives

$$\frac{1}{s\log n} \gg \frac{\Phi(|\bar{b}|)}{\rho_1 d} \gg \frac{Ls\log(n)^3}{d}. \tag{I.25}$$

In the following, we will simplify the above condition. Note that

$$\Phi(|\bar{b}|)/(\rho_1 d) = Q_1/\lambda_0 d^{-\varsigma} \leq d^{-\varsigma} \ll (s \log n)^{-1}$$

holds by using $\lambda_0 = \Theta(\mathrm{polylog}(n))$ according to Cond.(iv) and $\lambda_0^1 Q_1 \cdot d^{-\varsigma} = \Phi(|\bar{b}|)/(\rho_1 d)$ according to Cond.(v). Therefore, we can safely remove the first inequality in (I.25).

In the following, we aim to remove the condition $\sqrt{\rho_2 s}(t^\star \log n)^{3/2} \ll 1$ in Cond.(i). As $Q_1/\lambda_0 \gg d^\epsilon \sqrt{s} \log n \cdot \xi_1$ by Cond.(vi), we conclude that $\xi_1 \ll Q_1/\lambda_0 < 1$. By definition of $\xi_1$, this condition directly implies that $\rho_2 \ll n^{-1/2}$. Therefore, we can safely delete the condition $\sqrt{\rho_2 s}(t^\star \log n)^{3/2} \ll 1$ in Cond.(i).

To this end, we can summarize Cond.(ii), Cond.(iii), Cond.(v) and Cond.(vi) into one condition as follows:

$$\frac{Q_1}{\lambda_0} \cdot d^{-\varsigma} = \frac{\Phi(|\bar{b}|)}{\rho_1 d} \gg \max\Big\{ d^{\epsilon-\varsigma}\sqrt{s}\log n \cdot \xi_1, \frac{Ls\log(n)^3}{d} \Big\},$$

and Cond.(i) and Cond.(iv) can be summarized into

$$\lambda_0 = O(\mathrm{polylog}(n)), \quad \kappa_0 = O((\log n)^{-1/2}), \quad \eta^{-1} \ll N \cdot \Big(\frac{\rho_1 d}{\lambda_0} \wedge \Phi(|\bar{b}|)\Big).$$

Note that in the last condition, we have $\rho_1 d/\lambda_0 \gg \Phi(|\bar{b}|)$ according to the first equality in (I.4.1). Hence, we only need to keep $\eta^{-1} \ll N\Phi(|\bar{b}|)$. This completes the proof of Theorem G.13.

### I.4.2 PROOF OF THEOREM G.14

To prove this lemma, we need to upper bound the expectation term on the left-hand side of (G.10). Recall that $\widehat{\mathbb{E}}_{l,l'}$ is given by uniformly samples $l, l'$ from $[N]$, and that $\langle h_l, h_{l'} \rangle \leq 1$ always holds. We can upper bound the expectation term as follows:

$$\widehat{\mathbb{E}}_{l,l'}\Big[\Phi\Big(|\bar{b}_t|\sqrt{\frac{1 - \langle h_l, h_{l'}\rangle}{1 + \langle h_l, h_{l'}\rangle}}\Big)\langle h_l, h_{l'}\rangle\Big] \leq \frac{1}{N^2}\sum_{j=1}^n \sum_{l,l'\in\mathcal{D}_j} \Phi\Big(|\bar{b}|\sqrt{\frac{1 - \langle h_l, h_{l'}\rangle}{1 + \langle h_l, h_{l'}\rangle}}\Big)$$

$$= \frac{1}{N^2}\sum_{j=1}^n \sum_{l,l'\in\mathcal{D}_j} \Phi\Big(|\bar{b}|\sqrt{\frac{1 - \langle h_l, h_{l'}\rangle}{1 + \langle h_l, h_{l'}\rangle}}\Big) \cdot \mathbb{1}(\|h_l \circ h_{l'}\|_\infty = 1)$$

$$+ \frac{1}{N^2}\sum_{j=1}^n \sum_{l,l'\in\mathcal{D}_j} \Phi\Big(|\bar{b}|\sqrt{\frac{1 - \langle h_l, h_{l'}\rangle}{1 + \langle h_l, h_{l'}\rangle}}\Big) \cdot \mathbb{1}(\|h_l \circ h_{l'}\|_\infty \geq 2)$$

$$\leq \frac{1}{N^2}\sum_{j=1}^n \sum_{l,l'\in\mathcal{D}_j} \Phi\Big(|\bar{b}|\sqrt{\frac{1 - H_{l,j}H_{l',j}}{1 + H_{l,j}H_{l',j}}}\Big) + \frac{1}{N^2}\sum_{j=1}^n \sum_{l,l'\in\mathcal{D}_j} \mathbb{1}(\|h_l \circ h_{l'}\|_\infty \geq 2),$$

where in the identity, we split the summation according to whether how many non-zero entries are shared between two rows $h_l$ and $h_{l'}$ in the $H$ matrix. In the last inequality, we drop the indicator for the case $\|h_l \circ h_{l'}\|_\infty = 1$ and use the fact that $\Phi(\cdot) \leq 1$ for the case $\|h_l \circ h_{l'}\|_\infty \geq 2$. For the first term, we use the fact that $|\mathcal{D}_j|/N \leq \rho_1$ for all $j \in [n]$ to obtain

$$\frac{1}{N^2}\sum_{j=1}^n \sum_{l,l'\in\mathcal{D}_j} \Phi\Big(|\bar{b}|\sqrt{\frac{1 - H_{l,j}H_{l',j}}{1 + H_{l,j}H_{l',j}}}\Big) \leq \rho_1^2 \cdot \sum_{j=1}^n \frac{1}{|\mathcal{D}_j|^2} \sum_{l,l'\in\mathcal{D}_j} \Phi\Big(|\bar{b}|\sqrt{\frac{1 - H_{l,j}H_{l',j}}{1 + H_{l,j}H_{l',j}}}\Big)$$

$$\leq n\rho_1^2 \cdot \Phi\Big(|\bar{b}_t|\sqrt{\frac{1 - h_\star^2}{1 + h_\star^2}}\Big),$$

where the last inequality holds by the definition of $h_\star$ in (G.9). In addition, the second term is upper bounded by

$$\frac{1}{N^2}\sum_{j=1}^n \sum_{l,l'\in\mathcal{D}_j} \mathbb{1}(\|h_l\circ h_{l'}\|_\infty \geq 2) \leq \frac{1}{N^2}\sum_{l=1}^N \sum_{\substack{j\in[n]:\\H_{l,j}\neq 0}} \sum_{\substack{i\neq j:\\H_{l,i}\neq 0}} \sum_{l'=1}^N \mathbb{1}(H_{l',i}\neq 0)\cdot\mathbb{1}(H_{l',j}\neq 0)$$

$$\leq \max_{l\in[N]}\frac{1}{N} \sum_{\substack{i,j\in[n]:i\neq j\\H_{l,i}\neq 0, H_{l,j}\neq 0}} \sum_{l'=1}^N \mathbb{1}(H_{l',i}\neq 0)\cdot\mathbb{1}(H_{l',j}\neq 0)$$

$$\leq \max_{l\in[N]}\frac{1}{N} \sum_{\substack{i,j\in[n]:i\neq j\\H_{l,i}\neq 0, H_{l,j}\neq 0}} N\cdot\rho_1\cdot\rho_2 \leq s^2\rho_1\rho_2.$$

In the first inequality, we notice that if $\|h_l\circ h_{l'}\|_\infty \geq 2$, then there must exist two different feature indices $i\neq j$ such that both $h_l, h_{l'}$ are non-zero at these two indices. This is indeed reflected in the constraints $H_{l,j}\neq 0, H_{l,i}\neq 0$ and the two indicators $\mathbb{1}(H_{l',i}\neq 0)\cdot\mathbb{1}(H_{l',j}\neq 0)$. Therefore, summing over all posible $(i,j)$ pairs gives an upper bound for the second term. In the second inequality, we change the average over $l$ to be the maximum over $l$, and in the third inequality, we use the definition of $\rho_2$ and $\rho_1$ in (F.1) to upper bound sum of the double indicator term. The last inequality holds by noting that each row $h_l$ is $s$-sparse. Combining the above two bounds, we obtain that

$$\widehat{\mathbb{E}}_{l,l'}\left[\Phi\left(|\bar{b}_t|\sqrt{\frac{1-\langle h_l, h_{l'}\rangle}{1+\langle h_l, h_{l'}\rangle}}\right)\langle h_l, h_{l'}\rangle\right] \leq n\rho_1^2\cdot\Phi\left(|\bar{b}_t|\sqrt{\frac{1-h_\star^2}{1+h_\star^2}}\right) + \rho_1\rho_2 s^2$$

$$\leq Cn\rho_1^2\cdot\Phi(|\bar{b}_t|)^{\frac{1-h_\star^2}{1+h_\star^2}} + \rho_1\rho_2 s^2,$$

where in the last inequality, we use the Mills ratio

$$\Phi\left(|\bar{b}_t|\sqrt{\frac{1-h_\star^2}{1+h_\star^2}}\right) \leq \left(|\bar{b}_t|\sqrt{\frac{1-h_\star^2}{1+h_\star^2}}\right)^{-1}\cdot\frac{1}{\sqrt{2\pi}}\exp\left(-\frac{1-h_\star^2}{1+h_\star^2}\cdot\frac{\bar{b}_t^2}{2}\right)$$

$$\leq C|\bar{b}_t|^{-1}\cdot\frac{1}{\sqrt{2\pi}}\exp\left(-\frac{\bar{b}_t^2}{2}\right)^{\frac{1-h_\star^2}{1+h_\star^2}} \leq C\Phi(|\bar{b}_t|)^{\frac{1-h_\star^2}{1+h_\star^2}},$$

and the above inequalities hold as long as $(1-h_\star^2)/(1+h_\star^2)$ is on a constant level. Therefore, we have proved Theorem G.14.

### I.4.3 PROOF OF THEOREM G.15

Recall the definition $\mathcal{K}_t$ in (F.9) as

$$\mathcal{K}_1 := \left(n|\bar{b}|\Phi\left(\frac{-\bar{b}}{\sqrt{\frac{3}{4}\hbar_{4,\star}^2 + \frac{1}{4}}}\right)\right)^{1/4} + \left(\rho_2 sn|\bar{b}|\Phi\left(\frac{-\bar{b}}{\sqrt{\frac{2}{3}\hbar_{3,\star}^2 + \frac{1}{3}}}\right)\right)^{1/4}$$

$$+ \left(\Phi\left(-\frac{\bar{b}+\hbar_{4,1}\zeta_1}{\sqrt{1-\hbar_{4,1}^2}}\right) + (\rho_2 s)^{1/4}\right)\cdot(\log(n))^{1/4} + n^{1/4}\rho_2 s\log(n).$$

To upper bound the above terms, let us consider the following inequality for any $\tau\in(0,1)$ and $|\bar{b}|\geq 1$:

$$\Phi(\tau|\bar{b}|) \leq \frac{1}{\sqrt{2\pi}}\cdot\exp\left(-\frac{\tau^2|\bar{b}|^2}{2}\right)\cdot\frac{1}{\tau|\bar{b}|} \leq \frac{1}{\sqrt{2\pi}}\cdot\exp\left(-\frac{\tau^2|\bar{b}|^2}{2}\right)\cdot\frac{|\bar{b}|}{|\bar{b}|^2+1}\cdot 2\tau^{-1}$$

$$\leq \frac{1}{(\sqrt{2\pi})^{\tau^2}}\cdot\exp\left(-\frac{\tau^2|\bar{b}|^2}{2}\right)\cdot\left(\frac{|\bar{b}|}{|\bar{b}|^2+1}\right)^{\tau^2}\cdot 2\tau^{-1} \leq \frac{2}{\tau}\cdot\Phi(|\bar{b}|)^{\tau^2}, \tag{I.26}$$

where in the first and the last inequalities, we use the Mills' ratio bound that $x/(x^2 + 1) \leq \Phi(x)/p(x) \leq x^{-1}$ for all $x > 0$. Now, we can apply (I.26) to upper bound the first term in $\mathcal{K}_t$ as

$$\left( n\, |\bar{b}|\, \Phi\left( \frac{-\bar{b}}{\sqrt{\frac{3}{4}\, \hbar_{4,\star}^2 + \frac{1}{4}}} \right) \right)^{1/4} \leq C\left( n\, |\bar{b}| \right)^{1/4} \Phi(|\bar{b}|)^{\frac{1}{3\hbar_{4,\star}^2 + 1}} \leq C\left( n\, |\bar{b}| \right)^{1/4} \Phi(|\bar{b}|)^{\frac{1}{3h_\star^2 + 1}}, \quad \text{(I.27)}$$

where the last inequality holds because $h_\star \leq \hbar_{4,\star}$ by definition. Similarly, we can upper bound the second term as

$$\left( \rho_2 sn|\bar{b}|\Phi\left( \frac{-\bar{b}}{\sqrt{\frac{2}{3}\hbar_{3,\star}^2 + \frac{1}{3}}} \right) \right)^{1/4} \leq C(\rho_2 sn|\bar{b}|)^{1/4} \cdot \Phi(|\bar{b}|)^{\frac{3}{8\hbar_{3,\star}^2 + 4}} \leq C(\rho_2 sn|\bar{b}|)^{1/4} \cdot \Phi(|\bar{b}|)^{\frac{3}{8h_\star^2 + 4}}.$$

$$\text{(I.28)}$$

Here, the third term also follows from the above inequality as

$$\Phi\left( -\frac{\bar{b} + \hbar_{4,1}\zeta_1}{\sqrt{1 - \hbar_{4,1}^2}} \right) \leq \Phi\left( -\frac{\bar{b} + h_\star\zeta_1}{\sqrt{1 - h_\star^2}} \right) \leq C \cdot \Phi(|\bar{b}|)^{\frac{(1 - h_\star\zeta_1/|\bar{b}|)^2}{1 - h_\star^2}}, \quad \text{(I.29)}$$

where in the first inequality, we use the derivative in (F.12) and the fact that $\zeta_1/|\bar{b}| = \Theta(1) > 1$, which is given by the definition $\zeta_1 = (1 + \varepsilon)2\sqrt{\log n}$ in (E.11), to conclude that increasing $\hbar_{4,1}$ to $h_\star$ will only increase the value of the whole term. In the second inequality, we apply (I.26) with

$$\tau = \frac{1 - h_\star\zeta_1/|\bar{b}|}{\sqrt{1 - h_\star^2}} \in (0, 1).$$

Here, we claim $\tau \in (0, 1)$ because by condition $\zeta_1 h_\star < 1 - \nu$ for some constant $\nu > 0$, we have $1 - h_\star\zeta_1/|\bar{b}| > 0$, and by noting that $\zeta_1/|\bar{b}| > 1$, we have

$$\tau < \frac{1 - h_\star}{\sqrt{1 - h_\star}} = \sqrt{1 - h_\star} \leq 1.$$

In addition, since $|\bar{b}| \leq \sqrt{2\log n}$ by condition $\Phi(|\bar{b}|) \geq \rho_1 \geq n^{-1}$, we also have $\zeta_1/|\bar{b}| > 1$. Consequently, we obtain that

$$\frac{h_\star^{-1} - \zeta_1/|\bar{b}|}{\sqrt{h_\star^{-2} - 1}} \leq \frac{h_\star^{-1} - 1}{\sqrt{h_\star^{-2} - 1}} \leq 1$$

as $h_\star < 1$. Therefore, we can apply (I.26) with $\tau = (h_\star^{-1} - \zeta_1/|\bar{b}|)/\sqrt{h_\star^{-2} - 1} \in (0, 1)$ in the last inequality in (I.29). Now, we can combine (I.27), (I.28) and (I.29) to obtain the desired result in Theorem G.15.

## I.5 PROOFS FOR TECHNICAL LEMMAS

### I.5.1 PROOF OF THEOREM I.1

By Cauchy-Schwartz, it holds that

$$\sum_{\tau=1}^{t-1} \langle P_{u_{1:\tau}} z_\tau, u_t \rangle^2 \leq \sum_{\tau=1}^{t-1} \| P_{u_{1:\tau}} z_\tau \|_2^2 \cdot \| u_t \|_2^2.$$

One thing to be noted is that $z_\tau$ is independent of the filtration $\sigma(u_{1:\tau})$. Consiquently, when conditioned on $u_{1:\tau}$, $\| P_{u_{1:\tau}} z_\tau \|_2^2 \sim \chi_\tau^2$. By the concentration of $\chi^2$ distribution in Theorem J.1 with a union bound over all $\tau \in [T]$, we obtain that with probability at least $1 - n^{-c}$ for some universal constant $c, C > 0$ that

$$\| P_{u_{1:\tau}} z_\tau \|_2^2 \leq \tau + C\sqrt{\tau\log(nT)} + C\log(nT) \leq C(t + \log n), \quad \forall \tau \in [t-1], \quad t \in [T].$$

Therefore, we have that with probability at least $1 - n^{-c}$:

$$\sum_{\tau=1}^{t-1} \langle P_{u_{1:\tau}} z_\tau, u_t \rangle^2 \le C(t^2 + t \log n) \cdot \|u_t\|_2^2, \quad \forall t \in [T]. \tag{I.30}$$

For the second term, it follows from Theorem H.1 that with probability at least $1 - n^{-c}$:

$$\sum_{\tau=1}^{t-1} \left( \frac{\langle u_\tau^\perp, u_t \rangle}{\|u_\tau^\perp\|_2} \cdot \frac{\|w_\tau^\perp\|_2}{\|u_\tau^\perp\|_2} \right)^2 \le Cd \cdot \sum_{\tau=1}^{t-1} \frac{\langle u_\tau^\perp, u_t \rangle^2}{\|u_\tau^\perp\|_2^2} = Cd \cdot \|P_{u_{1:t-1}} u_t\|_2^2, \quad \forall t \in [T]. \tag{I.31}$$

For the last term $\|P_{w_{-1:t-1}}^\perp \widetilde{z}_t\|_2^2 \cdot \|u_t^\perp\|_2^2$, we also note that $\widetilde{z}_t$ is independent of the filtration $\sigma(w_{-1:t-1})$. Therefore, $\|P_{w_{-1:t-1}}^\perp \widetilde{z}_t\|_2^2 \sim \chi_{d-t+1}^2$. We have by Theorem J.1 with a union bound over all $t \in [T]$, and with probability at least $1 - n^{-c}$ that

$$\|P_{w_{-1:t-1}}^\perp \widetilde{z}_t\|_2^2 \le d - t + 1 + C\sqrt{(d-t+1)\log(nT)} + C\log(nT) \le Cd, \quad \forall t \in [T]. \tag{I.32}$$

Combining (I.31) and (I.32), we have with probability at least $1 - n^{-c}$ and for all $t \in [T]$:

$$\sum_{\tau=1}^{t-1} \left( \frac{\langle u_\tau^\perp, u_t \rangle}{\|u_\tau^\perp\|_2} \cdot \frac{\|w_\tau^\perp\|_2}{\|u_\tau^\perp\|_2} \right)^2 + \|P_{w_{-1:t-1}}^\perp \widetilde{z}_t\|_2^2 \cdot \|u_t^\perp\|_2^2 \le Cd \cdot \left( \|P_{u_{1:t-1}} u_t\|_2^2 + \|u_t^\perp\|_2^2 \right) = Cd \cdot \|u_t\|_2^2. \tag{I.33}$$

Now we combine (I.30) and (I.33) to obtain the desired result in Theorem I.1.

### I.5.2   PROOF OF THEOREM I.2

Recall that

$$u_t = E^\top \varphi(E y_t^\star; b_t) + F^\top \varphi(F y_t^\star + \theta \cdot v^\top \overline{w}_{t-1}; b_t) + \Delta E_t + \Delta F_t.$$

By the triangular inequality, we have

$$\|u_t\|_2 \le 2\sqrt{\|E^\top \varphi(E y_t^\star; b_t)\|_2^2 + \|F^\top \varphi(F y_t^\star + \theta \cdot v^\top \overline{w}_{t-1}; b_t)\|_2^2} + 2\sqrt{\|\Delta E_t\|_2^2 + \|\Delta F_t\|_2^2}$$

By Theorem G.4, we have

$$\sqrt{\|E^\top \varphi(E y_t^\star; b_t)\|_2^2 + \|F^\top \varphi(F y_t + \theta \cdot v^\top \overline{w}_{t-1}; b_t)\|_2^2} \le CLN\rho_1 \xi_t.$$

By Theorems F.12 and F.13, and the fact that $N_1 \le N$ and $N_2 \le N\rho_1$, we derive that

$$\sqrt{\|\Delta E_t\|_2^2 + \|\Delta F_t\|_2^2} \le CL\rho_1 N\sqrt{d}\beta_{t-1} + CL\rho_1\rho_2\sqrt{d}\beta_{t-1} \le CLN\rho_1\sqrt{d}\beta_{t-1}.$$

This completes the proof of Theorem I.2.

### I.5.3   PROOF OF THEOREM I.4

We recall from the definition of $\mathcal{K}_t$ that

$$\mathcal{K}_t = \left( n\,|\overline{b}|\,\Phi\left( \frac{-\overline{b}}{\sqrt{\frac{3}{4}\hbar_{4,\star}^2 + \frac{1}{4}}} \right) \right)^{1/4} + \left( \rho_2 s n|\overline{b}|\Phi\left( \frac{-\overline{b}}{\sqrt{\frac{2}{3}\hbar_{3,\star}^2 + \frac{1}{3}}} \right) \right)^{1/4}$$

$$+ \left( \Phi\left( -\frac{\overline{b} + \hbar_{4,t}\zeta_t}{\sqrt{1 - \hbar_{4,t}^2}} \right) + (\rho_2 s)^{1/4} \right) \cdot (t\log(n))^{1/4} + n^{1/4}\rho_2\,s\,t\log(n),$$

The terms that implicitly change with $t$ is $\zeta_t$ and $\hbar_{4,t}$. Recall that

$$\zeta_t = \zeta_1 + \mathbb{1}(t \ge 2) \cdot C(\beta_{t-1} + |\alpha_{-1,t-1}| + |\alpha_{-1,0}|)$$

with $\zeta_1 = \sqrt{2}(1+\varepsilon)\sqrt{2\log n}$. Moreover, by definition of $\hbar_{4,t}$, we can rewrite the term as

$$\Phi\left(-\frac{\bar{b}+\hbar_{4,t}\zeta_t}{\sqrt{1-\hbar_{4,t}^2}}\right) = \max_{j\in[n]}\left(\frac{1}{|\mathcal{D}_j|}\sum_{l\in\mathcal{D}_j}\Phi\left(-\frac{\bar{b}+H_{l,j}\zeta_t}{\sqrt{1-H_{l,j}^2}}\right)^4\right)^{1/4}. \tag{I.34}$$

To understand how the change in $\zeta_t$ affects the term, we take the derivatives for positive power $q$:

$$\frac{\mathrm{d}}{\mathrm{d}\zeta}\Phi\left(-\frac{\bar{b}+x\zeta}{\sqrt{1-x^2}}\right)^q\bigg|_{\zeta=\zeta_t} = q\Phi\left(-\frac{\bar{b}+x\zeta}{\sqrt{1-x^2}}\right)^{q-1}\cdot p\left(-\frac{\bar{b}+x\zeta}{\sqrt{1-x^2}}\right)\cdot\frac{\zeta x}{\sqrt{1-x^2}} > 0.$$

Here, we have the second derivative larger than 0 since $\zeta_t \geq \zeta_1 = \sqrt{2}(1+\varepsilon)\sqrt{2\log n} > |\bar{b}|$ by assumption. Now, our goal is to upper bound the derivative with respect to $\zeta$. We discuss in two cases:

- For $x \in \left[(1+|\bar{b}|/\zeta)/2, 1\right]$, we have $\bar{b}+x\zeta \in [(\bar{b}+\zeta)/2, \bar{b}+\zeta]$, $x \geq (1+1/\sqrt{2})/2$ and thus

$$\frac{\mathrm{d}}{\mathrm{d}\zeta}\Phi\left(-\frac{\bar{b}+x\zeta}{\sqrt{1-x^2}}\right)^q \leq q\cdot p\left(-\frac{\bar{b}+x\zeta}{\sqrt{1-x^2}}\right)\cdot\frac{\bar{b}+x\zeta}{\sqrt{1-x^2}}\cdot\frac{\zeta x}{\bar{b}+\zeta x}$$

$$\leq q\cdot \sup_{z\geq\sqrt{1-(1+1/\sqrt{2})^2/4}}p(z)\cdot z\cdot\frac{\zeta}{\bar{b}+\zeta} \leq q\cdot\sup_{z\geq 0}p(z)\cdot z = O(1).$$

- For $x \in [0, (1+|\bar{b}|/\zeta)/2]$, we have $\sqrt{1-x^2} \geq \sqrt{1-(1+|\bar{b}|/\zeta)^2/4} \geq \sqrt{1-(1+1/\sqrt{2})^2/4} = \Omega(1)$. Thus

$$\frac{\mathrm{d}}{\mathrm{d}\zeta}\Phi\left(-\frac{\bar{b}+x\zeta}{\sqrt{1-x^2}}\right)^q \leq q\cdot\frac{\zeta}{\sqrt{1-(1+1/\sqrt{2})^2/4}} = O(\sqrt{\log n}).$$

Therefore, we conclude that for $q = 1$, it holds for any $H_{l,j} \in [0,1]$ that

$$\Phi\left(-\frac{\bar{b}+H_{l,j}\zeta_t}{\sqrt{1-H_{l,j}^2}}\right) - \Phi\left(-\frac{\bar{b}+H_{l,j}\zeta_1}{\sqrt{1-H_{l,j}^2}}\right) \leq C\sqrt{\log n}\cdot(\beta_{t-1}+|\alpha_{-1,t-1}|+|\alpha_{-1,0}|). \tag{I.35}$$

Since $\|x\|_4 - \|y\|_4 \leq \|x-y\|_4 \leq m^{1/4}\|x-y\|_\infty$ for any $x,y \in \mathbb{R}^m$ by the triangle inequality, we conclude that the same upper bound in (I.35) holds for each $j$ in (I.34) as well. Therefore, the same upper bound also holds after taking the maximum over $j \in [n]$ in (I.34). Therefore, we obtain that

$$\mathcal{K}_t \leq t\cdot\left(\mathcal{K}_1 + C\sqrt{\log n}\cdot(\beta_{t-1}+|\alpha_{-1,t-1}|+|\alpha_{-1,0}|)\right).$$

This completes the proof of Theorem I.4.

## J AUXILIARY LEMMAS

### J.1 CONCENTRATION INEQUALITIES

**Lemma J.1** (Chi-square concentration, Lemma 1 in Laurent & Massart (2000)). *Let $X_1,\ldots,X_n$ be independent random variables such that $X_i \sim \mathcal{N}(0,1)$ for all $i$. Let $a \in \mathbb{R}_+^n$ be a vector with nonnegative entries. Then the following holds for any $\delta \in (0,1)$:*

$$\mathbb{P}\left(\left|\sum_{i=1}^n a_i X_i^2 - \|a\|_1\right| \geq 2\sqrt{\|a\|_2^2\log\delta^{-1}} + 2\|a\|_\infty\log\delta^{-1}\right) \leq \delta.$$

**Lemma J.2** (Tail probability for the maximum Gaussian random variables). *Let $X_1,\ldots,X_n$ be $\sigma^2$-subgaussian random variables with mean 0. Then for any $t > 0$,*

$$\mathbb{P}\left(\max_{i=1,\ldots,n}X_i \geq \sqrt{2\sigma^2\log n} + t\right) \leq \exp\left(-\frac{t^2}{2\sigma^2}\right).$$

*In particular, if $X_1,\ldots,X_n$ are independent standard normal random variables, then for any $c > 1$,*

$$\mathbb{P}\left(\max_{i=1,\ldots,n}X_i \geq c\sqrt{2\log n}\right) \leq n^{1-c^2}.$$

**Lemma J.3** (Bernstein's inequality). *Let $X_1, \ldots, X_n$ be independent random variables with $|X_i - \mathbb{E}[X_i]| \leq C$ for all $i \in [n]$. Then for any $\delta \in (0,1)$,*

$$\mathbb{P}\left(\left|\frac{1}{n}\sum_{i=1}^{n} X_i - \mathbb{E}X_i\right| \leq \sqrt{\frac{2 \cdot n^{-1}\sum_{i=1}^{n}\mathrm{Var}[X_i] \cdot \log \delta^{-1}}{n}} + \frac{C\log\delta^{-1}}{3n}\right) \geq 1 - \delta.$$

**Lemma J.4.** *Let $X_1, \ldots, X_n$ be independent standard normal random variables and define $M_n = \max_{1 \leq i \leq n} X_i$. Then for any fixed $\epsilon \in (0,1)$ and all sufficiently large $n$ with $2(1-\epsilon)^2 \log n \geq 1$ that*

$$\mathbb{P}\Big(M_n \leq (1-\epsilon)\sqrt{2\log n}\Big) \leq \exp\left(-\frac{n^{2\epsilon - \epsilon^2}}{3\sqrt{\pi \log n}}\right).$$

*Proof.* Since $X_1, \ldots, X_n \overset{\text{i.i.d.}}{\sim} N(0,1)$, it holds for any $x \in \mathbb{R}$

$$\mathbb{P}(M_n \leq x) = (1 - \Phi(x))^n,$$

where $\Phi(x)$ is the standard normal tail distribution function. In order to upper bound $(1 - \Phi(x))^n$ when $x = (1-\epsilon)\sqrt{2\log n}$, we use a well-known lower bound for the Gaussian tail. Specifically, for all $x > 0$ (see, e.g., Ledoux & Talagrand (2013) or Boucheron et al. (2013)),

$$\Phi(x) \geq \Delta(x) := \frac{x}{1+x^2}\,\frac{1}{\sqrt{2\pi}}\,e^{-x^2/2}.$$

Hence, further applying the fact that $1 - \Delta(x) \leq \exp(-\Delta(x))$, we get

$$\mathbb{P}(M_n \leq x) \leq \big(1 - \Delta(x)\big)^n \leq \exp\big(-n\Delta(x)\big).$$

Now, for $x = (1-\epsilon)\sqrt{2\log n}$, we have

$$\frac{x}{1+x^2} = \frac{(1-\epsilon)\sqrt{2\log n}}{1 + 2(1-\epsilon)^2\log n} \geq \frac{\sqrt{2}}{3(1-\epsilon)\sqrt{\log n}},$$

where the inequality holds for sufficiently large $n$ such that $2(1-\epsilon)^2\log n \geq 1$. Thus,

$$\Delta(x) \geq \frac{1}{3(1-\epsilon)\sqrt{\pi\log n}}\,n^{-(1-\epsilon)^2}.$$

Substituting this lower bound into our earlier inequality gives

$$\mathbb{P}\Big(M_n \leq (1-\epsilon)\sqrt{2\log n}\Big) \leq \exp\left(-\frac{1}{3(1-\epsilon)\sqrt{\pi\log n}}\,n^{1-(1-\epsilon)^2}\right)$$

$$= \exp\left(-\frac{n^{2\epsilon-\epsilon^2}}{3\sqrt{\pi\log n}}\right).$$

This completes the proof. $\qquad\square$

## J.2 EFRON-STEIN INEQUALITIES

Let $Z$ be a function of independent random variables $X_1, \ldots, X_n$ with domain $\mathcal{X}$:

$$Z = f(X_1, \ldots, X_n), \tag{J.1}$$

where $f : \mathcal{X}^n \to \mathbb{R}$ is a measurable function. Let $X_1', \ldots, X_n'$ be independent copies of $X_1, \ldots, X_n$. Define the modified versions of $Z$ where one coordinate is replaced by its independent copy:

$$Z^{(i)} = f(X_1, \ldots, X_{i-1}, X_i', X_{i+1}, \ldots, X_n).$$

Define the deviation terms:

$$V_+ = \mathbb{E}\left[\sum_{i=1}^{n}(Z - Z^{(i)})^2\,\mathbb{1}\{Z > Z^{(i)}\}\,\bigg|\,X_1, \ldots, X_n\right],$$

$$V_- = \mathbb{E}\left[\sum_{i=1}^{n}(Z - Z^{(i)})^2\,\mathbb{1}\{Z < Z^{(i)}\}\,\bigg|\,X_1, \ldots, X_n\right]. \tag{J.2}$$

The following lemma is borrowed from Theorem 5 in Boucheron et al. (2003) for the case where $V_+$ is dominated by some linear transformation of $Z$.

**Lemma J.5** (Efron-Stein for dominated variance). *For $Z$ and $V_+$ defined in (J.1) and (J.2), respectively, suppose that there exist positive constants $a$ and $b$ such that $V_+ \leq aZ + b$. Then there is a universal constant $C > 0$ such that for any $\delta \in (0, 1)$, with probability at least $1 - \delta$,*

$$Z \leq \mathbb{E}[Z] + C \cdot \sqrt{(a \cdot \mathbb{E}[Z] + b) \log(1/\delta)} + C \cdot a \log(1/\delta).$$

The following lemma is borrowed from Theorem 2 in Boucheron et al. (2003).

**Lemma J.6** (Efron-Stein for the moment generating function). *For all $\theta > 0$ and $\lambda \in (0, 1/\theta)$,*

$$\log \mathbb{E}\left[\exp\left(\lambda(Z - \mathbb{E}[Z])\right)\right] \leq \frac{\lambda\theta}{1 - \lambda\theta} \log \mathbb{E}\left[\exp\left(\frac{\lambda V_+}{\theta}\right)\right].$$

*On the other hand, for all $\theta > 0$ and $\lambda \in (0, 1/\theta)$,*

$$\log \mathbb{E}\left[\exp\left(-\lambda(Z - \mathbb{E}[Z])\right)\right] \leq \frac{\lambda\theta}{1 - \lambda\theta} \log \mathbb{E}\left[\exp\left(\frac{\lambda V_-}{\theta}\right)\right].$$

The following lemma is borrowed from Lemma 11 in Boucheron et al. (2003) for transforming the upper bound on moment generating function (MGF) bound into an exponential tail bound.

**Lemma J.7.** *Suppose for any $\lambda \in (0, 1/a)$, there exists a constant $V > 0$ such that:*

$$\log \mathbb{E}[\exp(\lambda(Z - \mathbb{E}[Z]))] \leq \frac{\lambda^2 V}{1 - \lambda a}.$$

*Then there exists some universal constant $C$ such that for any $\delta \in (0, 1)$, with probability at least $1 - \delta$:*

$$Z - \mathbb{E}[Z] \leq C \cdot \sqrt{V \log(1/\delta)} + C \cdot a \log(1/\delta).$$

With the above lemmas, we can derive the following Efron-Stein inequality for sub-exponential variance.

**Lemma J.8** (Efron-Stein inequality for sub-exponential variance). *Suppose either of the following two conditions is satisfied:*

1. *The variance $V_+$ for $Z$ satisfies that $\mathbb{E}[\exp(\lambda V_+)] \leq \mathbb{E}[\exp(\lambda V_+')]$ for any $\lambda > 0$, where $V_+'$ is $a$-subexponential with $a \in (0, 1)$:*

   $$Q(v) := \mathbb{P}(V_+' > V + v) \leq \exp(-v/a).$$

   *when $V_+$ exceeds some threshold $V > 0$.*

2. *The moment generating function of $V_+$ satisfies*

   $$\log \mathbb{E}[\exp(\lambda V_+)] \leq \lambda V + \frac{\lambda a}{1 - a\lambda}$$

   *for some $V > 0, 0 < a < 1$ and any $0 < \lambda < a^{-1/2}$.*

*Then, with probability at least $1 - \delta$, it holds that*

$$Z - \mathbb{E}[Z] \leq C \cdot \sqrt{V \log(\delta^{-1})} + C \cdot \sqrt{a} \log(\delta^{-1}).$$

*Similarly, if $V_-$ satisfies either of the two conditions, then with probability at least $1 - \delta$, it holds that*

$$\mathbb{E}[Z] - Z \leq C \cdot \sqrt{V \log(\delta^{-1})} + C \cdot \sqrt{a} \log(\delta^{-1}).$$

*Proof.* We just prove the first condition and the second condition can be implied by the proof. We explicit calculate the MGF for $V_+$. The case for $V_-$ can be handled similarly. Take parameter $\lambda \in (0, a^{-1/2})$, we have for the moment generating function of $V_+$ that

$$\mathbb{E}[\exp(\lambda V_+)] = \exp(\lambda V) \cdot \left(\lim_{v \to 0_-} Q(v) + \lambda \cdot \int_{0_+}^{\infty} \exp(\lambda \cdot v) \cdot Q(v) \mathrm{d}v\right)$$

$$\leq \exp(\lambda V) \cdot \left(1 + \lambda \cdot \int_{0_+}^{\infty} \exp\left(-(a^{-1} - \lambda) \cdot v\right) \mathrm{d}v\right)$$

$$= \exp(\lambda V) \cdot \left(1 + \frac{\lambda}{a^{-1} - \lambda}\right) = \exp(\lambda V) \cdot \left(1 + \frac{\lambda \cdot a}{1 - a\lambda}\right).$$

where we use $0_+$ and $0_-$ to denote the limit from the right and left side of $0$, respectively. Here, in the first line, we use integration by parts to obtain an integration term with respect to the tail probability $Q(v)$. In the final line, we have the denominator $1 - a\lambda > 0$ since $\lambda < a^{-1/2} < a^{-1}$ for $a \in (0, 1)$. Taking the logarithm on both side, we obtain that

$$\log \mathbb{E}[\exp(\lambda V_+)] \leq \lambda V + \log(1 + \lambda a/(1 - a\lambda)) \leq \lambda V + \frac{\lambda a}{1 - a\lambda}.$$

Now, we apply Theorem J.6 with $\lambda$ replaced by $\lambda/\theta$ for some $\theta \in (a\lambda, \lambda^{-1})$ to obtain that

$$\log \mathbb{E}[\exp(\lambda(Z - \mathbb{E}[Z]))] \leq \frac{\lambda\theta}{1 - \lambda\theta} \log \mathbb{E}\Big[\exp\Big(\frac{\lambda V_+}{\theta}\Big)\Big] \leq \frac{\lambda^2}{1 - \lambda\theta} \cdot \Big(V + \frac{a}{1 - a\lambda/\theta}\Big)$$

$$\leq \cdot \frac{\lambda^2(V + a)}{(1 - \lambda\theta)(1 - a\lambda/\theta)}.$$

Note that such a $\theta$ exists since $\lambda < a^{-1/2}$. In particular, we have by the constraint on $\lambda$ that $a\lambda < \sqrt{a} < \lambda^{-1}$. Let us just pick $\theta = \sqrt{a}$ and further restrict ourselves to $\lambda < a^{-1/2}/2$ to obtain that

$$\log \mathbb{E}[\exp(\lambda(Z - \mathbb{E}[Z]))] \leq \frac{\lambda^2(V + a)}{(1 - \lambda\sqrt{a})^2} \leq \frac{\lambda^2(V + a)}{(1 - 2\lambda\sqrt{a})}.$$

Now, we invoke Theorem J.7 and conclude that there exists universal constant $C > 0$ such that

$$Z - \mathbb{E}[Z] \leq C \cdot \sqrt{(V + a) \cdot \log(\delta^{-1})} + C \cdot \sqrt{a} \cdot \log(\delta^{-1}) \leq 2C \cdot \big(\sqrt{V \log(\delta^{-1})} + \sqrt{a} \cdot \log(\delta^{-1})\big).$$

A similar bound holds for the lower tail with the condition on $V_-$. Hence, we complete the proof. $\square$

**Lemma J.9** (Efron-Stein inequality for bounded variance). *Suppose that $\max\{V_+, V_-\} \leq V_0$ with probability at least $1 - \exp(-a)$ for some $a > n^{c_1}$ and $V_0 > n^{-c_2}$ for some universal constant $c_1, c_2 > 0$. Also assume that $\max\{V_+, V_-\}$ is uniformly bounded by $V_1$ with $V_1 \leq n^{c_3}$ for some universal constant $c_3 > 0$. Then, with probability at least $1 - \delta$, it holds that*

$$|Z - \mathbb{E}[Z]| \leq C \cdot \big(\sqrt{V_0 \log(\delta^{-1})} + \sqrt{a^{-1}V_1} \log(\delta^{-1})\big).$$

*Proof.* By Theorem J.6, we have for the moment generating function (MGF) of $V_+$ that

$$\log \mathbb{E}[\exp(\lambda(Z - \mathbb{E}[Z]))] \leq \frac{\lambda\theta}{1 - \lambda\theta} \cdot \log \mathbb{E}\Big[\exp\Big(\frac{\lambda V_+}{\theta}\Big)\Big]$$

$$\leq \frac{\lambda\theta}{1 - \lambda\theta} \cdot \log\Big(\exp\Big(\frac{\lambda V_0}{\theta}\Big) + \exp\Big(\frac{\lambda V_1}{\theta}\Big) \cdot \mathbb{P}(V_+ \geq V_0)\Big)$$

$$\leq \frac{\lambda}{1 - \lambda\theta} \cdot \Big(\lambda V_0 + \theta \exp\Big(\frac{\lambda(V_1 - V_0)}{\theta} - a\Big)\Big).$$

In the following, we take $\theta = 2\lambda(V_1 - V_0)/a$, and the above upper bound can be simplified as

$$\log \mathbb{E}[\exp(\lambda(Z - \mathbb{E}[Z]))] \leq \frac{\lambda^2}{1 - 2\lambda^2(V_1 - V_0)/a} \cdot \Big(V_0 + \frac{\exp(-a/2)}{2\lambda^2(V_1 - V_0)/a}\Big)$$

$$\leq \frac{\lambda^2}{1 - \lambda\sqrt{2(V_1 - V_0)/a}} \cdot \Big(V_0 + \frac{\exp(-a/2)}{2\lambda^2(V_1 - V_0)/a}\Big).$$

Similarly for $V_-$, we also have

$$\log \mathbb{E}[\exp(-\lambda(Z - \mathbb{E}[Z]))] \leq \frac{\lambda^2}{1 - \lambda\sqrt{2(V_1 - V_0)/a}} \cdot \Big(V_0 + \frac{\exp(-a/2)}{2\lambda^2(V_1 - V_0)/a}\Big).$$

Therefore, in the following, we only need to consider the upper tail and the lower tail can be directly implied. As long as $1/(2\lambda^2(V_1 - V_0)/a)$ is polynomially in $n$, we will have $\exp(-a/2)/(2\lambda^2(V_1 -$

$V_0)/a) \leq V_0$. Take $t$ to be the deviation of $Z$ from its mean, i.e., $t = |Z - \mathbb{E}[Z]|$, we have By Lemma 11 of Boucheron et al. (2003), we conclude by using the Chernoff bound that

$$\log \mathbb{P}(Z - \mathbb{E}[Z] \geq t) \leq \inf_{\lambda \in (0, \sqrt{a/2(V_1 - V_0)})} \left\{ \frac{\lambda^2 \cdot 2V_0}{1 - \lambda\sqrt{2(V_1 - V_0)/a}} - t\lambda \right\}$$

$$\leq -\frac{t^2}{2(4V_0 + t\sqrt{2a^{-1}(V_1 - V_0)}/3)},$$

where the last inequality holds as long as $t$ satisfies

$$1 - \left(1 + \frac{t\sqrt{2a^{-1}(V_1 - V_0)}}{2V_0}\right)^{-1/2} \geq \frac{\exp(-a/4)}{V_0}. \tag{J.3}$$

The lower bound holds similarly. A sufficient condition for (J.3) to hold is

$$t \geq \frac{8\exp(-a/4)}{\sqrt{2a^{-1}(V_1 - V_0)}}.$$

This condition will be automatically satisfied if we pick $t = C \cdot (\sqrt{V_0 \log(\delta^{-1})} + \sqrt{a^{-1}(V_1 - V_0)} \log(\delta^{-1})$. Therefore, we conclude that with probability at least $1 - \delta$, it holds that

$$|Z - \mathbb{E}[Z]| \leq C \cdot \left(\sqrt{V_0 \log(\delta^{-1})} + \sqrt{a^{-1}V_1} \log(\delta^{-1})\right).$$

This completes the proof. $\qquad \square$

**Lemma J.10.** *Let $w = (w_1, w_2, \ldots, w_d)$ be a random vector, and let $w^{(i)}$ denote the vector where the $i$-th coordinate $w_i$ is replaced by an independent copy $w'_i$, while all other coordinates remain unchanged. Suppose that $f : \mathbb{R}^d \to \mathbb{R}$ and $g : \mathbb{R}^d \to \mathbb{R}$ are both nondecreasing/nonincreasing functions with respect to the coordinate $w_i$. Then, we have the inequality:*

$$\mathbb{E}\big[f(w)g(w)\big] \geq \mathbb{E}\big[f(w)g(w^{(i)})\big].$$

*Proof of Theorem J.10.* By the monotonicity of $f$ and $g$, we have:

$$(f(w) - f(w^{(i)})) \cdot (g(w) - g(w^{(i)})) \geq 0.$$

Expanding the product and taking expectations, we obtain:

$$\mathbb{E}\big[f(w)g(w)\big] - \mathbb{E}\big[f(w)g(w^{(i)})\big] - \mathbb{E}\big[f(w^{(i)})g(w)\big] + \mathbb{E}\big[f(w^{(i)})g(w^{(i)})\big] \geq 0.$$

By the symmetry of expectations, the first and last terms are equal, and the second and third terms are also equal, so we obtain the desired inequality:

$$\mathbb{E}\big[f(w)g(w)\big] \geq \mathbb{E}\big[f(w)g(w^{(i)})\big].$$

This completes the proof. $\qquad \square$

## K  ADDITIONAL DISCUSSIONS AND RESULTS

### K.1  CLARIFICATION ON NEURON RESONANCE

Intuitively, once a neuron has already learned a single feature, then its activation frequency will match the feature's occurrence frequency. Our work, however, is about the *reverse* direction under a concrete training procedure:

*If a group of neurons are trained to activate at frequency p, under suitable conditions these neurons can provably recover features with frequency lying in a corresponding "resonance band".*

This is a non-trivial statement because at random initialization, the neurons do not align with any features, and the training dynamics must guide them to do so if they are tuned at the right frequency. This result justifies that a more active way of training Sparse Autoencoders (SAEs) is plausible, and provides a theoretical foundation for our GBA algorithm.

### K.2 CLARIFICATION THAT GBA IS NOT A HIERARCHICAL SAE

One might wonder whether GBA with multiple frequency groups is essentially the same as Matryoshka SAEs (Bussmann et al., 2025) or Hierarchical TopK SAEs (Balagansky et al., 2025) that are designed to recover hierarchical features. However, we would like to clarify that GBA is fundamentally different from these models, esentially because GBA uses a *single loss* for all groups, while hierarchical SAEs or Matryoshka SAEs use *multiple losses*, each applied to a subset of the neurons.

As a consequence, one limitation of GBA is that it is not guaranteed to separate hierarchical features. This does not violate our theoretical analysis, since we assume that the feature coefficient matrix $H$ has uniformly random supports, which rules out hierarchical feature structures.

To make this distinction more concrete, consider the following minimal example. Let $a, b \in \mathbb{R}^d$ be unit vectors with $b \neq \pm a$, and suppose the dataset consists only of the two inputs

$$x \in \{\, a + b, \ a - b \,\},$$

i.e., the high-level feature $a$ always co-occurs with either $b$ or $-b$. Under a single global reconstruction loss (the structure used by GBA), there is no penalty for representing the data by the two basis vectors $a+b$ and $a-b$ themselves. In other words, the model can minimize loss by learning neurons aligned with $a + b$ and $a - b$, without ever isolating $a$ or $b$ individually. By contrast, hierarchical or multi-loss SAEs (e.g., Matryoshka) impose intermediate reconstruction objectives or capacity constraints at different levels. A high-level group that must explain the input with very few neurons (or with its own reconstruction loss) is incentivized to capture the common component $a$ rather than the signed combinations $a \pm b$. Thus the multi-loss design can force a decomposition that separates high- and low-level factors.

Furthermore, this distinction explains why GBA is primarily compared against other single-loss SAEs (TopK, JumpReLU, $L_1$) in our experiments, rather than Matryoshka. Notably, despite not being designed for hierarchical feature recovery, GBA still demonstrates competitive performance on SAEBench compared to more advanced models like Matryoshka, as shown in Table 3. We believe that extending GBA to handle hierarchical feature decomposition is an interesting direction for future work, and it remains an insteresting research question on how to apply frequency-aware training in that context.

### K.3 ADDITIONAL RESULTS ON NEURON ANALYSIS

We further provide additional studies on the neurons learned by the GBA and TopK methods in terms of the three metrics used in the main experiment: maximum activation, Z-score, and maximum cosine similarity across different runs with different random seeds. All the other setup is the same as in Figure 4. These metrics are computed based on the validation part of Github dataset, with the hook position at the MLP output of layer 26. For the Z-score, we compute the largest value among the tokens in the validation set, and for the maximum cosine similarity, we compute the smaller value among the two additional runs. See §C.4 for rigorous definitions of these metrics. In addition, for each neuron $m$, we also compute the *activation fraction* (or activation rate), which is defined as the fraction of tokens where pre-activations of neuron $m$ are non-negative.

Thus, for each neuron $m$, we have four metrics: maximum activation, Z-score, maximum cosine similarity, and activation fraction. We generate scatter plots by plotting the Z-score against the other three metrics. The results for GBA and TopK are presented in Figure 15 and Figure 16, respectively.

**Z-score v.s. maximum activation.** In Figure 15 (left), we present the scatter plot of the Z-score versus the maximum activation of neurons, which is shown in the logarithmic scale with base 10. We observe an almost linear relationship between the two metrics, indicating that neurons with higher Z-scores also exhibit higher maximum activations. Notably, at the upper end of the distribution, a subset of neurons attains even higher Z-scores. This behavior suggests that these neurons capture a "cleaner" feature and fire exclusively when the feature is present. By the definition of the Z-score, for neurons with the same maximum activation, a higher Z-score implies a lower variance. In other words, these neurons' activations tend to be bimodal—predominantly near a baseline when the feature is absent and significantly higher when the feature is present. This is consistent with the dashboard results for individual neurons as we shown in Figure 17.

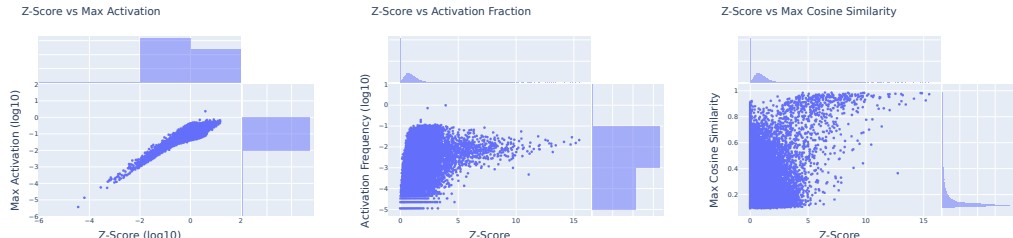

Figure 15: Scatter plots for all SAE neurons illustrating neuron properties for the GBA method: Z-score versus Maximum Activation, Fraction of Non-negative Pre-Activations (i.e., activation frequency), and Maximum Cosine Similarity across different runs with different random seed. The 66k-neuron SAE is trained on the `GitHub` dataset with a hook at the MLP output of layer 26.

**Z-score v.s. activation fraction.** In Figure 15 (middle), we present a scatter plot of the Z-score versus the activation fraction, which is shown in the logarithmic scale with base 10. Neurons with *higher Z-scores* generally exhibit an activation fraction near $10^{-3}$ to $10^{-1}$. This suggests that they predominantly capture infrequent yet salient features. Comparing to the TopK method, we observe that the GBA method is more effective in capturing infrequent features, which is primarily due to the fact that we purposefully assign groups with both high and low target activation frequency. Additionally, the neuron grouping mechanism effectively adapts to diverse feature occurrence frequencies, underscoring the adaptivity of our approach. Additionally, we observe several neurons with activation frequencies exceeding the HTF of 0.1 (by our default configurations). This behavior is facilitated by the bias-clamping mechanism, which prevents biases from becoming excessively negative, as discussed in §4.

On the contrary, we observe in Figure 16 (middle) that TopK is good at capturing frequent features, but for infrequent features, it has very low Z-scores compared to the GBA method. This fact again underscores the effectiveness of our approach in capturing infrequent features.

**Z-score v.s. MCS.** In Figure 15 (right), we present a scatter plot of the Z-score versus maximum cosine similarity across different runs with distinct random seeds. Recall that a higher maximum cosine similarity indicates more consistent feature recovery, and we observe that neurons with higher Z-scores tend to exhibit higher levels of consistency. This result supports the effectiveness of GBA in reliably extracting salient features.

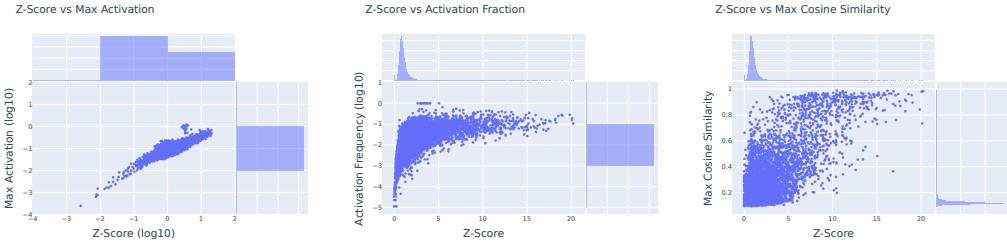

Figure 16: Scatter plots SAE neurons illustrating neuron properties trained using TopK with $K = 300$. The other configurations are the same as Figure 15

## K.4 WHY FREQUENCY-AWARE TRAINING? A TOY EXAMPLE

To illustrate the benefits of frequency-aware training, we present a simple toy example, where we will show that standard TopK/L1 SAEs can fail to recover the ground-truth features.

**Data Generation.** We randomly sample $n = 128$ features $\{v_i\}_{i=1}^n \in \mathbb{S}^{d-1}$ with $d = 42$ dimensions. We consider the dataset to be *heavily imbalanced* in the sense that different data could contain dramatically different number of active features. Speicifically, we consider two types of data points:

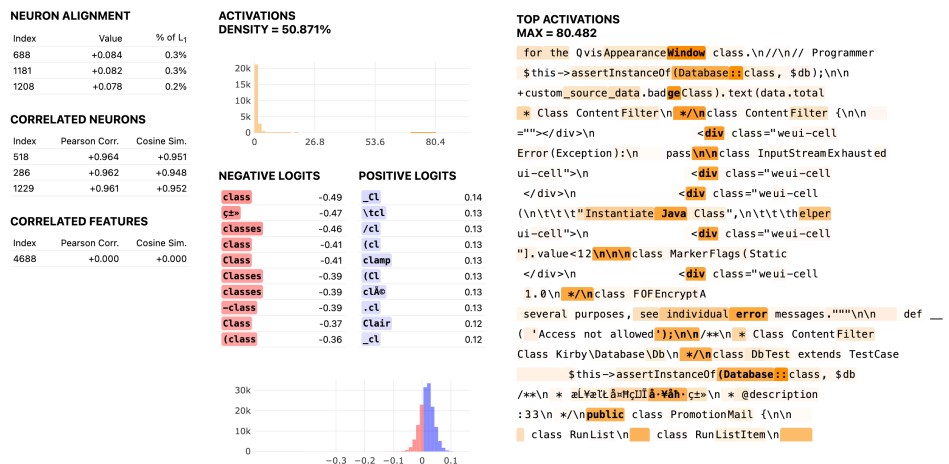

Figure 17: Feature dashboard for neuron 4688 in the GBA-SAE model trained on Pile Github at layer 26's MLP output position. This neuron exhibits a clear bimodal activation pattern, and is activated before outputting the "class" token.

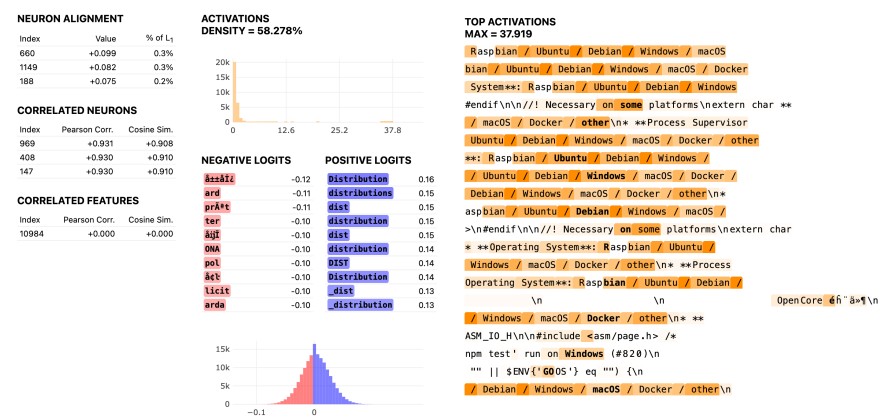

Figure 18: Feature dashboard for neuron 10984 in the GBA-SAE model trained on Pile Github at layer 26's MLP output position. This neuron activates uniquely when the model is going to generate names of operating system such as "Ubuntu", "Windows", and "macOS". This is a good example of a neuron that captures a concept rather than a specific token.

1. **Type A**: with probability $0.5$, we sample $s = 3$ features uniformly at random from $\{v_i\}_{i=1}^n$, take their sum and $\ell_2$-normalize as the data point $x$.

2. **Type B**: with probability $0.5$, we sample $s = 20$ features uniformly at random from $\{v_i\}_{i=1}^n$, take their sum and $\ell_2$-normalize as the data point $x$.

Thus, Type A data points are sparse combinations of a few features, while Type B data points are denser combinations of many features. This specific data generation resemble the real-world scenario where different data points could contain different number of active features. Note that the only challenge here is the *imbalance* in the number of active features, while the feature frequencies are all uniform at $f = (3 + 20)/2n \approx 0.09$.

**Why it could be a challenge for standard SAEs.** Standard SAEs like TopK regularized SAEs typically assume a *fixed sparsity level* across all data points. For example, a TopK SAE with $K = 10$ assumes that each data point can be represented by activating only $10$ neurons. While this $K$ value could be suitable for learing Type A data points (which only need 3 active features), it is too small for Type B data points (which need 20 active features). Hence, the model could struggle to learn real features from Type B data points.

**Training Setup.** We consider training TopK SAEs as our baseline method and make comparison with our proposed GBA method. For all SAEs, we set the number of neurons to be $M = 8192$. We consider the following configurations:

1. **TopK SAE**: We train TopK SAEs with $M = 8192$ neurons and sweep $K \in \{20, 30, 50\}$. These values correspond to different sparsity assumptions: $K = 20$ matches exactly the sparsity of the dense (Type B) samples, while $K = 30$ and $K = 50$ represent over-estimates. This setup evaluates the model's robustness to fixed-sparsity constraints when the true data sparsity varies significantly.

2. **GBA SAE**: We train two varients of GBA SAEs: (i) **Full coverage**: We set HTF=0.5, LTF=0.001 just like in our main experiments with 10 groups. This setup ensures that the target frequencies perfectly cover the feature frequency $f \approx 0.09$, allowing us to assess GBA's effectiveness when the frequency range is well-specified. (ii) **Misspecified coverage**: We set HTF=0.01, LTF=0.001 with 10 groups, which does not cover the feature frequency $f \approx 0.09$. This setup tests GBA's robustness to imperfect frequency range specifications.

**Results.** We present the results in Table 1 (in the main text). We observe that:

- **TopK SAE's performance is highly sensitive to $K$**: When $K = 20$, the model perfectly matches the sparsity of Type B samples, achieving a perfect FRR of $100\%$ at MCS $\geq 0.8$ and $98.4\%$ at MCS $\geq 0.9$. However, as $K$ increases to 30 and 50, the FRR drops significantly, especially at the higher MCS threshold of 0.9. This indicates that to recover features accurately, the sparsity level must be carefully tuned to the data distribution, which may not be feasible in practice.

- **GBA SAE with full coverage excels**: The GBA model with full frequency coverage achieves perfect FRR of $100\%$ at both MCS thresholds, demonstrating its ability to adaptively learn features across varying sparsity levels in the data. The $100\%$ FRR is not surprising here as predicted by our theoretical analysis. On the other hand, if we misspecify the frequency coverage, the FRR drops significantly. These results again validate that it is indeed the frequency-aware training that enables effective feature recovery.

## K.5 ADDITIONAL SAEBENCH EVALUATION

We provide additional comparison of GBA with other SAE variants on SAEBench (Karvonen et al., 2025) in Table 3.

For GBA, we always fix HTF $= 0.5$ and number of groups $K = 20$. To create GBA run with different $L_0$ sparsity, we vary the LTF from $10^{-3}$ to $10^{-4}$, and also also vary the number of neurons assigned to each group and $\gamma_+$, the coefficient that controls how much we increase the bias for dead

neurons in each group (See line 18 of Algorithm 1). Specifically, we allocate more neurons to groups with lower target frequencies while also decrease $\gamma_+$ if we want to achieve a higher overall sparsity level $L_0$. For this experiment, we train a groups of GBA models with final $L_0$ values ranging from 132.9 to 694.9.

**Why we do not target the very high sparsity (extremely low $L_0$) regime?**    In our sparsest run ($L_0 = 132.9$), we set LTF $= 10^{-4}$, $\gamma_+ = 10^{-4}$ and allocated a linearly increasing number of neurons to groups with lower target frequencies. Empirically, we find that

1. Further decreasing the LTF, $\gamma_+$ or assigning more neurons to low-frequency groups dramatically exacerbates the percentage of dead neurons, with the fraction of dead neurons exceeding $50\%$. This behavior is expected, as extremely low target frequencies (e.g., $< 10^{-4}$) are difficult to maintain stable given limited batch sizes, making neurons prone to becoming permanently inactive. This constraint is fundamentally different from TopK methods, where sparsity can be globally hardcoded by the parameter $K$.

2. Further tuning these hyperparameters does not effectively reduce the $L_0$ value. One evidence is from our ablation studies in Figure 6, where the lowest achievable sparsity is about $0.2\%$, which corresponds to $L_0 \approx 132$ for our setup with $66k$ neurons.

3. The $100 \sim 700$ region of $L_0$ already covers a wide range of sparsity levels that are of practical interest, whereas similar range of $L_0$ has also been used in prior SAE works (Gao et al., 2024).

Consequently, to evaluate the performance of GBA in a reasonable configuration setting, we do not make further attempt to push GBA into the extremely low $L_0$ regime in this experiment.

We provide the comparison results of GBA with other SAE varients in Figure 19. Here, we only train the GBA models with different $L_0$ values, and the results of the other SAE models are taken from Karvonen et al. (2025), where the SAE baselines are all trained with the same data and LLM architecture as ours.

GBA method demonstrates a distinct performance profile compared to standard Sparse Autoencoder (SAE) baselines. Key observations include:

1. **Explained Variance:** GBA consistently achieves the highest explained variance across all scrutinized $L_0$ levels (ranging from 100 to 400), significantly outperforming Standard and PAnneal baselines while maintaining a slight edge over BatchTopK and Matryoshka models.

2. **Absorption Score:** GBA achieves the lowest Absorption Score (approaching $10^{-2}$ on the log scale), which is significantly better than Standard, TopK, and GatedSAE (hovering around $10^{-1}$). This indicates that GBA features are consistently activated when their corresponding input features are present, demonstrating effective feature capture. We add more discussions on this point later.

3. **SCR Score:** GBA significantly underperforms on the Spurious Correlation Removal (SCR) metric when we push the $L_0$ value to 132.9. One possible reason for such a sharp drop is that many neurons in the model are allocated to extremely low-frequency groups (with target frequencies below $10^{-3}$), and the model can fail to capture spurious features that are used for the SCR task.

4. **Sparse Probing:** In the sparse probing classification task, GBA remains competitive with an accuracy of approximately $0.953 \sim 0.957$. However, it is marginally outperformed by other varients, which achieve slightly higher linear separability at similar sparsity levels.

5. **TPP Score:** GBA shows competitive performance on the TPP metric compared to other SAE varients, and is on par with BatchTopK and outperform JumpReLU and GatedSAE, demonstrating its ability to capture features with high causal disentanglement quality.

6. **Alive Fraction:** The method exhibits exceptional stability with a near-perfect alive fraction ($\approx 1.0$) for $L_0$ values above 300. However, for lower $L_0$ values (e.g., 132.9), the alive fraction decreases to around 0.75. The reason for this drop lies in how we configure GBA for achieving low $L_0$ values, which involves setting very low target frequencies for some groups, assigning more neurons to low-frequency groups, and using a small $\gamma_+$ value. All these factors contribute to an increased likelihood of neurons becoming permanently inactive.

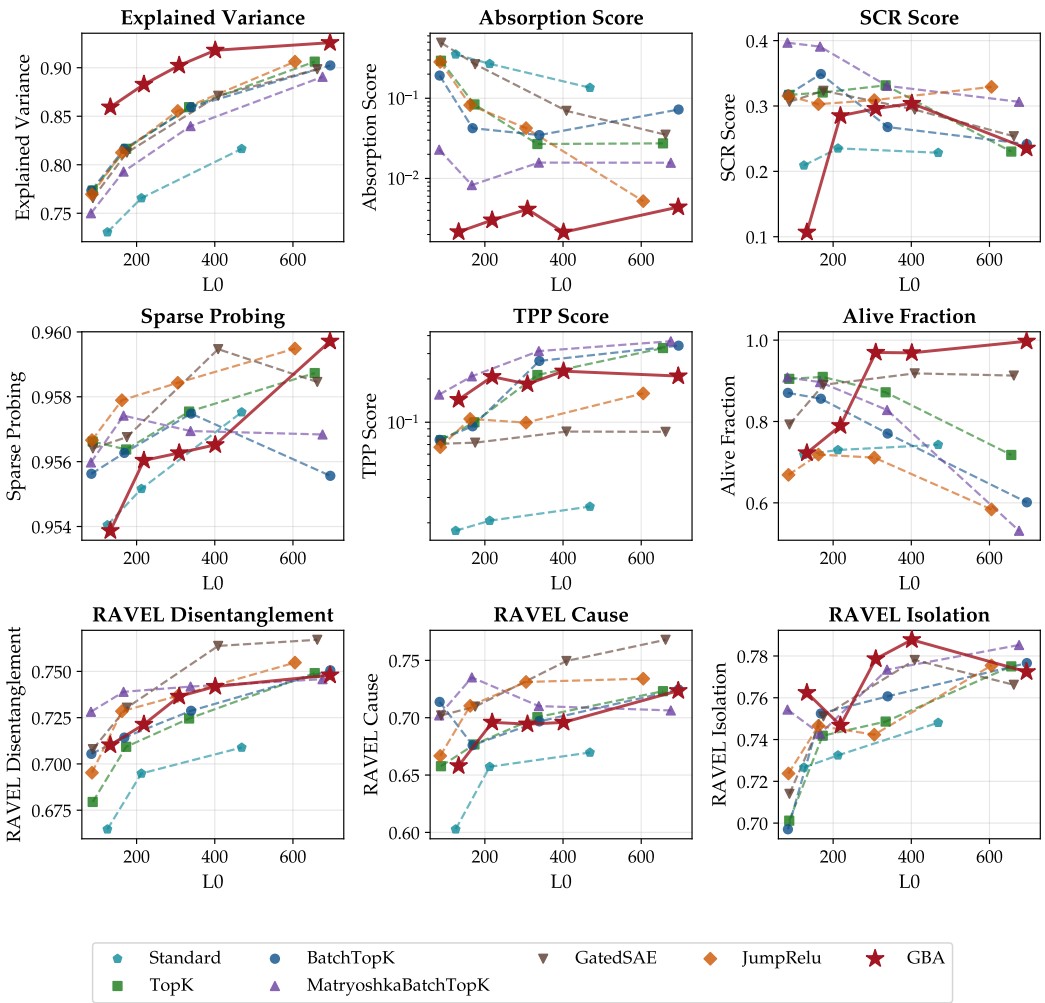

Figure 19: **SAEBench Evaluation Results.** We compare GBA with other SAE variants across multiple metrics. Here, we constrain the comparison to SAE models with $L_0$ values between 100 and 500 to ensure a fair evaluation. For SCR (Marks et al., 2024) and TPP metrics, we take the average of the scores over Top-20 and Top-50 neurons as scores evaluated for these numbers tend to be more stable (Karvonen et al., 2025) while avoiding biases from too limited neuron counts. All the metrics except Absorption Score are better when higher, while Absorption Score is better when lower. GBA demonstrates competitive performance across Explained Variance, Absorption Score, TPP Score, Alive Fraction (for high $L_0$), and RAVEL Isolation metrics.

7. **RAVEL Metrics:** GBA performs competitively on the RAVEL suite, particularly in *RAVEL Isolation* and *RAVEL Disentanglement*. The performance trend improves as $L_0$ increases, suggesting the features recovered are causally distinct and well-separated.

The tables for similar $L_0$ SAE comparison are also available in Table 5 and Table 6 for $L_0$ around 100 and 600, respectively.

**Further Discussion on Absorption Score.** GBA demonstrates particularly strong performance on the Absorption Score, achieving values significantly lower than other methods. To ensure this low score reflects genuine feature disentanglement rather than trivial artifacts, we additionally evaluate the **Mean F1 score** for the first-letter prediction task, following Chanin et al. (2024). Formally, the F1 score is defined as the harmonic mean of precision and recall:

$$F_1 = \frac{2 \cdot \text{Precision} \cdot \text{Recall}}{\text{Precision} + \text{Recall}},$$

where Precision = $\frac{TP}{TP+FP}$ and Recall = $\frac{TP}{TP+FN}$, with TP, FP, and FN denoting true positives, false positives, and false negatives, respectively. This metric evaluates the utility of the top-$k$ neurons—selected via cosine similarity with a task-specific probe—in predicting the first letter of a token. A high Mean F1 score confirms that the unabsorbed features are semantically meaningful.

As shown in Figure 20, GBA consistently achieves Mean F1 scores in the range of $0.6$–$0.7$ across different sparsity levels. Such an Mean F1 Score matches and is even slightly better than those reported in (Chanin et al., 2024). This suggests that GBA successfully recovers accurate, task-relevant features, thereby validating that its low Absorption Score reflects genuine disentanglement rather than artifacts. Furthermore, the Mean F1 score increases only mildly with $k$, implying that the top neuron alone captures most of the predictive power for this task—a strong indicator of minimal feature absorption.

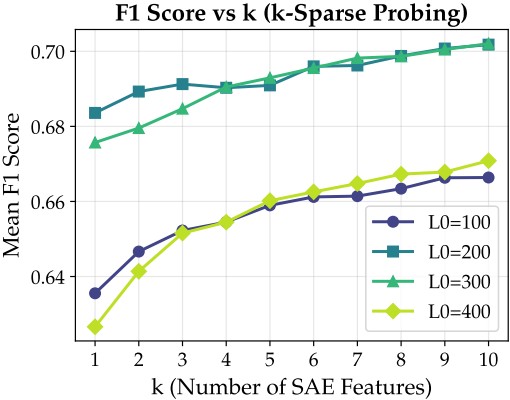

Figure 20: **Mean F1 Score Comparison.** We compare the Mean F1 score of the top-$k$ neurons for the first letter prediction task in our trained GBA SAEs with different $L_0$ values. Notably, across different sparsity levels, we find that GBA consistently achieves Mean F1 scores around $0.6 \sim 0.7$, while standard SAEs usually have Mean F1 scores below $0.5$ for these $L_0$ levels as reported in (Chanin et al., 2024). This indicates that GBA indeed learns SAE latents that are accurate in capturing meaningful features and the low Absorption Score is not due to trivial artifacts.

| Metric | GatedSAE | TopK | BatchTopK | Matryoshka | JumpReLU | GBA (ours) | Standard |
|---|---|---|---|---|---|---|---|
| $L_0$ | 175.2 | 173.4 | 168.6 | 166.7 | 162.5 | 132.9 | 125.3 |
| Explained Variance ↑ | 0.812 | 0.816 | 0.816 | 0.793 | 0.812 | **0.859** | 0.730 |
| Absorption Score ↓ | 0.2657 | 0.0838 | 0.0424 | 0.0083 | 0.0821 | **0.0022** | 0.3529 |
| SCR Metric ↑ | 0.323 | 0.321 | 0.349 | **0.391** | 0.303 | 0.107 | 0.209 |
| Sparse Probing ↑ | 0.957 | 0.956 | 0.956 | 0.957 | **0.958** | 0.954 | 0.954 |
| TPP Metric ↑ | 0.072 | 0.100 | 0.094 | **0.209** | 0.105 | 0.144 | 0.018 |
| Alive Fraction ↑ | 0.890 | **0.910** | 0.856 | 0.897 | 0.719 | 0.723 | 0.719 |
| RAVEL Disent. ↑ | 0.730 | 0.709 | 0.714 | **0.739** | 0.728 | 0.710 | 0.665 |
| RAVEL Cause ↑ | 0.710 | 0.677 | 0.676 | **0.735** | 0.710 | 0.658 | 0.603 |
| RAVEL Isolation ↑ | 0.751 | 0.742 | 0.752 | 0.743 | 0.747 | **0.762** | 0.727 |

Table 5: **Performance comparison of SAE models with $L_0$ between $100 \sim 200$ on SAEBench.** Arrows indicate whether higher (↑) or lower (↓) values are better. Bold indicates best performance, and underline indicates second best. GBA achieves the best performance in 3 out of 9 metrics, and for TPP metric GBA is also the second best.

## L  THE USE OF LARGE LANGUAGE MODEL

We acknowledge the use of a large language model (LLM) primarily to improve the grammar and clarity of this manuscript. The LLM was also used to assist with debugging and generating boiler-plate code snippets, which were reviewed and validated by the authors.

| Metric | Standard | BatchTopK | GBA (ours) | Matryoshka | GatedSAE | TopK | JumpReLU |
|---|---|---|---|---|---|---|---|
| $L_0$ | 835.1 | 695.6 | 694.9 | 675.7 | 662.3 | 655.7 | 605.2 |
| Explained Variance ↑ | 0.852 | 0.902 | **0.926** | 0.891 | 0.898 | 0.906 | 0.906 |
| Absorption Score ↓ | 0.0873 | 0.0724 | **0.0044** | 0.0157 | 0.0351 | 0.0274 | 0.0052 |
| SCR Metric ↑ | 0.239 | 0.242 | 0.235 | 0.306 | 0.254 | 0.230 | **0.329** |
| Sparse Probing ↑ | 0.958 | 0.956 | **0.960** | 0.957 | 0.958 | 0.959 | 0.959 |
| TPP Metric ↑ | 0.025 | 0.340 | 0.209 | **0.365** | 0.086 | 0.328 | 0.159 |
| Alive Fraction ↑ | 0.750 | 0.601 | **0.997** | 0.531 | 0.913 | 0.718 | 0.584 |
| RAVEL Disent. ↑ | 0.731 | 0.751 | 0.748 | 0.746 | **0.767** | 0.749 | 0.755 |
| RAVEL Cause ↑ | 0.680 | 0.725 | 0.724 | 0.706 | **0.768** | 0.723 | 0.734 |
| RAVEL Isolation ↑ | 0.781 | 0.777 | 0.772 | **0.785** | 0.766 | 0.775 | 0.775 |

Table 6: **Performance comparison of SAE models with $L_0$ between** $600 \sim 850$ **on SAEBench.** Arrows indicate whether higher (↑) or lower (↓) values are better. Bold indicates best performance, and underline indicates second best.

