# OpenReview forum: "Taming Polysemanticity in LLMs: Theory-Grounded Feature Recovery via Sparse Autoencoders"
_ICLR.cc/2026/Conference — ICLR 2026 Poster_

### Official Review · Reviewer_VpeD · 2025-10-29

**Soundness:** 1
**Presentation:** 2
**Contribution:** 2
**Rating:** 2
**Confidence:** 4

**Summary:**

This paper introduces a novel way of selecting SAE encoder bias (or, I think sometimes JumpReLU biases too) by adjusting the bias during training to force the SAE latent to fire within a pre-determined frequency range. The paper introduces a theorem stating that in order for an SAE latent to recover an underlying feature, it needs to fire with roughly the frequency of that feature. The paper turns this into an SAE architecture, called "Bias adaptation" (BA) or "Group Bias Adaptation" (BA), and evaluates the SAE on LLMs.

**Strengths:**

- The algorithm for tuning the bias is novel and simple
- The paper provides a theorem around when features can be recovered by an SAE

**Weaknesses:**

- This paper seems confused about how SAEs work currently. There is an implicit assumption that current SAEs do not naturally find the correct firing frequencies for latents, but this is easily disprovable. Toy model experiments with standard SAE architectures will show that current SAEs already naturally find the correct firing frequencies for latents.
- It is hard to tell what the authors are concretely doing in experiments. The background only talks about tied SAEs, but adds an encoder bias which does not make sense for tied SAEs. I suspect that the paper is not using tied SAEs at all, but it's very hard to tell. It is also confusing what specific bias is being used where, for instance, the authors talk about JumpReLU GBA SAEs, where it would make sense to control the JumpReLU bias, not the encoder bias.
- The experiments in this paper are not rigorous and confusingly do not compare to SOTA architectures. The authors train a JumpReLU version of their SAEs (JumpReLU is considered SOTA), but do not compare this to any SOTA architectures (BatchTopK or JumpReLU).
- The SAEs where SAEBench metrics are run are very strange, with L0 dramatically higher than I have ever seen before. SAEBench also tends to be noisy, and requires multiple seeds to be run to get a good signal, but this is not done.
- This paper is built on the idea that SAE latents need to be fire at certain frequencies to learn the underlying features, but then hardcodes arbitrary feature firing bands for the latents to fire at.

Overall, I believe the claim that SAEs latents should match the underlying firing frequencies of the features they track, but do not believe that the methods described in this paper are a valuable or valid way to adjust SAE bias thresholds. I also do not believe that the experiments are done in a valid and rigorous way.

**Questions:**

### Larger questions
- Why do SAEs not already naturally find the correct feature firing frequencies for features? In toy models SAEs seem to naturally learn the correct features and fire them the correct times, especially for toy models with independent features like you use. It sounds like your paper is claiming this only happens if you manually set the encoder bias? This seems hard to believe.
### Section 2
- Tied SAEs do not usually have an encoder bias, as setting a non-zero encoder bias breaks the tied SAE's ability to accurately recover features. Why was this added? If this is being used, it seems like it should harm the SAE. encoder bias only makes sense in untied SAEs.
### Section 3
- L150: "We construct s-sparse coefficient matrices H, which have uniform feature occurrence frequency" - does this mean that every input activation has the same number of active features, or just that each feature is sampled independently with a set frequency? Does every feature have the same frequency?
- Does the SAE have the same width as the number of true features?
- In section 2 it says that $b_m$ is a trainable parameter, but it seems you're not training it and setting it via a different method?
- Figure 2, right: The text says optimal recovery occurs when p = f, but this plot seems to show that optimal recovery occurs when p > f. Recovery is very low at f in this plot.

### Section 4
- Algorithm 1: Line 17 and 18, I think you've mixed up "min" and "max"

### Section 5
- L286: I don't understand this line. Up to now your SAEs have all been L1 SAEs with no mention of JumpReLU. What does it mean that "All methods employ JumpReLU activation"? How are you training the JumpReLU SAE? There are at least 2 different ways of training these (Deepmind's version and Anthropic's version). Is this a tied SAE? Why are you not comparing to a JumpReLU SAE if you're using JumpReLU SAEs for your method?
- In the appendix, you say: "JumpReLU decouples the neuron output magnitude from its bias". Are you controlling the *JumpReLU bias* with your method or the *encoder bias*? These are different things.
- Figure 4: Are all the SAEs you're comparing with tied SAEs? I'm confused if you're using tied SAEs at all, despite section 2 claiming you are using tied SAEs.
- Figure 4: You using JumpReLU, a state-of-the-art method for your own technique, but comparing against non-SOTA architectures (ReLU, TopK). This is not a valid comparison.
- Table 2: The L0 for these SAEs is unreasonably high. I have never seen anyone go above L0 of 400 for an SAE on Gemma-2-2b, yet these SAEs are all L0 600+, with your method having the highest L0. These experiments should be conducted at a reasonable L0, ideally multiple L0s.
- Table 2: SAEBench can be very noisy, so you should run multiple seeds and report mean and stdev. The results as-is do not mean much.

### Minor / formatting
- The use of the word "neuron" for the hidden activation of the SAE is confusing. I initially thought this was referring to model neurons. SAE hidden neurons are typically referred to as "features" or "latents" in the literature.
- L92: It looks like this is missing subtracting the decoder bias, based on your definition of an SAE in L132

---

> ### Author Response · Authors · 2025-11-21
> **Rebuttal (Part 1)**
>
> We thank the reviewer for the detailed and thoughtful comments.
> Below we first address the high-level concerns, then respond point-by-point to the specific questions.
>
> *Please note*: In the following, we incorporate coherent modifications to Appendix L to address your concerns. However, to keep the current line numbering consistent for all reviewers, we do not move these changes to the main sections yet.
> Some minor edits to the main text is marked with red color for visibility.
>
> ---
>
> ### Q1. "There is an implicit assumption that current SAEs do not naturally find the correct firing frequencies for latents, but this is easily disprovable."
>
> To be clear, **Our claim is not that standard SAEs *cannot* find the correct frequencies**. This is a misunderstanding of our claim. The research question we are tackling is: *Can we design a simple, theoretically grounded method to better learn monosemantic features?*
> In fact, previous methods have specific issues that limit their ability to do so. As we have pointed out in both Introduction (line 52-66) and Preliminaries (line 140-144):
> - The $L_1$ method is sensitive in the penalty coefficient, and suffers from activation shrinkage, while TopK enforces a fixed global sparsity level that overlooks the diverse feature occurrence patterns in real data. What this means is that to achieve good performance with these methods, one must carefully tune hyperparameters or have strong prior knowledge about the data distribution.
> - In addition, all the existing methods lack theoretical guarantees on feature recovery.
>
> Therefore, we propose Grouped Bias Adaptation (GBA): a simple, theoretically grounded procedure that explicitly controls per‑neuron activation frequencies based on the neuron resonance phenomenon revealed by our analysis and synthetic experiments. By aligning neuron firing rates with feature occurrence frequencies, GBA yields better reconstruction, sparsity, and interpretability in our empirical evaluations.
>
> ---
>
> ### Q2. "Toy model experiments with standard SAE architectures will show that current SAEs already naturally find the correct firing frequencies for latents."
>
> We are not absolutely certain **which prior work the reviewer is referring to here**. However, giving the context, we assume the reviewer is referring to Elhage et al. 2022 "Toy models of superposition." If so, we would like to clarify some key differences between their setup and ours:
> 1. **Model architecture:** Elhage et al. study models with the form of $\\mathrm{ReLU}(W^\\top Wx + b)$, which is not a standard SAE architecture, while we directly analyze standard SAEs with encoder/decoder structure.
> 2. **Learning guarantees:** Elhage et al. focus more on understanding the superposition phenomenon, but they do not provide theoretical guarantees for recovery of all monosemantic features.
>
> In contrast, our contributions are twofold:
>
> * **Theory:** Theorem 6.1 provides the first provable guarantee (to our knowledge) that an SAE training algorithm can recover all monosemantic features, and it explicitly characterizes a resonance band for the neuron activation frequency $p$ relative to the feature occurrence frequency $f$. In our simplified model this band is
>    $$
>    f \\;\\lesssim\\; p \\;\\lesssim\\; \\min\\{\\sqrt{f},\\, d f\\},
>    $$
>    i.e., neurons are most effective when $p$ lies in a band around (but not necessarily exactly equal to) $f$. This is a novel theoretical insight that goes beyond anecdotal observations, which is acknowledged by Reviewer fF2q and 73R2 as well.
>
> * **Empirics in both synthetic and LLM settings:** We empirically validate the neuron resonance phenomenon in the synthetic setting (Fig. 2) and show with real LLM SAEs that GBA achieves the reconstruction-sparsity-consistency frontier better than prior methods (Fig. 4-5) with competitive explanability metrics (Table 2).
>
> In addition, we would also like to point out previous work that shows some SAEs can fail in learning correct features in certain settings. For example, Bricken, et al. (2023) "Towards Monosemanticity: Decomposing Language Models With Dictionary Learning", shows that standard SAEs can suffer from feature splitting issue if the number of neurons scales up.
> Also, Paulo et al. (2025) "Sparse autoencoders trained on the same data learn different features" shows that different random seeds can lead to different learned features even when training on the same data with TopK SAEs.
> So it is not always the case that standard SAEs can learn the correct frequencies. Therefore, having a rigorous framework like Eq. (2.1) and going from pure empirics to a method with theoretical guarantees is important.

---

> ### Author Response · Authors · 2025-11-21
> **Rebuttal (Part 2)**
>
> ### Q3. Architectural clarity: tied SAEs, encoder biases, and JumpReLU: "The background only talks about tied SAEs, but adds an encoder bias which does not make sense for tied SAEs ... it would make sense to control the JumpReLU bias, not the encoder bias."
>
> **(a) Tied SAEs and per-neuron bias**
>
> Our SAE in Eq. (2.2) is a tied SAE with a per‑neuron (or per-latent) bias and a shared pre‑bias:
>
> $$
> y = W\\bigl(x - b_{\\mathrm{pre}}\\bigr) + b,\\qquad
> z = \\phi(y),\\qquad
> \\hat{x} = W^\\top \\operatorname{diag}(a) z + b_{\\mathrm{pre}}, \\quad \\text{where}\\quad (y, z, b) \\in \\mathbb{R}^M, (x, b_{\\mathrm{pre}}, \\hat{x}) \\in \\mathbb{R}^d
> $$
>
> For clarity, we explicitly state that our SAE uses tied weights; this point is mentioned multiple times in the manuscript (**see lines 129–138**).
>
> Regarding the comment that “adding an encoder bias does not make sense for tied SAEs”:
> - Adding per‑neuron bias is a standard, well‑motivated design choice and does not conflict with tied weights. These per-neuron biases set each unit’s activation threshold and are commonly used to achieve and control sparsity in both tied and untied SAEs (e.g., L1);
> - Prior work on tied SAEs (see our cited work Cunningham et al., 2024, Eq (1)-(3)) also includes biases in their tied architectures.
> The presence of a per-neuron bias preserves the tied decoder $W^T$ while allowing independent control of neuron firing probabilities, which is precisely the mechanism GBA exploits.
>    - For example, without the per-neuron bias, for $L_1$ SAE with ReLU activation, it is not clear how to ensure sufficient neuron sparsity.
>
> **(b) Which bias does GBA control vs. JumpReLU bias?**
> We sincerely express our gratitude for the reviewer's comments on this point that might cause confusion. In our implementation there is a single per‑neuron bias parameter $b_m$. Concretely:
>
> - For JumpReLU (thresholding), we just replace $\\phi$ with $\\mathrm{JumpReLU}$  and $b_m$ serves as the JumpReLU threshold in
> $$
> \\operatorname{JumpReLU}(y_m; b_m) =
> \\begin{cases}
> 0, & y_m + b_m < 0,\\\\[4pt]
> y_m & y_m + b_m \\ge 0
> \\end{cases}, \\text{ where } y_m = w_m^\\top (x - b_{\\mathrm{pre}})
> $$
> Thus, the usage of the bias $b_m$ is exactly as expected by the reviewer: it serves as the "encoder bias" if we are using ReLU activation, and as the "JumpReLU threshold" when using JumpReLU activation.
> Throughout the paper, we refer to this parameter as the "bias" $b_m$ for consistency, and did not use different terms for different activations to avoid confusion.
> The details are clarified in App. C.1, and there is a note "Minor notational discrepancy" explaining this.
> In the revision, we will clarify this point in the preliminary section (§2) to avoid confusion.
>
> ---
>
> ### Q4. Experimental setup and SOTA comparisons: "The experiments in this paper are not rigorous and confusingly do not compare to SOTA architectures..."
>
> **(a) Qwen experiments (§5, Fig. 4–5, Table 1)**
> We would like to clarify that in the Qwen2.5-1.5B experiments, **all methods employ JumpReLU activation** as we state in line 286.
> That is, TopK, L1, BA, and GBA all use JumpReLU as the nonlinearity, with the only difference being the sparsity mechanism (L1 penalty, TopK selection, or grouped bias adaptation).
> We believe that this is sufficient to justify the comparison, because (i) JumpReLU is one of the SOTA architectures (also aggreed by the reviewer), and (ii) the comparison is only about algorithmic differences, ruling out any possible architecture-specific advantages. Again, all methods are trained with the tied-weight SAE in Eq. (2.2) with biases serving as JumpReLU thresholds.
>
>
> **(b) SAEBench comparisons to “SOTA” SAEs**
>
> For interpretability metrics on Gemma2-2B, we *do* compare to state-of-the-art SAE architectures **JumpReLU SAE** and **Gated SAE** as implemented in SAEBench as you can see in **Table 2**.
>
> Hope this clarifies the concerns.

---

> ### Author Response · Authors · 2025-11-21
> **Rebuttal (Part 3)**
>
> ### Q5. High $L_0$ and multiple seeds on SAEBench: "The SAEs where SAEBench metrics are run are very strange, with L0 dramatically higher ... and requires multiple seeds to be run ..."
>
> The large $L_0 \\approx 600$ values in Table 2 are not unique to GBA; they are also present for the official GatedSAE, TopK, and JumpReLU SAEs at width 66k (as we have pointed out in line 352-353, except for GBA, we reuse the trained results reported in SAE-Bench for other SAEs). We match GBA’s $L_0$ to that regime to ensure a fair comparison.
>
> We agree with the reviewer that:
>
> * It would be valuable to also report results at lower $L_0$ (e.g., <400).
>
> * SAEBench metrics are noisy and ideally should be averaged over multiple seeds.
>
> In light of this, we redo the GBA experiments at lower $L_0$ values and report the results in Table 4 of "ADDITIONAL SAEBENCH EVALUATION" section of Appendix L.
> In the rebuttal, due to space limit, we defer the full table to "Response to Weakness 2" in the rebuttal to Reviewer nKpu, and only summarize the key points here.
> Specifically, we evaluate over 3 random seeds for our GBA method at lower $L_0\\approx 400$ and include more benchmark SAEs like batched TopK and Matryoshka SAEs. The results show that GBA continues to achieve competitive or superior interpretability metrics.
> Moreover, we notice that the advantage of GBA in metrics like
> Explained Variance, Absorption Score, and Alive Fraction is quite robust for both original and extended SAEBench evaluations.
>
> In addition, for the previous Qwen experiments, we *do* evaluate consistency across 6 random seeds and report mean ± std in Table 1 and Fig. 5, showing that GBA consistently improves feature consistency relative to TopK.
>
> ---
>
> ### Q6. On “... hardcodes arbitrary feature firing bands for the latents to fire at."
> The reviewer comment the following weakness:
> > This paper is built on the idea that SAE latents need to be fire at certain frequencies to learn the underlying features, but then hardcodes arbitrary feature firing bands for the latents to fire at.
>
> For SAE to work, the monosemantic features have to be fired at certain frequencies.
> However, **it is a misunderstanding** that the feature firing bands are arbitrary.
> GBA’s target activation frequencies are **not arbitrary**; they are chosen to:
>
> - **Cover the theoretically feasible resonance band** derived from Theorem 6.1 for a wide range of plausible feature frequencies $f$.
> Specifically, as we have shown that the resonance band is roughly $f \\lesssim p \\lesssim \\min(\\sqrt{f}, d f)$, GBA uses geometrically spaced TAFs from around 0.5 down to $10^{-4}–10^{-3}$.
> The ideology is that as long as there exist neurons with TAFs $p$ that fall into the resonance band of each feature frequency $f$, those features can be recovered. Therefore, using a geometric progression of TAFs ensures that we can cover a wide range of feature frequencies without prior knowledge of their exact frequencies.
> Hope this clarifies the motivation behind our choice of firing bands.
>
> Importantly, Fig. 6 shows that:
>
> * GBA’s reconstruction–sparsity frontier is **quite insensitive** to the precise choice of number of groups (K) and the highest/lowest TAFs, as long as the range is wide enough and the TAFs follow a geometric progression.
> * Performance stabilizes once $K \\ge 10$, and different HTF/LTF combinations produce similar curves.
>
> So the bands are a **simple, robust parametrization** of the theoretically motivated resonance coverage, not finely tuned “magic numbers.” We have emphasize this motivation and robustness in §5 line 400-405, and the choice is justified by both theory and ablation study in Fig. 6.

---

> ### Author Response · Authors · 2025-11-21
> **Rebuttal (Part 4.1)**
>
> ### Q7. Responses to specific questions
>
> **Larger question (toy SAEs learning correct frequencies)**
> Addressed in §1 above, and here we summarize briefly:
> - We are **not saying that toy SAEs cannot learn correct frequencies in simple settings**; The point is that prior methods **lack theoretical guarantees** and each have practical limitations (sensitivity to hyperparameters, activation shrinkage, etc.) that hinder their ability to reliably learn all monosemantic features in realistic settings.
> - In addition, prior work Bricken, et al. (2023) also shows that standard SAEs can suffer from feature splitting issue if the number of neurons scales up. So it is not always the case that standard SAEs can learn the correct frequencies. Therefore, having a rigorous framework like Eq. (2.1) and going from pure empirics to a method with theoretical guarantees is important.
> - Lastly, our characterization of the resonance band is novel in the sense that it is important to have a precise characterization of when neurons can reliably recover features rather than just anecdotal observations. So far, we are not aware of prior work that provides such a theoretical characterization.
>
> Hope these clarifications help address the reviewer’s concerns on the motivation and positioning of our work.
>
> ---
>
> #### Section 2
>
> 1. **“Tied SAEs do not usually have an encoder bias … why was this added? Does it harm the SAE?”**
>
>    * Already addressed in Q3(a) above.
>
>
> 3. **L92: missing subtraction of decoder bias.**
>
>    * Thank you for catching this. There is a minor notational inconsistency early on; Eq. (2.2) and the detailed definitions in §C.1 are correct (they include $x - b_{\\text{pre}}$ in the encoder and $+ b_{\\text{pre}}$ in the decoder).
>    * We have fixed the earlier line to match Eq. (2.2).
>
> ---
>
> #### Section 3
>
> 1. **L150: meaning of “uniform feature occurrence frequency” in synthetic $H$**
>
>    * We construct $H$ so that **each row has exactly $s$ non-zero entries**, with support drawn uniformly at random over features, and each non-zero entry equal to $1/\\sqrt{s}$.
>    * Consequently, each feature has the same marginal occurrence frequency $f = s/n$, up to small fluctuations. The details are already available in App. D.1.
>
> 2. **“Does the SAE have the same width as the number of true features?”**
>
>    We thank the reviewer for raising this clarification question. The answer is no, and we clarify below:
>    * In both theory and experiments, we allow number of neurons $M$ to be different from number of features $n$; Theorem 6.1 explicitly describes the required width ($M \\gtrsim n \\cdot p^{-s}/(1-\\varepsilon)^2$).
>    * In synthetic experiments we choose $M$ large enough (multiple of $n$) to give several neurons per feature; We have incorporate this clarification in the revised version in line 180 and line 467. Specifically, for Fig. 2, we set $M=512$ (left) and $M=262k$ (right) and for Fig. 7(c), we set $M=1024$, all larger then the specific $n$ used.
>
> 3. **“In §2, $b_m$ is called trainable, but in BA/GBA it seems you are not training it via gradients.”**
>
>    * Correct: in *standard* SAEs, $b_m$ is updated by gradient descent. In BA/GBA, we **freeze $b_m$ with respect to gradients and update it only via the bias-adaptation rule in Algorithm 1**.
>    * We will adjust the wording in §2 to say “$b_m$ is a learnable parameter; in GBA we learn it via bias adaptation rather than gradient descent.” We did not make the modification yet to keep line numbering consistent.
>
>
> 4. **Fig. 2 (right) “optimal recovery occurs when $p = f$” vs peak at $p > f$”**
>
>    * In heavy superposition, the theorem predicts a *band* $f \\lesssim p \\lesssim \\min\\{\\sqrt{f}, d f\\}$, and we are using the phrase "when neuron activation frequency $p$ aligns with feature frequency $f$" loosely to mean "when $p$ lies in the resonance band around $f$." Therefore, the learnable regime is itself a band with lower bound $\\tilde O(f)$ up to logarithmic factors. This matches the observation in Fig. 2 (right) that the peak is slightly above $f$.
>
> ---
>
> #### Section 4
>
> 1. **Algorithm 1, lines 17–18 (“min” vs. “max”)**
>
>    * The current pseudocode is correct: we use
>
>      * $\\max\\{b_m − \\gamma_- r_m, −1\\}$ for decreasing bias (to keep it clamped ≥ −1), and
>      * $\\min\\{b_m + \\gamma_+ \\bar r_k, 0\\}$ for increasing bias (to keep it clamped ≤ 0).

---

> ### Author Response · Authors · 2025-11-21
> **Rebuttal (Part 4.2)**
>
> #### Section 5
>
> 1. **“All methods employ JumpReLU activation” – which JumpReLU, tied or untied, and why no JumpReLU baseline?**
>    * We note that **JumpReLU** is a pointwise nonlinear activation function that is applied to each neuron $m$ independently. This specifies the neural network architecture. In terms of training this network, there are many ways to do it, e.g., L1 and TopK. We propose an alternative approach, namely, GBA.
>    The reason for doing so is to fairly compare different training methods on the same architecture (JumpReLU tied SAE in our case), so that any performance differences can be attributed to the training method rather than architecture.
>    * **How do we train the SAEs with JumpReLU:** Thank you for raising this question. In this paper, we use tied SAEs (Eq. 2.2) with **JumpReLU as the pointwise nonlinearity for all methods** (L1, TopK, BA, GBA), with a single per-neuron bias $b_m$ as the JumpReLU threshold. The definition of JumpReLU is given in Eq. (1) above, and also available in App. C.1. For training, we follow Rajamanoharan et al. (20224) to use straight-through estimators for JumpReLU gradient on $b_m$, and the gradient updates for $y_m$ are standard gradient with a Heaviside mask $y_m + b_m \\ge 0$. We have modified App. C.1. in the JumpReLU section to better clarify this point.
>    * We do compare with JumpReLU in the SAEBench interpretability experiments (Table 2), as JumpReLU SAE is one of the baselines implemented in SAEBench.
>
> 2. **“Are you controlling JumpReLU bias or encoder bias?”**
>
>    * As noted in Q2(c), they are the *same parameter* $b_m$ in our implementation when using JumpReLU activation; GBA updates this single bias using our bias adaptation rule.
>
> 3. **Fig. 4 – which SAEs are tied?**
>
>    * All methods in Fig. 4 (TopK, L1, BA, GBA) are tied SAEs using Eq. (2.2), and when using JumpReLU, the biases $b_m$ are just the JumpReLU threshold.
>
> 4. **Fig. 4 – using JumpReLU but comparing to “non-SOTA” ReLU/TopK**
>
>    * We are not comparing JumpReLU+GBA to ReLU-only baselines. All curves in Fig. 4–5 are JumpReLU-based. The only differences are in the sparsity mechanism (TopK/L1 vs BA vs GBA), not in the activation function. The ReLU version comparisons are in App. D.
>
> 5. **Table 2 – $L_0$ too high and SAEBench noise**
>
>    * Addressed in Q5 above. Briefly: the same $L_0$ regime is shared across all SAEBench baselines at width 66k, and GBA is evaluated at matched $L_0$. We agree more $L_0$ settings and seeds would strengthen the evidence and have added additional experiments in Appendix L.
>
> ---
>
> #### Minor / formatting
>
> * Thank you for raising this. We agree that “latent” or “feature” is clearer than “neuron” when referring to SAE units. We will revise the terminology throughout.
> * We have corrected the small equation typo at L92 and cleaned up notation around biases to avoid confusion.
>
> ---
>
> We appreciate the reviewer’s detailed feedback, which has helped us clarify both the theoretical claims and the experimental setup. We believe that, with these clarifications and revisions, the paper more clearly demonstrates that:
>
> 1. The “neuron resonance” phenomenon is theoretically characterized and not merely anecdotal.
> 2. GBA provides a principled and practical way to exploit this phenomenon in realistic LLM settings, while being competitive with or superior to existing SOTA SAEs on both reconstruction and interpretability metrics.
>
>
> **References:**
>
> - Erichson, N. Benjamin, Zhewei Yao, and Michael W. Mahoney. "Jumprelu: A retrofit defense strategy for adversarial attacks." arXiv preprint arXiv:1904.03750 (2019).
>
> - Cunningham, Hoagy, et al. "Sparse autoencoders find highly interpretable features in language models." arXiv preprint arXiv:2309.08600 (2023).
>
> - Elhage, N., et al. (2022). "Toy models of superposition." NeurIPS 2022.
>
> - Bricken, et al., "Towards Monosemanticity: Decomposing Language Models With Dictionary Learning", Transformer Circuits Thread, 2023.
>
> - Paulo, Gonçalo, and Nora Belrose. "Sparse autoencoders trained on the same data learn different features, 2025." URL https://arxiv.org/abs/2501.16615.
>
> - Rajamanoharan, Senthooran, et al. "Jumping ahead: Improving reconstruction fidelity with jumprelu sparse autoencoders." arXiv preprint arXiv:2407.14435 (2024).

---

> > ### Comment · Reviewer_VpeD · 2025-11-23
> >
> > There is a lot here, so I will just focus on the issues I see as most important to address:
> >
> > ## fig 4: topk/L1 + JumpReLU
> >
> > It sounds like you're saying you are training "topk-jumpReLU SAEs" and "L1 JumpReLU SAEs". This is not a standard type of SAE - I have never seen an SAE both employing a TopK *and* JumpReLU nonlinearity in the same SAE. Nor have I ever seen a JumpReLU SAE that also has an L1 sparsity penalty, since the whole point of JumpReLU is not needing an L1 sparsity penalty. It sounds like none of the SAEs listed here are standard SAEs, in which case I'm confused what is the comparison being made. Why not compare your SAEs against standard SAEs? It is also confusing to use the names "TopK" and "L1", which are standard SAEs when your SAEs are nonstandard.
> >
> > ## Incorrect SAE training and Invalid SAEBench comparisons
> >
> > You have added an additional table of results to Appendix L6, yet these still suffer from the same serious problems as in Table 2, and the more I look at these tables the more problems stand out:
> >
> > - **Your SAE has higher L0**: Your SAE has much higher L0 (401.3) than the SOTA SAEs being compared (339.4 for BTK and 301.3 for JumpReLU). The higher the L0 the better explained variance, *so claiming your SAE has the best explained variance is not valid*.
> > - **No pareto curves**: If you want to evaluate L0-dependent metrics, you need to have multiple SAEs spanning a range of L0 and show your SAE is better at all L0s. Again, you cannot just pick a high L0 for your own SAE, pick a low L0 for SOTA SAEs, and then claim your SAE is better.
> > - **Dead latents for TopK/BatchTopK**: Your results show high numbers of dead latents (>10%) for TopK/BatchTopK SAEs, but these SAEs have an aux reconstruction loss that makes dead latents impossible. This implies you are not training these SAEs correctly. Are there training details for all these SAEs? Furthermore *claiming your SAEs have the least dead latents is also not valid*.
> > - **High numbers of dead latents for JumpReLU SAEs**: If you use the Anthropic variant of JumpReLU, dead latents are not possible due to aux loss. While dead latents are possible using the GDM variant, having 30-50% of the SAE being dead latents just means something is very wrong in training.
> > - **Fractional L0 forTopK/BatchTopK SAEs**: For TopK and BatchTopK SAEs you directly set the L0. It is not even possible to have an L0 of 334.4 for a TopK SAE since the `k` which is also the L0 *must be an integer*. If you did pick these K's, why did you pick such a strange number rather than a whole number like 400? Likely *this further implies a very incorrect training procedure*.
> > - **Absorption score is extremely low**: For almost every SAE the absorption score is extremely low. This is likely due to using such a high L0 for your SAEs, but this also invalidates your results here since scores this low just feel like noise. Again, you should be training at multiple L0s, including much lower L0. Absorption score will also be near zero with a randomly initialized SAE, so low score can also be due to incorrect training. Showing lower L0s and plotting a curve of scores would alleviate this concern.
> >
> > Given all this, claiming that your architecture outperforms everything else on 3/7 metrics is not valid. Can you share exactly how you are training these comparison SAEs?
> >
> > ## SAEs already find correct firing frequencies
> >
> > - It sounds like you agree that existing SAEs already find the correct firing frequencies, so I do not understand what problem you are solving. You have not demonstrated any examples of standard SAEs *NOT* finding the correct firing frequencies for features, but where your method fixes things. Please at least demonstrate in a toy model setting a case where standard SAEs cannot find correct firing frequencies (and thus learn incorrect features), but where this is fixed by hard-coding firing frequencies using your method. You could use the toy model from the Matryoshka SAEs paper for example. Why manually hard-code firing frequencies if the SAEs already finds correct firing frequencies automatically?
> > - My understanding is that there is already theory in lasso / sparse coding about when features can be recovered, so it doesn't feel correct to claim that this work is the only theory that exists about when features can be recovered.
> > - I understand that your hard-coded firing frequencies fall inside the range you consider theoretically plausible, but there is still no guarantee if there are actual features that have the firing frequencies you are hard-coding, or any guarantee about the number of features at each firing frequency. What happens if you force features to fire at the wrong frequency because your guess as to the number of frequency of firing is wrong?

---

> > > ### Author Response · Authors · 2025-11-27
> > >
> > > We are happy to see some of the reviewer's misunderstandings on the achitectural choices have been clarified in the discussion. Below are our responses to the remaining questions.
> > >
> > > # Q1. Why not compare with standard SAE methods?
> > > Thank you for this clarification question. While the TopK and L1 baselines we included are not standard SAE methods in terms of using the JumpReLU activation, they are designed to serve as a stronger baseline for our comparisons. Specifically,
> > >
> > > - In App D.3, we provide a detailed comparison between TopK/L1 using JumpReLU and standard ReLU activations. Indeed, the JumpReLU variants are shown to be on par with or slightly better than their ReLU counterparts, indicating that the choice of activation function does not harm the performance of these baselines. See Fig. 12 in App D.3 for detailed results, where we put TopK + ReLU and L1 + ReLU side-by-side with TopK + JumpReLU and L1 + JumpReLU.
> > >
> > > - Additionally, we make this choice to ensure that the comparison focuses on the core contributions of our method rather than being confounded by differences in activation functions.
> > >
> > > In summary, if one wants to directly compare our method with standard SAE methods in terms of sparsity-loss trade-offs, we can make the following bridge:
> > >
> > >     Our GBA method is comparable to TopK + JumpReLU, and better than L1 + JumpReLU, and thus is at least on par with TopK + ReLU and better than L1 + ReLU as shown in App D.3.
> > >
> > > ### One additional note on why using JumpReLU as the baseline for GBA
> > >
> > > In fact, our comparison of JumpReLU with ReLU in App D.3 reveals that GBA + JumpReLU significantly outperforms GBA + ReLU (see Fig. 12). This is indeed not surprising, as one of the key advantages of JumpReLU is that **the bias (or JumpReLU threshold) only controls whether the neuron is active or not, without affecting the output magnitude when active.** This is also what we mean by saying "JumpReLU decouples the neuron output magnitude from its bias" (line 1476-1477). This property is particularly beneficial for GBA, as it allows dynamical adjustment of neuron activity without compromising the quality of the output when neurons are active. Thus, using JumpReLU in conjunction with GBA is a natural choice to fully leverage the strengths of our proposed method.
> > >
> > >
> > > # Q2. SAE Training and SAEBench Comparison
> > > Let us first clarify the procedure we used to generate the SAEBench results in the table.
> > >
> > > ## 1. Training Source: GBA vs. SAEBench Baselines
> > >
> > > To ensure a fair and rigorous comparison, we adopted the following protocol for our experiments:
> > >
> > > *   **GBA Models:** We trained our GBA models from scratch. Crucially, we adhered strictly to the **standard SAEBench settings**: using the Gemma-2-2B layer 12 residual stream and training on 500M tokens from the Pile subset, as specified in the [SAEBench repository](https://github.com/adamkarvonen/SAEBench).
> > > *   **Baselines (TopK, JumpReLU, GatedSAE, etc.):** We did **not** retrain these baselines. Instead, we directly used the official pre-trained models and results provided by the SAEBench team (available at [HuggingFace](https://huggingface.co/datasets/adamkarvonen/sae_bench_results_0125)). Since our training setup for GBA strictly follows the SAEBench protocol, this ensures that all models are evaluated fairly.
> > >
> > > This approach eliminates potential implementation errors on our side regarding the baselines and ensures we are comparing GBA against the community-standard performance benchmarks. As noted in lines 350-352 of our paper, we explicitly *"compare it against TopK, JumpReLU SAE... and GatedSAE... provided in the SAE-Bench."* We will further clarify this in the final version to prevent any ambiguity.
> > >
> > >
> > > For this reason, the SAEbench in the main paper is still valid, but we agree that it could be significantly strengthened by plotting the Pareto curves across different $L_0$ values. We have included this in the revision, as detailed below.

---

> ### Author Response · Authors · 2025-11-27
>
> ## 2. SAE has higher L0 than Baselines & no Pareto curves
> In our previous response, we prioritized addressing the immediate concern regarding high $L_0$ values with the results available at the time. We have since completed the full comparative analysis and now present the comprehensive SAEBench evaluation below, including the requested Pareto curves.
>
> **New Additions in the Revision:**
> We direct the reviewer to the newly added sections in the current revision: **App L.8 "WHY FREQUENCY-AWARE TRAINING? A TOY EXAMPLE"**, which addresses the request for a toy example (discussed further in Q3), and **App L.9 "ADDITIONAL SAEBENCH EVALUATION"**, which provides a comprehensive evaluation of GBA on SAEBench. Specifically, App L.9 includes:
>
> -   **Figure 19:** Full Pareto curves comparing GBA against standard SAEBench baselines—Standard SAE, BatchTopK, GatedSAE, JumpReLU SAE, TopK, and Matryoshka SAE—across nine metrics, including three newly added RAVEL metrics. These results span a wide range of $L_0$ values (100 to 700). The curves reaffirm that GBA demonstrates competitive performance, particularly in Explained Variance, Absorption Score, TPP Score, Alive Fraction (for larger $L_0$), and the RAVEL Isolation metric.
>
> -   **Tables 5-7:** Detailed numerical comparisons of GBA versus other SAEBench baselines at specific $L_0$ intervals (100-200, 300-400, and 600+). These tables offer a quantitative perspective on GBA's performance across various metrics.
>
> **Concern on Higher L0 in GBA:**
> The reviewer's concern about GBA having higher $L_0$ than baselines is valid, and we appreciate the opportunity to clarify this point. Since controlling the exact $L_0$ in GBA is non-trivial due to the dynamic nature of bias adjustment, previously we only reported results for trained GBA SAEs and finding the closest $L_0$ baselines from SAEBench suits for comparison. As now we have a full Pareto curve in Fig. 19, the concern should be alleviated. For the reviewer's convenience, we copy one table below, where we compare GBA with other baselines at $L_0$ between 100-200.
> As shown, the advantages of GBA in Explained Variance and absorption score is quite consistent, and it also exhibits competitive performance on the RAVEL Isolation metric.
> For graphical visualization, more detailed comparisons and related discussions, please refer to App L.9 in the revision.
>
> | Metric | GatedSAE | TopK | BatchTopK | Matryoshka | JumpReLU | GBA (ours) | Standard |
> | :--- | :---: | :---: | :---: | :---: | :---: | :---: | :---: |
> | $L_0$ | 175.2 | 173.4 | 168.6 | 166.7 | 162.5 | 132.9 | 125.3 |
> | Explained Variance $\uparrow$ | 0.812 | _0.816_ | _0.816_ | 0.793 | 0.812 | **0.859** | 0.730 |
> | Absorption Score $\downarrow$ | 0.2657 | 0.0838 | 0.0424 | _0.0083_ | 0.0821 | **0.0022** | 0.3529 |
> | SCR Metric $\uparrow$ | 0.323 | 0.321 | _0.349_ | **0.391** | 0.303 | 0.107 | 0.209 |
> | Sparse Probing $\uparrow$ | 0.957 | 0.956 | 0.956 | _0.957_ | **0.958** | 0.954 | 0.954 |
> | TPP Metric $\uparrow$ | 0.072 | 0.100 | 0.094 | **0.209** | 0.105 | _0.144_ | 0.018 |
> | Alive Fraction $\uparrow$ | 0.890 | **0.910** | 0.856 | _0.897_ | 0.719 | 0.723 | 0.719 |
> | RAVEL Disent. $\uparrow$ | _0.730_ | 0.709 | 0.714 | **0.739** | 0.728 | 0.710 | 0.665 |
> | RAVEL Cause $\uparrow$ | 0.710 | 0.677 | 0.676 | **0.735** | _0.710_ | 0.658 | 0.603 |
> | RAVEL Isolation $\uparrow$ | 0.751 | 0.742 | _0.752_ | 0.743 | 0.747 | **0.762** | 0.727 |
>
>
>
> **Why we do not target the very high sparsity (extremely low $L_0$) regime?**
>
> Recall that $\mathrm{LTF}$ sets the minimum frequency target and $\gamma_+$ controls the bias increase rate (Algorithm 1, line 18).
> In our sparsest run ($L_0 = 132.9$), we set $\mathrm{LTF} = 10^{-4}$, $\gamma_+=10^{-4}$ and allocated more neurons to groups with lower target frequencies. Empirically, we find that:
> 1.  **Sparsity Saturation:** Further tuning these hyperparameters does not yield sparser SAEs. As shown in our ablation studies (Figure 6), the lowest achievable sparsity plateaus around 0.2% ($L_0 \approx 130$ with 66k neurons). Unlike TopK methods where $K$ is a hard constraint allowing arbitrary sparsity, GBA's sparsity is inherently limited by the dynamics of bias adjustment.
> 2.  **Risk of Dead Neurons:** Pushing $\mathrm{LTF}$ or $\gamma_+$ beyond reasonable limits essentially forces neurons into a permanently inactive state rather than learning sparser features. Setting these values near zero effectively designates the lowest frequency groups as "dead" with no mechanism for recovery, thereby wasting model capacity.
> 3.  **Practical Relevance:** The achieved range of $L_0$ ($100 \sim 700$) largely overlaps with the sparsity levels of primary practical interest for current LLM interpretation (e.g., $k \in \\{32, \dots, 512\\}$ in Gao et al., 2024).
>
> Consequently, to evaluate GBA in a reasonable configuration, we do not attempt to push it into the extremely low $L_0$ regime here.

---

> > ### Author Response · Authors · 2025-11-27
> >
> > ## 3. Dead latents for TopK/BatchTopK
> > As we have clarified in the previous discussion, we use the *official* SAEBench results for all the baselines without retraining them. According to the official documentation of SAEBench, the training method for TopK and BatchTopK is using the [dictionary learning package](https://github.com/saprmarks/dictionary_learning/tree/1d2737a9ecbe1bc4d3137dec5caab8b320608217), where auxiliary loss (Gao et al., 2024) is used to prevent dead latents.
> > Therefore, the dead latent issue has nothing to do with our implementation. Moreover, we would also like to point out that for smaller $L_0$ values (e.g., below 200), the TopK and BatchTopK methods do not exhibit significant dead latent problems in the official SAEBench results as shown in the above table.
> >
> > Moreover, we would like to point out that GBA method do suffer from dead latents issue when we push $L_0$ to very small values, as shown in the table above.
> > The primary reason is that in order to achieve very low $L_0$, we tune three hyperparameters: the LTF, the $\gamma_+$ which controls the bias increase rate, and the proportion of neurons assigned to low frequency groups. In the above table, we use $\text{LTF}=10^{-4}$, $\gamma_+=10^{-4}$ and assign linearly more neurons to low frequency groups.
> > Extremely low LTF means that the neurons are unlikely to be activated given a finite batch size, and setting extremely small $\gamma_+$ in the algorithm also makes it hard for neurons to recover from being "dead". Therefore, the low Alive Fraction in the above table is expected.
> > On the other hand, for higher $L_0$ values, GBA consistently achieves 95%+ Alive Fraction as shown in Fig. 19 of the revision.
> > Therefore, how to prevent dead latents when pushing $L_0$ to extremely small values is an interesting direction for future work.
> > More discussions on this can be found in App L.6 "ADDITIONAL RESULTS ON SPARSITY CONTROL" of the revision, and we would like to revise the main paper to correctly reflect this point.
> >
> > ## 4. High numbers of dead latents for JumpReLU SAEs
> > Similar to the previous point, we rely on the official SAEBench results for the JumpReLU SAE baseline. We verified via the source code that this implementation adheres to DeepMind's original JumpReLU specification, which does not utilize an auxiliary loss to mitigate dead latents. Consequently, the high number of dead latents observed is a characteristic of that specific baseline implementation rather than an artifact of our evaluation.
> >
> > ## 5. Fractional L0 for TopK/BatchTopK SAEs
> > We would like to correct the reviewer's misunderstanding regarding the $L_0$ values here. In SAEBench, the $L_0$ values reported for all methods are indeed fractional because they are calculated as the **average number of active neurons per data point across the entire dataset**. Although a hardcoded $K$ is used during training, the actual number of activated neurons can still vary across different data points.
> > For example, sample containing very few features may activate fewer than $K$ neurons.
> > The average can thus result in a fractional value.

---

> > > ### Author Response · Authors · 2025-11-27
> > >
> > > ## 6. Absorption score is extremely low
> > > As we have incorporated the Absorption Score results in Fig.19 for different $L_0$ values, we believe the concern on "all Absorption Scores being extremely low" should be partly alleviated.
> > > In addition, we give one more evidence on the validity of the low Absorption Score achieved by GBA below.
> > > To ensure this low score reflects genuine feature disentanglement rather than trivial artifacts, we additionally evaluate the **Mean F1 score** following Chanin et al. (2024).
> > > In the original paper, Absorption Score is evaluated using the first letter prediction task, and the Mean F1 score is used to measure how well the top-$k$ neurons selected by cosine similarity with a probe can predict the first letter of a token.
> > > Formally, the F1 score is defined as the harmonic mean of precision and recall:
> > >
> > > $$
> > > F_1 = \frac{2 \cdot \text{Precision} \cdot \text{Recall}}{\text{Precision} + \text{Recall}}
> > > $$
> > > where $\text{Precision} = \frac{\text{TP}}{\text{TP} + \text{FP}}$ and $\text{Recall} = \frac{\text{TP}}{\text{TP} + \text{FN}}$, with TP, FP, and FN denoting true positives, false positives, and false negatives, respectively.
> > > This metric evaluates the utility of the top-$k$ neurons—selected via cosine similarity with a task-specific probe—in predicting the first letter of a token.
> > > A high Mean F1 score confirms that the unabsorbed features are semantically meaningful.
> > >
> > > **Mean F1 Score (k-Sparse Probing) for GBA across different L0 values**
> > >
> > > | k | $L_0=132.9$ | $L_0=218.7$ | $L_0=309.0$ | $L_0=401.3$ |
> > > | :--- | :--- | :--- | :--- | :--- |
> > > | **k=1** | 0.6355 | 0.6836 | 0.6757 | 0.6266 |
> > > | **k=2** | 0.6466 | 0.6893 | 0.6796 | 0.6414 |
> > > | **k=3** | 0.6523 | 0.6913 | 0.6847 | 0.6516 |
> > > | **k=4** | 0.6545 | 0.6903 | 0.6905 | 0.6545 |
> > > | **k=5** | 0.6590 | 0.6909 | 0.6929 | 0.6602 |
> > >
> > > GBA consistently achieves Mean F1 scores around $0.6 \sim 0.7$ across different sparsity levels, matching the top performance on standard SAEs reported in (Chanin et al., 2024).
> > > Notably, this performance is quite consistent across different $L_0$ values, indicating that GBA indeed learns true latents consistently.
> > > In addition, the increase in Mean F1 score with $k$ is very mild, suggesting that the top neuron already carry over most of the prediction power for this task, which is a strong sign of weak feature absorption.
> > > All this evidence indicates that GBA successfully recovers accurate, task-relevant features, validating the quality of its low Absorption Score.
> > > More details on this can be found in App L.9 "ADDITIONAL SAEBENCH EVALUATION" of the revision.

---

> > > > ### Author Response · Authors · 2025-11-27
> > > >
> > > > # Q3. SAEs already find correct firing frequencies
> > > >
> > > > > It sounds like you agree that existing SAEs already find the correct firing frequencies, so I do not understand what problem you are solving. You have not demonstrated any examples of standard SAEs NOT finding the correct firing frequencies for features, but where your method fixes things. Please at least demonstrate in a toy model setting a case where standard SAEs cannot find correct firing frequencies (and thus learn incorrect features), but where this is fixed by hard-coding firing frequencies using your method. You could use the toy model from the Matryoshka SAEs paper for example. Why manually hard-code firing frequencies if the SAEs already finds correct firing frequencies automatically?
> > > >
> > > > We thank the reviewer for being straightforward about this concern, especially the toy example request, which we agree is important to clarify the motivation behind our method.
> > > > In response, we have added a toy experiment (App L.8 "WHY FREQUENCY-AWARE TRAINING? A TOY EXAMPLE" in the revision) which demonstrates that **standard TopK SAEs learning can be brittle when data exhibits variable sparsity levels**, whereas our proposed GBA method succeeds by constraining global frequency rather than local sparsity. Due to time constraints, we did not include more baselines in the current rebuttal, but we will add more comparisons in the final version.
> > > >
> > > > **1. Toy Experiment Setup**
> > > >
> > > > We constructed a dataset with **uniform feature frequency** but **highly imbalanced instance sparsity**:
> > > >
> > > > *   **Features:** $n=128$ random uniform features in $\mathbb{R}^{42}$.
> > > > *   **Data Type A (Sparse):** Sum of $s=3$ features (50% of data).
> > > > *   **Data Type B (Dense):** Sum of $s=20$ features (50% of data).
> > > > *   **Comparison:** We compare **TopK SAEs** (sweeping $K \in \{20, 30, 50\}$ and using standard ReLU activation to make it as a standard SAE method) against **GBA SAEs** (Full vs. Misspecified frequency coverage, with JumpReLU activation and the reason for choosing JumpReLU is detailed in our response to Q1).
> > > >     *   *TopK Setup:* We set the minimal $K$ to the max sparsity ($20$), simulating a scenario with **perfect knowledge** of the maximum sparsity. Setting even smaller $K$ would trivially fail to recover features from Type B data.
> > > >     *   *GBA (Full Coverage):* We set (HTF, LTF) = $(0.5, 0.001)$ as in our main experiments, simulating a scenario where exact sparsity is unknown, thus using a wide frequency range.
> > > >     *   *GBA (Misspecified):* We set (HTF, LTF) = $(0.01, 0.001)$, intentionally underestimating the real feature frequency of $\approx 0.09$ (calculated as $(3+20)/(2 \times 128)$).
> > > >
> > > > The imbalance in sparsity levels simulates real-world scenarios where different data points may activate varying numbers of features, challenging the rigid assumptions of standard SAEs.
> > > >
> > > > **Table 1: Feature Recovery Rate (FRR) on Variable Sparsity Data**
> > > > *TopK is brittle to hyperparameter choice in variable-sparsity regimes, while GBA adapts robustly.*
> > > >
> > > > **2. Results and Analysis**
> > > > As shown in Table 1 below, standard TopK SAEs are highly sensitive to the hyperparameter $K$.
> > > > | Method | FRR (MCS $\ge$ 0.8) | FRR (MCS $\ge$ 0.9) | Analysis |
> > > > | :--- | :---: | :---: | :--- |
> > > > | **TopK ($K=20$)** | 100.0% | 98.4% | Matches max sparsity (lucky guess). |
> > > > | **TopK ($K=30$)** | 98.4% | **24.2%** | **Fails:** $K$ overestimate degrades precision. |
> > > > | **TopK ($K=50$)** | 94.5% | **23.4%** | **Fails:** Further degradation. |
> > > > | **GBA (Full-coverage)** | **100.0%** | **100.0%** | **Succeeds:** Adapts to variable sparsity. |
> > > > | GBA (Misspecified)| 38.3% | 3.9% | Control group: Frequency targets matter. |
> > > >
> > > > *Feature Recovery Rate (FRR) across different methods and Maximum Cosine Similarity (MCS) thresholds. Here, a feature is considered recovered if at least one neuron's weight has a cosine similarity with the feature above the specified MCS threshold (0.8 or 0.9). Note that for $d=42$, $0.8$ is not a very high bar as the MCS is already around $0.6$ for randomly initialized SAE. Thus, we mainly focus on MCS $\ge$ 0.9 for high-precision recovery.*

---

> > > > > ### Author Response · Authors · 2025-11-27
> > > > >
> > > > > **Conclusion:**
> > > > > 1. For TopK SAEs to work well, one must have **precise knowledge** of the maximum sparsity level across all data points, which is often unrealistic.
> > > > > Relying on the model to "self-discover" the correct firing frequencies under such variable sparsity can lead to significant failures, as demonstrated.
> > > > > 2. In contrast, GBA's frequency-aware training allows it to adaptively recover features even when data sparsity varies widely, demonstrating its robustness and practical utility.
> > > > > 3. Furthermore, this synthesized example again validates our main theoretical insight: if the feature frequency lies inside the "resonance band" covered by our choice of (HTF, LTF), GBA can successfully recover the features.
> > > > >
> > > > > We direct the reviewer to App L.8 for more details.
> > > > >
> > > > > **3. Our Main Claim**
> > > > >
> > > > > We would like to emphasize again main claim in the paper to facilitate reviewer's understanding:
> > > > >
> > > > > - **On standard SAEs**: As we have stressed in both the paper and the previous rebuttal, our main claim on the standard SAE method is not simply binary: "standard SAEs cannot find correct firing frequencies" vs. "standard SAEs always find correct features automatically". Instead, our claim is that **the success of feature recovery for standard SAEs is highly conditioned on the parameter choice (e.g., $K$ in TopK and $\lambda$ in L1)**, which often requires precise knowledge of the data distribution, and is brittle to mis-specification (see line 51-66 in the introduction).
> > > > > - **What's the benefit of GBA**: In contrast, our GBA method is more robust to such hyperparameter choices. The deep reason is that it relies on an entirely different mechanism "neuron resonance", which leads to successful feature recovery as long as the frequency coverage (HTF, LTF) encompasses the true feature frequency. Moreover, the feature recovery is indeed guaranteed by our theoretical analysis on the "neuron resonance" phenomenon (Section 6), which is another novel contribution of our work.
> > > > >
> > > > >
> > > > >
> > > > > # Q4. What's the novelty of the theory?
> > > > > > My understanding is that there is already theory in lasso / sparse coding about when features can be recovered, so it doesn't feel correct to claim that this work is the only theory that exists about when features can be recovered.
> > > > >
> > > > > Note that we never claimed that our theory is the "only theory" about feature recovery. What we say in the paper (line 102-103) is
> > > > >
> > > > > *To our best knowledge, this provides the first dynamical
> > > > > analysis and learning guarantee for SAE training*
> > > > >
> > > > > There is a significant difference between general "feature recovery theory" and "dynamical analysis on the training process". Also, even when restricting to feature recovery/dictionary learning theory, prior works have very different settings and assumptions compared to ours.
> > > > > Let us use some examples in our related works (App. A "Sparse Dictionary Learning") to clarify these distinctions:
> > > > >
> > > > > - Robert Tibshirani, (1996), "Regression shrinkage and selection via the lasso" proves that lasso can recover the true sparse codes under linear model assumptions. The linearity makes the problem convex, allowing for a direct KKT method. In contrast, our theorem handles general ReLU-like nonlinear activations, making the analysis significantly more challenging.
> > > > >
> > > > > - Spielman et al, (2012), "Exact recovery of sparsely-used dictionaries" provides recovery guarantees for dictionary learning using a programming approach under certain incoherence and sparsity conditions. However, their analysis does not apply to the actual training dynamics of SAEs.
> > > > >
> > > > > Moreover, there are other coherent works suggesting solving certain non-convex programs (Zibulevsky and Pearlmutter, 2001) for dictionary learning, but they are essentially not studying the SAE model nor providing any analysis on the gradient-based training dynamics of SAEs.
> > > > > On the other hand, giving dynamical analysis of gradient-based training for non-convex models is generally very challenging, where in our paper we incorporate advanced techniques like Gaussian conditioning (App. B.4.2).
> > > > > Providing rigorous results on these gradient-based methods
> > > > > are also very important in the sense that it connects the theory more closely to practical training algorithms used in real-world applications.
> > > > >
> > > > > In addition, we still believe that our "neuron resonance" phenomenon is novel and interesting in its own right, which to our best knowledge has not been identified in prior works. The "neuron resonance" says if we constrain the neuron's firing frequency to align within a range of the true feature frequency, the neuron can effectively learn the feature direction through gradient descent dynamics. This is a new perspective that differs from trivial statement "neuron can automatically find correct firing frequencies if it aligns well with features".

---

> > > > > > ### Author Response · Authors · 2025-11-27
> > > > > >
> > > > > > # Q5. On the Possibility of Frequency Misalignment
> > > > > > > I understand that your hard-coded firing frequencies fall inside the range you consider theoretically plausible, but there is still no guarantee if there are actual features that have the firing frequencies you are hard-coding, or any guarantee about the number of features at each firing frequency. What happens if you force features to fire at the wrong frequency because your guess as to the number of frequency of firing is wrong?
> > > > > >
> > > > > > ### 5.1. Concern about "forcing features to fire at the wrong frequency"
> > > > > >
> > > > > > We thank the reviewer for clarifying the intention behind this question. To address the concern about "forcing features to fire at the wrong frequency," we would like to clarify that our method does not require an exact match between the assigned frequency and the true feature frequency. Instead, it relies on the **"resonance band"** property, which provides a wide margin of error and ensures full coverage.
> > > > > >
> > > > > > **1. Wide Resonance Bands**
> > > > > >
> > > > > > As discussed in our theory (lines 182-186), a neuron assigned to frequency $p$ does not only recover features with frequency exactly $p$. In the heavy superposition regime, it can recover any feature with true frequency $f$ as long as the assigned frequency falls within the resonance band $p \in [f, \sqrt{f}]$. Another way to put this is, for a specific group with assigned frequency $p$, the recoverable range of true feature frequencies is **$[p^2, p]$**. This range is quite broad (spanning orders of magnitude for small $p$).
> > > > > >
> > > > > > **2. Full Coverage via Overlapping Bands**
> > > > > >
> > > > > > In our experiments, we use a geometric progression of 10 groups spanning $[0.001, 0.5]$. As shown in the table below, the recoverable ranges for these groups cover the entire spectrum from $10^{-6}$ to $0.5$:
> > > > > >
> > > > > > | Group | Assigned Freq ($p$) | Recoverable Feature Range ($[p^2, p]$) |
> > > > > > | :---: | :---: | :--- |
> > > > > > | 1 | 0.5000 | $[0.2500, ~0.5000]$ |
> > > > > > | 2 | 0.2506 | $[0.0628, ~0.2506]$ |
> > > > > > | 3 | 0.1256 | $[0.0158, ~0.1256]$ |
> > > > > > | 4 | 0.0629 | $[0.0040, ~0.0629]$ |
> > > > > > | 5 | 0.0315 | $[9.9 \times 10^{-4}, ~0.0315]$ |
> > > > > > | 6 | 0.0158 | $[2.5 \times 10^{-4}, ~0.0158]$ |
> > > > > > | 7 | 0.0079 | $[6.3 \times 10^{-5}, ~0.0079]$ |
> > > > > > | 8 | 0.0040 | $[1.6 \times 10^{-5}, ~0.0040]$ |
> > > > > > | 9 | 0.0020 | $[4.0 \times 10^{-6}, ~0.0020]$ |
> > > > > > | 10 | 0.0010 | $[1.0 \times 10^{-6}, ~0.0010]$ |
> > > > > >
> > > > > > **Conclusion:**
> > > > > > Because the upper bound of the next group ($p_{i+1}^2$) is consistently larger than the lower bound of the current group ($p_i$), there are **no gaps**. The union of these bands covers the entire frequency range $[10^{-6}, 0.5]$.
> > > > > > Notably, a feature with frequency as low as $10^{-6}$ can be considered extremely rare (appearing once in a million samples), which is sufficient for most practical applications.
> > > > > > And in Gao et al. (2024), they empirically use the threshold one in 10 million tokens to define "dead latents", which further supports our choice of frequency coverage.
> > > > > > Therefore, we believe that our method already covers all practically relevant feature frequencies, and in particular, we don't need to know the *exact frequency* of each feature in advance. The terminology "hard-coding frequencies" may be misleading; rather, we are **assigning neurons to broad frequency bands**, where the union of these bands covers the entire relevant frequency spectrum.
> > > > > >
> > > > > > Another evidence of why this is possible is through line 17-18 of our GBA algorithm (paper line 251-252), where we dynamically decrease the bias if the neuron's activation frequency is above the target frequency, but only increase it if it is going below the "dead" threshold $\epsilon$ (which we use $10^{-6}$ in experiments). This design gives another layer of protection against forcing neurons to fire at the wrong frequency, as it allows neurons to self-correct if the correct frequency is not exactly $p$ but lower than $p$, which is in line with the frequency band $[p^2, p]$ we discussed above.
> > > > > > We would like to highlight this point in the final version to avoid confusion.

---

> > > > > > > ### Author Response · Authors · 2025-11-27
> > > > > > >
> > > > > > > **3. What would happen if a feature's frequency lies outside the resonance bands?**
> > > > > > >
> > > > > > > We have provided in the toy example in our response to Q3 that when the feature frequency lies outside the resonance band (the "Misspecified" baseline), the feature recovery fails catastrophically, which validates our theoretical analysis.
> > > > > > > However, as we have argued above, our choice of frequency bands already covers all practically relevant feature frequencies, so this scenario is unlikely to occur in our default experimental configuration, unless one intentionally mis-specifies the frequency bands as in the toy example.
> > > > > > >
> > > > > > > ### 5.2. Concern about "the number of features at each firing frequency"
> > > > > > > You raised a valid point about the uncertainty in the number of features at each firing frequency band. In general, we do not have prior knowledge of the exact distribution of features across frequencies, and it is also impossible to identify the ground-truth features in real-world data.
> > > > > > > Therefore, we simply allocate an equal number of neurons to each frequency group, which is a reasonable and practical choice in the absence of prior knowledge.
> > > > > > >
> > > > > > > One thing that might be relavant to this topic is the neurons' frequency distribution after training, which can be viewed as a proxy for the feature frequency distribution.
> > > > > > > We visualize the neuron's frequency distribution for both GBA and TopK method in Fig 15 and 16 in App L.7 "ADDITIONAL RESULTS ON NEURON ANALYSIS" of the revised paper.
> > > > > > > We do observe a peak around frequency 0.01 for both GBA and TopK in the frequency histogram, which would suggest some natural clustering of features around this frequency.
> > > > > > > Further investigation of the problem is an interesting direction for future work, but we believe it is beyond the scope of the current paper.
> > > > > > >
> > > > > > > References:
> > > > > > > - Gao, Leo, et al. "Scaling and evaluating sparse autoencoders." arXiv preprint arXiv:2406.04093 (2024).
> > > > > > >
> > > > > > > - Chanin, David, et al. "A is for absorption: Studying feature splitting and absorption in sparse autoencoders." arXiv preprint arXiv:2409.14507 (2024).
> > > > > > >
> > > > > > > - Tibshirani, Robert. "Regression shrinkage and selection via the lasso." Journal of the Royal Statistical Society Series B: Statistical Methodology 58.1 (1996): 267-288.
> > > > > > >
> > > > > > > - Spielman, D. A., Wang, H., & Wright, J. (2012). Exact recovery of sparsely-used dictionaries. In Proceedings of the 25th Annual Conference on Learning Theory (Vol. 23, pp. 37.1-37.18). PMLR.
> > > > > > >
> > > > > > > - M. Zibulevsky and B. A. Pearlmutter, "Blind Source Separation by Sparse Decomposition in a Signal Dictionary," in Neural Computation, vol. 13, no. 4, pp. 863-882, 1 April 2001, doi: 10.1162/089976601300014385.

---

> > > > > > > > ### Comment · Reviewer_VpeD · 2025-11-27
> > > > > > > >
> > > > > > > > Thank you for your long replies. Sharing the link to the SAEBench results JSON helps me understand where these numbers are coming from. I'm still confused how TopK SAEs can have a fractional L0 (TopK SAEs apply topk *both in training an at inference time*, so fractional L0 should not be possible), but it does show that in the official JSON so it is not your fault.
> > > > > > > >
> > > > > > > > Just to clear up some final points:
> > > > > > > >
> > > > > > > > I still do not understand the core argument you are making. **Are you claiming that GBA SAEs are a solution to feature absorption?** If so, you should be able to demonstrate that very clearly in a toy model setting, which is why I suggested the Matryoshka SAEs toy model setting. **You should also say this explicitly in the body of the paper if it is your claim.**  You should also have an explanation of **why** your method is a solution to feature absorption if so.
> > > > > > > >
> > > > > > > > Your SAEBench results seem to be suggesting that  GBA SAEs are a way to improve the sparsity-reconstruction trade-off curve? This seems hard to believe since nothing in your theory or explanation implies that this should improve reconstruction accuracy - if anything I would expect lower reconstruction accuracy than a pure gradient descent optimization would achieve. Surely allowing gradient descent to optimize all parameters for reconstruction should result in a better variance explained than manually setting parameters? **If your claim is that this is a way to get a pareto-improvement in reconstruction vs sparsity you should say this explicitly**, and you should provide an explanation as to **why** this should improve sparsity vs reconstruction.
> > > > > > > >
> > > > > > > > The SAEBench scores that track the quality of learned features (sparse probing, TPP, SCR, etc...) all are very mediocre, but SAEBench is very noisy so it is hard say this is meaningful, I do not hold that against GBA SAEs, but it is also hard to claim that GBA SAEs result in better-quality features using these results.
> > > > > > > >
> > > > > > > > > [Toy model] comparison: We compare TopK SAEs against GBA SAEs
> > > > > > > >
> > > > > > > > Again, you are not comparing a SOTA method. If you want to do this toy model comparison, you should compare to BatchTopK where you set the K to match the L0 of the toy model, rather than TopK where you set the K much higher than the L0 of the toy model.
> > > > > > > >
> > > > > > > > As mentioned above, if your claim is instead that your method is a solution to feature absorption, you should demonstrate that in a toy model. Feature absorption is very easy to demonstrate in toy settings.

---

> > > > > > > > > ### Author Response · Authors · 2025-12-01
> > > > > > > > >
> > > > > > > > > We thank the reviewer for the continued engagement. We are glad the SAEBench data sources and $L_0$ definitions are cleared up. We further provide the following clarifications.
> > > > > > > > >
> > > > > > > > > ### 1. The Core Argument (Larger Scope)
> > > > > > > > >
> > > > > > > > > > *"I still do not understand the core argument you are making … Are you claiming that GBA SAEs are a solution to feature absorption? ... If your claim is that this is a way to get a pareto-improvement... you should say this explicitly."*
> > > > > > > > >
> > > > > > > > > We wish to correct a misunderstanding: our main claim is **not** that GBA is designed solely as a "feature absorption solver" or a "reconstruction optimizer." Focusing merely on specific metrics like absorption score or the loss-sparsity tradeoff overlooks the fundamental contribution of this work.
> > > > > > > > >
> > > > > > > > > As stated explicitly in our **Introduction (Lines 98-107)**, our contribution is three-fold:
> > > > > > > > >
> > > > > > > > > 1.  Discover and investigate the **neuron resonance phenomenon**, revealing the principle that governs successful feature learning in SAEs.
> > > > > > > > > 2.  **Theoretical Grounding via Training Dynamics:** We provide the first dynamical analysis proving that controlling latent activation frequency allows an SAE to recover **ALL** monosemantic features if they come from our defined statistical model (Theorem 6.1). This addresses the fundamental problem: *under what conditions can an SAE latent provably recover features?*
> > > > > > > > > 3.  **Comprehensive Empirical Validity:** We translate this theoretical insight into a practical algorithm (GBA) and demonstrate through comprehensive evaluation on large-scale LLMs that it achieves competitive performance across a suite of metrics (sparsity-loss tradeoff, consistency, and SAEBench metrics).
> > > > > > > > >
> > > > > > > > > We urge the reviewer to view our contributions through this lens: GBA offers a **new perspective** on SAE training. Metrics like feature absorption and reconstruction error are *proxies* for measuring success. GBA excels at these proxies not because it optimizes them directly, but because it leverages a theoretically grounded mechanism (frequency control) to recover the true underlying features more reliably.
> > > > > > > > >
> > > > > > > > > ### 2. Mechanism of Improvement (Smaller Scope)
> > > > > > > > >
> > > > > > > > > > *"Surely allowing gradient descent to optimize all parameters for reconstruction should result in a better variance explained than manually setting parameters?"*
> > > > > > > > >
> > > > > > > > > **2.1. We would like to clarify several points here.**
> > > > > > > > >
> > > > > > > > > * Firstly, we are not "manually setting parameters"—GBA still uses gradient descent to optimize the encoder/decoder weights, while the bias is adjusted dynamically based on latent activation frequency.
> > > > > > > > >
> > > > > > > > > * Secondly, we have shown that GBA achieves good sparsity-loss tradeoffs (Figure 4) and good explained variance in the SAEBench tasks. We use these results to show that GBA does not sacrifice reconstruction quality.
> > > > > > > > >
> > > > > > > > > * Lastly, dictionary learning is fundamentally different from standard supervised learning tasks in the sense it is highly non-convex and achieving good reconstruction alone does not guarantee good feature recovery. Overall, GBA is theoretically proven to recover monosemantic features under our statistical model and empirically validated to also be good at reconstruction, we believe this is a significant achievement already clearly demonstrated in our paper.
> > > > > > > > >
> > > > > > > > >
> > > > > > > > > **2.2. Why does GBA improve metrics like Absorption and Variance Explained?**
> > > > > > > > >
> > > > > > > > > This is a downstream effect of **better alignment**:
> > > > > > > > > * **Strong Alignment:** GBA guides neurons into the "resonance band," ensuring they align closely and consistently with the true features (consistency is supported by our high **Consistency scores** in Figure 5 and Table 1, and feature recovery is also validated by the toy demo where GBA can recover all features).
> > > > > > > > > * **Absorption:** Feature absorption occurs when sparse
> > > > > > > > > decomposition/splitting of hierarchical features is not robust. Because GBA forces latents to operate at the correct frequency of the underlying features, it has the ability to **consistently** recover the monosemantic features and naturally resists absorbing features.
> > > > > > > > > * **Variance Explained:** By recovering the actual generating factors (true features) rather than suboptimal dense combinations, the model reconstructs the signal more efficiently.

---

> > > > > > > > > > ### Author Response · Authors · 2025-12-01
> > > > > > > > > >
> > > > > > > > > > ### 3. The Toy Model Scope
> > > > > > > > > >
> > > > > > > > > > > *"Again, you are not comparing a SOTA method... you should compare to Batch TopK where you set the K to match the LO of the toy model"*
> > > > > > > > > >
> > > > > > > > > > We believe the current toy model comparison is appropriate and sufficient for the scope of our paper.
> > > > > > > > > >
> > > > > > > > > > **Our Claim:** Standard SAE methods (like TopK) are **brittle regarding parameter choice**, requiring strong prior knowledge of the actual data distribution (e.g., the exact $L_0$). GBA is **robust**, working without such precise prior knowledge.
> > > > > > > > > >
> > > > > > > > > > **Why the current setup fits:**
> > > > > > > > > > 1.  **Addressing the request:** The experiment directly answers your request to *"demonstrate... a case where standard SAEs cannot find correct firing frequencies."* TopK is a standard SAE baseline used throughout the literature and our paper.
> > > > > > > > > > 2.  **The "Prior Knowledge" Problem:** Suggesting we set $K$ to match the $L_0$ of the toy model assumes we *know* the $L_0$, which is indeed required for TopK to work well. This is exactly the problem we highlight: in real-world scenarios, we often do not know the true sparsity level of features in advance, and tuning these hyperparameters is non-trivial.
> > > > > > > > > > 3.  **GBA's Advantage:** GBA succeeds **without** knowing these exact sparsity levels, simply by covering the frequency spectrum. Adding Batch TopK does not change this fundamental conclusion: Batch TopK still requires setting a target sparsity (prior knowledge), whereas GBA adapts dynamically.
> > > > > > > > > >
> > > > > > > > > > We hope this final response clarifies the positioning of our work: it is a theoretically rigorous proposal for frequency-based feature recovery that yields robust empirical results across standard metrics.

---

### Official Review · Reviewer_73R2 · 2025-10-31

**Soundness:** 3
**Presentation:** 3
**Contribution:** 3
**Rating:** 6
**Confidence:** 3

**Summary:**

This paper studies when SAEs can recover interpretable monosemantic features from LLM activations and introduces a frequency-aware training scheme. The authors model activations as sparse nonnegative mixtures of latent features X = H V, then analyze when features are identifiable and recoverable under this statistical framework. They observe a “neuron resonance” effect: neurons reliably learn a feature when their activation frequency p lies in a band around that feature’s occurrence frequency f. Building on this, they propose Group Bias Adaptation (GBA), which partitions SAE neurons into groups with target activation frequencies and adapts biases to match those targets. Theoretically, they prove recovery guarantees for a simplified single-group Bias Adaptation variant under decomposable data and Gaussian feature assumptions, and argue the conditions extend group-wise to GBA. Empirically, on Qwen2.5-1.5B and Gemma2-2B layers, GBA matches TopK on the reconstruction–sparsity frontier and improves cross-seed consistency over TopK, while keeping a high alive-neuron fraction and competitive SAE-Bench interpretability metrics.

**Strengths:**

- The paper gives a formal recovery guarantee for a Bias Adaptation variant and motivates Group Bias Adaptation from the “neuron resonance” effect, then connects the two cleanly with Algorithm 1 and the resonance-to-algorithm narrative.
- Consistency is reported over six seeds with explicit MCS definitions and tables. GBA exceeds TopK across selection schemes, and the results seem robust.
- SAE-Bench metrics on Gemma2-2B show GBA competitive or best on multiple axes while keeping a very high alive-neuron fraction.

**Weaknesses:**

- The formal guarantees assume a Gaussian feature dictionary V; the paper notes this is for convenience and claims methods work when V is non-Gaussian, but it does not provide a systematic analysis of how violations affect results. This remains under-discussed.
- Hyperparameter choices for HTF/LTF are supported by theory and ablations, but there is no measurement or estimation of the actual feature-frequency distribution in a target LLM to motivate ranges.

**Questions:**

1. How sensitive are your guarantees and empirical results to violations of the Gaussian and incoherence assumptions on V?
2. Can you provide an empirical estimate or proxy for the distribution of feature occurrence frequencies in the target models, and explain how this supports your HTF/LTF choices?

---

> ### Author Response · Authors · 2025-11-21
> **Rebuttal (Part 1)**
>
> We thank the reviewer for the careful reading and for the detailed and constructive comments. We are glad that you found the theory–algorithm connection, the consistency analysis, and the SAE-Bench results compelling. We address your two main concerns and questions below.
>
> *Please note*: In the following, we incorporate coherent modifications to Appendix L to address your concerns. However, to keep the current line numbering consistent for all reviewers, we do not move these changes to the main sections yet.
> Some minor edits to the main text is marked with red color for visibility.
>
> ---
>
> ### Q1. Assumptions on the feature dictionary $V$ (Gaussianity and incoherence)
>
> *“The formal guarantees assume a Gaussian feature dictionary $V$; the paper notes this is for convenience and claims methods work when $V$ is non-Gaussian, but it does not provide a systematic analysis of how violations affect results. … How sensitive are your guarantees and empirical results to violations of the Gaussian and incoherence assumptions on $V$?”*
>
> Thank you for raising this important point. We clarify the role of the Gaussian assumptions below.
>
> **(a) What parts of the proof use Gaussianity?**
>
> In our current presentation (Sec. 6 and App. B), we specialize to the case where rows of $V$ are i.i.d. Gaussian. This is primarily used to obtain clean, closed-form concentration bounds on:
>
> 1. (App. B.4.1) Inner products between different feature directions, so we can control interference between features when multiple are active, and
> 2. (App. B.4.2) Pre-activations $y = w^\top x$ and their sparsity under the data model $X = HV$.
>
> In specific, we employ the **Gaussian conditioning** technique, which allows us to decompose a high dimensional Gaussian random vector into components that are explicitly dependent on the conditioning event and components that are independent Gaussian noise. We find that this decomposition is crucial for rigorously analyzing the behavior of the pre-activations and their sparsity patterns.
> Theoretically extending these probabilistic techniques to non-Gaussian settings is a non-trivial task and out of the scope of this paper.
>
>
> **(b) Empirical extension to more general $V$**
>
> An important generalization of this is to consider $V$ drawn from unit sphere, as the layer normalization in practical LLM models effectively enforces feature vectors to have similar norms.
> In fact, our synthetic experiments in Sec. 3 is conducted with $V$ drawn from a uniform distribution on the unit sphere, and the results demonstrated in Fig. 2 show similar resonance behavior as predicted by the theory. Also, the empirical results on real LLM activations (Sec. 7) further confirm that the proposed GBA method works well when $V$'s rows are just the features learned by the LLM, which are non-Gaussian and not perfectly incoherent.
>
> We incorporate the above discussion into the "DISCUSSIONS FOR GAUSSIAN FEATURE ASSUMPTION" Section at the end of the appendix, and will later move it to relevant sections in the revision.

---

> ### Author Response · Authors · 2025-11-21
> **Rebuttal (Part 2)**
>
> ### Q2. Feature frequency distribution and HTF/LTF choices
>
> *“Hyperparameter choices for HTF/LTF are supported by theory and ablations, but there is no measurement or estimation of the actual feature-frequency distribution in a target LLM to motivate ranges. … Can you provide an empirical estimate or proxy for the distribution of feature occurrence frequencies in the target models, and explain how this supports your HTF/LTF choices?”*
>
> Thanks for this insightful question and suggestion. We agree that having an empirical view of the feature frequency distribution in the target LLMs would strengthen the motivation for our HTF/LTF choices. We address this in two parts:
>
> **(a) Using neuron activation frequencies as a proxy for feature frequencies**
>
> Our theoretical model explicitly links feature occurrence frequency $f$ and neuron activation frequency $p$: in the monosemantic regime, a neuron that has converged to feature $i$ should activate with frequency $p \approx f_i$. This suggests we can use the neuron's empirical activation frequencies as a proxy for the *unobserved* feature frequencies in the LLM.
>
> We have incorporated a few new figures to the "ADDITIONAL RESULTS ON NEURON ANALYSIS" section at the end of the appendix, where we provide plots histograms of neuron activation frequencies for TopK SAEs on Qwen2.5-1.5B model.
> Note that computing actual distribution of feature occurrence frequencies in the target models is impossible because we don't know the ground truth feature dictionary $V$. Our proxy is to use TopK methods here and compare it with our GBA results. The reason is that  TopK does not explicitly enforce target frequencies, so the distribution of activation frequencies learned by TopK also serves as a natural reflection of the underlying feature frequency distribution in the data.
> And our method achieves comparable performance to TopK, which further supports the validity of our proxy.
>
>
> In addition, we provide two SAE dashboard visualizations at the end of the paper, with additional dashboards available in the supplementary material (dashboard/). Each dashboard includes per-neuron peak activations and activation-frequency histograms to offer further qualitative evidence of feature selectivity and usage.
>
> **(b) Justifying the HTF/LTF choices**
>
> However, we would like to note that our design of HTF/LTF is primarily motivated by covering a wide range of feature frequencies, rather than matching the exact feature frequency distribution.
> Although a careful design of the neuron group could potentially benefit neuron utilization and reconstruction performance, our empirical evidence still shows that simple geometric spacing with a wide HTF/LTF range is a practical and robust choice in the absence of precise knowledge of the feature frequency distribution.
> This also explains why our GBA method performs well across different models and layers without the need for extensive hyperparameter tuning.
>
> ---
>
> We again thank the reviewer for highlighting these points.
> We agree that making the dependence on dictionary assumptions more explicit and adding an empirical view of feature frequencies will make the paper clearer and more convincing.
> We encourage the reviewer to check out the updated appendix "L REVISION" section where we have added the relevant discussions and figures.

---

### Official Review · Reviewer_nKpu · 2025-11-01

**Soundness:** 2
**Presentation:** 3
**Contribution:** 2
**Rating:** 4
**Confidence:** 2

**Summary:**

This paper proposes an alternative SAE training approach designed around adapting the bias of subset of neurons / SAE Latents to create latents which track features of varying frequencies. This approach is motivated by an observed phenomena called "neuron resonance" - neurons learn monosemantic features when their frequencies are similar. They benchmark their method against other SAEs using SAE bench.

**Strengths:**

- Clarity: The paper is mostly well written and has useful explanatory diagrams. The key points / insights and core questions are made clear to the reader.
- Originality: Directly attempting to mediate SAE latent activation frequency is an interesting proposal.
- Significance: Training GBA SAEs on language models and not just synthetic examples makes the results much more interesting / potentially relevant to use in the real world.

**Weaknesses:**

- It is trivially true that if a latent is going to track a feature, it needs to fire at a similar frequency. Other SAE approaches hope that sparsity inducing loss will match features to their underlying frequencies indirectly which has the benefit of not requiring that you set target activation frequencies. How should we decide what these target activation frequencies should be? Ideally, SAEs are as unsupervised as possible and this approach risks creating challenging hyper-parameter tuning.
- The paper doesn't compare there SAEs to Matrioshka SAEs ("Learning Multi-Level Features with Matryoshka Sparse Autoencoders") which are indirectly related by virtue of learning higher and lower level features which may relate to more or less frequent features. More generally, the L0 of SAEs in Table 2 is fairly high. Ideally, use of SAE bench should show results for a range of sparsity levels. If the point of these SAEs is to avoid using multiple SAEs with different sparsity levels (and often capture features of different frequencies), this should be addressed more directly.

**Questions:**

- SAE quality is often measured by inspection of SAE latent dashboard - max activating examples, activation frequencies etc. The absence of feature dashboards is conspicuous -> can you please share feature dashboards? Ideally, comparing features with those found by other architectures and looking for signs that wSAEs are providing a more useful lens on the analysis of model activations. It seems plausible that they might.

---

> ### Author Response · Authors · 2025-11-21
> **Rebuttal (Part 1)**
>
> Thank you for your thoughtful review and for highlighting both the clarity and potential significance of our work. We particularly appreciate your comments on the motivation for controlling SAE latent activation frequencies. We address your main concerns in turn.
>
> *Please note*: In the following, we incorporate coherent modifications to Appendix L to address your concerns. However, to keep the current line numbering consistent for all reviewers, we do not move these changes to the main sections yet.
> Some minor edits to the main text is marked with red color for visibility.
>
>
> ---
>
> ## Response to Weakness 1:
> > It is trivially true that if a latent is going to track a feature, it needs to fire at a similar frequency. Other SAE approaches hope that sparsity inducing loss will match features to their underlying frequencies indirectly which has the benefit of not requiring that you set target activation frequencies. How should we decide what these target activation frequencies should be? Ideally, SAEs are as unsupervised as possible and this approach risks creating challenging hyper-parameter tuning.
>
> This piece of comment is a bit long and contains some critical misunderstandings. In the following, we will address this comment sentence by sentence.
>
> ### (a). “It is trivially true that if a latent is going to track a feature, it needs to fire at a similar frequency.”
>
>
> We fully agree with the *forward* direction: once a neuron has already learned a single feature, then its activation frequency will match the feature’s occurrence frequency. This is indeed intuitive and not our contribution.
>
> Our work, however, is about the *reverse* direction under a concrete training procedure:
>
> > If we **force** a neuron to have an activation frequency $p$, under suitable conditions this neuron will **provably** converge to a single feature whose occurrence frequency $f$ lies in a corresponding “resonance band”.
>
> Concretely:
>
> * We study an SAE trained with **bias adaptation** (BA), where activation frequencies are controlled via biases rather than via generic sparsity penalties.
> * Under a generative model $X = HV$, we prove **Theorem 6.1**, which shows that if the neuron frequency $p$ lies in the theoretically characterized range (the resonance band), then **every monosemantic feature is recovered** by at least one neuron within a constant number of iterations.
> * In other words, our theorem is about *training dynamics and recovery guarantees*: not just “if a neuron already tracks feature (i), its firing rate is $f_i$”, but “if we choose $p$ in a certain range and run BA, then the neuron *will learn* a feature of that frequency.”
>
> We also emphasize that our results **do not require knowing feature frequencies $f$ in advance**. The theory shows that each feature with frequency $f$ is recovered when some neurons have target frequency $p$ lying in a band $f \\lesssim p \\lesssim \\min(\\sqrt{f}, d f)$ (up to log factors). This is why, in GBA, we use *geometrically spaced* target frequencies: this grid of $p$’s covers the feasible range and makes it likely that each feature falls within a suitable resonance band for some group.
>
> We have clarify this point in "CLARIFICATION ON NEURON RESONANCE" subsection under section L in the appendix, and will later move it to introduction and remind the reader in relevant sections in the revision.

---

> ### Author Response · Authors · 2025-11-21
> **Rebuttal (Part 2)**
>
> ### (b). Contribution beyond “hoping” SAEs adapt to feature frequencies. ("Other SAE approaches ... frequencies.")
>
> You correctly note that existing SAEs typically impose a sparsity penalty (e.g., L1) or constraint (TopK), pick a global hyperparameter (e.g., $\\lambda$ or $K$), and then **hope** that training and optimization serendipitously align neurons with the underlying feature frequency spectrum. We note that parameters $\\lambda$ and $K$ also need to be tuned, with the goal that these sparsity-inducing schemes implicitly align neurons with the underlying feature frequency spectrum.
>
>
>
> Our contribution is to move from this “hope-based” view to a **theoretically-grounded, frequency-aware training mechanism**. We directly probe the frequency alignment between neurons and features. In particular,
>
> * We propose **bias adaptation (BA)** and its practical extension **GBA**, which explicitly maintain neuron activation frequencies at prescribed values via a simple control loop on the biases.
> * We provide what is, to our knowledge, the **first rigorous training-dynamics analysis of SAE feature recovery**, showing that BA recovers *all* monosemantic features under a well-defined statistical model. This significance of this contribution is also acknowledged by Reviewer fF2q and Reviewer 73R2.
>
> * We then show experimentally that GBA scales to LLMs (Qwen2.5-1.5B, Gemma2-2B) and achieves a Pareto frontier in reconstruction–sparsity–consistency trade-offs, competitive with or better than widely used baselines (TopK, L1, JumpReLU, GatedSAE), while maintaining >99% neuron aliveness on SAEBench.
>
> Thus, while prior methods treat feature-frequency adaptation as an emergent byproduct of generic sparsity, our work:
>
> 1. Identifies **neuron resonance** as a key mechanism,
> 2. **Characterizes** the feasible frequency band where successful feature learning is guaranteed, and
> 3. Designs a **practical algorithm (GBA)** that operationalizes this mechanism and empirically validates it on real LLMs.
>
>
> ---
>
> ### (c). Target frequencies and hyperparameter tuning ("How should we ... hyper-parameter tuning")
>
> You raise a valid concern that specifying target activation frequencies might introduce new tuning burdens. Our ablation in Fig. 6 directly addresses this:
>
> * We show that performance is **stable** once the number of groups $K \\ge 10$ and the highest target frequency (HTF) is around $0.5$, with lowest target frequency (LTF) between $10^{-3}$ and $10^{-4}$. Note that this is a mild assumption. If HTF exceeds $0.5$, this means that there exists a monosemantic feature that is shared by at least half of the text tokens, which is not realistic in practice.
> Another justification is that, at initialization, with zero biases, all neurons activate 50% of the time in expectation (as we have mentioned in line 401-402).
> And setting LTF to $10^{-4}$ is also a mild assumption, which means we ignore too rare features that fire less than $10^{-4}$ of the time, as it is hard to infer whether this neuron is learning a meaningful feature or just close to dead.
> * The usage of **geometrically spaced** frequencies is well justified by our characterization of resonance bands, where we show only logarithmically many frequency groups are needed to ensure sufficient coverage of the feature frequency spectrum.
> * Based on those experiments, we propose a **simple, dataset-agnostic rule** (Sec. 5): HTF = $0.5$, LTF (= $10^{-3}$–$10^{-4}$, and $K \\ge 10$. We use the *same* rule across datasets (Pile-Github, Wiki, and also the evaluation datasets used in SAEBench) **without task-specific tuning**.
> * The rule is robust across different datasets and tasks, as demonstrated in our experiments.
>
> We will highlight this rule more prominently to emphasize that GBA is **nearly tuning-free in practice** and does not require feature-level supervision.

---

> ### Author Response · Authors · 2025-11-21
> **Rebuttal (Part 3)**
>
> ## Response to Weakness 2:
>
> > The paper doesn't compare there SAEs to Matrioshka SAEs ("Learning Multi-Level Features with Matryoshka Sparse Autoencoders") which are indirectly related by virtue of learning higher and lower level features which may relate to more or less frequent features. More generally, the L0 of SAEs in Table 2 is fairly high. Ideally, use of SAE bench should show results for a range of sparsity levels. If the point of these SAEs is to avoid using multiple SAEs with different sparsity levels (and often capture features of different frequencies), this should be addressed more directly.
>
>
> ### (a). Comparison to Matryoshka SAEs and $L_0$ in SAEBench ("The paper doesn't compare there SAEs to Matrioshka SAEs...the L0 of SAEs in Table 2 is fairly high.")
>
> We thank the reviewer for bringing forward the Matryoshka SAEs. However, we would like to bring to attention some important distinctions that the Matryoshka SAE method has from our approach in terms of the goals that they aim to achieve.
>
> * Matryoshka uses **nested dictionaries** and each dictionary has an individual loss for recovering the inputs at different hierarchy levels. This is primarily aimed at decomposing the hierarchy of features in the data.
> * GBA places **multiple frequency-sensitive groups in a single SAE**, and only a single loss is applied to all groups.
>
> Crucially, the "single-loss" structure means that GBA is not guaranteed to decouple hierarchy levels.
> Think of the case where we have one high-level feature $a$ and two low-level features $(b, c)$. Suppose in the data we always have $a$ co-occur with either $b$ or $c$.
> The "single-loss" structure means that the model can represent $a+b$ and $a+c$ as two distinct features, mixing hierarchy levels, and there is no incentive to separate $a$, $b$, and $c$ into different groups if the loss is perfectly minimized by the mixed representations.
>
> This fact does not violate our theoretical study, as our assumption on the coefficient matrix $H$ having uniformly random supports also rules out hierarchical feature structures (line 419-420).
> This is the reason why we primarily focus on comparing GBA to other single-loss SAEs (TopK, JumpReLU, L1) rather than Matryoshka, and demonstrate the advantages of frequency-aware training in this context.
> How to extend GBA to handle hierarchical feature decomposition is an interesting direction for future work.
>
> Nevertheless, we have also incorporated an enhanced SAEBench evaluation in Appendix L.9, where we include Matryoshka as an additional baseline for comparison.
> For the reviewer's convenience, we would like to copy a relavant result table here with lower $L_0$ regimes (100-200):
>
> | Metric | GatedSAE | TopK | BatchTopK | Matryoshka | JumpReLU | GBA (ours) | Standard |
> | :--- | :---: | :---: | :---: | :---: | :---: | :---: | :---: |
> | $L_0$ | 175.2 | 173.4 | 168.6 | 166.7 | 162.5 | 132.9 | 125.3 |
> | Explained Variance $\uparrow$ | 0.812 | _0.816_ | _0.816_ | 0.793 | 0.812 | **0.859** | 0.730 |
> | Absorption Score $\downarrow$ | 0.2657 | 0.0838 | 0.0424 | _0.0083_ | 0.0821 | **0.0022** | 0.3529 |
> | SCR Metric $\uparrow$ | 0.323 | 0.321 | _0.349_ | **0.391** | 0.303 | 0.107 | 0.209 |
> | Sparse Probing $\uparrow$ | 0.957 | 0.956 | 0.956 | _0.957_ | **0.958** | 0.954 | 0.954 |
> | TPP Metric $\uparrow$ | 0.072 | 0.100 | 0.094 | **0.209** | 0.105 | _0.144_ | 0.018 |
> | Alive Fraction $\uparrow$ | 0.890 | **0.910** | 0.856 | _0.897_ | 0.719 | 0.723 | 0.719 |
> | RAVEL Disent. $\uparrow$ | _0.730_ | 0.709 | 0.714 | **0.739** | 0.728 | 0.710 | 0.665 |
> | RAVEL Cause $\uparrow$ | 0.710 | 0.677 | 0.676 | **0.735** | _0.710_ | 0.658 | 0.603 |
> | RAVEL Isolation $\uparrow$ | 0.751 | 0.742 | _0.752_ | 0.743 | 0.747 | **0.762** | 0.727 |
>
> Notably, even though GBA is not designed for hierarchical feature recovery, it still demonstrates superior performance on "Explained Variance", "Absorption Score", and  "RAVEL Isolation" metrics, and remain competitive in other interpretability metrics (e.g., TPP), demonstrating its effectiveness in learning meaningful and disentangled features.

---

> ### Author Response · Authors · 2025-11-21
> **Rebuttal (Part 4)**
>
> ### (b). Comparison to multiple SAEs with different sparsitly levels ("If the point of these SAEs is to avoid using multiple SAEs with different sparsity levels ...")
>
> We assume the reviewer refers to Balagansky et al. (2023) where they propose the Hierarchical TopK SAE. A key structural distinction is that hierarchical SAEs use multiple losses (one per dictionary/level) to encourage explicit, decoupled recovery of features at different hierarchical scales, whereas GBA uses a single reconstruction loss across all frequency groups. This is the same point we made above regarding Matryoshka SAEs.
> Because of this, GBA is not a direct replacement for multi‑loss methods: the single‑loss objective does not by itself incentivize separation of hierarchical levels, even if groups use different target frequencies.
>
> On the other hand, we also note that if we only use a single loss across all neurons but separate groups with different Top‑$K$ values, this is not a meaningful strategy either.
> In this case, if one only varies Top‑K per group while keeping a single loss, the model would effectively view this new SAE model as equivalent to a single Top‑K SAE with K equal to the sum of the groups’ K values.
> The above discussion has already been incorporated into the "ADDITIONAL SAEBENCH EVALUATION" subsection under Appendix L.
>
> ---
> ## Response to Question:
> > SAE quality is often measured by inspection of SAE latent dashboard - max activating examples, activation frequencies etc. The absence of feature dashboards is conspicuous -> can you please share feature dashboards? Ideally, comparing features with those found by other architectures and looking for signs that wSAEs are providing a more useful lens on the analysis of model activations. It seems plausible that they might.
>
> We agree that feature dashboards are an important part of SAE evaluation. In Figure 1 (right), we actually show the activation of a GBA-trained neuron on sentences using the feature dashboard tool.
> However, given the numerous neurons and limited space, we only included one example in the main paper.
> In addition, we provide two SAE dashboard visualizations at the end of the paper, with additional dashboards available in the supplementary material (dashboard/). Each dashboard includes per-neuron peak activations and activation-frequency histograms to offer further qualitative evidence of feature selectivity and usage, showcasing neurons that capture both token-level and topic-level features.
>
>
> **References:**
>
> Balagansky, Nikita, et al. "Train One Sparse Autoencoder Across Multiple Sparsity Budgets to Preserve Interpretability and Accuracy." Proceedings of the 2025 Conference on Empirical Methods in Natural Language Processing. 2025.

---

### Official Review · Reviewer_fF2q · 2025-11-06

**Soundness:** 3
**Presentation:** 2
**Contribution:** 4
**Rating:** 6
**Confidence:** 2

**Summary:**

The paper aims to provide theoretical grounding for understanding and improving monosemantic feature recovery in Sparse AutoEncoders (SAEs). First, authors show how the phenomenon of "neuron resonance" -- i.e., where SAEs best recover features when their feature (neuron) activation frequency falls within a certain "resonance band" around the ground-truth monosemantic feature occurrence frequency -- manifests in a toy setting, then theoretically characterize this phenomenon more generally. The primary contribution is a novel training algorithm based on neuron resonance, group bias adaptation (GBA), which provably (under a given statistical model) recovers monosemantic features. Empirically, GBA achieves performance that is competitive with prior SAE baselines across a variety of common evaluations, while substantially improving the consistency of learned features across seeds and hyperparameter settings and reducing the proportion of dead features.

**Strengths:**

- The paper provides the first (to my knowledge) theoretical treatment of feature recovery in SAEs. Given the very limited current theoretical understanding of SAE training, this is an important and timely contribution to the field, which has so far relied overwhelmingly on empirical analysis alone.
- Experiments across a variety of settings show clear empirical benefits of GBA relative to leading baselines, in addition to their improved theoretical grounding with respect to feature recovery.

**Weaknesses:**

The paper's greatest weakness is poor organization and clarity.
- **Organization:** The main paper should be relatively self-contained, including all the necessary high-level information to understand key contributions. A huge amount of critical content -- including comparison with related work, proposed evaluation metrics, and key information about experiments that are strictly necessary for understanding empirical results -- is provided only in the appendix. For instance:
    - Introductions and basic definitions of novel evaluation metrics in appendix C.2 should be moved into corresponding sections where these terms are used (even if some lower-level details remain in appendices).
    - The related work section (appendix A) *must* appear in the main paper. (This case in particular feels like an instance where authors might have simply moved the entirety of a necessary main-paper section into the appendix in order to save space; but this could easily have been done much more appropriately by compressing the introduction or moving some experiments/ablations to the appendix instead.)
- **Clarity:** Even after looking through the appendix, I am still unable to find definitions of several important experimental details, new terms, etc. necessary to provide a complete assessment (see the Questions section, below).

Please note: **I am completely open to raising my rating (and confidence score) if authors are able to provide this missing information** -- given my (currently incomplete) understanding of the contributions, I feel that the paper would very likely merit a higher score once I have the necessary information to make a full assessment.

**Questions:**

**Essential missing information required for full assessment** (as discussed above, I cannot find any definitions of the following in either the main paper or appendix, making it impossible to interpret key results):
- Missing from section 3 (and appendices):
    - What are m and $\mu$ appearing on the y-axis of fig 2?
- Missing from sec 5 (and appendices):
    - What are $\alpha$ and the corresponding "top $\alpha$ selection rule"?
    - What are Highest Target Frequency (HTF) and Lowest Target Frequency (LTF)?

General questions:
- In sec 3:
    - What/where is the supposed "phase transition" at $d = \sqrt{n}$? Wouldn't this correspond to the rightmost column of the right heatmap (where $d \approx \sqrt{n} = 256$), which doesn't seem to exhibit a "phase transition" with respect to the preceding columns? I would appreciate it if authors could provide:
        - (a) an explanation of what, precisely, "phase transition" is intended to mean here;
        - (b) a clear criterion for categorizing a given result as showing a phase transition, rather than simply a gradual change in feature recovery around good hyperparameter settings;
        - (c) an explanation of how neuron resonance is expected to correspond to either a phase transition or a more gradual change (such as mentioned in (b)), and how this relates to the feasible frequency range in sec 6; and
        - (d) a description of what, specifically, in figure 2 is taken to be evidence of a phase transition. (My understanding is that the "narrow band" in the right $d < \sqrt{n}$ plot is interpreted as the phase transition, where the larger high-FRR region in the left plot is *not* interpreted as such, but rather a more gradual change -- is this correct?)
    - Sec 3 experiments are based on the setting with uniform feature occurrences, unlike the "feature spectrum" (multi-group) settings discussed in later sections. What would the experiments in sec 3 look like given such a feature spectrum (either discretized into groups or varying continuously over a geometric distribution)?
- In sec 4:
    - Is there any particular reason for partitioning neuron TAFs based on a geometric distribution rather than, e.g., a [Zipfian distribution](https://en.wikipedia.org/wiki/Zipf%27s_law) (as is often observed in the context of linguistic features)?

---

> ### Author Response · Authors · 2025-11-21
> **Rebuttal (Part 1)**
>
> We thank the reviewer for the careful and constructive review, especially the acknowledgment of our contributions. Below we address each point.
>
> *Please note*: In the following, we incorporate coherent modifications to Appendix L to address your concerns. However, to keep the current line numbering consistent for all reviewers, we do not move these changes to the main sections yet.
> Some minor edits to the main text is marked with red color for visibility.
>
> ---
>
> ## Q1. Organization and missing definitions
>
> **Related work in the appendix.**
> We agree that related work should appear in the main paper.
> In the submission version, we placed it in the appendix to save space in the main text for the core contributions and technical details, given the strict page limit.
> In the camera ready version, given one more page limit, we will move §A (“Related works”) into the main text, shortening the introduction and moving some ablations to the appendix to stay within the page limit.
> Note that we do not do this in the current version just in order to maintain the line numbers for the reviewers' convenience.
>
> **Evaluation metrics and terminology.**
> You are right that we leave the detailed definitions of TopK and JumpReLU activation, as well as the evaluation metrics such as Feature Recovery Rate (FRR), Maximum Activation, Neuron Z-score, and Max Cosine Similarity (MCS), to the appendix §C.2. This was done to streamline the main text and focus on the high-level ideas and results, while keeping the technical definitions accessible for interested readers.
> In addition, in the main text, we have already provided intuitive explanations of these notions when they are first introduced, along with references to the appendix for formal definitions.
> For instance, we incorporate the FRR explanation in line 157 of the main text and refer to the appendix for the formal definition.
> We believe this strikes a balance between clarity and conciseness.
>
> In the revision we will:
>
> * Add a short “Evaluation metrics” subsection in the main text (at the start of §3 and §5), where we briefly define:
>   * **Feature Recovery Rate (FRR)** for synthetic experiments: the fraction of ground-truth features $v_i$ such that at least one neuron’s weight $w_m$ has cosine similarity with $v_i$ above a threshold $\tau_{\text{align}}$.
>   * **Max Cosine Similarity (MCS)**, neuron percentage, maximum activation, neuron Z-score, etc., in the real-data experiments.
>
> * Inline the definitions of **Highest Target Frequency (HTF)**, **Lowest Target Frequency (LTF)**, and the **top-$\alpha$ selection rule** in §4–5 where they are first used.
>
> We have drafted the relevant text for these changes in the Sec. L at the end of the appendix, and will later move them to the main text in the revision.
> We believe these changes will directly address your concern that “critical content is only in the appendix” and make the main narrative self-contained.

---

> > ### Author Response · Authors · 2025-11-21
> > **Rebuttal (Part 4)**
> >
> > ## Q4. Uniform feature frequency in §3 vs. “feature spectrum” settings
> >
> > You are correct that §3 uses a uniform feature occurrence rate $f = s/n$, whereas later sections discuss multi-group “feature spectra.” We chose the uniform setting in §3 intentionally to isolate the resonance phenomenon in the simplest identifiable regime.
> >
> > - On the theoretical side, §6 also assume uniform feature frequencies in the statement; but it can be easily extended to multiple group case as we mentioned in the remark.
> >
> > - On the empirical side, we also conduct synthetic experiments where we have two groups of features, where one group has a high occurrence frequency $f_{\\text{high}}$ (we term "common features") and the other has a low occurrence frequency $f_{\\text{low}}$ (we term "rare features").
> > We direct the reviewer to Figure 7(c). When the group imbalance ratio $f_{\\text{low}}/f_{\\text{high}}$ becomes extreme, the FRR of single group GBA degrades dramatically for the rare features, which aligns with our intuition that a single group cannot cover both frequencies well.
> > On the other hand, GBA with 4 different groups still maintains a descent FRR across different imbalance ratios, showing the advantage of multiple groups in covering a wide feature spectrum.
> >
> > As a remark, we conduct two-group synthetic experiments to highlight the contrast between single-group and multi-group GBA. It is expected that the same resonance phenomenon and theoretical insights extend to more general feature spectra with multiple frequency groups.
> >
> > ---
> >
> > ## Q5. Why a geometric (not Zipfian) distribution of target frequencies in §4
> >
> > The decision to use **geometrically spaced TAFs** (e.g., 10%, 5%, 2.5%, …) is motivated by both theory and practicality:
> >
> > 1. **Theoretic guided group allocation.** The theoretical resonance condition depends on $p$ being within *at least a multiplicative* band around $f$ (can be even wider though if we have less superposition). A geometric grid guarantees that for any feature frequency $f$ within [LTF, HTF], there exists some group with TAF $p_k$ within a constant factor of $f$, regardless of the exact exponent of the empirical feature distribution.
> >
> > 2. **Better coverage in log-frequency space.** Geometric spacing minimizes the number of groups $K$ needed to cover a wide frequency range [LTF, HTF] to just logarithmic in the ratio HTF/LTF. Empirically, we show in Figure 6 that having $K > 10$ groups is sufficient for covering much of the frequency spectrum.
> > However, a Zipfian spacing would potentially require many more groups to achieve similar coverage, as the frequency decay is slower than geometric. This would dilute the number of neurons per group, and we might risk missing features in that frequency range.
> >
> > 3. **Model-agnostic choice.** Zipf’s law is common but not universal, and different layers/activation subspaces of LLMs can deviate from a single Zipf exponent. Hard-coding a Zipfian TAF schedule risks mis-allocating groups if the actual spectrum is flatter or steeper; geometric spacing remains robust.
> >
> > 4. **Empirical insensitivity once coverage is sufficient.** Figure 6 shows that once HTF and LTF span a wide range and number of groups $K > 10$, performance becomes quite stable no matter how we change the exact group configuration. This is another layer of evidence that geometric spacing is a practical and robust choice.
> >
> > We will explicitly add a sentence in §4 explaining that we use geometric spacing as a generic way to cover a broad log-frequency range without assuming a specific Zipf exponent, and that our ablations show GBA is not sensitive to the exact spacing once coverage is adequate.
> > For now, we have drafted the relevant text in the "DISCUSSIONS FOR GEOMETRIC SPACING OF TARGET ACTIVATION FREQUENCIES" section at the end of the appendix, and will later move it to §4 in the revision.
> >
> > ---
> >
> > Again, we appreciate your detailed feedback and agree that improving the organization and moving key definitions into the main text will significantly clarify the paper. We hope the clarifications above address your questions, and we will make all of the described edits in the revised version.

---

> ### Author Response · Authors · 2025-11-21
> **Rebuttal (Part 2)**
>
> ## Q2. Specific missing definitions
>
> ### (a) Symbols on the y-axis of Figure 2
>
> We apologize for the confusing labeling. Figure 2 is intended to show a **heatmap of FRR as a function of neuron activation frequency $p$ and feature dimension $d$** for the synthetic model. The $\mu$ symbol on the y-axis stands for $10^{-6}$, which is a standard scientific notation for "micro", and the $m$ symbol stands for $10^{-3}$, which is a standard scientific notation for "milli". Thus, the y-axis values represent neuron activation frequencies ranging from $10^{-6}$ to $10^{-1}$.
> We defer the reviewer to https://en.wikipedia.org/wiki/Milli- for the table of scientific prefixes.
> We have incorporated this explanation into the caption of Figure 2.
>
>
> ### (b) “Top-$\alpha$ selection rule” in §5
>
> We would like to thank you to point out that the “top-$\alpha$” terminology might not be immediately clear. To help make this precise, we would like to direct you to the explanation in the main text together with a more formal definition below:
>
> * In the caption of Figure 5, we mention
>     > We take *Max Activation* and *Z-Score* as the selection criteria and plot results for subsets of neurons that rank in the top-$\alpha$ proportion under each criterion, with $\\alpha \\in \\{0.3\\%, 0.05\\%\\}$ (i.e., top-200 and top-30 neurons out of 66k).
>
>     Here,
>
>     - *Max Activation* of a neuron is its maximum activation across all evaluation samples;
>     - *Z-Score* is defined formally in §C.2, which is measured by taking that neuron's activation on samples as an empirical distribution, and then following the standard Z-score formula.
> * For a given fraction $\alpha \in (0,1]$, the **top-$\alpha$ selection rule** means: Sort neurons by the chosen scalar metric (max activation or Z-score) and take the top $\alpha$ fraction of them for evaluation.
>
> The rationale is that in practice, **only a small fraction of neurons are expected to capture meaningful features**, so we focus on the most “active” or “significant” neurons according to the metric. Therefore, we vary $\alpha$ for both extreme subsets (e.g., top 0.3% or 0.05%) and general subsets (e.g., top 10%, 25%, or 50%) to see if our claims on the neuron consistency robustly hold.  And the conclusion is that our claims are indeed robust with respect to $\alpha$. See, e.g., Table 1, where our GBA method achieves higher consistency than TopK method for all $\alpha$.
>
>
> ### (c) Highest Target Frequency (HTF) and Lowest Target Frequency (LTF)
>
> In GBA, we partition the $M$ neurons into $K$ groups, each with a **target activation frequency (TAF)** $p_k$, arranged as a geometric sequence between a largest and smallest value:
>
> * **HTF (Highest Target Frequency)**: $p_1$, the largest TAF in the sequence (e.g., 0.5).
> * **LTF (Lowest Target Frequency)**: $p_K$, the smallest TAF (e.g., $10^{-3}$–$10^{-4}$).
>
> The definition is briefly mentioned in lines 376-377. However, we agree that it would be clearer to define HTF and LTF more explicitly when they are first introduced. The targeted frequences $\\{ p_k\\}_{k=1}^K$, defined as a geometric sequence between a largest and smallest value, are part of the inputs of the GBA algorithm. Intuitively, for each $p_k$, we expect that the neurons in group $G_k$ have average activation frequency roughly equal to $p_k$. In practice, when the input data has sufficient diversity, GBA can correctly identify monosemantic features that activate with frequencies within the range of $p_1$ and $p_K$.
> In addition, our theoretical analyis and synthetic experiments also suggest the existence of a wide "resonance band" where the neuron activation frequency $p$ does not need to exactly match the feature occurrence frequency $f$ for successful learning.
> With this geometric spacing of TAFs between HTF and LTF, we can ensure that most features within this spectrum can be reliably learned when their occurrence frequencies lie between HTF and LTF.

---

> ### Author Response · Authors · 2025-11-21
> **Rebuttal (Part 3)**
>
> ## Q3. “Phase transition” in §3 and §6
>
> You raise several important points about our use of the phrase “phase transition.” We address them below:
>
> ### (a) What we mean by “phase transition”
>
> We mean by “phase transition” the change in the scaling of the admissible neuron-frequency band. Specifically the resonance band we present **in the main text** is:
> $$
> 1/n \\lesssim p \\lesssim \\min\\{1/\\sqrt{n},\\ d/n\\}.
> $$
> Specifically, the **upper bound on $p$** exhibits different dominant terms depending on the ratio $d/\\sqrt{n}$:
> - Heavy superposition ($d \\lesssim \\sqrt{n}$): the upper bound is dominated by $d/n$, so the learnable band grows with $d$.
> - Light superposition ($d \\gtrsim \\sqrt{n}$): the upper bound is dominated by $1/\\sqrt{n}$, so the learnable band is relatively wide and constant in $d$.
>
> **Phase transition:** As $d$ crosses $ \\sqrt{n} $ the dominant upper‑bound term switches from $d/n$ to $1/\\sqrt{n}$, producing a visible change in the resonance band's shape at $d\\approx\\sqrt{n}$. Figure 7(b) demonstrates the theoretical trapezoid learnable region, matching the empirical heatmap in Figure 2 (Right). Intuitively, this transition of the resonance bound reflects a shift in the driving mechanism: heavy superposition (smaller $d$) increases interference and forces tighter alignment between neuron activation frequency $p$ and feature frequency $f$, while light superposition (larger $d$) reduces interference and allows a wider range of $p$ to learn features of frequency $f$.
>
> The theoretical bounds are stated up to logarithmic factors; for large $n$ (e.g., $n=65{,}536$) these log factors are non‑negligible and can shift the empirical boundary in Figure 2 (Right). In that plot the upper bound visibly flattens near $d\\approx100$, which is consistent with the predicted change in scaling once logarithmic correction terms are accounted for (note $\\sqrt{n}\\approx256$, so $\\log n$–scale factors can substantially reduce the effective $d$–scale where the dominant term switches). By contrast, for the small‑$n$ case ($n=128$, $\\sqrt{n}\\approx11$) the high‑dimensional asymptotics and log‑asymptotics do not hold, so the transition is much more gradual and the theoretical prediction is less precise. These finite‑size and logarithmic effects therefore explain why the right heatmap shows a sharper transition while the left heatmap appears smoother.
>
>
> ### (b) Further clarification on the learnable region criterion
>
> As the reviewer also mentions about "an explanation of how neuron resonance is expected to correspond to either a phase transition or a more gradual change ... the feasible frequency range in sec 6", in addition to the above explanation of the "phase transition" at $d \\approx \\sqrt{n}$, we would also like to clarify the  **criterion for the learnable region** where features can be learned reliably.
> Empirically, for finite $(n,d)$, we observe a **rapid but continuous** drop in FRR as $p$ moves outside the band predicted by the theorem; so the transition is not a literal discontinuity but a sharp change in FRR when the theoretical conditions fail.
> From the visualization in Figure 2, we believe the transition from "learnable" to "unlearnable" regions are already quite sharp and clearly visible.
>
> ### (c) Relation to neuron resonance and feasible frequency range
>
> Neuron resonance is the statement that **features with occurrence frequency $f$ are most reliably recovered when neuron activation frequencies $p$ lie in this band**. The theoretical condition $f \\lesssim p \\lesssim \\min\\{\\sqrt{f}, d f\\}$ gives the **feasible frequency range** for resonance:
>
> * As we move from light to heavy superposition (decreasing $d$), the upper limit $p_{\\max}$ shrinks from $\\sqrt{f}$ to $d f$, so the range of $p$ that can “resonate” with a feature of frequency $f$ becomes narrower.
>
>
>
> ### (d) What exactly in Figure 2 is evidence of the transition?
>
> Figure 2 contains two heatmaps:
>
> * **Left (light superposition, $d > \\sqrt{n}$)**: the bright high-FRR region occupies a relatively **wide band in $p$** above $f$.
> * **Right (heavy superposition, $d < \\sqrt{n}$)**: the high-FRR region collapses into a **much narrower diagonal band**, indicating that $p$ must track $f$ more tightly.
>
> The “evidence” we have in mind is the contrast between these two regimes: the same feature frequency $f$ yields a wider resonance band in light superposition and a narrow band in heavy superposition, aligning with the theory’s change in upper bound at $d \\approx \\sqrt{n}$.
> We add a more explicit explanation of phase transition in Figure 2 at the end of the appendix in the "ENHANCED CAPTION FOR FIGURE 2: EVIDENCE OF PHASE TRANSITION" section in Appendix L, and will later move it to the main text in the revision.

---

### Author Response · Authors · 2025-11-27
**Summary of New Additions in the Revision**

We have significantly expanded **Appendix L** to address reviewer questions regarding theoretical assumptions, provide deeper empirical analysis, and include comprehensive benchmarking. We direct the reviewers to the following newly added sections:

* **Theoretical Clarifications & Assumptions (App L.1, L.4, L.5, L.6):**
    * (Reviewer 73R2) **App L.1** discusses the Gaussian feature assumption used in our theory, clarifying that while chosen for clean concentration bounds, our method empirically generalizes to non-Gaussian settings (e.g., unit sphere features and real LLM activations).
    * (Reviewer fF2q) **App L.4** provides the rationale for the geometric spacing of target activation frequencies, explaining how it ensures efficient coverage of the feature frequency spectrum in log-space.
    * (Reviewer VpeD & Reviewer nKpu) **App L.5** clarify the subtle differences between "neuron can activate at specific frequency if it converges to a feature direction" versus "when the neuron is trained with a target frequency, it can converge to a feature with similar occurrence frequency".
    * (Reviewer nKpu) **App L.6** explicitly distinguish GBA from Hierarchical/Matryoshka SAEs.

* **Enhanced Metrics & Captions (App L.2, L.3):**
    * (Reviewer fF2q) **App L.2** adds formal definitions for all evaluation metrics used (FRR, MCS, Z-score, etc.) that will be later integrated into the main text.
    * (Reviewer 73R2) **App L.3** provides an enhanced caption for **Figure 2**, explicitly detailing the empirical evidence for the theoretical phase transition between light and heavy superposition regimes.

* **Extended Empirical Analysis (App L.7):**
    * (Reviewer 73R2 &. Reviewer nKpu) **App L.7 "ADDITIONAL RESULTS ON NEURON ANALYSIS"** presents a granular analysis of learned neurons. **Figures 15 & 16** provide scatter plots correlating Z-scores with Maximum Activation and Cross-seed Consistency, revealing that GBA neurons capturing more infrequent features compared to TopK. We also include feature dashboards (**Figures 17 & 18**) demonstrating clear bimodal activation patterns and semantic coherence.

* **New Controlled Experiments (App L.8):**
    * (Reviewer VpeD) **App L.8 "WHY FREQUENCY-AWARE TRAINING? A TOY EXAMPLE"** addresses the request for a controlled demonstration. We introduce a synthetic setting with imbalanced feature sparsity to show how standard TopK could fail under mild sparsity misspecification, while GBA with frequency coverage achieves 100% Feature Recovery Rate.

* **Comprehensive Benchmarking (App L.9):**
    * (Reviewer VpeD & Reviewer nKpu) **App L.9 "ADDITIONAL SAEBENCH EVALUATION"** provides a rigorous evaluation of GBA on the SAEBench suite.
        * **Figure 19** displays full Pareto curves comparing GBA against six standard baselines (Standard, BatchTopK, Gated, JumpReLU, TopK, Matryoshka) across nine metrics. The results confirm GBA’s competitive performance, particularly in **Explained Variance**, **Absorption Score**, and **RAVEL Isolation**.
        * **Tables 5-7** provide detailed numerical comparisons at specific $L_0$ intervals (100–200, 300–400, and 600+), offering a quantitative view of GBA's performance across diverse sparsity regimes.

---

### Meta-Review · Area_Chair_jiLb · 2026-01-11

**Summary:**

The authors introduce a new sparse autoencoder (SAE) training algorithm called Group Bias Adaptation. This algorithm is motivated by a natural observation that they call "neuron resonance" based on observing specific features' occurrence frequencies. They theoretically analyze the algorithm under a certain sampling model for the data, and show empirical performance on practical LLMs of sizes up to 2 billion parameters.

The paper received several high quality reviews, but the initial scores were mixed (6,6,4,2). The authors provided (very) lengthy rebuttals --- in particular, they had an extensive back-and-forth with the reviewer who gave the lowest score. Some of the concerns are described in more detail below, but my opinion is that the negative review stemmed from a rather-deep disagreement on the motivation behind the paper, and I felt that the authors' responses were generally satisfactory.

My own reading of the paper is a bit more positive than the average score might suggest: it is a very-well-executed paper which is one of the first theoretically-grounded efforts towards understanding SAE training algorithms, and may shed light on designing autoencoder-style interpretability methods moving forward. Therefore, I tentatively recommend acceptance.

**Reviewer Concerns:**

fF2q asked several clarification questions, which the authors addressed adequately.

fF2q had a question around the claimed "phase transition" in the resonance band. I think this is a bit of an overloaded term in the field, and would suggest the authors just call it a regime shift instead.

Several reviewers asked for additional experiments; the authors added lots more.

nKPu asked about SAE latent dashboards, and the authors addressed this to a limited extent.

73R2 asked if the theory would work under a less stringent data model, and asked for additional evidence of algorithm robustness. I think the authors gave a reasonable response here.

VpED had some questions about high level motivation and what core problem the paper meant to address. There was lengthy back-and-forth here but I think the responses were generally satisfactory. There were lingering disagreements on whether the L0 parameter was set too high, and whether GBA solves feature absorption.

**Reviewer Scores:**

Reviewer fF2q explicitly said that if some clarification questions were answered (which the authors responded to, suitably) then they would have raised their score.

Reviewer 73R2 asked if the theory would work under a less stringent data model, and asked for additional evidence of algorithm robustness. I think the authors gave a reasonable response here. They would have kept or increased their score.

Reviewer nKUP had some important initial objections (one of which was that the resonance phenomenon was trivial). The authors provided a good response, showing that the resonance was indeed being leveraged _within the algorithm_ via dynamic bias adaptation. The reviewer may have increased the score. They also asked for additional comparisons with Matryoshka SAEs, which the authors provided via additional experiments.

Reviewer VpeD was the most negative review, with lots of disagreements with the authors and a lengthy back and forth. There are probably fundamental disagreements here, and I'm not sure if they would have changed their score.

---

### Decision · Program_Chairs · 2026-01-26

Accept (Poster)